# Decomposable Neural Symbolic Regression

## Abstract

Symbolic regression (SR) models complex systems by discovering mathematical expressions that capture underlying relationships in observed data. However, most SR methods prioritize minimizing prediction error over identifying the governing equations, often producing overly complex or inaccurate expressions. To address this, we present a decomposable SR method that generates interpretable multivariate expressions leveraging transformer models, genetic algorithms (GAs), and genetic programming (GP). In particular, our explainable SR method distills a trained "opaque" regression model into mathematical expressions that serve as explanations of its computed function. Our method employs a Multi-Set Transformer to generate multiple univariate symbolic skeletons that characterize how each variable influences the opaque model's response. We then evaluate the generated skeletons' performance using a GA-based approach to select a subset of high-quality candidates before incrementally merging them via a GP-based cascade procedure that preserves their original skeleton structure. The final multivariate skeletons undergo coefficient optimization via a GA. We evaluated our method on problems with controlled and varying degrees of noise, demonstrating lower or comparable interpolation and extrapolation errors compared to two GP-based methods, three neural SR methods, and a hybrid approach. Unlike them, our approach consistently learned expressions that matched the original mathematical structure. Similarly, our method achieved both a high symbolic solution recovery rate and competitive predictive performance relative to benchmark methods on the Feynman dataset.

## 1 Introduction

Deep learning-based systems have achieved remarkable success across various domains due to their ability to model complex, nonlinear functions. However, these systems are often described as "opaque" models[1] in the literature, emphasizing the high complexity of the functions they learn and their many required parameters. Their lack of interpretability and traceability poses challenges in applications where understanding the underlying mechanisms is crucial (Linardatos et al., 2021).

In scientific research, particularly in the physical sciences, interpretability is essential for uncovering governing equations that describe observed phenomena (Camps-Valls et al., 2023; Lee & Kumar, 2023). Data-driven methods play a crucial role in this process by identifying patterns and relationships directly from empirical data. Symbolic regression (SR) emerges as a powerful technique, offering an alternative to opaque models for discovering interpretable mathematical representations from data (Filho et al., 2020). Unlike conventional regression techniques that assume predefined model structures, SR searches the space of mathematical expressions to identify compact and accurate equations describing the observed data (Kronberger et al., 2024).

SR methods have been successfully applied to real-world problems such as dynamical system modeling (Schmidt & Lipson, 2009), astrophysics (Cranmer et al., 2020; Delgado et al., 2022), fluid mechanics (Reinbold et al., 2021), and agriculture [Ref. 1, Ref. 2].[2] In certain applications, SR methods have achieved performance comparable to state-of-the-art opaque models, showcasing their potential in accurately capturing the complexities of observed data (Orzechowski et al., 2018). However, for large-scale datasets, the computational cost of direct SR discovery often becomes prohibitive as efficiency degrades with increasing

---

[1]This term is preferred over "black-box": acm.org/diversity-inclusion/words-matter
[2]These references are hidden for double-blind review purposes.

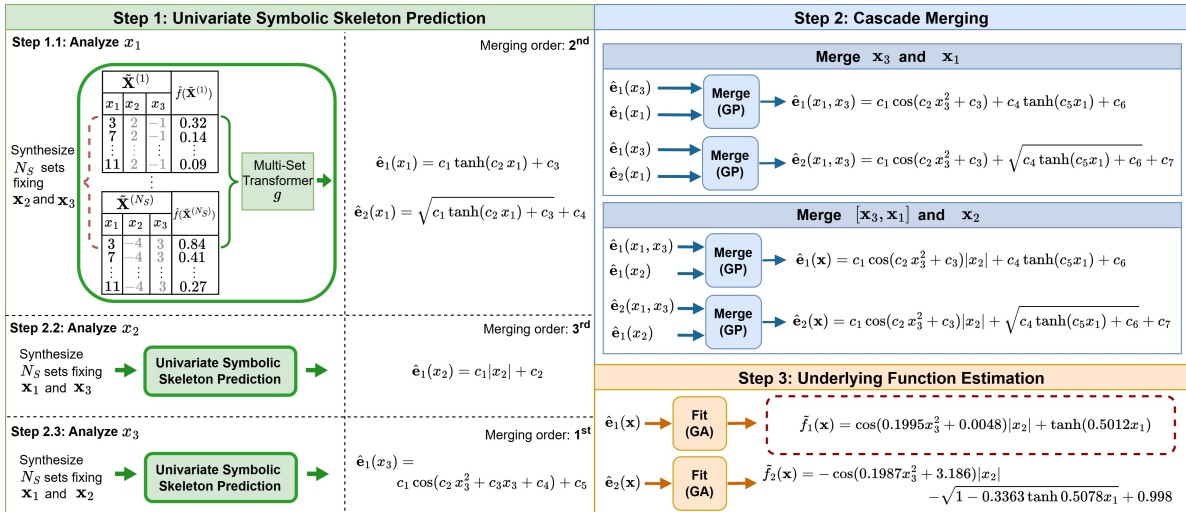

Figure 1: Overview of the SeTGAP symbolic discovery process.

data volume. As a result, some approaches use bagging to process data in manageable subsets (Kamienny et al., 2022). Instead, high-performing opaque models trained on large datasets can easily encapsulate or approximate the underlying system's behavior. As such, in conjunction with SR methods, they can be used as a proxy to distil mathematical expressions. This provides a bridge for discovering or approximating governing equations for complex phenomena, such as modeling neutrino oscillation behavior from the high-resolution data gathered by next-generation telescopes and large-scale simulations (KM3NeT Collaboration, 2025).

Furthermore, most existing SR approaches focus on minimizing prediction error rather than extracting governing equations (Bertschinger et al., 2024). This often leads to expressions that fit the data well but are overly complex and difficult to interpret (La Cava et al., 2021). Thus, recent neural SR techniques (Biggio et al., 2021; Kamienny et al., 2022; Bertschinger et al., 2024) use transformer-based models to generate symbolic skeletons or mathematical expressions. However, they process all variables simultaneously and often fail to capture the functional form between each variable and the system's response correctly [Ref. 1][2].

To tackle these limitations, we introduce a novel multivariate SR method called **S**ymbolic R**e**gression using **T**ransformers, **G**enetic **A**lgorithms, and genetic **P**rogramming (SeTGAP), illustrated in Fig. 1. SeTGAP is a post-hoc explainable SR approach designed to provide explanations for opaque models. Hence, it extracts concise and human-readable mathematical expressions that serve as symbolic surrogates of an already trained opaque model's learned function. In Sect. 3.2, we describe how a Multi-Set Transformer generates multiple univariate symbolic skeletons that capture the functional relationships between each independent variable and the opaque model's response. A genetic algorithm (GA)-based selection process then filters out low-quality skeletons, retaining the most informative ones. In Sec. 3.3, these skeletons are merged through an incremental genetic programming (GP)-based procedure. Sec. 3.3.1 details how, given two skeletons to be merged, a pool of candidate skeleton combinations is produced, ensuring that the merged expressions remain aligned with the original structures. Sec. 3.3.2 explains the selection of the most appropriate combination from the pool of skeleton combinations. Sec. 3.3.3 outlines our cascade approach, which incorporates one variable at a time into the merging process. Finally, Sec. 3.3.4 details a GA-based refinement of the numerical coefficients in the resulting multivariate expression. Our experiments show that SeTGAP consistently recovers the correct functional form of the underlying equations, unlike existing GP-based and neural SR methods, which often fail to do so. In addition, SeTGAP achieves lower or comparable interpolation and extrapolation errors across various problem settings, including scenarios with controlled and varying noise levels.

## 2   Related Work

SR constitutes an NP-hard problem that becomes increasingly complex as the number of observations, operators, and variables increases (Virgolin & Pissis, 2022). As such, brute-force approaches become infeasible.

Most SR approaches are based on GP (Kronberger et al., 2024). They evolve candidate solutions, or programs, to identify suitable functional forms and optimize their parameters. This process involves applying genetic operators such as mutation, crossover, and selection to populations of expressions (Cranmer, 2023; Makke & Chawla, 2022; Orzechowski et al., 2018; Schmidt & Lipson, 2009). A candidate's fitness is evaluated based on its ability to minimize prediction error, guiding the evolution toward more accurate models.

GP's flexibility in generating programs of varying lengths allows it to explore a vast search space. One significant drawback is called "code bloat," the uncontrolled growth of program size during evolution. This phenomenon leads to increasingly complex solutions, inflating computational costs and reducing model interpretability (Dignum & Poli, 2007; Poli et al., 2007). The redundancy in the search space further exacerbates this issue, as many solutions represent nearly equivalent functions within the studied domain but differ syntactically (Ebner, 1999; Amir Haeri et al., 2017). This redundancy increases the difficulty of identifying optimal solutions and contributes to GP's bias toward generating larger expressions (Langdon et al., 1999).

Another limitation is that fitness evaluation typically focuses on the overall output of a program, neglecting evaluation of the intermediate components that contribute to the final result. Thus, Arnaldo et al. (2014) proposed to optimize the generated programs before the selection step by optimizing the fitness contributions of its subexpressions. Conversely, Jiang & Xue (2023) proposed Control Variable Genetic Programming (CVGP), which introduces independent variables into the analysis sequentially through a series of controlled experiments. In each stage, a GP learns simplified expressions by holding all but one variable constant, progressively refining candidate equations as more variables are incorporated. This iterative procedure enables CVGP to recover complex symbolic expressions more reliably than end-to-end approaches.

GP has been combined with deep learning techniques for SR. Mundhenk et al. (2021) introduced a recurrent neural network (RNN) to generate initial GP populations. The best-performing individuals are then used to retrain the RNN. Similarly, Petersen et al. (2021) presented a method that utilizes RNNs to generate mathematical expressions in a reinforcement learning framework. In addition, Udrescu & Tegmark (2020) proposed a recursive algorithm called AI Feynmann that uses NNs to identify properties such as symmetry, separability, and compositionally, which are exploited to define sub-problems that are easier to solve.

Several methods have proposed training NNs and pruning irrelevant parts of them until a simple equation can be distilled from the network weights (Martius & Lampert, 2016; Sahoo et al., 2018; Tsoi et al., 2025; Werner et al., 2021). One of their main limitations is that they do not leverage past experiences, and each problem is learned from scratch. Transformer-based methods have been proposed recently as an alternative. Biggio et al. (2021) presented a Set Transformer-based model (Lee et al., 2019) and pre-trained it on a synthetic dataset of multivariate equations to predict symbolic skeletons from input-output data pairs. During inference, the observed data is used to predict the equation's skeleton using the pre-trained transformer. Next, the constants within the skeleton are optimized using the BFGS algorithm (Fletcher, 1987). To extend this approach, Landajuela et al. (2022) presented a unified deep SR (uDSR) framework that combines multiple SR strategies. It first attempts to decompose the target problem into subproblems using AI-Feynman (Udrescu & Tegmark, 2020). For each subproblem, an RNN-based controller (Petersen et al., 2021) generates candidate mathematical expressions via a neural SR model based on that proposed by Biggio et al. (2021). At each optimization step, candidates are evolved and refined using GP. The refined expressions are fed back to update the RNN. The final solution assembles the subproblem expressions using a linear model.

Alternatively, Kamienny et al. (2022) proposed an end-to-end (E2E) transformer model that directly predicts complete mathematical expressions, bypassing skeleton prediction. Learned constants can be fine-tuned using a non-convex optimizer such as BFGS. Shojaee et al. (2023) introduced Transformer-based Planning for Symbolic Regression (TPSR), a decoding strategy that uses Monte Carlo Tree Search (MCTS) to guide equation generation via lookahead planning. TPSR is model-agnostic and works with any pre-trained transformer-based SR model, enabling optimization for non-differentiable objectives like reward and complexity. Their experiments show that applying TPSR on the E2E model boosts performance on benchmark datasets.

In addition, Bertschinger et al. (2024) proposed an SR neuro-evolution approach that trains a population of transformer models using two objective functions: prediction error and symbolic loss. Finally, in [Ref. 1][2], we noted that SR methods analyzing all variables simultaneously often fail to identify the functional relationships between each variable and the system's response. To address this, we introduced the Multi-Set

Transformer, trained on synthetic data to identify symbolic skeletons shared across multiple input-response sets. Then, we proposed a *post-hoc* explainability method that extracts univariate skeletons, providing interpretable approximations of the function learned by an opaque regression model.

## 3 Deep Evolutionary Symbolic Regression

We consider a system with response $y \in \mathbb{R}$ and $t$ explanatory variables $\mathbf{x} = \{x_1, \ldots, x_t\}$. We assume that its underlying function $f(\mathbf{x}) = f(x_1, \ldots, x_t)$ can be constructed using a finite number of unary and binary operations. The response is expressed as $y = f(\mathbf{x}) + \varepsilon_a$, where $\varepsilon_a$ represents the error term due to aleatoric uncertainty. Below, we define the SR problem formally and describe our SeTGAP methodology.

### 3.1 Problem Definition

Let us define our SR problem formally:

**Definition 1.** *Given a dataset $(\mathbf{X}, \mathbf{y})$, with inputs $\mathbf{X} = \{\mathbf{x}_1, \mathbf{x}_2, \ldots, \mathbf{x}_{N_R}\} \subseteq \mathbb{R}^t$ and target responses $\mathbf{y} = \{y_1, y_2, \ldots, y_{N_R}\} \subseteq \mathbb{R}$, the SR problem seeks to discover a function $\tilde{f} : \mathbb{R}^t \to \mathbb{R}$ that approximates the unknown underlying function $f(\mathbf{x})$. The function $\tilde{f}$ is represented as a composition of a finite number of unary and binary operators, such that $\tilde{f}(\mathbf{x}) \approx f(\mathbf{x})$ while capturing its functional structure and behavior.*

Thus, $\tilde{f}(\mathbf{x})$ can be expressed as $\tilde{f}(\mathbf{x}) = f(\mathbf{x}) + \varepsilon_a + \varepsilon_e$, where $\varepsilon_e$ is the error due to epistemic uncertainty, which is attributable to a lack of knowledge about $f$ and can be reduced by acquiring additional information and improving the predictive model. In this context, solving the SR problem implies minimizing $\varepsilon_e$ by selecting a suitable representation for $\tilde{f}(\mathbf{x})$ and identifying an optimal set of parameters $\boldsymbol{\theta}_{\tilde{f}}$ for such representation.

### 3.2 Univariate Symbolic Skeleton Prediction

We deviate from SR approaches that prioritize the minimization of the prediction error of the learned functions. We argue that a correctly identified functional form $\tilde{f}$ inherently leads to a low estimation error and emphasizes interpretability and faithfulness to the underlying system's governing principles. Existing works on Symbolic Skeleton Prediction (SSP) (Bertschinger et al., 2024; Biggio et al., 2021; Petersen et al., 2021; Valipour et al., 2022) attempt to address this issue by generating multivariate symbolic skeletons that describe the behavior of $f$, with numerical coefficients subsequently optimized to minimize prediction error.

#### 3.2.1 Background: Multi-Set Skeleton Prediction

Given a mathematical expression, its symbolic skeleton is an expression that replaces the numerical constants with placeholders. For example, if $f(\mathbf{x}) = 5 \log x_1 (\sin(x_2^2) + 1) - 4$, its skeleton is expressed as $\mathbf{e}(\mathbf{x}) = \kappa(f(\mathbf{x})) = c_1 \log x_1 (\sin(x_2^2) + c_2) + c_3$, where $\kappa(\cdot)$ represents the skeleton function and $c_i$ are placeholders. In prior work [Ref. 1][2], we showed that SSP methods struggle to identify the correct functional form of all variables in a system. Thus, we introduced a univariate skeleton prediction method that produces univariate skeletons explaining the relationships between each variable and the system's response. Following the previous example, its univariate skeletons are: $\mathbf{e}(x_1) = \kappa(f(\mathbf{x}); x_1) = c_4(\log x_1) + c_5$, and $\mathbf{e}(x_2) = \kappa(f(\mathbf{x}); x_2) = c_6 \sin(x_2^2) + c_7$. Here, $\kappa(\cdot; x_v)$ considers the remaining variables $\mathbf{x} \setminus x_v$ irrelevant when describing the functional form between $x_v$ ($v \in [1, \ldots, t]$) and the system's response. In this case, the placeholders $c_i$ may represent numeric constants or functions of other variables.

This method uses a trained regression model $\hat{f}$ (e.g., a neural network) that approximates the underlying function $f$ by capturing the association between $\mathbf{X}$ and $\mathbf{y}$. Then, each variable of the system, $x_v$, is analyzed separately. To do this, $N_s$ artificial sets of input–response pairs $\{(\tilde{\mathbf{X}}^{(1)}, \tilde{\mathbf{y}}^{(1)}), \ldots, (\tilde{\mathbf{X}}^{(N_s)}, \tilde{\mathbf{y}}^{(N_s)})\}$ are generated. To isolate the influence of variable $x_v$, the set $\tilde{\mathbf{X}}^{(s)}$ ($s \in [1, \ldots, N_s]$) is constructed such that the variable $x_v$ (i.e., the $v$-th column $\tilde{\mathbf{X}}_v^{(s)}$) is allowed to vary while the other variables are fixed to random values. Thus, $\tilde{\mathbf{y}}^{(s)}$ denotes the estimated response of inputs $\tilde{\mathbf{X}}^{(s)}$ using the trained model $\hat{f}$ as $\tilde{\mathbf{y}}^{(s)} = \hat{f}(\tilde{\mathbf{X}}^{(s)})$.

Fixing the columns corresponding to the variables $\mathbf{x} \setminus x_v$ ensures that $\tilde{\mathbf{y}}^{(s)}$ depends only on the column $\tilde{\mathbf{X}}_v^{(s)}$. However, this process may project the function into a space where its functional form becomes less

recognizable. To address this, the influence of $x_v$ on the system's response is analyzed using $N_s$ sets of input–response pairs, each reflecting a different effect of the variables $\mathbf{x} \setminus \{x_v\}$. As such, each set $(\tilde{\mathbf{X}}^{(s)}, \tilde{\mathbf{y}}^{(s)})$ is generated independently by fixing $\mathbf{x} \setminus \{x_v\}$ at different values. The relationship between each $\tilde{\mathbf{X}}_v^{(s)}$ and $\tilde{\mathbf{y}}^{(s)}$ can be described by a univariate function $f_v^{(s)}$. Note that functions $f_v^{(1)}, \ldots, f_v^{(N_S)}$ have been derived from the same function $f(\mathbf{x})$ and should share the same symbolic skeleton $\mathbf{e}(x_v)$, which is unknown. The task of predicting a skeleton $\hat{\mathbf{e}}(x_v)$ that describes the shared function form of input $\tilde{\mathbf{D}}_v = \{\tilde{\mathbf{D}}_v^{(1)}, \ldots, \tilde{\mathbf{D}}_v^{(N_s)}\}$ (i.e., $\hat{\mathbf{e}}(x_v) \approx \mathbf{e}(x_v)$), s.t. $\tilde{\mathbf{D}}_v^{(s)} = (\tilde{\mathbf{X}}_v^{(s)}, \tilde{\mathbf{y}}^{(s)})$, is known as multi-set symbolic skeleton prediction (MSSP).

The MSSP problem has been addressed by [Ref. 1][2] by designing a Multi-Set Transformer. The model's function and parameters are denoted by $g(\cdot)$ and $\Theta$, respectively. It was trained on a large dataset of synthetically generated MSSP problems to produce accurate estimated skeletons. Given an input collection $\tilde{\mathbf{D}}_v$, the estimated skeleton obtained for variable $x_v$ is computed as $\tilde{\mathbf{e}}(x_v) = g(\tilde{\mathbf{D}}_v, \Theta)$. Key details about the model architecture, training procedure, and vocabulary are provided in Appendix A.

### 3.2.2 Extended Univariate Symbolic Skeleton Prediction

Unlike previous work, we generate up to $n_{\text{cand}}$ distinct candidate skeletons, as described in Algorithm 1. We generate a $\tilde{\mathbf{D}}_v$ collection and feed it into $g$ to obtain $n_B$ skeletons using a diverse beam search (DBS) strategy (Vijayakumar et al., 2018) to promote variability among the $n_B$ generated skeletons. This process is repeated $n_{\text{cand}}$ times, yielding a total of $n_{\text{cand}} n_B$ skeletons. Each repetition yields a new $\tilde{\mathbf{D}}_v$ collection using different combinations of fixed values of $\mathbf{x} \setminus \{x_v\}$, increasing input diversity and potentially leading to different skeletons. Then, we discard identical and some mathematically equivalent skeletons, determined using basic trigonometric rules (see Appendix B). For example, we consider the skeleton $c_1 \sin(c_2 x_v + c_3)$ is equivalent to $c_4 \cos(c_5 x_v + c_6)$ if $c_1 = c_4$, $c_2 = c_5$, and $c_3 = c_6 - \pi/2$. The motivation to identify such equivalences is to avoid redundancy and reduce computational cost. However, this simplification does not affect the final results. A more comprehensive treatment of equivalence identification is left for future work.

If the generated skeleton list, $\texttt{genSks}_v$, exceeds $n_{\text{cand}}$ elements, we evaluate their performance and select the top $n_{\text{cand}}$ candidates. To do this, an additional collection $\tilde{\mathbf{D}}_v$ is generated. Since a skeleton expression for variable $\mathbf{x}_v$ is expected to describe the functional form of all sets in $\tilde{\mathbf{D}}_v$, we choose a random set $\tilde{\mathbf{D}}_v^{(\text{test})} = (\tilde{\mathbf{X}}_v^{(\text{test})}, \tilde{\mathbf{y}}^{(\text{test})}) \subset \tilde{\mathbf{D}}_v$ and use it to fit the coefficients of each skeleton $\hat{\mathbf{e}}_k(x_v)$, where $k \in \{1, \ldots, |\texttt{genSks}_v|\}$. The coefficient fitting problem is described as follows. Let $f_{est}(x_v) = \texttt{setConstants}(\hat{\mathbf{e}}_k(x_v), \mathbf{c})$ be the function obtained when replacing the $n_c$ coefficients in $\hat{\mathbf{e}}_k(x_v)$ with the numerical values in a given set $\mathbf{c} \in \mathbb{R}^{n_c}$. Then, the objective is to find an optimal set $\mathbf{c}^*$ that minimizes the following error:

$$\mathbf{c}^* = \underset{\mathbf{c}}{\arg\min} \frac{1}{|\tilde{\mathbf{X}}_v^{(\text{test})}|} \sum_{(\mathbf{x}_j, y_j) \in (\tilde{\mathbf{X}}_v^{(\text{test})}, \tilde{\mathbf{y}}^{(\text{test})})} \left( \texttt{setConstants}(\hat{\mathbf{e}}_i(\mathbf{x}_j), \mathbf{c}) - y_j \right)^2,$$

which minimizes the mean squared error (MSE) between $f_{est}(\tilde{\mathbf{X}}_v^{(\text{test})})$ and $\tilde{\mathbf{y}}^{(s)}$. Note that in generating $\tilde{\mathbf{D}}_v$, we fixed the values of $\mathbf{x} \setminus x_v$, so we can assume that all coefficients in $\mathbf{c}$ are numerical values. The learned coefficients are then discarded, as they serve only to evaluate the fit of a univariate skeleton to the data.

This problem is solved using a genetic algorithm (GA) (Holland, 1992). The individuals of our GA are arrays of $n_c$ elements that represent potential $\mathbf{c}$ sets. Then the optimization process is carried out by function $\texttt{fitCoefficients}(\hat{\mathbf{e}}_k(x_v), \tilde{\mathbf{D}}_v^{(\text{test})})$. This optimization process is repeated for all system variables to derive their univariate skeleton expressions with respect to the system's response.

### 3.3 Merging Univariate Symbolic Skeletons

The next step is to merge the univariate skeleton candidates to produce multivariate expressions. This process is carried out in a cascade fashion until a final expression incorporating all variables is formed. Below, we outline the specific contribution of each subsection:

- **Sec. 3.3.1 (Merging Skeleton Expressions):** Given a generic symbolic skeleton $\mathbf{e}_1(\mathbf{x}_S)$ involving a subset of variables $S$, it details a recursive algorithm ($\texttt{merge}$) to merge it randomly with a distinct univariate skeleton $\mathbf{e}_2(x_q)$ while strictly preserving their independent univariate structures.

---

**Algorithm 1** Univariate Skeleton Generation

1: **function** GENERATEUNIVSKS($v, n, N_s, \hat{f}, g, n_{\text{cand}}, n_B$)
2:     $\text{genSks}_v \leftarrow []$
3:     **for** each $i \in (1, n_{\text{cand}})$ **do**
4:         $\tilde{\mathbf{D}}_v \leftarrow \texttt{generateCollection}(v, n, N_s, \hat{f})$
5:         $\text{genSks}_v.\texttt{append}(g(\tilde{\mathbf{D}}_v, \Theta; n_B))$
6:     $\text{genSks}_v \leftarrow \text{removeDuplicates}(\text{genSks}_v)$          $\triangleright$ $\text{genSks}_v = \{\hat{\mathbf{e}}_1(x_v), \ldots, \hat{\mathbf{e}}_{|\text{genSks}_v|}(x_v)\}$
7:     $\tilde{\mathbf{D}}_v^{(\text{test})} \leftarrow \texttt{generateCollection}(v, n, N_s, \hat{f})$
8:     $\text{MSEVals}_v \leftarrow \text{zeros}(|\text{genSks}_v|)$
9:     **for** each $k \in (1, n_{\text{cand}})$ **do**
10:         $\text{MSEVals}_v[k] \leftarrow \texttt{fitCoefficients}(\hat{\mathbf{e}}_k(x_v), \tilde{\mathbf{D}}_v^{(\text{test})})$
11:     $\text{genSks}_v \leftarrow \texttt{sortSkeletons}(\text{genSks}_v, \text{MSEVals}_v)$
12:     **if** $|\text{genSks}_v| > n_{\text{cand}}$ **then**
13:         $\text{genSks}_v, \text{MSEVals}_v \leftarrow \text{genSks}_v[1 : n_{\text{cand}}], \text{MSEVals}_v[1 : n_{\text{cand}}]$
14:     **return** $\text{genSks}_v, \text{MSEVals}_v$

---

Table 1: Summary of notation used in the multivariate merging cascade

| Symbol / Variable | Description |
|---|---|
| $S$ | Index set of variables currently integrated into the merged model; e.g., in a 5-variable system ($t = 5$), $S = \{3, 1, 4\}$ |
| $q$ | Index of the variable to be merged ($q \notin S$); e.g., $q = 0$ |
| $S'$ | Index set of all variables present during the merging process; i.e., $S' = \{S \cup q\}$ |
| $\mathbf{x}_S, x_q$ | The aggregated variables vector (e.g., $\mathbf{x}_S = \{x_3, x_1, x_4\}$) and the variable to be merged (e.g., $x_q = x_0$), respectively |
| $\mathbf{e}_1(\mathbf{x}_S), \mathbf{e}_2(x_q)$ | Candidate symbolic skeletons chosen for the structural combination step |
| $\nu_{i,j}, T_{i,j}$ | Generic unary operators and sub-trees defining the recursive structural decomposition form, in which any skeleton can be expressed; e.g., $\mathbf{e}_1(\mathbf{x}_S) = c_0 + \sum_i c_i \prod_j \nu_{i,j}(T_{i,j}(\mathbf{x}_S))$ |
| $\bowtie$ | Symbolic composition operator defining skeleton expression merging |
| $\texttt{exShort}, \texttt{exLong}$ | Dynamic lables assigned to $\mathbf{e}_1(\mathbf{x}_S)$ and $\mathbf{e}_2(x_q)$ during merging, based on three depth |
| $\texttt{candSks}$ | Stochastically generated population containing unique candidate merging skeletons |
| $\tilde{\mathbf{D}}_{S'}^{(\text{test})}$ | Localized evaluation dataset where active variables vary and inactive variables are fixed |

- **Sec. 3.3.2 (Selecting Combined Skeleton Expression):** The previous step yields one of multiple possible skeleton combinations of $\mathbf{e}_1(\mathbf{x}_S)$ and $\mathbf{e}_2(x_q)$. Here, we use $\texttt{merge}$ iteratively to construct a diverse pool of candidate merging skeletons, $\texttt{candSks}$, and introduce a constrained evolutionary optimization to evaluate their fitness, decoupling structural exploration from parameter optimization.
- **Sec. 3.3.3 (Cascade Merging):** Establishes an incremental cascade framework where variables are incorporated sequentially, one at a time. At each stage, it applies the evolutionary selection routine from Sec. 3.3.2 to preserve the most viable combined skeletons, effectively bypassing combinatorial explosion.
- **Sec. 3.3.4 (Underlying Function Estimation):** Performs a final global optimization pass on the real multivariate dataset to determine the numerical parameter values for the top-performing merged skeletons.

To facilitate readability, Table 1 summarizes the primary notations, index conventions, and symbolic operators utilized throughout this section. Fig. 1 illustrates a successful instance of the merging process, while Fig. 9 provides a more detailed trace of an unsuccessful execution path.

### 3.3.1 Merging Skeleton Expressions

Given two skeleton expressions, multiple mathematically valid ways to combine them may exist. Here, we explore how to generate such combinations. We start with the following proposition:

**Proposition 1.** *Let $f(\mathbf{x})$ be a scalar-valued function defined by a finite composition of real-valued unary and binary operators applied to scalar sub-expressions. Then, $f(\mathbf{x})$ can always be expressed as: $f(\mathbf{x}) = c_0 + \sum_i c_i \prod_j \nu_{i,j}(T_{i,j}(\mathbf{x}))$, where $c_0, c_i \in \mathbb{R}$, $\nu_{i,j}$ is a unary operator (including the identity function $I(f(\mathbf{x})) = f(\mathbf{x})$), and $T_{i,j}(\mathbf{x})$ is a sub-expression. Moreover, each $T_{i,j}(\mathbf{x})$ can be recursively decomposed in the same structure as $f$, continuing until the decomposition reduces to variables or constants.*

Proposition 1 is proved in Appendix C. Then, let $\mathbf{e}_1(\mathbf{x}_S)$ and $\mathbf{e}_2(x_q)$ be two candidate skeletons to be merged. Here, $S$ is an index set, $S \subset \{1, \ldots, t\}$, specifying the variables that $\mathbf{e}_1$ depends on; i.e., $\mathbf{x}_S = \{x_r | r \in S\}$. In contrast, $x_q$ is a variable distinct from those in $\mathbf{x}_S$ ($q \notin S$). Following from Proposition 1, the skeletons can be expressed as $\mathbf{e}_1(\mathbf{x}_S) = c_0 + \sum_i c_i \prod_j \nu_{i,j}(T_{i,j}(\mathbf{x}_S))$ and $\mathbf{e}_2(x_q) = c_0' + \sum_i c_{i'}' \prod_{j'} \nu_{i',j'}'(T_{i',j'}'(x_q))$. Below, we explain how the subtrees of both expressions can be merged recursively.

The key idea is that a constant placeholder in $\mathbf{e}_1(\mathbf{x}_S)$ may be replaced by a subtree of $\mathbf{e}_2(x_q)$, and vice versa. Given two skeletons, $\mathbf{e}_1(\mathbf{x}_S) = c_1 T_1(\mathbf{x}_S)$ and $\mathbf{e}_2(x_q) = c_2 T_2(x_q)$, a straightforward way to merge them is by recognizing that part of $c_1$ may be a function of $x_q$, while part of $c_2$ may be a function of $\mathbf{x}_S$. Thus, their combination, denoted by the operation $\bowtie$, is given by $\mathbf{e}_3(\mathbf{x}_S \cup x_q) = \mathbf{e}_1(\mathbf{x}_S) \bowtie \mathbf{e}_2(x_q) = c_3(c_4 + T_1(\mathbf{x}_S))(c_5 + T_2(x_q))$. Expanding this expression conforms to the expected functional form stated in Proposition 1. Note that the skeletons with respect to the corresponding initial variable sets remain unchanged; that is, $\kappa(\mathbf{e}_1(\mathbf{x}_S); \mathbf{x}_S) = \kappa(\mathbf{e}_3(\mathbf{x}_S \cup x_q); \mathbf{x}_S) = c_1 T_1(\mathbf{x}_S)$ and $\kappa(\mathbf{e}_2(x_q); x_q) = \kappa(\mathbf{e}_3(\mathbf{x}_S \cup x_q); x_q) = c_q T_2(x_q)$.

Applying the same principle to a more general case in which $\mathbf{e}_1(\mathbf{x}_S) = c_1 + c_2 \prod_j T_{1,j}(\mathbf{x}_S)$ and $\mathbf{e}_2(x_q) = c_3 + c_4 \prod_{j'} T_{2,j'}(x_q)$, we obtain $\mathbf{e}_3(\mathbf{x}_S \cup x_q) = \mathbf{e}_1(\mathbf{x}_S) \bowtie \mathbf{e}_2(x_q) = c_5 + c_6 \prod_j (c_{7,j} + T_{1,j}(\mathbf{x}_S)) \prod_{j'} (c_{8,j'} + T_{2,j'}(\mathbf{x}_S))$. This case is referred to as the "wrapped–product merge" for future reference. Nevertheless, more combinations are possible if the candidate skeletons share functions with compatible mathematical structures. For example, if $\mathbf{e}_1(x_1, x_2) = c_1 \sin(c_2 x_1 x_2 + c_3)$ and $\mathbf{e}_2(x_3) = c_4 \sin(c_5 x_3 + c_6)$, we could obtain $\mathbf{e}_3(x_1, x_2, x_3) = \mathbf{e}_1(x_1, x_2) \bowtie \mathbf{e}_2(x_3) = c_7(c_8 + \sin(c_9 x_1 x_2 + c_{11}))(c_{12} + \sin(c_{13} x_3 + c_{14}))$, as shown before, but also $\mathbf{e}_3(x_1, x_2, x_3) = c_{15} \sin(c_{16} x_1 x_2 + c_{17} x_3)$ and $\mathbf{e}_3(x_1, x_2, x_3) = c_{18} \sin(c_{19} x_1 x_2 x_3 + c_{20})$, which yield to the same skeletons with respect to the initial variable sets.

Algorithm 2 implements our recursive merging procedure, $\mathtt{merge}(\mathtt{ex}_1, \mathtt{ex}_2)$, which takes as inputs two skeleton expressions $\mathtt{ex}_1$ and $\mathtt{ex}_2$, and constructs multivariate skeleton candidates by respecting the decomposition guaranteed by Proposition 1. The high-level invariant maintained throughout the recursion is that every intermediate expression is represented in a form consistent with the proposition; i.e., as a constant plus a sum of terms, each being a product of sub-expressions affected by unary operators. The procedure begins by extracting the immediate subtrees (the summands or multiplicative factors) of each input expression using $\mathtt{getSubtreesLists}(\mathtt{ex}_1, \mathtt{ex}_2)$, which returns the subtree lists ordered so that $\mathtt{exShort}$ contains fewer or equal elements than $\mathtt{exLong}$. The lists are shuffled to introduce diversity in the merging order. If $\mathtt{exShort}$ lacks an explicit constant term, the routine appends a constant placeholder and ensures it is the last element; this placeholder enables absorption of unmatched terms and enforces the proposition's canonical form.

When both inputs are sums (Line 4), the algorithm iterates the subtrees (i.e., summands) in $\mathtt{exShort}$, attempting to merge each with compatible subtrees from $\mathtt{exLong}$; i.e., subtrees that share the same unary or binary operator. The compatibility check is performed by $\mathtt{findComp}$, and a subset of matching subtrees is randomly selected for merging, as depicted in Fig. 2. A random subset $\mathtt{selectedArgs}$ of these compatible subtrees is chosen (Line 10). If $\mathtt{selectedArgs}$ is empty, the loop continues; if it contains a single element, that element is merged recursively with the current summand, $\mathtt{exShort}[i]$, via $\mathtt{merge}$. However, if multiple subtrees are selected, the only way to maintain the form required by Proposition 1 is to treat their sum as a single, indivisible block, effectively a constant term with respect to $x_q$, which is then multiplied by $\mathtt{exShort}[i]$ (Line 16). The last element of $\mathtt{exShort}$ (the constant placeholder) then absorbs any remaining unmatched summands from $\mathtt{exLong}$, again preserving the canonical structure (Line 7).

If neither expression is a sum or a product but they share the same outer operator, we keep that operator and merge their inner arguments recursively. This is shown in Line 20, where $\mathtt{ex1.func}$ is the SymPy attribute that returns the outermost operator of $\mathtt{ex1}$, and $\mathtt{ex1.args}$ returns its tuple of inner arguments. The call $\mathtt{ex1.func}(\ldots)$ constructs a new expression by applying this same operator to the result of merging the arguments $\mathtt{ex1.args}$ and $\mathtt{ex2.args}$. The $\mathtt{merge}$ call is recursive because each argument can itself be a composite expression, and the same merging logic can be applied at deeper levels of the expression tree.

If both $\mathtt{ex1}$ and $\mathtt{ex2}$ are multiplications (Line 22), we distinguish two subcases. If $\mathtt{isAllSymbols}(\mathtt{exShort})$ is true, each factor of $\mathtt{exShort}$ represents a symbol, and the merged expression is built as wrapped–product merge, $c_1 \prod_i (c_{2,i} + \mathtt{exShort}[i]) \prod_j (c_{3,j} + \mathtt{exLong}[j])$ (Line 24), which fits into the topology of Proposition 1. If non-symbol factors are present, the algorithm attempts one-to-one merges between factors. It iterates through the factors of $\mathtt{exShort}$, uses $\mathtt{findComp}$ to locate compatible candidates in $\mathtt{exLong}$ (Line 31), selects

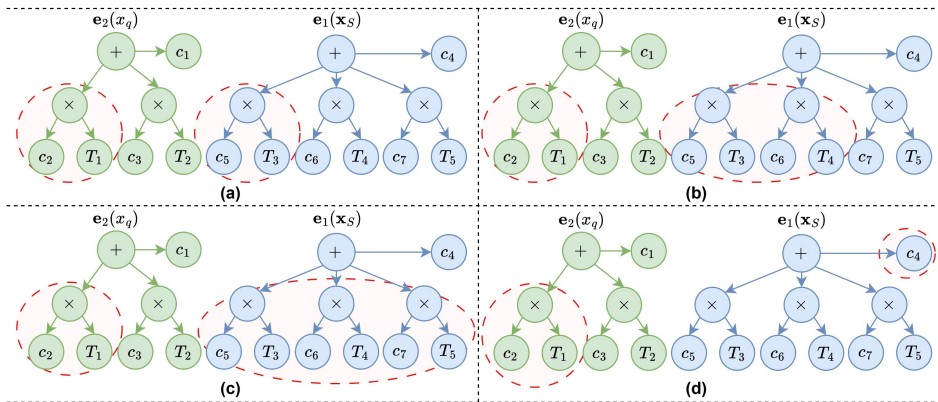

Figure 2: Example of a selected subtree of $\mathbf{e}_2(x_q)$ within a sum merging with one or more subtrees from $\mathbf{e}_1(\mathbf{x}_S)$, illustrating four out of the nine possible cases.

---

**Algorithm 2** Recursive Skeleton Merging

```
1:  function MERGE(ex₁, ex₂)
2:      exShort, exLong ← getSubtreesLists(ex₁, ex₂)                          ▷ Return lists of subtrees in sum
3:      exShort.shuffle(), exLong.shuffle()
4:      if isSum(ex₁) and isSum(ex₂) then
5:          for each i ∈ (1, |exShort|) do
6:              if i = |exShort| and |exLong| > 0 then
7:                  exShort[i] ← exShort[i] + ∑_j exLong[j]
8:              else
9:                  args ← findComp(exLong, exShort[i])                       ▷ Find exLong args compatible w/exShort[i]
10:                 selectedArgs ← sample(args, randInt(0, |args|))
11:                 if |selectedArgs| = 0 then continue
12:                 exLong.remove(selectedArgs)
13:                 if |selectedArgs| = 1 then
14:                     exShort[i] ← merge(exShort[i], selectedArgs[0])
15:                 else
16:                     exShort[i] ← exShort[i] ∑_j selectedArgs[j]
17:          mergedEx ← ∑_i exShort[i]
18:      else
19:          if ex₁.func = ex₂.func then
20:              if !isMult(ex₁) then                                          ▷ If compatible, merge inner arguments
21:                  mergedEx ← ex₁.func(merge(ex₁.args, ex₂.args))
22:              else
23:                  if isAllSymbols(exShort) then
24:                      mergedEx ← c₁ ∏_i(c_{2,i} + exShort[i]) ∏_j(c_{3,j} + exLong[j])
25:                  else
26:                      mergedEx ← null
27:                      for each i ∈ (1, |exShort|) do
28:                          if i = |exShort| and |exLong| > 0 then
29:                              mergedEx ← c₁ ∏_{i'=0}^{i}(c_{2,i'} + exShort[i']) ∏_j(c_{3,j} + exLong[j])
30:                          else
31:                              args ← findComp(exLong, exShort[i])
32:                              if |args| = 0 then continue
33:                              selectedArg ← choice(args)
34:                              if random(0, 1) < 0.5 then continue
35:                              exLong.remove(selectedArg)
36:                              exShort[i] ← merge(exShort[i], selectedArg)
37:                      if mergedEx is null then
38:                          mergedEx ← ∏_i exShort[i]
39:          else
40:              mergedEx ← ex₁ ex₂
41:      return mergedEx
```

---

a single candidate uniformly at random (Line 33), and accepts the recursive merge with probability 0.5 to introduce variability (Line 34–36). When analyzing the last `exShort` factor, Line 29 implements the absorption of any remaining unmatched `exLong` factors using a constant-wrapped product. Otherwise, if `exLong` has no remaining factors, the merged expression is the product of the updated `exShort` factors.

If the outermost operators differ (Line 40), the merged expression is the product $ex_1\, ex_2$. This preserves the canonical form because the product of two canonical sub-expressions expands into a sum of products of

---

**Algorithm 3** Skeleton Combination with Genetic Programming

---

1: **function** SELECTCOMBINATION(candSks, rep, maxG, $\tilde{\mathbf{D}}_{S'}^{(\text{test})}$)
2:     Pop $\leftarrow$ []
3:     **for** each $s \in (1, |\text{candSks}|)$ **do**
4:         exps $\leftarrow$ []
5:         **for** $i = 1$ to rep **do** exps.append(assignValues(candSks[$s$]))
6:         Pop.append(exps)
7:     **for** each gen $\in (1, \text{maxG})$ **do**
8:         fitnesses, bestFitnesperSk $\leftarrow$ evalMSE(Pop, $\tilde{\mathbf{D}}_{S'}^{(\text{test})}$)
9:         **for** each $p \in (1, |\text{Pop}|)$ **do**
10:             Pop[$p$] $\leftarrow$ evolveSks(Pop[$p$], fitnesses)    ▷ Expressions derived from the same skeleton combination are evolved together
11:     **return** candSks[argmin(bestFitnesperSk)], max(bestFitnesperSk)

---

unary-applied subexpressions (see Appendix C), as required by Proposition 1. Finally, the function returns the merged expression, which integrates elements from both inputs while preserving their initial structures. An illustrative example of the merging process, including its intermediate steps, is provided in Appendix D.

### 3.3.2 Selecting Combined Skeleton Expressions

Algorithm 2 generates a single combination of skeletons. Here, we extend this process to construct a population of candidate skeletons and select the top-performing one. Thus, we employ an evolutionary strategy to assess the performance of each skeleton combination. The population of skeletons, `candSks`, is constructed by repeatedly applying Algorithm 2. Since the merging process is stochastic, each application of the algorithm may yield a different skeleton combination. The process continues until a population size $P_{\max}$ is reached or a patience criterion is met. As such, if no new valid skeleton is found after a certain number of attempts, the generation process halts, assuming that all viable combinations have been explored.

We aim to identify the most promising combination in `candSks` by evaluating how well each skeleton can fit the test data. Algorithm 3 provides a high-level overview of this process. We generate test data $\tilde{\mathbf{D}}^{(\text{test})} = (\tilde{\mathbf{X}}^{(\text{test})}, \tilde{\mathbf{y}}^{(\text{test})})$. Unlike in Sec. 3.2, here the columns in $S' = \{S \cup q\}$ (i.e., the indices of variables present in the combined skeletons) are allowed to vary, while all other variables are fixed to random values. The set $\tilde{\mathbf{D}}_{S'}^{(\text{test})} = (\tilde{\mathbf{X}}_{S'}^{(\text{test})}, \tilde{\mathbf{y}}^{(\text{test})})$ is then used for evaluation. Each skeleton in `candSks` is replicated `rep` times, with randomly assigned coefficient values, to form an initial pool of candidate expressions `Pop`.

The population then undergoes an evolution process. In each iteration, the fitness of every expression in `Pop` is evaluated using `evalMSE`, which computes the MSE between $\tilde{\mathbf{y}}^{(\text{test})}$ and the output produced by the expression when evaluated on $\tilde{\mathbf{X}}^{(\text{test})}$. After the evaluation, `evolveSks` updates each subpopulation in `Pop` independently. As such, expressions derived from the same skeleton combination are evolved together, and no crossover is allowed between expressions coming from different skeletons. This grouping enforces the structural constraint because crossover is implemented as an exchange of coefficient vectors. In other words, when two individuals are selected for crossover, their skeleton structure is kept fixed and only their numeric coefficient assignments are swapped. Likewise, mutation acts only on coefficients, so the expression trees remain unchanged throughout evolution. Finally, after `maxG` generations, the algorithm returns the skeleton whose evolved instances achieved the highest fitness. This approach bears similarities to GP, with the distinction that the search for viable skeletons is decoupled from the evolutionary optimization process. Rather than evolving the expressions dynamically, as in standard GP, we enumerate all admissible skeleton combinations via Algorithm 2 and then apply our constrained evolution, as described in Algorithm 3, where the structure of all mathematical expressions is fixed to the precomputed combined skeletons.

### 3.3.3 Cascade Merging

The construction of multivariate skeletons follows an incremental merging process in which univariate skeleton candidates are combined progressively. For each variable $x_v$ Algorithm 1 generates up to $n_{\text{cand}}$ candidate skeletons $\text{genSks}_v$ along with their corresponding MSE values $\text{MSEVals}_v$. To determine the merging order, the variables are ranked according to the minimum MSE value obtained across their generated skeletons. This ordering strategy is a heuristic founded on the intuition that prioritizing skeletons with stronger relationships to the data reduces the risk of propagating structural uncertainty. While future work will explore

a more formal ordering approach, a permutation analysis is provided in Appendix K to evaluate the impact of different variable sequences on the final recovered expressions.

Let $S$ denote the indices of the variables whose skeletons have already been merged. Initially, $S$ contains only the index of the variable that corresponds to the skeleton with the lowest MSE value. At each iteration, a new variable $x_q$ is selected according to the ranking, and its univariate skeletons are merged with those corresponding to $S$. Specifically, Algorithms 2 and 3 are applied to combine each skeleton generated for $\mathbf{x}_S$ with each skeleton generated for $x_q$, producing at most $n_{\text{cand}}^2$ skeleton combinations. Since Algorithm 3 returns both the selected merged skeleton and its fitness, we apply a greedy selection strategy, retaining only the $n_{\text{cand}}$ highest-performing skeletons at each step. This process repeats until skeletons incorporating all $t$ variables have been generated, which serve as interpretations that describe the functional form of model $\hat{f}$.

We argue that our cascade approach ensures a structured integration of variables into the final multivariate skeleton. Directly constructing a multivariate skeleton from all univariate candidates would prioritize minimizing overall prediction error, potentially obscuring individual contributions and leading to overfitting. Instead, incorporating one variable at a time allows for a more interpretable learning process, where each newly added variable's effect is evaluated in the context of the previously analyzed ones.

### 3.3.4 Underlying Function Estimation

We utilize the $N_e$ multivariate skeletons $\hat{\mathbf{e}}_1(\mathbf{x}), \ldots, \hat{\mathbf{e}}_{N_e}(\mathbf{x})$ produced by our cascade merging procedure, where $N_e \leq n_{\text{cand}}$. The goal is to construct functions $\tilde{f}_i(\mathbf{x})$ that approximate the underlying function $f(\mathbf{x})$ using the corresponding skeleton $\hat{\mathbf{e}}_i(\mathbf{x})$, $\forall i \in (1, \ldots, N_e)$. This defines a coefficient fitting problem similar to the one presented in Sec. 3.2. Unlike earlier sections, we minimize the MSE of the response calculated by evaluating $\tilde{f}_i(\mathbf{x})$ on the original dataset $(\mathbf{X}, \mathbf{y})$. This is feasible because $\hat{\mathbf{e}}(\mathbf{x})$ contains all system variables and there is no need to generate a synthetic set of estimated points using $\hat{f}$. Hence, the new coefficient fitting problem consists of finding an optimal set of coefficient values that minimizes the prediction MSE $\mathbf{c}^* = \underset{\mathbf{c}}{\operatorname{argmin}} \frac{1}{|\mathbf{X}|} \sum_{(\mathbf{x}_j, y_j) \in (\mathbf{X}, \mathbf{y})} \left( \texttt{setConstants}(\hat{\mathbf{e}}_i(\mathbf{x}_j), \mathbf{c}) - y_j \right)^2$, such that $\tilde{f}_i(\mathbf{x}) = \texttt{setConstants}(\hat{\mathbf{e}}_i(\mathbf{x}), \mathbf{c}^*)$. This optimization is carried out using the GA-based function $\texttt{fitCoefficients}(\hat{\mathbf{e}}_i(\mathbf{x}), (\mathbf{X}, \mathbf{y}))$, introduced in Sec. 3.2.

### 3.4 Comparison to Decomposition-based Methods

In this section, we point out the key methodological differences between SeTGAP and two other decomposition-based SR frameworks, namely CVGP (Jiang & Xue, 2023) and ScaleSR (Chu et al., 2024).

### 3.4.1 Comparison to CVGP

It is worth pointing out that our cascade-fashion approach shares similarities with the CVGP method. CVGP analyzes the $t$ system variables in a fixed order: $x_1, x_2, \ldots, x_t$. First, multiple datasets (trials) are generated where only $x_1$ varies while others remain fixed. A GP algorithm extracts a skeleton involving $x_1$. Then, new trials are generated where $x_1$ and $x_2$ vary, and the previously discovered skeleton is fixed while GP extends it to include both variables. This process repeats until all variables are included. Our approach differs in both motivation and implementation. SeTGAP starts by independently identifying univariate skeletons for each variable, focusing on interpreting the individual functional relationships learned by the opaque model $\hat{f}$. Once fixed, it aims to discover mathematically valid ways to merge them into coherent multivariate expressions. In contrast, CVGP builds successive expressions atop a single skeleton for $x_1$. If this initial skeleton is poorly fitted, subsequent expressions tend to compensate for the misspecification rather than recover accurate functional relationships for the remaining variables.

### 3.4.2 Comparison to ScaleSR

Another related decomposition-based approach is ScaleSR. Note that while ScaleSR describes itself as a neural SR framework, its search mechanism relies on Monte Carlo tree search rather than the transformer-based architectures typically associated with modern neural SR. Methodologically, both ScaleSR and SeTGAP

Table 2: Equations used for experiments

| Eq. | Underlying equation | Reference | Domain range |
|---|---|---|---|
| E1 | $(3.0375 x_0 x_1 + 5.5 \sin(9/4(x_0 - 2/3)(x_1 - 2/3)))/5$ | Jin et al. (2020) | $[-5, 5]^2$ |
| E2 | $5.5 + (1 - x_0/4)^2 + \sqrt{x_1 + 10} \sin(x_2/5)$ | [Ref. 1][2] | $[-10, 10]^2$ |
| E3 | $(1.5 e^{1.5 x_0} + 5 \cos(3 x_1))/10$ | Jin et al. (2020) | $[-5, 5]^2$ |
| E4 | $((1 - x_0)^2 + (1 - x_2)^2 + 100(x_1 - x_0^2)^2 + 100(x_3 - x_2^2)^2)/10000$ | Rosenbrock-4D | $[-5, 5]^4$ |
| E5 | $\sin(x_0 + x_1 x_2) + \exp(1.2 x_3)$ | [Ref. 1][2] | $x_1 \in [-10, 10], \ x_2 \in [-5, 5],$ $x_3 \in [-5, 5], \ x_4 \in [-3, 3]$ |
| E6 | $\tanh(x_0/2) + |x_1| \cos(x_2^2/5)$ | [Ref. 1][2] | $[-10, 10]^3$ |
| E7 | $(1 - x_1^2)/(\sin(2\pi x_0) + 1.5)$ | Werner et al. (2021) | $[-5, 5]^2$ |
| E8 | $x_0^4/(x_0^4 + 1) + x_1^4/(x_1^4 + 1)$ | Trujillo et al. (2016) | $[-5, 5]^2$ |
| E9 | $\log(2 x_1 + 1) - \log(4 x_0^2 + 1)$ | Trujillo et al. (2016) | $[0, 5]^2$ |
| E10 | $\sin(x_0 e^{x_1})$ | Bertschinger et al. (2023) | $x_1 \in [-2, 2], x_2 \in [-4, 4]$ |
| E11 | $x_0 \log(x_1^4)$ | Bertschinger et al. (2023) | $[-5, 5]^2$ |
| E12 | $1 + x_0 \sin(1/x_1)$ | Bertschinger et al. (2023) | $[-10, 10]^2$ |
| E13 | $\sqrt{x_0} \log(x_1^2)$ | Bertschinger et al. (2023) | $x_1 \in [0, 20], x_2 \in [-5, 5]$ |

utilize an opaque model as a data proxy to decompose multivariate problems into single-variable sub-tasks. However, their data generation strategies differ significantly. When analyzing an independent variable $x_v$, ScaleSR generates a single synthetic dataset by varying it alongside previously learned variables while fixing all remaining unlearned variables to a single set of randomly selected constant values.

Fixing unlearned variables to single constant values introduces a substantial risk of functional projection artifacts, where the underlying structure becomes obscured, as discussed in Sec. 3.2.1. For instance, consider the function $f(x) = \sin\left(\frac{x_1}{10 x_2} + \frac{\pi}{2}\right)^2$. If $x_2$ is held constant at a specific value, such as 10, the function is projected into a narrow domain region where it may become flat or not expressive enough for an SR method to identify its sinusoidal behavior. SeTGAP avoids this limitation by varying only the specific variable under analysis across multiple $N_s$ independent sets. This frames the task as an MSSP problem where the Multi-Set transformer must identify a single skeleton that characterizes the functional form of all $N_S$ generated sets.

Furthermore, ScaleSR's skeleton merging mechanism differs fundamentally from ours. It discovers univariate skeletons sequentially by modeling the numerical coefficients of previously discovered variables as functions of the newly integrated variable. For example, if the skeleton for $x_1$ is determined to be $c_1 x_1 + c_2$, the next stage attempts to solve for $c_1$ and $c_2$ as mathematical functions of $x_2$ explicitly. This sequential layering creates a strict structural dependency, similar to CVGP. That is, any misidentification or slight error in an early skeleton propagates heavily into subsequent stages, forcing later variables to numerically compensate for the misspecification rather than recovering the true functional form. SeTGAP resolves this by discovering univariate skeletons independently before assembling them through a structure-preserving cascade.

# 4 Experimental Results

## 4.1 Non-linear Synthetic Problems

We evaluated SeTGAP on 13 synthetic SR problems inspired by previous work. Table 2 lists these equations along with their domain ranges. The 13 synthetic SR problems were chosen to cover a range of functional forms and difficulties. Equations E1, E3, E4, E7, E8, and E9 correspond to expressions frequently used in prior SR studies (Trujillo et al., 2016; Jin et al., 2020; Werner et al., 2021), while E10–E13 were adapted to the multivariate setting from the suite proposed by Bertschinger et al. (2023). We also included E2, E5, and E6 [Ref. 1][2] to increase the proportion of non-separable problems, a class where neural SR approaches have been observed to struggle. All equations were evaluated over extended input ranges (e.g., $[-5, 5]$ and $[-10, 10]$) rather than the narrow domains commonly used in earlier works, thereby increasing problem difficulty. The datasets generated from these equations consisted of 10,000 points and each variable was sampled according to a uniform distribution.

The regression models $\hat{f}$ tested are assumed to be opaque functions trained to approximate the target response. In our experiments, we implemented $\hat{f}$ as feedforward neural networks whose architectures were tuned independently for each problem to minimize MSE, as is standard in regression tasks. Different problems vary in complexity, and accordingly, the number of hidden layers. They featured three hidden layers for

Table 3: Comparison of predicted equations (E1—E4) with rounded numerical coefficients — Iteration 1

| Method | E1 | E2 | E3 | E4 |
|---|---|---|---|---|
| **PySR** | $0.61x_0x_1$ | $0.41x_2 + ||x_0 - 3.51| - 1.95| + 4.11$ | $0.34e^{x_0}|\sinh(0.47x_0)|$ | $0.21x_0^2 - 0.18x_1 + 0.21x_2^2 - 0.18x_3 - 0.76$ |
| **TaylorGP** | $0.64x_0x_1$ | $-0.5x_0 + 0.001x_1 + 0.39x_2 + 8.62$ | $0.23x_0e^{x_0}$ | $0.29x_0^2$ |
| **NeSymReS** | $0.59x_0x_1 + \cos(0.01(x_1 - x_0 - 0.08)^2)$ | $-x_0 + 0.40x_2 + e^{e^{-0.001x_1}} + 5.88$ | $9.10e^{0.72x_0}\cos(0.15x_1)$ | — |
| **E2E** | $1.08(0.56x_0x_1 - 0.03x_0 + 0.02x_1 - \sin(0.01x_0^2 + 8.6x_0 + 0.45) - 0.01)$ | $0.06x_0^2 - 0.51x_0 - 0.22x_1\cos(0.18x_2 + 1.43) - 0.01x_1 + 0.01x_2 - 3.25\cos(0.18x_2 + 1.43) + 6.56$ | $0.14e^{1.52x_0} + 0.52\cos(3.45x_1 + 0.05) + 0.11$ | $0.001|8.99(-0.88x_1 + (x_0 + 0.01)^2 + 0.62)^2 + 9.72(-x_3 + 0.98(x_2 + 0.01)^2 + 0.01)^2| + 0.0023$ |
| **uDSR** | $(-0.006x_0^2 - 0.001x_0x_1^2 + 0.61x_0x_1 + 0.001x_0 - 0.001x_1^2 + 0.031)\cos(e^{-4x_1^2 + x_1})$ | $0.063x_0^2 - 0.501x_0 + 0.029x_1x_2 - 0.001x_2^3 + 0.351x_2 + \sin(0.25x_2) + 6.498$ | $0.013x_0^4 - 0.032x_0^3 - 0.001x_0^2x_1 - 0.41x_0^2 + 0.001x_0x_1^2 + 0.002x_0x_1 - 0.902x_0 - 0.087x_1^4 - 0.001x_1^3 + 0.397x_1^2 + 0.006x_1 + e^{x_0} - \cos(x_1) + \cos(2x_1) - 0.477$ | $0.01x_0^4 - 0.02x_0^2x_1 + 0.01x_1^2 + 0.01x_2^4 - 0.02x_2^2x_3 + 0.01x_3^2$ |
| **TPSR** | $0.146x_1(4.161x_0 - 0.071) - 0.036\sin(5.687x_1 - 1.848) - 0.002$ | $\left(0.014 + \frac{1}{x_0 + 14.596}\right)(0.039x_0 + 3.247)\left(-0.059x_1 - 83.999 + \frac{0.061}{x_2}\right)(0.028x_2 - 0.409\arctan(0.14474x_2 + 0.0015) - 0.334)$ | $0.15e^{1.5x_0} + 0.5\cos(3.0x_1)$ | $0.175x_1 - 0.124x_3 + 0.3626 + (8.858 - 5.843x_1)(0.001x_1 + 0.073) + 0.0016\left(x_0 + 0.595(0.944x_0 - 1)^2\right)^3$ |
| **SeTGAP** | $0.61x_0x_1 + 1.15\sin((2.24x_0 - 1.5)(x_1 - 0.68))$ | $0.06x_0^2 - 0.5x_0 + (3.37\sqrt{0.1x_1 + 1} - 0.19)(\sin(0.2x_2) + 0.01) + 6.49$ | $0.15e^{1.5x_0} + 0.5\sin(3x_1 - 4.71)$ | $0.01x_0^4 - 0.02x_0^2x_1 - 0.001x_0 + 0.01x_1^2 + 0.01x_2^4 + 0.01x_3^2 - (0.02x_2^2 + 0.004)(x_3 - 0.11) - 0.02$ |

problem E2, five for E1, E4, E5, and E7, and four for all other cases, with each layer containing 500 ReLU-activated nodes. Tuning these models is not part of SeTGAP's process, but ensures that $\hat{f}$ provides a reasonable approximation of the underlying function. In addition, the Multi-Set Transformer $g$, used to infer univariate skeletons, follows the design and hyperparameters established in [Ref. 1][2]. The generated collections $\tilde{\mathbf{D}}_v$ consist of $N_S$ sets of $n = 3000$ elements, with these parameters tuned in prior work.

For SeTGAP, the hyperparameters in Algorithm 1 include the beam size $n_B$ and the number of univariate skeleton candidates $n_{\text{cand}}$. We set $n_B = 3$ and $n_{\text{cand}} = 3$, as higher values did not yield more distinct skeletons across all problems. The GAs in Secs. 3.2 and 3.3 share the same configuration and loss function: minimizing MSE. The GA runs with a population of 500 individuals and terminates when the objective function change remains below $10^{-6}$ for 30 consecutive generations. It uses tournament selection, binomial crossover, and generational replacement. Algorithm 3 uses the number of instance expressions per candidate skeleton combination `rep` and the maximum number of generations `maxG`, with values set to `rep` = 150 and `maxG` = 300. In addition, the initial population `candSks` was generated with a maximum size $P_{max} = 5000$, though none of the cases reached this limit. This setup was chosen for its consistent and effective optimization results across all problems. Please refer to Appendix E for additional ablation results.

For each problem, SeTGAP generates up to $n_{\text{cand}}$ multivariate expressions, but for brevity, we report only the one with the lowest MSE. These results are compared against expressions obtained from two GP-based methods, PySR (Cranmer, 2023) and TaylorGP (He et al., 2022), three neural SR approaches, NeSym-ReS (Biggio et al., 2021), E2E (Kamienny et al., 2022), and TPSR (Shojaee et al., 2023), and a hybrid approach, uDSR (Landajuela et al., 2022). The comparison did not include the SR neuro-evolution method proposed by Bertschinger et al. (2024), as training and evolving their large transformer architectures in a multivariate setting is computationally prohibitive. NeSymReS could not be applied to E4 and E5 due to its limitation to systems with at most three variables. The only hyperparameters tuned were the beam size and the number of BFGS restarts, both initially set to 5 and 4, respectively. However, it was found that a beam size above 2 and more than 2 BFGS restarts were unnecessary. The transformer architecture used by uDSR follows a similar configuration. For E2E, we limited the number of generated candidates to a maximum of $K = 10$, as increasing this value provided no observable improvement. TPSR employed E2E as its backbone, and its regularization parameter $\lambda$, which controls the trade-off between accuracy and expression complexity, was set to 1, as smaller values led to unnecessarily complex expressions.

For GP-based methods, we set an iteration limit of 10,000, though all cases converged earlier. Population sizes of 100, 200, 500, and 1000 were tested, with no observed benefits beyond 500. All methods, including uDSR, were configured to use the same set of operators as those present in the vocabularies used by the neural SR methods. In addition, uDSR was configured to include polynomial terms up to degree four and executed for up to 2 million expression evaluations per sub-problem. Although our approach may appear

Table 4: Comparison of predicted equations (E5—E8) with rounded numerical coefficients — Iteration 1

| Method | E5 | E6 | E7 | E8 |
|---|---|---|---|---|
| PySR | $e^{1.2x_3} + \sin(x_0 + x_1 x_2)$ | $\tanh(e^{x_2})$ | $(0.56 - 0.59x_0^2)/(\sinh(\sinh(\tanh(e^{\sinh(\sin(6.28x_1))}))))$ | $(\tanh(\cosh(x_0) - 1.04) + \tanh(\cosh(x_1) - 1.04))$ |
| TaylorGP | $0.51e^{x_3}e^{\sin(0.87x_3)}$ | $-\frac{\sin(0.34x_2^2)}{-\sqrt{|x_2| + \sin(\sqrt{|x_2|})}}$ | $\sqrt{|x_1|} - x_1^2$ | $2$ |
| NeSymReS | — | $-0.39x_0 + x_1\sin(\frac{x_0}{x_1} - 0.001x_2)$ | $\frac{0.12x_0 + x_1^2}{\cos(3.1(-0.02x_1-1)^2) - 0.31}$ | $\cos(\sin(1.69x_0)/(x_0 x_1)) + 0.71$ |
| E2E | $e^{1.2x_3} - 0.91\cos((2.62x_0 + 0.15)(24.66x_1 + 1.24)) - 0.05$ | $0.01x_1(-7.5\cos(15.41x_1 + 0.21) - 0.18) + 0.69\operatorname{atan}(0.75x_0 + 0.05) + 0.47$ | $(-0.03x_0 - 0.03)(0.34x_1 - 0.35)(41.59(1 - 0.5\sin(6.74x_0 + 0.23))^2 + 40)$ | $2.01 - 1.05e^{-0.06|x_0 2.73 - 0.14||0.59x_1 + 0.1|}$ |
| uDSR | $0.001x_0^2 - 0.001x_0 x_2 x_3 + 0.001x_0 x_2 - 0.002x_0 x_3 + 0.002x_1 x_3 + 0.003x_1 + 0.002x_2^2 + 0.004x_2 x_3 + 0.003x_2 + 0.152x_3^4 + 0.568x_3^3 + 0.506x_3^2 + 0.609x_3 + \sin(x_0 + x_1 x_2) + 1.074$ | $-0.002x_0^3 + 0.001x_0^2 - 0.005x_0 x_1 - 0.003x_0 x_2 + 0.234x_0 + 0.046x_1^2 - 0.001x_1 x_2 + 0.005x_1 + 0.003x_2^4 - 0.257x_2^2 - 0.012x_2 + 3\cos(x_2) - \cos(2x_2) + 2.512$ | $-0.001x_0^3 x_1 - 0.017x_0^3 + 0.002x_0^2 x_1 + 0.003x_0^2 - 0.019x_0 x_1^2 + 0.017x_0 x_1 + 0.283x_0 - 0.001x_1^4 - 0.89x_1^2 - 0.042x_1 + e^{\sin(6x_0)} - 0.372$ | $0.008x_0^4 - 0.179x_0^2 + 0.001x_0 x_2 + 0.001x_0 - 0.004x_1^4 + 0.107x_1^2 + 0.004x_1 - e^{\cos(x_1)} - \cos(x_0) + \cos(x_1) + 2.721$ |
| TPSR | $1.032e^{-0.001x_2 + 1.191x_3} - 0.054 + 0.063\cos(-2.276x_0 + x_1 + 5.52)$ | $(0.003 - 0.024(\cos(39.378(1 - 0.005x_2)^2 - 38.823) + 0.56)^2)(-3.997x_0 + 51.732x_2 + 16.608\cos(1.089x_1 - 0.479) - 135.856)$ | $0.681 - 0.848(-x_1 - 0.134\cos(x_0 + 2.153) - 0.069)^2$ | $2.002 - 1.474e^{-0.009|(39.99x_0 - 0.28)(1.08x_1 + 0.03)|}$ |
| SeTGAP | $0.999e^{1.2x_3} - \sin(x_0 + x_1 x_2 + 9.42)$ | $\cos(0.2x_2^2 + 0.05)|x_1| + \tanh(0.5x_0)$ | $\frac{4.53 - 4.54x_1^2}{4.54\sin(6.28x_0 + 6.28) + 6.81}$ | $2 - \frac{19.76}{19.31x_0^4 + 0.12x_0^3 + 0.42x_0^2 + 19.72} - \frac{5.33}{5.44x_1^4 - 0.09x_1^2 + 5.34}$ |

Table 5: Comparison of predicted equations (E9—E13) with rounded numerical coefficients — Iteration 1

| Method | E9 | E10 | E11 | E12 | E13 |
|---|---|---|---|---|---|
| PySR | $\log(\frac{x_1 + 0.5}{0.5 + 2x_0^2})$ | $\sin(x_0 e^{x_1})$ | $x_0\log(x_1^4)$ | $\sin(\frac{x_0}{x_1/0.12})|x_0| + 0.99$ | $\sqrt{x_0}\log(x_1^2)$ |
| TaylorGP | $\log(\frac{9.79}{|x_0|}) - 2.36e^{-x_1}$ | $x_0 e^{-\sqrt{|x_1|}}$ | $4x_0\log(|x_1|)$ | $(x_0 + \sqrt{|x_1|} - 0.91)\sin(\frac{0.73}{x_1})$ | $\sqrt{e^{\sqrt{x_0}}}\log(|x_1|) + \log(|x_1|) + 0.58$ |
| NeSymReS | $1.12\log(|x_1/x_0|) - 1.37$ | $\sin(x_0 e^{x_1})$ | $x_0\log(x_1^4)$ | $(x_0 + x_1)\sin(1/x_1)$ | $0.31x_0 + 3.19\log(x_1^2) - 3.25$ |
| E2E | $2 - 0.60\log(13.36(0.004 - x_0)^2(1 - 0.13/(-0.06x_1 - 0.02))^2 + 0.8)$ | $-0.98\sin((0.06 - 2.86e^{1.03x_1})(0.32x_0 + 0.002)) - 0.007$ | $-0.74x_0(-5.62\log(0.07|-6.94x_1 + 0.13| + 0.01) - 3.74)$ | $(0.79x_0 - 0.04)\sin(4.5/(3.4x_1 + 0.08))$ | $(-90.0 + \frac{9}{0.12|3.4x_1 + 0.12| + 0.04})(0.09 - 0.1\log(0.17x_0 + 3.29))$ |
| uDSR | $\log\left(\frac{x_1(2x_1+1)}{4.0x_0^2 x_1 + 1.0x_1}\right)$ | $\sin(x_0 e^{x_1})$ | $x_0\log(1.0x_1^4)$ | $(x_0 x_1\sin(1/x_1) + x_1)/x_1$ | $\log(x_1^2)\log(0.004x_0^3 + 0.06x_0^2 + 0.001x_0 x_1 + 1.189x_0 - 0.003x_1 + 1.422)$ |
| TPSR | $0.401x_1 - 0.025|8.656x_0 - 120.665\arctan(-0.611x_0 - 0.004) + 0.098| + 0.94$ | $1.0\sin(1.002x_0 e^{x_1})$ | $(0.022 - 0.601x_0)(-0.223x_1^4 - 7.502 - \frac{36.162}{-4.305|x_1| - 0.971})$ | $0.001x_0(17.945x_1 - 0.666 + \frac{1.203}{x_1}) - 0.008x_0 + 0.941$ | $\left(-0.008 + \frac{0.742}{5.792|1.369x_1 + 0.005| + 1.035}\right)\left(7.888 - \frac{28.241}{x_0 + 4.024}\right)(x_1 - 0.995)(1.462x_0 + 44.99)(0.112x_1 + 0.111)$ |
| SeTGAP | $-\log(13.95x_0^2 + 3.48) + |\log(8.32x_1 + 4.18)| - 0.18$ | $\sin(x_0 e^{0.999x_1})$ | $1.998x_0\log(x_1^2)$ | $x_0\sin(1/x_1) + 1$ | $2\sqrt{x_0}\log|x_1|$ |

similar to CVGP, as discussed in Sec. 3.3.3, we did not include it in our experiments. This is because CVGP relies heavily on a data oracle and does not apply to a standard SR setting where a fixed initial dataset is available, making a fair comparison infeasible. In addition, ScaleSR was originally evaluated only on the basic Nguyen benchmark, achieving unsatisfactory results. Given its methodological limitations and its demonstrated limited applicability, it was excluded from our experimental evaluation as it is not suited for the complex systems considered in this study. Tables 3–5 present the learned expressions, with shaded cells indicating that the learned expression's functional form matches that of the underlying function. To ensure a fair comparison, the evaluation was repeated nine additional times, each with a newly generated dataset using a different random seed. All expressions obtained across problems and iterations are reported in Appendix F. Table 6 summarizes the results across all iterations, showing the ratio of successful runs.

We evaluated the extrapolation capability of the learned expressions by testing them on an extended domain range. The original domain range, or interpolation range, for a variable $x_v$ is denoted as $[x_v^\ell, x_v^u]$, while its extrapolation range is defined as $[2x_v^\ell, x_v^\ell[ \cup ]x_v^u, 2x_v^u]$. This evaluation was repeated for each of the 10 expressions learned by each method. We also evaluated the extrapolation capability of the opaque models $\hat{f}$ trained for each problem. Each extrapolation set comprised 10,000 points sampled uniformly within this range. Table 7 shows the rounded mean and standard deviation of the extrapolation MSE across these runs. Bold entries indicate the method with the lowest mean error and a statistically significant difference, determined by Tukey's honestly significant difference test at the 0.05 significance level. Appendix I presents the results of SeTGAP evaluated on functions from the SRBench++ benchmark (de Franca et al., 2025).

Regarding the fairness of the experimental comparisons, it is necessary to contextualize the differences between the operational pipelines of standard SR and the proposed framework. Conventional SR methods map the fixed dataset directly to an expression via a data-to-expression pipeline. In contrast, SeTGAP implements a distillation pipeline where the same fixed dataset is first used to train the opaque model $\hat{f}$, which

Table 6: Success rates in identifying the correct functional form across 10 iterations (E1–E13)

| Method | E1 | E2 | E3 | E4 | E5 | E6 | E7 | E8 | E9 | E10 | E11 | E12 | E13 |
|---|---|---|---|---|---|---|---|---|---|---|---|---|---|
| PySR | 6/10 | 0/10 | 7/10 | 0/10 | 8/10 | 2/10 | 0/10 | 0/10 | 1/10 | 9/10 | 9/10 | 7/10 | 8/10 |
| TaylorGP | 0/10 | 0/10 | 0/10 | 0/10 | 0/10 | 0/10 | 0/10 | 0/10 | 0/10 | 6/10 | 9/10 | 1/10 | 6/10 |
| NSymRes | 0/10 | 0/10 | 3/10 | 0/10 | 0/10 | 0/10 | 0/10 | 0/10 | 0/10 | 10/10 | 10/10 | 0/10 | 1/10 |
| E2E | 0/10 | 0/10 | 2/10 | 1/10 | 0/10 | 0/10 | 0/10 | 0/10 | 0/10 | 4/10 | 0/10 | 0/10 | 0/10 |
| uDSR | 0/10 | 0/10 | 1/10 | 9/10 | 0/10 | 0/10 | 0/10 | 0/10 | 6/10 | 10/10 | 9/10 | 9/10 | 0/10 |
| TPSR | 0/10 | 0/10 | 6/10 | 0/10 | 1/10 | 0/10 | 0/10 | 0/10 | 0/10 | 10/10 | 0/10 | 0/10 | 0/10 |
| SeTGAP | 10/10 | 10/10 | 10/10 | 10/10 | 10/10 | 10/10 | 10/10 | 10/10 | 10/10 | 10/10 | 10/10 | 10/10 | 10/10 |

Table 7: Extrapolation MSE Comparison

| Eq. | NN | PySR | TaylorGP | NeSymRes | E2E | uDSR | TPSR | SeTGAP |
|---|---|---|---|---|---|---|---|---|
| E1 | 431.1 ± 0.39 | 0.37 ± 0.40 | 1.22 ± 1.31 | 3.37 ± 3.49 | 1.22 ± 0.13 | 1.94 ± 1.32 | 0.65 ± 0.61 | **0.12 ± 0.09** |
| E2 | 30.77 ± 0.37 | 98.99 ± 64.70 | 238.50 ± 75.63 | 8.59e+7 ± 1.91e+8 | 68.85 ± 68.22 | 1.32e+03 ± 3.19e+02 | 1.38e+7 ± 4.04e+7 | **0.08 ± 7.62e-2** |
| E3 | 2.14e+4 ± 594.9 | 2.07e+3 ± 6.19e+3 | 3.15e+4 ± 3.96e+3 | 4.65e+4 ± 9.21e+3 | 1.45e+5 ± 3.68e+5 | 4.84e+04 ± 1.16e+04 | 9.42e+7 ± 2.83e+8 | **10.28 ± 17.89** |
| E4 | 2869 ± 15.4 | 2.21e+5 ± 9.04e+4 | 6.62e+3 ± 1.06e+3 | — | 876.20 ± 2.32e+3 | 5.726e+04 ± 5.035e+02 | 2.44e+5 ± 7.26e+5 | **1.62 ± 1.89** |
| E5 | 9.29e+4 ± 1.19e+3 | **5.01e-2 ± 0.15** | 2.22e+4 ± 2.99e+4 | — | 2.05e+3 ± 2.81e+3 | 5.726e+04 ± 5.035e+02 | 5.73e+4 ± 1.72e+5 | 1.02 ± 1.33 |
| E6 | 200.8 ± 0.52 | 18.26 ± 34.46 | 116.10 ± 3.86 | 189.20 ± 40.71 | 130.20 ± 25.14 | 2.631e+04 ± 1.876e+03 | 213.9 ± 191.3 | **3.83 ± 5.89** |
| E7 | 2330 ± 14.28 | 570.70 ± 1.11e+3 | 1.04e+3 ± 819.30 | 1.62e+3 ± 819.30 | 1.78e+3 ± 525.90 | 1.384e+03 ± 1.071e+02 | 1.28e+3 ± 656.6 | **3.60e-2 ± 4.49e-2** |
| E8 | 1.68e-2 ± 4.66e-5 | **3.32e-4 ± 4.76e-4** | 1.18 ± 2.12 | 0.12 ± 1.91e-2 | 0.13 ± 0.38 | 5.731e+02 ± 2.295e+02 | 1.25e-2 ± 1.83e-2 | **3.38e-8 ± 6.31e-8** |
| E9 | 6.09e-2 ± 3.73e-4 | 779.40 ± 2.33e+3 | 0.30 ± 0.15 | 1.03 ± 4.76e-2 | 0.43 ± 0.17 | 1.06e+01 ± 2.65e+01 | 0.77 ± 1.11 | **2.72e-6 ± 4.09e-6** |
| E10 | 2.39 ± 3.38e-2 | **7.3e-11 ± 1.5e-10** | 0.17 ± 0.21 | **0.00 ± 0.00** | 0.36 ± 4.76e-2 | **0.00 ± 0.00** | **4.96e-12 ± 1.45e-11** | 3.72e-4 ± 5.62e-4 |
| E11 | 240.2 ± 3.09 | 1.94e-2 ± 5.83e-2 | 3.08 ± 8.02 | **0.00 ± 0.00** | 98.80 ± 104.80 | **2.35e-11 ± 5.34e-11** | 200.8 ± 125.0 | 9.36e-5 ± 2.81e-4 |
| E12 | 4.35 ± 0.24 | 0.80 ± 2.41 | 2.97 ± 1.57 | 2.57e-2 ± 0.00 | 5.31 ± 1.66 | 6.78e-03 ± 2.03e-02 | 15.42 ± 20.69 | **1.11e-6 ± 2.22e-6** |
| E13 | 1.43 ± 5.5e-2 | 2.72 ± 8.16 | 9.43 ± 23.11 | 60.55 ± 15.63 | 182.90 ± 330.00 | 1.47e+05 ± 4.41e+05 | 26.17 ± 22.79 | **8.59e-7 ± 1.78e-6** |

Table 8: MSE comparison using SeTGAP with noisy data

| Eq. | Interpolation ($\sigma_a = 0$) | Interpolation ($\sigma_a = 0.01$) | Interpolation ($\sigma_a = 0.03$) | Interpolation ($\sigma = 0.05$) | Extrapolation ($\sigma_a = 0$) | Extrapolation ($\sigma_a = 0.01$) | Extrapolation ($\sigma_a = 0.03$) | Extrapolation ($\sigma_a = 0.05$) |
|---|---|---|---|---|---|---|---|---|
| E1 | 7.159e-03 | 1.259e-02 | 1.236e-01 | 5.455e-01 | 1.012e-01 | 4.433e-01 | 1.645e+00 | 3.845e+00 |
| E2 | 7.024e-03 | 4.459e-03 | 1.563e-02 | 4.754e-02 | 1.300e-01 | 5.232e-02 | 1.017e-01 | 3.504e-01 |
| E3 | 1.063e-03 | 1.022e-03 | 7.219e-03 | 2.003e-02 | 2.961e-01 | 2.055e+02 | 7.467e+01 | 1.652e+02 |
| E4 | 1.158e-04 | 2.969e-03 | 2.061e-01 | 2.228e-02 | 4.121e-02 | 1.022e+01 | 7.420e+01 | 3.105e+01 |
| E5 | 7.211e-06 | 6.807e-03 | 5.288e-01 | 1.694e-01 | 3.069e-01 | 1.097e+01 | 8.402e+01 | 2.312e+02 |
| E6 | 5.619e-03 | 1.147e-02 | 1.726e-02 | 4.819e-02 | 4.508e-02 | 4.651e+00 | 2.081e-01 | 5.598e-01 |
| E7 | 2.655e-06 | 1.076e-02 | 6.822e-02 | 1.903e-01 | 6.447e-05 | 2.517e-01 | 1.247e+00 | 3.520e+00 |
| E8 | 1.742e-06 | 2.413e-05 | 2.103e-04 | 5.828e-04 | 1.817e-10 | 1.686e-07 | 7.539e-09 | 1.672e-07 |
| E9 | 4.907e-08 | 2.160e-04 | 1.943e-03 | 5.408e-03 | 2.780e-07 | 3.063e-04 | 2.756e-03 | 7.943e-03 |
| E10 | 2.282e-06 | 3.298e-05 | 2.950e-04 | 8.192e-04 | 1.857e-04 | 2.233e-04 | 4.793e-04 | 3.379e-03 |
| E11 | 1.178e-04 | 1.838e-02 | 1.654e-01 | 4.595e-01 | 9.308e-04 | 1.453e-01 | 1.308e+00 | 3.632e+00 |
| E12 | 8.029e-06 | 4.948e-04 | 4.436e-03 | 1.232e-02 | 7.273e-06 | 1.031e-03 | 9.257e-03 | 2.571e-02 |
| E13 | 2.583e-07 | 4.200e-03 | 3.991e-02 | 1.047e-01 | 1.015e-06 | 9.599e-03 | 9.348e-02 | 2.389e-01 |

is subsequently queried by the explainable SR method to extract the final expression $\tilde{f}$. Because $\hat{f}$ is bound by the constraints, sample size, and distribution of the original training data, SeTGAP does not possess an unfair advantage regarding the underlying system information. If the initial dataset is limited or poorly distributed, the opaque model will naturally yield an inadequate approximation of function $f$, and the resulting distilled expression will inherently reflect that mismatch.

Furthermore, we tested SeTGAP under increasingly noisy conditions. We considered a normal error term, defined as $\varepsilon_a = \mathcal{N}(0, \sigma_a \sigma_{\mathbf{y}})$, and four noise levels: $\sigma_a = \{0, 0.01, 0.03, 0.05\}$. Here, $\sigma_{\mathbf{y}}$ denotes the standard deviation of the response variable so that the noise is scaled relative to the dispersion of each problem. The obtained interpolation and extrapolation MSE are shown in Table 8, where shaded cells indicate an incorrectly identified functional form. The corresponding learned expressions are provided in Appendix G.

Finally, in [Ref. 2][2], we studied the behavior of SeTGAP under the influence of varying levels of epistemic uncertainty. As an illustrative example, consider an initial incomplete dataset generated by $f(x) = 10 + 5\cos(\frac{\mathbf{x}^2}{5})$ subject to heteroskedastic noise $\epsilon_a = \mathcal{N}(0, \frac{1}{2}(1 - \frac{\mathbf{x}^2}{100}))$. An adaptive sampling (AS) mechanism called Adaptive Sampling with Prediction-Interval Neural Networks (ASPINN) (Morales & Sheppard, 2025) was utilized to intelligently select samples that maximize uncertainty reduction using prediction interval-generating neural networks and Gaussian processes. This setup allows for an analysis of how the

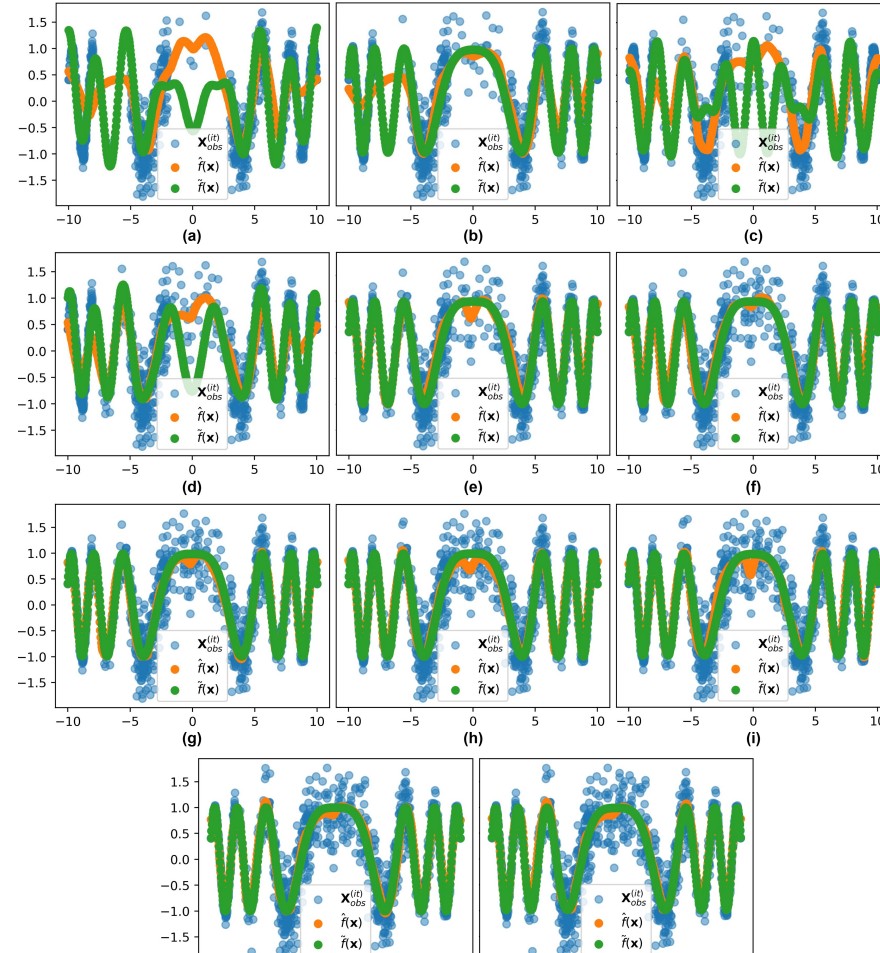

Table 9: Identified expressions during the AS process [Ref. 2]

| it | $\tilde{f}_{it}$ | MSE |
|----|------------------|-----|
| 1 | $0.145\mathbf{x}\sin(2.643\mathbf{x}+6.235)+$ $0.564\sin(1.732\mathbf{x}+10.919)$ | 0.325 |
| 5 | $0.012-0.983\cdot$ $\sin(0.197\mathbf{x}^2+4.911)$ | 0.222 |
| 10 | $-1.271\sin(1.201\mathbf{x}+6.256)^2$ $-0.495\sin(3.242\mathbf{x}$ $+10.966)+0.655$ | 0.409 |
| 15 | $-0.016\mathbf{x}^2\sin(2.849\mathbf{x}+1.598)+$ $0.053\mathbf{x}\sin(1.122\mathbf{x}-0.075)$ $+0.762\sin(1.745\mathbf{x}+4.757)$ | 0.277 |
| 20 | $0.993\sin(0.199\mathbf{x}^2+1.612)$ $+0.003$ | 0.220 |
| 25 | $0.994\sin(0.200\mathbf{x}^2+1.608)$ $+0.004$ | 0.222 |
| 30 | $0.991\sin(0.199\mathbf{x}^2$ $+1.623)$ | 0.223 |
| 35 | $-0.997\sin(0.200\mathbf{x}^2$ $+17.315)$ | 0.221 |
| 40 | $0.986\sin(0.198\mathbf{x}^2$ $+1.688)+0.004$ | 0.223 |
| 45 | $1.0\sin(0.200\mathbf{x}^2+1.612)$ $+0.002$ | 0.228 |
| 50 | $0.990\sin(0.199\mathbf{x}^2-17.217)$ $+0.004$ | 0.231 |

Figure 3: $\hat{f}_{it}(\mathbf{x})$ vs. $\tilde{f}_{it}(\mathbf{x})$ throughout the AS process for `cosqr`. $it =$ **(a)** 1 **(b)** 5 **(c)** 10 **(d)** 15 **(e)** 20 **(f)**25 **(g)** 30 **(h)** 35 **(i)** 40 **(j)** 45 **(k)** 50. [Ref. 2]

learned expressions evolve over AS iterations and whether they successfully converge toward the target function. Table 9 summarizes the expressions recovered by SeTGAP at intervals $it \in \{1, 5, 10, \ldots, 50\}$, where the dataset is augmented by five samples per iteration. For each expression, we report the predicted MSE over the entire domain computed using the learned expression $\tilde{f}_{it}$, with highlighted cells indicating an exact functional match to the target function $f$. The results demonstrate that the estimated expressions consistently recover the true functional form by iteration $it = 20$. In certain cases, such as in $it = 5$ (Fig. 3.b), we uncovered the correct underlying functional form even when the prediction model $\hat{f}_{it}$ exhibited notable inaccuracies. However, these discoveries tended to be unstable, as subsequent iterations often produced alternative expressions. Further configuration details and synthetic examples, alongside an application to a real-world precision agriculture problem, are provided in [Ref. 2][2].

## 4.2 Benchmark Problems

Note that the objective of this paper is not to introduce a state-of-the-art SR method but is focused on improved interpretability of opaque models. Specifically, we aim to highlight the advantages of a decomposable symbolic regression approach compared to conventional strategies. Thus, in the previous section, we showed that by focusing on accurate variable-wise identification of functional forms, it is possible to merge the gen-

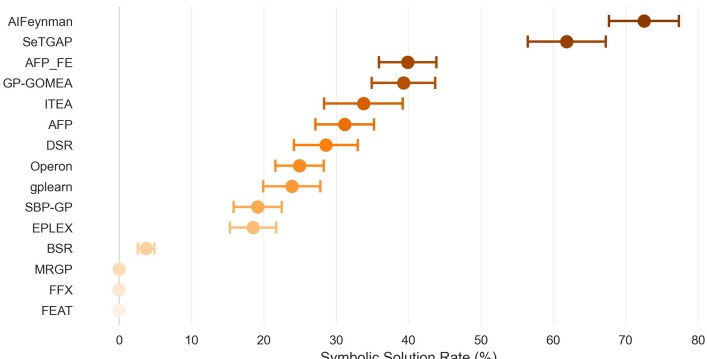

Figure 4: Comparison of the symbolic solution recovery rate distribution obtained on the Feynman dataset

erated univariate skeletons into complete mathematical expressions that recover the underlying equations while preserving the initially identified skeletons.

Nevertheless, benchmarking remains an important component of SR research. It allows us to contextualize our approach within the broader SR landscape and assess its behavior on established benchmark problems. To this end, we evaluated our method on the Feynman dataset, which has been extensively used for benchmarking in the SRBench framework (La Cava et al., 2021). For the sake of fairness, we filter out problems from the Feynman dataset that are not compatible with the representational capacity of our Multi-Set Transformer. In particular, our model was trained on a large synthetic dataset composed of expressions containing up to seven operators, with at most two unary operators and a single level of unary operator nesting [Ref. 1][2]. Equations requiring operators that were absent from the training set, namely $\mathtt{Pow}(\cdot, 2/3)$, $\mathtt{Pow}(\cdot, 5)$, and $\mathtt{arcsin}$, were also excluded. This yields 78 SR problems, listed in Appendix J.

For each problem, SeTGAP is executed 10 times using the configuration described in Sec. 4.1, with each run evaluated on a different data partition, following SRBench settings. The obtained results are compared against those reported for the following benchmarked methods: Age-Fitness Pareto Optimization (AFP) (Schmidt & Lipson, 2010), AFP with co-evolved fitness estimates (AFP_FE) (Schmidt & Lipson, 2009), AIFeynman (Udrescu & Tegmark, 2020), Bayesian Symbolic Regression (BSR) (Jin et al., 2019), Deep Symbolic Regression (DSR) (Petersen et al., 2021), $\varepsilon$-lexicase selection (EPLEX) (La Cava et al., 2019a), Feature Engineering Automation Tool (FEAT) (La Cava et al., 2019b), Fast Function Extraction (FFX) (McConaghy, 2011), GP version of the Gene-pool Optimal Mixing Evolutionary Algorithm (GP-GOMEA) (Virgolin et al., 2021), gplearn[3], Interaction–Transformation Evolutionary Algorithm (ITEA) (de França & Aldeia, 2021), Multiple Regression Genetic Programming (MRGP) (Arnaldo et al., 2014), Operon (Kommenda et al., 2020), Semantic BackPropagation-based GP (SBG-GP) (Virgolin et al., 2019).

Figures 4 and 5 report aggregated results across all problems. Fig. 4 summarizes the average symbolic solution recovery rate. Here, for each problem, we calculate the fraction of runs that recovered the underlying equation, then report the mean and standard error across all problems. Figures 5.a and 5.b present predictive performance in terms of $R^2$ and MSE, respectively, where for each problem we first compute the mean metric across runs, then report the mean and standard error of these per-problem averages across all problems. Tables 51– 50 report the mean and standard deviation of MSE and $R^2$ for each Feynman problem.

## 5   Discussion

SeTGAP can be viewed as a *post-hoc* interpretability method, as it extracts mathematical expressions that align with the functional response learned by a given opaque regression model. Our decomposable approach learns and preserves functional relationships between input variables and the system's response, and increments them progressively, allowing for an interpretable evolution. This prevents the resulting expressions from focusing solely on error minimization, encouraging alignment with the true functional form instead.

---

[3]gplearn: `https://gplearn.readthedocs.io/`

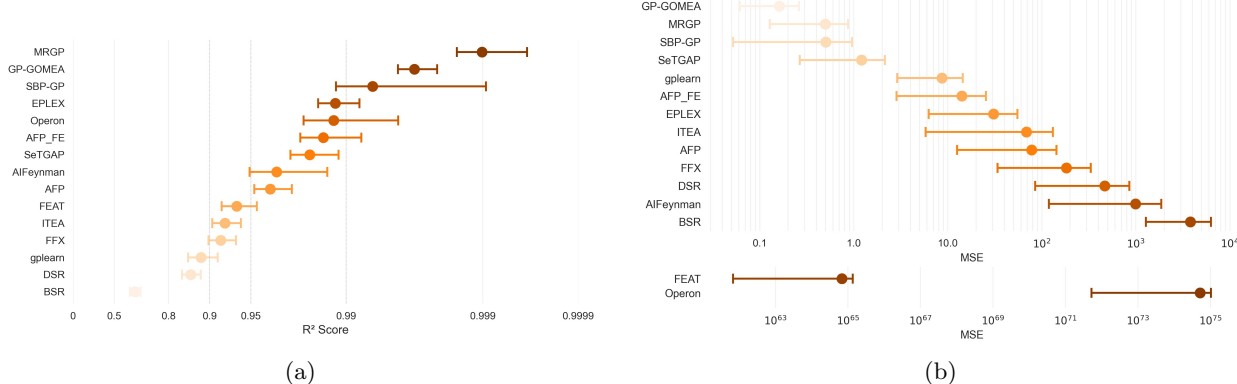

Figure 5: Comparison of the **(a)** $R^2$ and **(b)** MSE distributions obtained on the Feynman dataset

As established in [Ref. 1][2], the Multi-Set Transformer is highly effective at identifying the correct univariate skeleton for each variable within a multivariate system. Traditional SR methods often fail because they attempt to discover the entire multivariate structure simultaneously, which leads to a combinatorial explosion of the search space and a high risk of getting trapped in local minima. The fundamental hypothesis of SeT-GAP is that if the univariate functional forms of all independent variables have been successfully identified, a multivariate expression equivalent to the system's underlying function must exist within a search space that respects these pre-identified structures. While prior work focuses on minimizing empirical error, often at the cost of structural accuracy, SeTGAP uses the initially identified skeletons as structural constraints. This ensures that the final expression remains aligned with the individual functional relationships identified by the Multi-Set Transformer, even when noise or low variable sensitivity might lead other methods to select incorrect but numerically close operators. As a consequence, we transform a high-dimensional symbolic search into a series of guided, incremental operations.

From Tables 3–5, we confirmed that SeTGAP successfully learned mathematical expressions equivalent to the underlying functions in Table 2 across all tested problems and all 10 iterations. In contrast, none of the competing methods correctly identified the underlying functions in more than seven out of the 13 problems in any of the 10 iterations. Notably, some methods only captured the functional form of the most influential variables; i.e., those contributing most to the response value. For instance, E2E recovered correctly the term $0.06x_0^2 - 0.51x_0 + 6.5$ for $x_0$ in E2 but failed for the other variables. E2E and TPSR were the only methods that produced expressions longer than reported, requiring simplification via a symbolic manipulation library[4].

Table 7 confirms that SeTGAP achieved lower or comparable extrapolation MSE values across all problems. These results suggest that other methods, which optimize purely for in-domain MSE, tend to overfit and learn expressions lacking the structural flexibility needed for extrapolation. In contrast, SeTGAP's decompositional approach learns functional forms that better capture underlying relationships, enabling superior generalization beyond the training domain. For example, in problem E8, most SR methods effectively minimized prediction MSE. TaylorGP, prioritizing parsimony, evolved the expression $\tilde{f}(\mathbf{x}) = 2$, effectively smoothing the data but offering no meaningful solution. This illustrates that the simplest solution isn't always best, as small data variations may reflect functional components important for generalization. Table 7 also shows cases where NeSymRes achieved zero extrapolation error across all 10 iterations for E10 and E11, while SeTGAP had low but nonzero MSE. This is because some methods learned expressions that exactly matched the underlying form, eliminating the need for coefficient fitting. PySR, for instance, identified $\sqrt{x_0} \log(x_1^2)$, an expression with no numerical coefficients, for E13 in its first iteration. SeTGAP, by contrast, produced $2.000137\sqrt{x_0} \log|x_1|$, where the fitted constant caused minor errors. Similarly, uDSR achieved near-zero extrapolation error for E4 consistently, whereas SeTGAP obtained higher MSE despite recovering the functional form correctly. We also compare SeTGAP's extrapolation with that of the opaque NNs from which its symbolic expressions are derived. Despite being trained on the same in-domain data, the original networks consistently show significantly higher extrapolation errors, as shown in Table 7. This

---

[4]SymPy: https://www.sympy.org/

highlights SeTGAP's value not only as a regression method but also as an interpretability tool that extracts functional approximations with superior generalization.

Since SeTGAP was the only method that consistently identified the correct functional form across all tested problems, we evaluated its robustness under different noise levels, as shown in Table 8. As expected, interpolation and extrapolation errors increased with higher noise levels. Interpolation errors remained low across all cases, while extrapolation errors showed a few exceptions due to poor coefficient fitting or incorrect functional form identification. For example, E3 and E5 exhibited high extrapolation errors because their functions contain exponential terms. Even when the learned expressions matched the expected functional form, small coefficient errors in the exponential term led to significant deviations for larger values of $x_1$. A similar case was observed in E1, where extrapolation errors were high for $\sigma_a \geq 0.03$ despite correctly identifying the functional form. The ability to recover the correct functional form in the presence of noise can be attributed to two factors. First, the Multi-Set Transformer was trained with small noise levels, making it more resilient to perturbations. Second, the regression NN $\hat{f}$ used during inference to generate the multiple input sets $\tilde{\mathbf{D}}_v$ smooths the estimated response values, mitigating the impact of noise. In the remaining cases, we observed two outcomes. SeTGAP correctly identified the functional forms of individual variables, but noise hindered the detection of relationships between variables. Otherwise, incorrect but reasonable univariate skeletons were identified, leading to expressions with small errors; e.g., for E9 and $\sigma_a = 0.05$, SeTGAP produced $\tilde{f}(\mathbf{x}) = 5.965\sqrt{0.63\log(9.4x_1 + 6.4) + 1} - \log(11.26x_0^2 + 2.82) - 7.71$.

It is important to note that most equations in the Feynman dataset are linearly or additively separable, making them less challenging for our decomposable approach. In contrast, the equations described in Sec. 4.1 involve more complex variable interactions that better highlight the advantages of our approach. Furthermore, some Feynman equations employ variable ranges that are too narrow to enable accurate identification of the true functional forms. For example, equation `I.9.18` ($F = \frac{G\,m_1\,m_2}{(x_2-x_1)^2+(y_2-y_1)^2+(z_2-z_1)^2}$) uses ranges of $[3, 4]$ for $x_1$, $y_1$, and $z_1$, and $[1, 2]$ for $x_2$, $y_2$, and $z_2$. Such limited variation restricts the model's ability to infer the correct functional behavior. Consequently, the only method reported to recover this equation successfully was AIFeynman, which leveraged additional physical information in the form of variable dimensional units.

From Fig. 4, SeTGAP ranks second in symbolic solution recovery rate, with approximately 61.2% successful recoveries. Furthermore, Fig. 5.a shows that SeTGAP appears to rank seventh in terms of the average $R^2$ aggregated across all Feynman problems, and Fig. 5.b shows an apparent fourth place in terms of the average MSE aggregated across the same set of problems. These figures follow the standard reporting format commonly used in the SRBenchmark. However, aggregating results across problems with different characteristics and scales may lead to misleading interpretations. A more reliable assessment can be obtained through statistical significance tests. Specifically, in Appendix J, Tables 49 and 50 show that, according to Tukey's HSD test, SeTGAP achieves the lowest or statistically comparable $R^2$ in 73 out of 78 cases. Similarly, Tables 51 and 52 indicate that SeTGAP achieves the lowest or statistically comparable MSE in 46 out of 78 cases according to the Kruskal–Wallis test followed by Dunn's post hoc pairwise comparisons with Holm correction. Here, as observed earlier in Table 7, SeTGAP often produces small but nonzero MSE values even when it correctly identifies the underlying functional form of a problem. For example, for problem `II.34.29a`, whose underlying function is $\frac{x_0 x_1}{4\pi x_2}$, SeTGAP identified expressions, such as $\frac{0.08002397 x_0 x_1}{x_2}$ and $\frac{0.08038262 x_0 x_1}{x_2}$. In these cases, the structural form of the equations is correctly recovered and the resulting $R^2$ values are consistently equal to 1.0. Nevertheless, the resulting MSE values, with mean and standard deviation $1.04 \times 10^{-5} \pm 6.32 \times 10^{-6}$, while small, are not statistically comparable to those achieved by some competing methods. For instance, GP-GOMEA obtains MSE values on the order of $1.01 \times 10^{-32} \pm 1.16 \times 10^{-33}$ for the same problem. Overall, these results demonstrate that, owing to its decomposable approach, SeTGAP achieves both a high symbolic solution recovery rate and competitive predictive performance, unlike compared methods that primarily focus on minimizing prediction error, as measured by MSE or maximizing $R^2$, while yielding complex expressions that do not often correspond to the systems' underlying functions.

SeTGAP is limited by the complexity of skeletons and the set of unary and binary operators used to train the Multi-Set Transformer. As shown in Appendix I, SeTGAP fails on problem F4 of the SRBench++ benchmark. The univariate skeleton of the underlying function with respect to $x_0$ is $(c_1 x_0 + \sin(x_0 + c_2))/(c_3 x_0^2 + c_4)$, which requires eight operators, including three unary operators (`sin`, `sqr`, and `inv`). This level of complex-

ity exceeds the capabilities of our approach, since the Multi-Set Transformer used in the experiments was trained only on expressions containing up to seven operators and at most two unary operators. This limitation, inherent to any neural SR method, can be addressed through transfer learning, where the model is fine-tuned on more complex tasks to expand its expressivity. Beyond structural complexity, the method's performance is also tied to the uncertainty of the underlying opaque model $\hat{f}$. In instances where the initial neural network provides a poor approximation of the target function, the Multi-Set Transformer may fail to identify the correct univariate skeletons for all variables. Another limitation is the computational cost due to multiple intermediate optimization processes, as detailed in Appendix H, making SeTGAP less efficient than all compared approaches. However, in applications like scientific discovery, where the goal is to derive interpretable and reliable mathematical expressions rather than simply optimizing predictive accuracy, the additional computational effort is warranted. In this work, we do not aim to achieve competitive computational complexity, but rather to demonstrate that a decomposable approach reconstructs the correct functional form of a system more effectively. Future work will focus on developing fast and scalable merging strategies capable of synthesizing high-dimensional expressions efficiently.

## 6   Conclusion

Symbolic regression aims to find interpretable equations that represent the relationships between input variables and their response. As such, SR represents a promising avenue for building interpretable models, a key aspect of modern machine learning. By distilling opaque regression models into transparent mathematical expressions, SR provides insight into their inner workings and enhances confidence in their reliability.

In this work, we introduced SeTGAP, a decomposable SR method that integrates transformers, genetic algorithms, and genetic programming. It first generates multiple univariate skeletons that capture the functional relationship between each variable and the system's response. These skeletons are systematically merged using evolutionary approaches, ensuring interpretability throughout the process. Experimental results demonstrated that SeTGAP consistently recovered the correct functional forms across all tested problems, unlike the compared methods. In addition, it exhibited robustness against varying noise levels. SeTGAP also demonstrated a high rate of symbolic solution recovery while maintaining competitive predictive performance on the Feynman dataset relative to several benchmark methods.

Future work will investigate how the order of skeleton merging influences the learned expressions and their performance. We also aim to extend SeTGAP beyond unary and binary operators to identify more complex functions, such as differential operators and integral transforms. This expansion would significantly enlarge the search space, increasing the challenge of exploration. To address this, we plan to refine the Multi-Set Transformer's structure, incorporating prior knowledge to better guide the search process. Finally, while effective in consistently recovering ground-truth functions in low-dimensional settings, the evolutionary merging process becomes computationally infeasible as dimensionality increases. Future work will focus on developing faster, scalable merging strategies, grounded in a formal theoretical framework, to determine when correctly inferred skeletons can be reliably combined into coherent multivariate expressions.

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

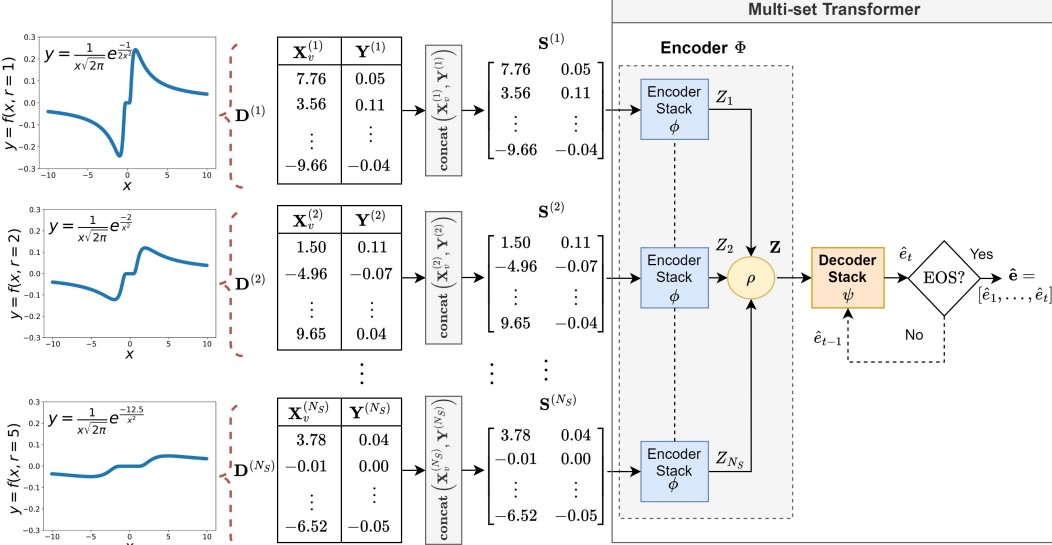

Figure 6: An example of an MSSP problem using the Multi-Set Transformer.

# A  Multi-Set Transformer

This section provides background on the Multi-Set Transformer introduced in our prior work [Ref. 1][2], summarizing its formulation and role within the symbolic skeleton prediction task.

## A.1  Multi-Set Transformer Architecture

Recall from Sec. 3.2.1 that each input set $\tilde{\mathbf{D}}_v^{(s)} = (\tilde{\mathbf{X}}_v^{(s)}, \tilde{\mathbf{y}}^{(s)})$ is defined as a set of $n$ input–response pairs. Here, we rearrange $\tilde{\mathbf{D}}_v^{(s)}$ into a two-column matrix $\mathbf{S}^{(s)}$ by concatenating $\tilde{\mathbf{X}}_v^{(s)}$ and $\tilde{\mathbf{y}}^{(s)}$ column-wise. The Multi-Set Transformer comprises two primary components: an encoder and a decoder (Fig. 6). The encoder maps the information of all input sets into a unique latent representation $\mathbf{Z}$. To do so, an encoder stack $\phi$ transforms each input set $\mathbf{S}^{(s)}$ into a latent representation $Z^{(s)} \in \mathbb{R}^d$ (where $d$ is context vector length or the "embedding size") individually. Our encoder, denoted as $\Phi$, comprises the use of the encoder stack $\phi$ to generate $N_S$ individual encodings $Z^{(1)}, \ldots, Z^{(N_S)}$, which are then aggregated into $\mathbf{Z}$:

$$\mathbf{Z} = \Phi\left(\mathbf{S}^{(1)}, \ldots, \mathbf{S}^{(N_S)}, \boldsymbol{\theta}_e\right) = \rho\left(\phi\left(\mathbf{S}^{(1)}, \boldsymbol{\theta}_e\right), \ldots, \phi\left(\mathbf{S}^{(N_S)}, \boldsymbol{\theta}_e\right)\right) = \rho\left(Z^{(1)}, \ldots, Z^{(N_s)}, \boldsymbol{\theta}_e\right), \qquad (1)$$

where $\rho(\cdot)$ is a pooling function, and $\boldsymbol{\theta}_e$ represents the trainable weights of the encoder stack. We define $\phi$ as a stack of $\ell$ ISAB blocks so that it encodes high-order interactions among the elements of an input set in a permutation-invariant way. Furthermore, unlike the Set Transformer's encoder, we include a PMA layer in $\phi$ to aggregate the features extracted by the ISAB blocks, whose dimensionality is $n \times d$, into a single $d$-dimensional latent vector. Finally, the function $\rho(\cdot)$ that is used to aggregate the latent representations $Z^{(s)}$ is implemented using an additional PMA layer.

On the other hand, the objective of the decoder, denoted as $\psi$, is to generate sequences conditioned on the representation $\mathbf{Z}$ generated by $\Phi$. This objective is aligned with that of the standard transformer decoder (Vaswani et al., 2017); thus, the same architecture is used for our Multi-Set Transformer. Specifically, $\psi$ consists of a stack of $M$ identical blocks, each of which is composed of three main layers: a multi-head self-attention layer, an encoder–decoder attention layer, and a position-wise feedforward network.

Let $\hat{\mathbf{e}} = \{\hat{e}_1, \ldots, \hat{e}_{N_{out}}\}$ denote the output sequence produced by the Multi-Set Transformer, which represents the symbolic skeleton as a sequence of indexed tokens in prefix notation. For instance, the skeleton $\frac{c}{x}e^{\frac{c}{x^2}}$ would be expressed as the sequence of tokens $\{\texttt{mul}, \texttt{div}, \texttt{c}, \texttt{x}, \texttt{exp}, \texttt{div}, \texttt{c}, \texttt{square}, \texttt{x}\}$ in prefix notation. In

Table 10: Vocabulary used to pre-train the Multi-Set Transformer.

| Token | Meaning | Index |
|-------|---------|-------|
| SOS | Start of sentence | 0 |
| EOS | End of sentence | 1 |
| c | Constant placeholder | 2 |
| x | Variable | 3 |
| abs | Absolute value | 4 |
| add | Sum | 6 |
| cos | Cosine | 9 |
| cosh | Hyperbolic cosine | 10 |
| div | Division | 11 |
| exp | Exponential | 12 |
| log | Logarithmic | 13 |
| mul | Multiplication | 14 |
| pow | Power | 15 |
| sin | Sine | 16 |
| sinh | Hyperbolic sine | 17 |
| sqrt | Square root | 18 |
| tan | Tangent | 19 |
| tanh | Hyperbolic tangent | 20 |
| -3 | Integer number | 21 |
| -2 | Integer number | 22 |
| -1 | Integer number | 23 |
| 0 | Integer number | 24 |
| 1 | Integer number | 25 |
| 2 | Integer number | 26 |
| 3 | Integer number | 27 |
| 4 | Integer number | 28 |
| 5 | Integer number | 29 |
| E | Euler's number | 30 |

addition, each token in this sequence is transformed into a numerical index according to a pre-defined vocabulary that contains all unique symbols that appear in the dataset being processed. The vocabulary used in this work is provided in Table 10. According to this, the previous sequence in prefix notation would be expressed as the following sequence of indices: $\{0, 14, 11, 2, 3, 12, 11, 2, 18, 3, 1\}$.

During inference, each element $\hat{e}_i$ ($i \in [1, N_{out}]$) is generated in an auto-regressive manner. Specifically, the decoder $\psi$ produces a probability distribution over the elements of the vocabulary as follows:

$$\sigma\left(\psi\left(\mathbf{Z}, \boldsymbol{\theta}_d | \hat{e}_1, \ldots, \hat{e}_{i-1}\right)\right) = P\left(\hat{e}_i | \hat{e}_1, \ldots, \hat{e}_{i-1}, \mathbf{Z}\right),$$

where $\boldsymbol{\theta}_d$ represents the trainable weights of the decoder stack. This distribution is obtained by applying a softmax function $\sigma$ to the decoder's output. The element $\hat{e}_i$ is thus selected from the obtained probability distribution by using a sampling decoding strategy, which samples a token from the distribution to allow diversity in the generated sequence. Hence, the generation process can be written as:

$$\hat{e}_i = \text{sample}\left(\sigma\left(\psi\left(\mathbf{Z}, \boldsymbol{\theta}_d | \hat{e}_1, \ldots, \hat{e}_{i-1}\right)\right)\right).$$

The decoder keeps generating new sequence elements until the "end-of-sentence" token (EOS) is produced ($\hat{e}_i = 1$, according to Table 10) or the maximum output length allowed, denoted as $N_{max}$, is reached.

## A.2    Multi-Set Transformer Training

The Multi-Set Transformer is trained using a large collection of synthetically generated mathematical expressions, each stored in prefix notation. To control expression complexity, we impose a maximum length of 20 elements, following recommendations by Lample & Charton (2020) and Biggio et al. (2021). The training dataset contains one million symbolic skeletons, while an additional 100,000 are used for validation. The method used to generate this large dataset of symbolic skeletons is described in detail in [Ref. 1][2]. There, we outlined how skeletons are randomly drawn from a symbolic grammar that defines a space of valid expressions. Each skeleton is designed to be syntactically valid and structurally diverse.

Algorithm 4 outlines the core training routine of the Multi-Set Transformer. The function computed by the model is denoted as $g(\cdot)$, and $\boldsymbol{\Theta}$ denotes its weights. Note that $\boldsymbol{\Theta} = [\boldsymbol{\theta}_e, \boldsymbol{\theta}_d]$ contains the weights of the

---

**Algorithm 4** Multi-Set Transformer Training

---

1: **function** TRAINMODEL($\mathbf{Q}$, $g$, $N_S$, $n$, $B$)
2:    **for** each $t \in$ range(1, maxEpochs) **do**
3:       Batches $\leftarrow$ getBatches($N_T$, $B$)
4:       **for** each batch $\in$ Batches **do**
5:          $\mathbf{E}^B, \hat{\mathbf{E}}^B \leftarrow [\,], [\,]$
6:          **for** each $j \in$ batch **do**
7:             $\mathbf{D}_j, \mathbf{e}_j =$ generateSets($\mathbf{Q}[j]$, $N_S$, $n$)
8:             $\hat{\mathbf{e}}_j =$ forward($g$, $\mathbf{D}_j$, $\mathbf{e}_j$)
9:             $\mathbf{E}^B$.append($\mathbf{e}_j$)
10:            $\hat{\mathbf{E}}^B$.append($\hat{\mathbf{e}}_j$)
11:          $L \leftarrow \mathcal{L}(\mathbf{E}^B, \hat{\mathbf{E}}^B)$
12:          update($g$, $L$)
13:    **return** $g, \Theta$

---

encoder and the decoder stacks. At each training iteration, a mini-batch of expression indices is drawn by shuffling the dataset. For each selected index $j$, the corresponding skeleton expression $\mathbf{e}_j$ is retrieved. A data collection $\mathbf{D}_j$ is then generated using the function generateSets, which produces multiple input–response sets derived from $\mathbf{e}_j$. The model processes each pair $(\mathbf{D}_j, \mathbf{e}_j)$ through the forward function to obtain a predicted skeleton $\hat{\mathbf{e}}_j$, following a "teacher forcing" strategy. The batch of predicted and ground-truth skeletons is used to compute the cross-entropy loss $\mathcal{L}(\mathbf{E}^B, \hat{\mathbf{E}}^B)$, and the model is updated via standard backpropagation and gradient descent routines encapsulated in the update function.

The function generateSets, detailed in [Ref. 1][2], plays a central role during training. Given a skeleton expression, it generates a multi-set collection of $N_S$ input–response sets, each containing $n$ samples. For each set, the skeleton is instantiated by sampling a new set of numerical constants. The resulting expression is then evaluated over a randomly sampled support vector to generate synthetic data. Repeating this process produces a collection of sets that reflect different realizations of the same symbolic structure, enabling the model to generalize across variations of functional forms that share the same skeleton.

Finally, as reported in [Ref. 1][2], we used a one-factor-at-a-time approach to tune the hyperparameters of the Multi-Set Transformer; the resulting configuration includes $N_S = 10$ input sets, each containing $n = 3000$ input–response pairs, a batch size of $B = 16$, and training performed using the Adadelta optimizer (Zeiler, 2012) with an initial learning rate of 0.0001. The architecture comprises $\ell = 3$ ISAB blocks in the encoder, $M = 5$ decoder blocks, an embedding size of $d = 512$, and $h = 8$ attention heads.

## B  Skeleton Equivalency Identification

This section lists representative cases of mathematically equivalent skeletons that our method can identify to avoid redundancy and reduce computational cost. The cases shown in Table 11 go beyond basic algebraic simplifications handled by SymPy, focusing instead on structural and parametric identities in trigonometric, hyperbolic, exponential, and logarithmic forms. This list is not exhaustive, and a more systematic approach to symbolic equivalence will be explored in future work.

At any point in SeTGAP's process, whenever a candidate skeleton matches a pattern in the first column of Table 11, it is rewritten into the corresponding form shown in the second column. If it already matches the form of the second column, it is left unchanged. After this normalization step, skeletons are compared using SymPy.equal(), so that skeletons that map to the same representation are detected as equivalent.

## C  Proof of Proposition 1

*Proof.* We prove the proposition by structural induction on the composition of operations that define $f(\mathbf{x})$. We begin with the base cases. If $f(\mathbf{x})$ is a constant, the required form is satisfied trivially by setting $c_0 = c$ with no additional terms. If $f(\mathbf{x}) = x_i$ is a single-variable function, the structure is preserved by setting $c_0 = 0$, $c_1 = 1$, and $T_{1,1}(\mathbf{x}) = x_i$.

For the inductive hypothesis, assume the decomposition holds for functions $h(\mathbf{x})$ and $u(\mathbf{x})$, which are composed of unary or binary operations. That is, each function can be written in the form $h(\mathbf{x}) =$

Table 11: List of Equivalent Skeletons

| Skeleton A | Skeleton B | Condition / mapping of constants |
|---|---|---|
| $c_1 \cos(c_2 f(x) + c_3)$ | $c_1 \sin(c_2 f(x) + c_4)$ | $c_3 = c_4 + \pi/2$ |
| $c_1 \sin(c_2 f(x)) + c_3 \cos(c_2 f(x))$ | $c_5 \sin(c_2 f(x) + c_6)$ | $c_5 = \sqrt{c_2^2 + c_3^2},\ c_6 = \mathrm{atan2}(c_3, c_1)$ |
| $c_1 \cos(c_2 f(x) + c_3) + c_4 \sin(c_2 f(x) + c_5)$ | $c_6 \sin(c_2 f(x) + c_7)$ | $c_6 = \sqrt{(c_1 \cos(c_3) + c_4 \cos(c_5))^2 + (c_1 \sin(c_3) + c_4 \sin(c_5))^2}$ $c_6 = \mathrm{atan2}(c_1 \sin(c_3) + c_4 \sin(c_5), c_1 \cos(c_3) + c_4 \cos(c_5))$ |
| $c_1 \sin(f(x)) + c_1 \sin(g(x))$ | $c_2 \sin(c_3(f(x) + g(x))) \cos(c_3(f(x) - g(x)))$ | $c_2 = 2c_1,\ c_3 = 0.5$ |
| $c_1 \sinh(f(x))$ | $c_2(\exp(f(x)) - \exp(-f(x)))$ | $c_2 = c_1/2$ |
| $c_1 \tanh(f(x))$ | $c_1 \frac{\exp(c_2 f(x)) - 1}{\exp(c_2 f(x)) + 1}$ | $c_2 = 2$ |
| $c_1 \log(f(x)^{c_2})$ | $c_3 \log(f(x))$ | $c_3 = c_1 c_2$ |
| $\log(\exp(c_1 f(x) + c_2))$ | $c_1 f(x) + c_2$ | Log-exp cancellation |
| $\log(c_1 \exp(f(x)))$ | $c_2 + f(x)$ | $c_2 = \log(c_1)$ |
| $c_1 f(c_2 x + c_3)$ | $c_1 f(c_2(x + c_4))$ | $c_4 = c_3/c_2$ |
| $\frac{c_1}{c_2 + c_3 f(x)}$ | $\frac{c_4}{1 + c_5 f(x)}$ | $c_4 = c_1/c_2,\ c_5 = c_3/c_2$ |

$c_0' + \sum_i c_i' \prod_j \nu_{i,j}'(T_{i,j}'(\mathbf{x}))$ and $u(\mathbf{x}) = c_0'' + \sum_i c_i'' \prod_j \nu_{i,j}''(T_{i,j}''(\mathbf{x}))$, with terms $T_{i,j}'$ and $T_{i,j}''$ themselves recursively decomposable in the same way.

For the inductive step, we consider the result of applying unary or binary operations to such functions. For a unary operation $f(\mathbf{x}) = \nu(h(\mathbf{x}))$, the expression satisfies the required structure by treating the composition as a single term ($i = 1$, $j = 1$) where $T_{1,1}(\mathbf{x}) = h(\mathbf{x})$, $\nu_{1,1} = \nu$, $c_0 = 0$, and $c_1 = 1$. Since $h(\mathbf{x})$ itself satisfies the recursive form by the inductive hypothesis, $f(\mathbf{x})$ does as well.

Now consider a binary operation $f(\mathbf{x}) = h(\mathbf{x}) \circ u(\mathbf{x})$, where $\circ$ is a binary operator. For the addition operation ($\circ = +$), substituting the expressions and grouping constants and summation terms yields:

$$f(\mathbf{x}) = (c_0' + c_0'') + \sum_i c_i' \prod_j \nu_{i,j}'(T_{i,j}'(\mathbf{x})) + \sum_i c_i'' \prod_j \nu_{i,j}''(T_{i,j}''(\mathbf{x})).$$

This expansion matches the desired structure $f(\mathbf{x}) = c_0 + \sum_i c_i \prod_j \nu_{i,j}(T_{i,j}(\mathbf{x}))$, considering that $c_0 = c_0' + c_0''$ and that the summation terms come directly from the sub-expressions of $h(\mathbf{x})$ and $u(\mathbf{x})$.

For the multiplication operation ($\circ = \cdot$), the product expansion gives:

$$f(\mathbf{x}) = c_0' c_0'' + c_0' \sum_i c_i'' \prod_j \nu_{i,j}''(T_{i,j}''(\mathbf{x})) + c_0'' \sum_i c_i' \prod_j \nu_{i,j}'(T_{i,j}'(\mathbf{x}))$$
$$+ \sum_{i,j} c_i' c_i'' \prod_j \nu_{i,j}'(T_{i,j}'(\mathbf{x})) \prod_j \nu_{i,j}''(T_{i,j}''(\mathbf{x})).$$

Each term fits into the structure $f(\mathbf{x}) = c_0 + \sum_i c_i \prod_j \nu_{i,j}(T_{i,j}(\mathbf{x}))$: the constant term can be expressed as $c_0 = c_0' \cdot c_0''$; the second and third terms are sums over products of unary operator applications, consistent with the required form; and the fourth term comprises products of sub-expressions from $h(\mathbf{x})$ and $u(\mathbf{x})$, which can be grouped as new terms $\prod_j \nu_{i,j}(Ti, j(\mathbf{x}))$. For other scalar-valued binary operations that are algebraically reducible, such as subtraction and division, we use the identities $h(\mathbf{x}) - u(\mathbf{x}) = h(\mathbf{x}) + (-u(\mathbf{x}))$ and $h(\mathbf{x})/u(\mathbf{x}) = h(\mathbf{x}) \cdot u(\mathbf{x})^{-1}$. Negation and inversion are unary operations, and since the unary case has been established, the result follows. Binary operations that involve non-scalar interactions (e.g., convolution or vector operations) are beyond the scope of this structural decomposition.

Thus, any function $f(\mathbf{x})$, defined by finite compositions of unary and binary operations, can always be expressed in the required form. Since the base case holds and the inductive step is proven, the proposition is established by structural induction. □

# D  Merging Skeleton Expressions: Example

In this appendix, we include an example demonstrating how all conditions in Algorithm 2 operate during a random merging process. The example considers two skeleton candidates: $e_1(x_1) =$

Table 12: Ablation of $n_B$.

| $n_B = 1$ | $n_B = 2$ | $n_B = 3$ | $n_B = 4$ |
|---|---|---|---|
| $14.02 \pm 19.64$ | $7.086 \pm 9.201$ | $1.021 \pm 1.325$ | $1.119 \pm 0.922$ |

Table 13: Ablation of $n_{\text{cand}}$

| $n_{\text{cand}} = 1$ | $n_{\text{cand}} = 2$ | $n_{\text{cand}} = 3$ | $n_{\text{cand}} = 4$ |
|---|---|---|---|
| $4.487 \pm 4.919$ | $4.448 \pm 4.354$ | $1.021 \pm 1.325$ | $1.564 \pm 1.912$ |

Table 14: Ablation of `rep`

| `rep` $= 50$ | `rep` $= 100$ | `rep` $= 150$ | `rep` $= 200$ |
|---|---|---|---|
| $8.564e+01 \pm 1.884e+02$ | $4.776e+01 \pm 1.252e+02$ | $1.021 \pm 1.325$ | $1.095 \pm 1.536$ |

Table 15: Ablation of $\text{GA}_{\text{pop}}$

| $\text{GA}_{\text{pop}} = 300$ | $\text{GA}_{\text{pop}} = 400$ | $\text{GA}_{\text{pop}} = 500$ | $\text{GA}_{\text{pop}} = 600$ |
|---|---|---|---|
| $7.238e+02 \pm 1.213e+03$ | $1.137e+02 \pm 1.294e+02$ | $1.021 \pm 1.325$ | $0.959 \pm 1.827$ |

$c_7 + c_8 x_1^3 + c_9 \sin(c_{10} x_1^2) \sin(c_{11} \sqrt{x_1} + c_{12} e^{x_1}) \tan(c_{13} x_1 \log(c_{14} x_1))$ and $e_2(x_2) = c_1 + c_2 x_2^2 + c_3 \sin(c_4 x_2) \tan(c_5 x_2 \log(c_6 x_2))$. Figure 7 presents the sequence of intermediate combinations, with key steps referencing the corresponding line numbers in Algorithm 2.

## E Ablation Studies

This section presents ablation studies aimed at assessing the effect of varying key hyperparameters on performance. Problem E5 was chosen for testing since it involves the largest number of variables among the tested problems and produced the highest extrapolation MSE values in Table 7. We adopted a one-at-a-time strategy, where a single hyperparameter was varied while the others were fixed to the default configuration: $n_B = 3$, $n_{\text{cand}} = 3$, `rep` $= 150$, and $\text{GA}_{\text{pop}} = 500$. In other words, whenever one hyperparameter was modified, all others were kept at these default values.

For $n_B$, we tested the values $[1, 2, 3, 4, 5]$; for $n_{\text{cand}}$, we tested $[1, 2, 3, 4, 5]$; for `rep`, we tested $[50, 100, 150, 200]$; and for the population size used for all GA optimizations, denoted as $\text{GA}_{\text{pop}}$, we tested $[300, 400, 500, 600]$. Each setting was evaluated over 10 independent runs, each on a dataset generated with a different random seed. Fig. 8 shows the distribution of the extrapolation MSE values across runs, while Tables 12–15 report the rounded mean and standard deviation of the extrapolation MSE. In the box plots, the central line marks the median, box edges correspond to the 25th and 75th percentiles, whiskers span the full range excluding outliers, and outliers are defined as points outside $1.5 \times \text{IQR}$ beyond the interquartile range.

As observed in Tables 12–15, extrapolation errors do not decrease substantially when using hyperparameter values larger than $n_B = 3$, $n_{\text{cand}} = 3$, `rep` $= 150$, or $\text{GA}_{\text{pop}} = 500$. Since higher values incur additional computational cost without clear performance gains, increasing them is not justified.

## F Comparison of Predicted Mathematical Expressions

This appendix presents in Tables 16–42 the complete set of mathematical expressions learned by each symbolic regression method across iterations 2 to 9 of dataset generation. Results corresponding to the first iteration were reported in Tables 3–5.

In each iteration, a new dataset was generated using a unique initialization seed, ensuring that all methods were trained and evaluated on the same data for a fair comparison. Each table reports the best expression found by each SR method, with shaded cells indicating cases where the learned expression's functional form matches the underlying ground-truth function. These results provide insight into the stability and consistency of each method in discovering accurate symbolic representations across different dataset realizations.

**Merging skeletons**

$$c_1 + c_2 x_2^2 + c_3 \sin(c_4 x_2) \tan(c_5 x_2 \log(c_6 x_2)) \qquad c_7 + c_8 x_1^3 + c_9 \sin(c_{10} x_1^2) \sin(c_{11}\sqrt{x_1} + c_{12} e^{x_1}) \tan(c_{13} x_1 \log(c_{14} x_1))$$

exShort $= [c_1, c_2 x_2^2, c_3 \sin(c_4 x_2) \tan(c_5 x_2 \log(c_6 x_2))]$     exLong $= [c_7, c_8 x_1^3, c_9 \sin(c_{10} x_1^2) \sin(c_{11}\sqrt{x_1} + c_{12} e^{x_1}) \tan(c_{13} x_1 \log(c_{14} x_1))]$

Merge sums. Line 5.

  1st summand: $c_3 \sin(c_4 x_2) \tan(c_5 x_2 \log(c_6 x_2))$

    A compatible summand from exLong was identified and randomly accepted. Recursion starts. Line 14.

**Merging skeletons**

$$c_3 \sin(c_4 x_2) \tan(c_5 x_2 \log(c_6 x_2)) \qquad c_9 \sin(c_{10} x_1^2) \sin(c_{11}\sqrt{x_1} + c_{12} e^{x_1}) \tan(c_{13} x_1 \log(c_{14} x_1))$$

exShort $= [c_3, \sin(c_4 x_2), \tan(c_5 x_2 \log(c_6 x_2))]$     exLong $= [c_9, \sin(c_{10} x_1^2), \sin(c_{11}\sqrt{x_1} + c_{12} e^{x_1}) \tan(c_{13} x_1 \log(c_{14} x_1))]$

Merge multiplications. Line 27.

  1st factor: $\tan(c_5 x_2 \log(c_6 x_2))$

    A compatible factor from exLong was identified and randomly accepted. Recursion starts. Line 36.

**Merging skeletons**

$$\tan(c_5 x_2 \log(c_6 x_2)) \qquad \tan(c_{13} x_1 \log(c_{14} x_1))$$

exShort $= [c_5 x_2 \log(c_6 x_2)]$     exLong $= [c_{13} x_1 \log(c_{14} x_1)]$

Merge compatible unary operators. Recursion starts. Line 21.

**Merging skeletons**

$$c_5 x_2 \log(c_6 x_2) \qquad c_{13} x_1 \log(c_{14} x_1)$$

exShort $= [c_5 x_2, \log(c_6 x_2)]$     exLong $= [c_{13} x_1, \log(c_{14} x_1)]$

Merge multiplications. Line 27.

  1st factor: $\log(c_6 x_2)$

    A compatible factor from exLong was identified and randomly accepted. Recursion starts. Line 36.

**Merging skeletons**

$$\log(c_6 x_2) \qquad \log(c_{14} x_1)$$

exShort $= [c_6 x_2]$     exLong $= [c_{14} x_1]$

Merge compatible unary operators. Recursion starts. Line 21.

**Merging skeletons**

$$c_6 x_2 \qquad c_{14} x_1$$

exShort $= [c_6, x_2]$     exLong $= [c_{14}, x_1]$

Merge multiplications of symbols. Line 24.

**Merged expression:** $c_{15}(c_{16} + x_1)(c_{17} + x_2)$

**Merged expression:** $\log(c_{15}(c_{16} + x_1)(c_{17} + x_2))$

  2nd factor: $c_5 x_2$

    A compatible factor from exLong was identified and randomly accepted. Recursion starts. Line 36.

**Merging skeletons**

$$c_5 x_2 \qquad c_{13} x_1$$

exShort $= [c_5 x_2]$     exLong $= [c_{13}, x_1]$

Merge multiplications of symbols. Line 24.

**Merged expression:** $c_{18}(c_{19} + x_1)(c_{20} + x_2)$

**Merged expression:** $c_{18}(c_{19} + x_1)(c_{20} + x_2) \log(c_{15}(c_{16} + x_1)(c_{17} + x_2))$

**Merged expression:** $\tan(c_{18}(c_{19} + x_1)(c_{20} + x_2) \log(c_{15}(c_{16} + x_1)(c_{17} + x_2)))$

  2nd factor: $\sin(c_4 x_2)$

    A compatible factor from exLong was identified (out of 2) and randomly accepted. Recursion starts. Line 36.

**Merging skeletons**

$$\sin(c_4 x_2) \qquad \sin(c_{11}\sqrt{x_1} + c_{12} e^{x_1})$$

exShort $= [c_4 x_2]$     exLong $= [c_{11}\sqrt{x_1} + c_{12} e^{x_1}]$

Merge compatible unary operators. Recursion starts. Line 21.

**Merging skeletons**

$$c_4 x_2 \qquad c_{11}\sqrt{x_1} + c_{12} e^{x_1}$$

exShort $= [c_4, x_2]$     exLong $= [c_{11}\sqrt{x_1}, c_{12} e^{x_1}]$

Merge expressions whose outermost operators differ. Line 40.

**Merged expression:** $c_{21} x_2 (c_{11}\sqrt{x_1} + c_{12} e^{x_1})$

**Merged expression:** $\sin(c_{21} x_2 (c_{11}\sqrt{x_1} + c_{12} e^{x_1}))$

  All factors from exShort where analyzed. There are remaining unmatched factors from exLong. Line 29.

**Merged expression:** $c_{22}(c_{23} + \sin(c_{21} x_2 (c_{11}\sqrt{x_1} + c_{12} e^{x_1}))) \dots$
$$(c_{24} + \tan(c_{18}(c_{19} + x_1)(c_{20} + x_2) \log(c_{15}(c_{16} + x_1)(c_{17} + x_2))))(c_{25} + \sin(c_{10} x_1^2))$$

2nd summand: $c_2 x_2^2$

  A compatible summand from exLong was identified and randomly accepted. Recursion starts. Line 14.

**Merging skeletons**

$$c_2 x_2^2 \qquad c_8 x_1^3$$

exShort $= [c_2, x_2^2]$     exLong $= [c_8, x_1^3]$

Merge multiplications of symbols. Line 24.

**Merged expression:** $c_{26}(c_{27} + x_2^2)(c_{28} + x_1^3)$

**Merged expression:** $c_{29} + c_{26}(c_{27} + x_2^2)(c_{28} + x_1^3) + c_{22}(c_{23} + \sin(c_{21} x_2 (c_{11}\sqrt{x_1} + c_{12} e^{x_1}))) \dots$
$$(c_{24} + \tan(c_{18}(c_{19} + x_1)(c_{20} + x_2) \log(c_{15}(c_{16} + x_1)(c_{17} + x_2))))(c_{25} + \sin(c_{10} x_1^2))$$

Figure 7: Example of a random merging process, according to Algorithm 2.

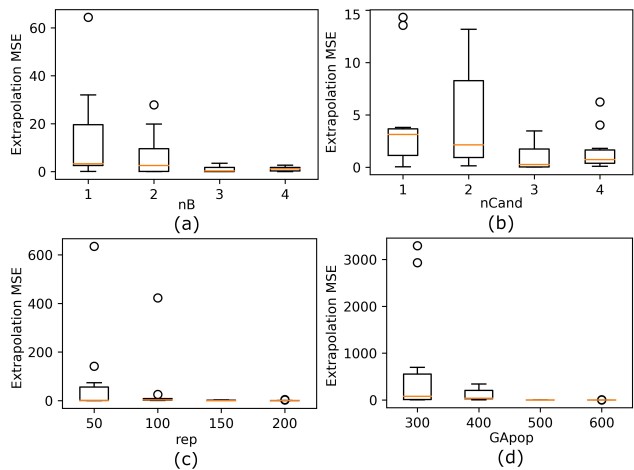

Figure 8: Ablation experiments for **(a)** $n_B$, **(b)** $n_{\text{cand}}$, **(c)** `rep`, and **(d)** GA$_{\text{pop}}$.

Table 16: Comparison of predicted equations (E1—E4) with rounded numerical coefficients — Iteration 2

| Method | E1 | E2 | E3 | E4 |
|---|---|---|---|---|
| PySR | $0.607x_0x_1$ | $0.726x_0 - 0.759x_2 + 12.944\tanh(0.107x_2) - 4.572 + 11.225e^{-0.099x_0}$ | $0.15e^{1.5x_0} + 0.5\cos(3.0x_1)$ | $0.091\cosh(x_2) + \cosh(0.064x_0^2 - 0.052x_1 - 0.052x_3 + 1.154) - 1.914$ |
| TaylorGP | $0.597x_0x_1$ | $2.95(\sqrt{e^{\sqrt{\log(2x_1)}}} + 0.206 + e^{-0.294x_0} + e^{\tanh(x_2)} + e^{\tanh(\log(x_0))} + \sin^{0.5}(1.821(( -e^{\log(x_0\tanh(x_0)+x_1)^{0.5}} - e^{-0.294x_0})^{0.5} + 0.179)^{0.5} + e^{\tanh(x_2)} + e^{\tanh(\log(x_0))} + \sqrt{\sin(x_2)} + e^{-0.294x_0}) + e^{-0.294x_0})^{0.5}$ | $-x_0 - (-x_0 - 0.868(e^{x_0})^{0.5} + e^{x_0})^{0.5} + e^{x_0} - \cos(\cos^{0.5}(\frac{x_1}{x_0})) + 0.092$ | $\sqrt{x_0}\log(x_0) + x_2 - e^{\tanh(\log(x_2))}$ |
| NeSymReS | $0.586x_0x_1 + \cos(0.593(x_0 - 0.979x_1)^2)$ | $-x_0 + 0.001e^{0.597x_1} + e^{\sin(0.175x_2)} + 7.267$ | $0.581e^{x_0} - 0.463\sin(1.104(0.003x_1 + 1)^2)$ | — |
| E2E | $(0.081x_0 - 17.975)(0.051\sqrt{0.008x_1 + 1} + 0.006)\cos(1.744(0.045(0.001 |15810201.601(0.292x_0 + 1)^3 + 0.016|+1)^2 +1)^{0.5} - 0.007) + (2.709x_0 - 0.014) (0.223x_1 + 0.002)$ | $-0.002x_1 + ((0.016x_0 + 0.004)(1.231x_0 - 9.407) + \sin(0.235x_2 - 0.031) - 0.007) (0.004x_0 + 0.087x_1 - 0.081x_2 + 3.316) + 6.19$ | $0.177\sqrt{(0.58e^{3.132x_0} + 1} + 0.486\cos(4.706(x_1 + 0.016)^2 - 0.066) - 0.104$ | $0.002|31.163x_3 + 5.524(x_0 + 0.023)^4 + 5.498(0.985x_1 - (x_2 - 0.038)^2 + 0.042)^2 - 1.464|$ |
| uDSR | $(0.001x_0^3 - 0.007x_0^2 - 0.001x_0x_1^2 +0.611x_0x_1 + 0.002x_1^2 - 0.002x_1 -0.006)\cos(e^{-5x_1^2+x_1})$ | $0.063x_0^2 - 0.5x_0 + 0.029x_1x_2 +0.001x_1 - 0.001x_1^3 +0.352x_2 + \sin(0.25x_2) + 6.499$ | $0.074x_1^4 + 0.237x_0^3 + 0.001x_0^2x_1 + 0.032x_0^2 -0.001x_0x_1 - 0.112x_0 - 0.095x_1^4 + 0.001x_1^3 +0.949x_1^2 - 0.005x_1 + e^{\cos(x_1/\log(2))} - 2.133$ | $(0.007x_0^4 - 0.013x_0^2x_1 + 0.007x_1^2 +0.007x_2^4 - 0.013x_2^2x_3 + 0.007x_3^2)e^{\sin(e)}$ |
| TPSR | $-0.071x_1(-8.596x_0 - 0.119) - 0.113\cos(x_0 - 0.782) - 0.009$ | $(-1.362 + 0.321/(-9.597e^{2.221x_0 - 6.275x_2} - 0.665)) (-0.044x_2 + 1.552 + 1/((0.008x_0 + 0.06) (0.043x_0 - 0.981) - 0.109))$ | $0.15e^{1.5x_0} + 0.5\cos(3.0x_1)$ | $0.044x_1 + 0.01(-x_2 - 0.013)^4 + 0.063(-0.355x_0^2 + 0.418x_1 + 0.209x_3 - 1)^2 - 0.126$ |
| SeTGAP | $0.61x_0x_1 + 1.041\sin((2.243x_0 - 1.512)(x_1 - 0.669) + 0.005)$ | $-0.497x_0 + 0.063x_0^2 + (0.648\sqrt{0.1x_1 + 1} + 0.002)(4.913\sin(0.205x_2) + 0.003) + 6.497$ | $0.151e^{1.499x_0} + 0.499\sin(3x_1 + 1.574)$ | $-0.02x_0^2x_1 - 0.004x_0^4 + 0.01x_0^4 + 0.01x_1^2 -0.02x_2^2x_3 + 0.01x_2^4 + 0.01x_3^4 + 0.01$ |

Table 17: Comparison of predicted equations (E5—E8) with rounded numerical coefficients — Iteration 2

| Method | E5 | E6 | E7 | E8 |
|---|---|---|---|---|
| PySR | $e^{1.2x_3}$ | $\frac{\cos(0.2x_2^2)}{\cos(1.947\tanh(\tanh(0.281x_1)))} + \tanh(x_0)$ | $-x_1^2\sinh(0.492\cos(6.274x_0 + 1.574) + 0.778) + 1.049$ | $2.036\cosh(\tanh(x_0)) + 3.34\cosh(\tanh(\tanh(x_1\tanh(x_1)))) - 5.5$ |
| TaylorGP | $x_3^{0.5}e^{x_3} + \sin^{0.5}(0.056e^{x_3})$ | $\tanh(x_0) + 0.327 + \frac{4.831\sin(x_2)}{x_2}$ | $-0.974x_1^2 + |x_1|^{0.5} + 0.008$ | $\frac{\tanh(x_1)}{\cos(\tanh(x_0))}$ |
| NeSymReS | — | $0.149x_0 + x_1\sin(0.29x_1 + \frac{x_2}{x_1})$ | $\frac{0.289x_0 + x_1^2}{\cos(2.667(0.131x_1 - 1)^2) - 2.044}$ | $\cos(\frac{\sin(\frac{x_0}{x_1})}{x_0}) + 0.629$ |
| E2E | $0.96e^{1.218x_3} - 0.004 + 0.947\sin(221.795x_1 + 27.668)$ | $(5.971|0.169x_1 - 0.005|+0.14) \cos((1.531 - 17.089x_2)(-0.012x_2 - 0.009)) + 0.8\arctan(0.695x_0 + 0.196) - 0.01$ | $(0.009\sin(7.746x_0 + 0.025) + 7.454) (0.002(0.055x_0 - 1)^3 - 0.863(x_1 - 0.001)^2 + 0.696) /(6.13\sin(7.746x_0 + 0.025) + 9.52)$ | $-1.35\arctan(-0.096|2.433x_1 + (0.328x_0 - 0.01)(-2.094x_1 + (0.008 - 1.796x_0) (0.465 - 13.81x_1)(-0.835x_1 - 0.038) +0.128) + 0.113|-0.04) - 0.035$ |
| uDSR | $-0.001x_0^2 - 0.001x_0x_1 - 0.002x_0x_3 - 0.002x_0 - 0.001x_1^2 - 0.001x_1x_2 - 0.001x_1x_3^2 - 0.002x_1x_3 - 0.002x_2x_3 + 0.004x_2 + 0.153x_3^4 + 0.57x_3^3 + 0.499x_3^2 +0.609x_3 + \sin(x_0 + x_1x_2) + 1.107$ | $-0.001x_0^3 + 0.004x_0^2 - 0.01x_0x_1 - 0.001x_0x_2 + 0.24x_0 + 0.036x_1^2 - 0.002x_1x_2 - 0.012x_1 + 0.003x_2^4 - 0.262x_2^2 + 0.004x_2 + 3\cos(x_2) - \cos(2x_2) + 2.632$ | $-0.001x_0^4 - 0.002x_0^3x_1 - 0.015x_0^3 - 0.003x_0^2x_1^2 - 0.006x_0^2x_1 + 0.039x_0^2 - 0.008x_0x_1^2 + 0.019x_0x_1 + 0.404 +0.235x_0 - 0.002x_1^4 - 0.827x_1^2 + 0.03x_1 + \cos(x_0x_1) + x_0\sin(x_0^2)\cos(x_0)$ | $0.008x_0^4 - 0.18x_0^2 - 0.004x_1^4 + 0.106x_1^2 + 0.001x_1 - e^{\cos(x_1)} - \cos(x_0) + \cos(x_1) + 2.728$ |
| TPSR | $0.003 + 0.987e^{0.001x_1x_2 + 1.206x_3} - 0.036\cos(0.188x_0 - x_1 + 1.229)$ | $2795.262 - 2795.746\sin(0.008 (-0.169x_0 + 0.025x_2 + 1)^2 + 1.529)$ | $-0.003(-0.124x_0 - 21.52)(x_1 - 0.202) (19.963 - 0.322x_1 - 0.484 \cos(6.286x_0 + 10.97) + 1)^2 - 56.092)$ | $2.008 - 1.496e^{-0.644|(0.071x_0 - 0.001)(7.973x_1 - 0.124)|}$ |
| SeTGAP | $e^{1.2x_3} - \sin(x_0 + x_1x_2 + 9.43)$ | $-(0.997|x_1|+0.008)\sin(0.201x_2^2 - 1.611) + \tanh(0.492x_0) - 0.008$ | $\frac{2.785x_1^2 + 0.05\sin(6.283x_0 - 3.149) - 2.793}{2.77\sin(6.283x_0 - 3.149) - 4.167}$ | $2 - \frac{12.162}{-0.026x_1 + 0.132x_1^2 + 0.084x_1^4 + 12.008x_1^4 + 12.144} - \frac{14.373}{14.113x_0^4 + 0.232(-x_0)^2 + 14.361}$ |

Table 18: Comparison of predicted equations (E9—E13) with rounded numerical coefficients — Iteration 2

| Method | E9 | E10 | E11 | E12 | E13 |
|---|---|---|---|---|---|
| PySR | $\tan(1.044\cos(0.627x_0)+0.242)+$ $\tan(\cos(e^{\cos(0.893x_1^{0.5})}))-2.117$ | $\sin(x_0 e^{x_1})$ | $2x_0\log(x_1^2)$ | $x_0\sin(\frac{1.0}{x_1})+1.0$ | $(x_0(\tan(\sinh(\sinh(\cosh(\tanh(0.309\cosh(x_1))))))+7.703))(x_0+8.766)$ |
| TaylorGP | $-0.906\log(|x_0|)+\log(|x_1|)\tanh(x_1)$ $-|x_0+0.119|^{0.5}$ | $\sin(x_0 e^{x_1})$ | $4x_0\log(|x_1|)$ | $0.752x_0\tanh(\frac{0.985}{x_1})+0.752$ | $2\sqrt{x_0}\log(|x_1|)$ |
| NeSymReS | $\log(|\frac{x_1}{x_0}|)-1.272$ | $\sin(x_0 e^{x_1})$ | $x_0\log(x_1^4)$ | $(x_0+x_1)\sin(\frac{1}{x_1})$ | $0.231x_0+\log(x_1^2)$ |
| E2E | $((1.9-0.6\log(13.214(x_0+0.001)^2$ $+0.6))(0.23x_1+0.231)-0.6)/$ $(0.23x_1+0.231)$ | $-0.02x_0-0.982\sin($ $(29.612x_0+0.106)$ $(10.592e^{0.571x_1}$ $-0.033)(0.017\cos($ $0.07e^{0.902x_1}-4.53)$ $+0.002))-0.003$ | $x_0(6.669\log(0.19$ $(|22.158x_1+0.255|+$ $0.111)^{0.5}-0.01)+0.1)$ | $0.997-7.62$ $\sin(\frac{(0.131x_0-0.001)(0.004x_1-6.7)}{6.661x_1+0.137})$ | $(0.091\log(0.174x_0+0.772)$ $+0.086)(27.6\log(0.711$ $(x_1+0.043)^2+0.13)$ $-0.099)$ |
| uDSR | $-0.002x_0^4-0.001x_0^3-0.052x_0^2$ $-0.001x_0 x_1+0.024x_0-0.008x_1^4$ $+0.099x_1^3-0.504x_1^2+1.45x_1$ $+e^{x_0-e^{x_0}}+e^{\cos(x_0)}-3.3$ | $\sin(x_0 e^{x_1})$ | $x_0\log(1.0x_1^4)$ | $(1.0x_0 x_1\sin(1/x_1)+x_1)/x_1$ | $\log(x_1^2)\log(0.004x_0^3+$ $0.06x_0^2+1.191x_0+$ $0.001x_1+1.42)$ |
| TPSR | $\left(-3.757-\frac{2.69}{10.828(x_0-0.002)^2+13.341}\right)$ $(-1.817\arctan(x_1+2.012)-10.316)$ $-0.045|39.196x_0+0.018|$ $(0.003x_1-0.256)|-48.763$ | $1.0\sin(0.9976x_0 e^{x_1})$ | $-1.567x_0+(0.007+1/(-1.1$ $|12.472x_1-0.03|-0.988))$ $(12.636-27.802(x_1-0.002)^2)$ $(1.515x_0+0.013)-0.05$ | $6.9697135x_1+$ $1.654205-\frac{4.6\cdot10^{-5}}{x_1}$ | $(-0.053+$ $\frac{1}{0.228|6.18x_1+0.015|+0.15})$ $(0.088-0.092x_1^2)$ $\left(1.026-\frac{1.844}{x_0+1.941}\right)$ $(-2.037x_0-40.447)$ |
| SeTGAP | $-\log(30.042x_0^2+7.509)+$ $|\log(13.19x_1+6.59)|+0.13$ | $\sin(x_0 e^{x_1})$ | $2x_0\log(x_1^2)$ | $x_0\sin(\frac{1}{x_1})+1.0$ | $2.0\sqrt{x_0}\log(|x_1|)$ |

Table 19: Comparison of predicted equations (E1—E4) with rounded numerical coefficients — Iteration 3

| Method | E1 | E2 | E3 | E4 |
|---|---|---|---|---|
| PySR | $0.607x_0 x_1+$ $1.1\cos(-2.25x_0 x_1+$ $1.5x_0+1.5x_1+0.571)$ | $e^{e^{-\sin(0.128x_0)}}+$ $1.214e^{\sqrt{e^{\tanh(x_2)}}}+\tanh(\frac{x_1}{x_2})$ | $0.15e^{1.499x_0}+$ $0.497\cos(3.001x_1+\frac{1.557}{x_0+22.291})$ | $-0.02x_2^2(\frac{2.628x_1\sin(x_2)}{x_2}+x_3)-$ $0.007(x_1-13.401)(\cosh(x_0)+\cosh(x_2))$ |
| TaylorGP | $0.6x_0 x_1$ | $-0.503x_0+0.001x_1+0.211x_2+8.607$ | $0.346e^{x_0}\log(1.348(-0.247x_0+e^{x_0}$ $-0.247e^{0.5|\sin(x_1)|^{0.5}})^{0.5})+0.107$ | $x_0+x_2-|x_0|^{\frac{1}{4}}|x_2|^{\frac{1}{4}}+0.778$ |
| NeSymReS | $0.586x_0 x_1+$ $\cos(0.585(0.991x_0-x_1)^2)$ | $-x_0+0.384e^{0.024x_1}+e^{\sin(0.175x_2)}+6.813$ | $0.582e^{x_0}-$ $0.749\sin(0.57(1-0.001x_1)^2)$ | — |
| E2E | $0.255x_0(2.399x_1+0.034)+$ $1.11\cos(3089.014x_0+$ $1035.398)-0.006$ | $(0.776-0.111x_0)(0.171x_1+(0.174x_0+0.019)$ $(0.001x_2-0.07)(-0.499x_0+$ $0.042\cos(1.689x_1+50.388)$ $+42.271)+0.017)-2.94\cos((2.184x_1+86.243)$ $(0.025\sin(0.116x_2-0.038)+0.016))+6.62$ | $0.113e^{1.594x_0}+0.511\cos(3.08$ $(x_1+0.032)^2+0.008)+0.136$ | $0.0$ |
| uDSR | $(0.001x_0^3 x_1+0.002x_0^3+2.455x_0^2 x_1$ $+0.002x_0^2+0.002x_0 x_1^3-0.002x_0 x_1^2$ $-1.229x_0 x_1-0.045x_0-0.004x_1^3$ $-0.007x_1^2+0.103x_1+0.104)$ $\sin\left(\frac{x_0}{2x_0+(2x_0^2-2x_0)/x_0}\right)/x_0$ | $\log(662.215e^{0.062x_0^2+0.001x_0 x_2}$ $e^{-0.503x_0+0.029x_1 x_2-0.002x_2^3+0.541x_2}+\cos(x_1/x_2))$ | $0.075x_0^4+0.001x_0^3 x_1+0.235x_0^3+$ $0.023x_0^2-0.003x_0 x_1-0.108x_0$ $+0.014x_1^4-0.228x_1^2-$ $\sin(x_1)\sin(2x_1)+0.706$ | $0.01x_0^4-0.02x_0^2 x_1+0.01x_1^2+0.01x_2^4$ $-0.02x_2^2 x_3+0.01x_3^2-0.526+\sin(e^e)$ |
| TPSR | $-0.141x_1(0.053-4.272x_0)+$ $0.002\sin(6.904x_1-7.997)$ $+0.05$ | $(-0.08+\frac{1}{-468.708e^{3.582x_0-9.1x_2-11.692}})$ $(-0.03x_1-7.596)((-0.427x_0-0.022x_2-$ $(-0.006x_0-0.303)(1.156x_0+2.022)-0.294)^2$ $+1.84)(-0.008x_1+1.435+\frac{1}{0.002x_0+0.394})$ | $0.15e^{1.5x_0}+0.5\cos(3.0x_1)$ | $0.011\left(0.705-(-x_0-0.011)^2\right)^2+$ $0.049(-0.003x_0-0.243x_1$ $+0.41x_2^2-0.407x_3+1)^2-0.065$ |
| SeTGAP | $0.607x_0 x_1+$ $1.123\sin((2.243x_0-1.496)$ $(x_1-0.654)+0.014)$ | $-0.5x_0+0.062x_0^2+$ $(3.163\sqrt{0.1x_1+1}+0.001)\sin(0.2x_2)+6.513$ | $0.151e^{1.497x_0}-$ $0.5\sin(3.004x_1-7.854)$ | $-0.004x_0-0.02x_0^2 x_1+$ $0.017(-x_0)^2+0.009(-x_0)^4+0.008(-x_1)^2+$ $0.01(-x_2)^4+0.011(-x_3)^2-(0.013x_3^2+0.003)$ $(1.575x_3+0.614)-0.024$ |

Table 20: Comparison of predicted equations (E5—E8) with rounded numerical coefficients — Iteration 3

| Method | E5 | E6 | E7 | E8 |
|---|---|---|---|---|
| PySR | $e^{1.2x_3}+\sin(x_0+x_1 x_2)$ | $|x_1|\cos(0.2x_2^2)+\tanh(0.502x_0)$ | $-0.412x_1^2\cosh(1.055e^{-0.743\sin(6.278x_0)})+1$ | $2.046\cosh(\tanh(x_0))+$ $2.046\tanh(\frac{x_1}{\tanh(\sinh(1.875x_1))})$ $-3.196$ |
| TaylorGP | $2e^{x_3}-\log(3e^{x_3}+0.856\sqrt{e^{(x_3)}|x_3|}-$ $\log(2.32e^{x_3}+0.856\sqrt{e^{(x_3)}|x_3|})-0.26)-0.26$ | $\frac{x_1\sin(x_2)}{x_2\tanh(x_1)}+\cos(x_2)+\tanh(x_0)$ | $-0.974x_1^2+\sqrt{|x_1|}$ | $1.148|x_0|^{\frac{1}{4}}|x_1|^{\frac{1}{4}}-0.105$ |
| NeSymReS | — | $0.144x_0+x_1\sin(0.065x_1+\frac{x_2}{x_1})$ | $\frac{0.072x_0+x_1^2}{\cos(3.099(0.012x_1-1)^2)-0.252}$ | $\cos\left(\frac{\sin(\frac{x_0}{x_1})}{x_0}\right)+0.69$ |
| E2E | $0.871e^{1.253x_3}+0.981\cos(88.561(0.014|3.137x_1+$ $31752866.296(0.104\arctan(0.087x_2+$ $5.508)-1)^2-2.465|-1)^{0.5}-0.006)+0.13$ | $(0.002\sin(0.816x_1+0.616)+$ $0.151)(0.036x_1+\cos(0.711(0.014$ $-x_2)^2-0.573)+3.995)$ | $(0.002(1-0.256\sin(0.183(-32.468x_0+$ $0.047x_1+14.284|-0.293))^3$ $-0.026)(-0.043x_0+$ $19.294(x_1+0.017)^2-0.543)$ | $0.865$ |
| uDSR | $-0.001x_0^2+0.001x_0 x_2-0.001x_0 x_3+0.002x_0-$ $0.001x_1^2 x_3+0.002x_1^2-0.001x_1 x_3-0.004x_1+$ $0.003x_2^2+0.005x_2 x_3-0.002x_2+0.153x_3^4+1.084$ $+0.567x_3^3+0.508x_3^2+0.619x_3+\sin(x_0+x_1 x_2)$ | $-0.002x_0^3+0.001x_0^2-0.006x_0 x_1+$ $0.004x_0 x_2+0.26x_0+0.032x_1^2+$ $0.003x_1 x_2-0.003x_1+0.003x_2^2-$ $0.266x_2^2+0.022x_2+3\cos(x_2)-$ $\cos(2x_2)+2.831$ | $0.001x_0^4+0.001x_0^3 x_1-0.014x_0^3-0.002x_0^2 x_1$ $-0.015x_0^2-0.002x_0 x_1^3-0.009x_0 x_1^2+0.002x_0 x_1$ $+0.231x_0+0.001x_1^4+0.002x_1^3-0.913x_1^2$ $-0.004x_1+(x_0\cos(x_0)-\sin(x_0))\sin(x_0^2)+0.933$ | $0.008x_0^4-0.18x_0^2+0.001x_0 x_2+$ $0.001x_0-0.004x_1^4+0.108x_1^2+$ $0.002x_1-e^{\cos(x_1)}-\cos(x_0)+$ $\cos(x_1)+2.722$ |
| TPSR | $(x_1+923.05)(0.001e^{1.2x_3}-0.01)+8.903$ | $(0.016\cos(0.771(x_1+0.145)^2$ $-0.875)-0.084)(-2.126x_0-$ $43.836\cos((3.209-\frac{8.357}{0.028x_2+0.476})$ $(0.058x_2+0.044))-17.68)$ | $-0.039x_0+294.76(-1+0.139/$ $(0.036-3.266e^{-7}/(-5.83e^{-9}$ $(-x_1-0.092)^2-6.046e^{-7})))^2-167.907$ | $(-0.239x_1-4.712)$ $(0.455x_1-8.997)$ $(\arctan(4.665(0.02-x_1)^2$ $(x_0-0.039)^2+31.821)-1.521)$ |
| SeTGAP | $1.006e^{1.201x_3}+1.006\sin(1.008x_0+$ $(1.005x_1-0.006)(x_2+0.029)+12.451)-0.007$ | $1.001\cos(0.2x_2^2+0.01)$ $|x_1|+1.0\tanh(0.5x_0)+0.002$ | $\frac{5.18x_1^2-5.18}{5.18\sin(6.283x_0+3.142)-7.77}$ | $2.0-\frac{14.485}{0.142x_0^2+14.37(-x_0)^4+14.466+}$ $\frac{13.074}{-13.081x_1^4-13.074}$ |

Table 21: Comparison of predicted equations (E9—E13) with rounded numerical coefficients — Iteration 3

| Method | E9 | E10 | E11 | E12 | E13 |
|---|---|---|---|---|---|
| PySR | $1.172\sqrt{|x_1|} - 1.518\sinh(\sinh(\tan(\tanh($ $1.739e^{\cos(\cos(0.217x_0)} + \cos(\tanh(x_0))))))) + 2.232$ | $\sin(x_0 e^{x_1})$ | $2x_0\log(x_1^2)$ | $x_0\sin\left(\frac{1.0}{x_1}\right) + 1$ | $\sqrt{x_0}\log(x_1^2)$ |
| TaylorGP | $-\log(|x_0|) + \tanh(\log(|x_1|)) - \sqrt{|\log(\frac{0.568}{|x_0|})|}$ | $\sin(x_0 e^{x_1})$ | $4.545x_0\log(|x_1|)$ | $\frac{2.77\log\left(\sqrt{|x_0+x_1|}\right)}{\tanh\left(\frac{0.5(x_0+x_1)}{x_1}\right)}$ | $2\log(|x_1|)\sqrt{|x_0|}$ |
| NeSymReS | $\log\left(\frac{0.286|x_1|}{|x_0|}\right)$ | $\sin(x_0 e^{x_1})$ | $x_0\log(x_1^4)$ | $(x_0+x_1)\sin\left(\frac{1}{x_1}\right)$ | $0.241x_0 + \log(x_1^2)$ |
| E2E | $-0.098 - 0.052/((-0.003(0.029x_1 - 1)^3 + 0.005 + 6.94/(-0.001x_0 - 51.126x_1 + (3.21 -0.085|32.081x_0 - 1.081| )(2.068x_1 - 69.713) + 126.399)))$ | $1.0\sin(0.035 x_0(26.26e^{1.16x_1} -0.1)) - 0.001$ | $(0.058|15.4\arctan( 0.258x_1 + 0.008) - 0.009| -0.005)(-0.006x_0 + 59.4 \sin(0.168x_0 - 0.001) + 0.011)$ | $1.0 + 7.5\sin(0.001x_0 (0.1 - \frac{24.6}{-0.159x_1 - 0.005}))$ | $1.08\log(0.02(-x_0 - 0.825)^2 (-x_1 - 0.057)^2 (-x_1 - 0.056)^2 + 0.101) + 0.09$ |
| uDSR | $\log\left(\frac{x_1(2x_1^2 + x_1)}{4.0x_0^2 x_1^2 + 1.0x_1^2}\right)$ | $1.0\sin(x_0 e^{x_1})$ | $x_0\log(1.0x_1^4)$ | $x_0\sin(1/x_1) + 1.0$ | $\frac{\log(x_1^2)\log(0.004x_0^3 + 0.061x_0^2 + 1.182x_0}{-0.001x_1 + 1.433)}$ |
| TPSR | $0.396x_1 + 0.004\left(1.464 + \frac{1}{19.402(0.002 - x_0)^2 + 28.549}\right) (0.873x_0 - 46.945)(-10.284x_0 + x_1 - 555.305) -149.911$ | $1.0\sin(1.001x_0 e^{x_1})$ | $(0.001 - 0.12x_0)(-0.971x_1^2 - 42.941 - 103.498(-0.605 |3.087x_1 - 0.011| - 0.633))$ | $(-1.292 - 0.14/(-0.179 (0.01x_0 + x_1 + 0.073)^2 - 0.106)(-0.005x_1 + \frac{0.195 - 0.883x_0}{x_1 + 0.066} - 0.809)$ | $(-0.104 + 1/(0.01|82.844x_1 -0.068| + 0.111)(49.998 - 50.042\cos(0.042x_1)) (159.942 + \frac{170.493}{-0.07x_0 - 1.16})$ |
| SeTGAP | $-0.999\log(11.668x_0^2 + 2.91) + 1.006|\log(13.4x_1 + 6.834)| - 0.861$ | $1.0\sin(1.0x_0 e^{x_1})$ | $2.0x_0\log(x_1^2)$ | $1.0x_0\sin\left(\frac{1}{x_1}\right) + 1.0$ | $\sqrt{1.0x_0}\log(x_1^2)$ |

Table 22: Comparison of predicted equations (E1—E4) with rounded numerical coefficients — Iteration 4

| Method | E1 | E2 | E3 | E4 |
|---|---|---|---|---|
| PySR | $0.608x_0 x_1 + 1.1\cos(1.076(1.045x_0 -0.697)(2x_1 -1.333) - 1.571)$ | $(0.198x_1 + 2.985)\sin(0.201x_2) + \cosh(0.836e^{\tanh(2.02e^{0.162x_0})} - 4.76)$ | $13.457e^{\frac{1.753x_0 - 5.251}{\cosh(\tanh(\cos(1.525x_1)))}}$ | $0.091x_1(\cos(0.643x_0) - 1.803) - 0.044x_3 \cosh(0.65x_2) + 0.091\cosh(x_0) +0.091\cosh(x_2) - 0.027$ |
| TaylorGP | $0.602x_0 x_1$ | $1.946\sqrt{x_0(\tanh(x_0) - 0.837)} + 1.946(\log(e^{x_2} + 11.111\sin(\cos(e^{x_0})) + 11.111\sqrt{|x_0(x_2 + 3.147\sqrt{|x_0|} + 0.984)|}))$ | $\frac{-x_0 + e^{x_0} -}{\sqrt{|x_0 - e^{x_0} - \sqrt{|x_0|} + 0.785\sqrt{|e^{x_0} - 0.083|}}}$ | $\frac{x_2}{\tanh(x_2)} + \log(|x_0|)\sqrt{|x_0|} - 1.571$ |
| NeSymReS | $0.586x_0 x_1 + \cos(0.586(x_0 -0.988x_1)^2)$ | $-x_0 + e^{\sin(x_2)} + 7.289$ | $0.584e^{x_0} - 0.411$ | — |
| E2E | $0.03x_0(19.977x_1 + 0.327) - 1.1\cos(31.768 x_1 + 2.118) + 0.016$ | $0.075x_0^2 - 0.443x_0 + 0.014x_1 x_2 + 0.002x_1 - 0.002x_2^2 + 0.083x_2 + 3.07\sin(0.227x_2 + 0.025) + 6.23$ | $0.107x_0 + 0.156(0.318x_0 + 1)^2(x_0 - 0.002)^2 (|0.279x_0 + 0.578| + 0.133)^2 - 0.501\sin(0.091x_0 + 3.572x_1 - 1.273) + 0.152$ | $(0.095 - 0.001x_1)(0.1(0.559x_1 - (x_0 - 0.003)^2 + 0.037)^2 + 0.073) + 0.036 (-0.524x_3 + (0.204x_0 - 0.001)(0.142x_2 +0.013) + (0.017x_2 + 0.001) (32.565x_2 + 1.169) + 0.024)^2 - 0.001$ |
| uDSR | $(0.224x_0 x_1 + 0.001x_0 - 0.003x_1 -0.004)e^{\cos(e^{x_0(1-3x_0)})}$ | $\log(663.667e^{0.062x_0^2 - 0.001x_0 x_1} e^{0.001x_0 x_2 - 0.502x_0 + 0.03x_1 x_2} e^{-0.004x_1 - 0.002x_2^3 + 0.541x_2} + 1)$ | $0.013x_0^4 - 0.032x_0^3 - 0.416x_0^2 - 0.002x_0 x_1 -0.901x_0 - 0.013x_1^4 + 0.225x_1^2 -0.001x_1 + e^{x_0} + \cos(x_1)\cos(2x_1) - 1.295$ | $0.01x_0^4 - 0.02x_0^2 x_1 + 0.01x_1^2 + 0.01x_2^4 - 0.02x_2^2 x_3 + 0.01x_3^2$ |
| TPSR | $-0.069x_0(-8.798x_1 - 0.074) + 0.036\sin(5.606 x_1 + 2.258) + 0.009$ | $\frac{0.001(31.887x_1 + x_2 + 12721.866)}{e^{(-1.482(0.06x_0 + 0.651))} e^{\frac{78.544(0.06x_0 + 0.651)}{-41.665e^{0.01x_1 + 0.693x_2} - 29.007}}}$ | $0.15e^{1.5x_0} + 0.5\cos(3.0x_1)$ | $11.1441\left(1 - 0.944(-0.074x_2 - 1)^2\right)^2 -0.214569$ |
| SeTGAP | $0.607x_0 x_1 + 1.1\sin((2.25x_0 - 1.489) (x_1 - 0.663))$ | $-0.5x_0 + 0.063x_0^2 + (3.148\sqrt{(0.1x_1 + 1)} + 0.012) \sin(0.204x_2) + 6.493$ | $0.147e^{1.508x_0} + 0.5\sin(2.999x_1 + 1.571) + 0.002$ | $-0.02x_0^2 x_1 + 0.011x_0^4 + 0.009x_1^2 + 0.011x_2^2 - 0.013(-x_0)^2 + 0.009(-x_2)^4 - (0.035x_2^2 - 0.002)(0.584x_3 - 0.46) - 0.016$ |

Table 23: Comparison of predicted equations (E5—E8) with rounded numerical coefficients — Iteration 4

| Method | E5 | E6 | E7 | E8 |
|---|---|---|---|---|
| PySR | $e^{1.2x_3} + \sin(x_0 + x_1 x_2)$ | $\sqrt{(x_1^2)}\cos(0.2x_2^2) + \tanh(0.5x_0)$ | $\frac{-1.0x_1^2 + \cos(0.011x_1\sin(6.283x_0))}{\sin(6.283x_0) + 1.5}$ | $4.404\tanh(\cosh(0.869x_0) \sqrt{\tanh(\cosh(x_1))}) \tanh(\cosh(0.869x_1)) - 2.427$ |
| TaylorGP | $e^{x_3}\sqrt{|x_3|} + \tanh(0.515e^{x_3} - 0.331\sin(\sqrt{e^{x_3}|x_3 - 0.662|}))$ | $\left(-\log(|x_2|) + |x_1|^{0.5}\right)(\cos(x_2) + 0.604) + \cos(x_2) + \tanh(0.142x_0)$ | $-0.974x_1^2 + |x_1|^{0.5}$ | $1.108|x_0|^{\frac{1}{4}}|x_1|^{\frac{1}{4}}$ |
| NeSymReS | — | $0.173x_0 + x_1\sin\left(0.284x_1 + \frac{x_2}{x_1}\right)$ | $\frac{0.059x_0 + x_1^2}{\cos(1.577(-0.668x_1 - 1)^3) - 1.862}$ | $\cos\left(\frac{\sin\left(\frac{x_0}{x_1}\right)}{x_0}\right) + 0.644$ |
| E2E | $0.96e^{1.21x_3} - 0.963\sin(-0.001x_0 \left(0.03 + \frac{90.6}{0.216x_1 - 0.235}\right) + 0.555x_1 + 0.03x_2 + (0.737 - 6.778x_0) (0.007x_1 - 1.38) + 0.898) + 0.055$ | $0.011x_1 - (1.57 - 0.001x_2)\arctan(-0.155x_0 + (0.055\sin(0.173x_1 + 0.031) - 0.005) (53.8\cos(0.171x_1 - 0.172(x_2 + 0.075)^2 + 0.011) + 0.006) + 0.004) - 0.006$ | $\left(11.393(x_1 - 0.128)^2 + 11.849\right) (-0.02(-0.021\cos(1254.632 (1 - 0.528x_0)^2 + 0.173) - 1)^2 -0.022)$ | $2.0 - 0.9(0.006|(3.121x_1 +0.11)(24.018(x_0 - 0.003)^2 + 6.0)| + 0.5)$ |
| uDSR | $-0.001x_0 x_3 + 0.001x_0 + 0.001x_1^2 x_3^2 - 0.002x_1^2 + 0.001x_1 x_2 + 0.003x_1 x_3 + 0.001x_1 + 0.001x_2 + 0.153x_3^4 + 0.569x_3^3 + 0.497x_3^2 + 0.599x_3 + \sin(x_0 + x_1 x_2) + 1.103$ | $-0.002x_0^3 - 0.007x_0 x_1 + 0.006x_0 x_2 + 0.26x_0 +0.039x_1^2 - 0.001x_1 x_2 - 0.012x_1 + 0.003x_2^4 -0.262x_2^2 - 0.035x_2 + 3\cos(x_2) - \cos(2x_2) + 2.743$ | $0.002x_0^4 + 0.001x_0^3 x_1 - 0.029x_0^3 - 0.002x_0^2 x_1^2 + 0.002x_0^2 x_1 - 0.024x_0^2 - 0.009x_0 x_1^2 - 0.008x_0 x_1 - 0.55x_0 - 0.002x_1^4 - 0.003x_1^3 - 0.865x_1^2 + \log(e^{-x_0(\sin(x_0(x_0 + 1))\cos(x_0) - 1)}) +0.01x_1 + 0.951$ | $-0.004x_0^4 + 0.107x_0^2 - 0.001x_0 x_2 +0.008x_1^4 - 0.18x_1^2 - 0.002x_1 -e^{\cos(x_0)} + \cos(x_0) - \cos(x_1) +2.729$ |
| TPSR | $0.988e^{1.203x_3} + 0.055\sin(x_0 -8.658x_1 + 1.093) + 0.029$ | $(0.001 + \frac{1}{x_0 - 411.157})(31.572 - 0.001x_2) (0.214\cos(2.182x_1 + 2.001) - 5.403) ((4.295 - 1.096x_0)(-0.044x_1 - 0.586) +14.767\cos(0.533x_2 - 0.05) + 9.177)$ | $-0.026x_0 - 1.254 + \frac{0.067 - 0.027(-x_1 - 0.023)^2}{0.006\sin(7.183x_0 + 2.841) + 0.03}$ | $2.01 - 1.478 e^{-0.00384|x_1(90.031x_0 + 2.401)|}$ |
| SeTGAP | $0.997e^{1.201x_3} + 0.999\sin(x_0 + 1.0x_1 x_2 + 0.017) + 0.004$ | $0.997\cos(0.202x_2^2 + 6.236)|x_1| + 1.0\tanh(0.5x_0)$ | $\frac{5.94x_1^2 - 5.979}{-5.941\sin(6.283x_0 - 6.283) - 8.91} -0.004$ | $2.0 - \frac{15.123}{15.039x_1^4 + 15.129} - \frac{15.086}{15.06x_0^4 + 15.09}$ |

Table 24: Comparison of predicted equations (E9—E13) with rounded numerical coefficients — Iteration 4

| Method | E9 | E10 | E11 | E12 | E13 |
|---|---|---|---|---|---|
| PySR | $1.141\sqrt{x_1} + 1.141(-0.895x_0 + (0.059x_0^2 - 1.08)\tanh(3.763x_0))\tanh(x_0)$ | $\sin(x_0 e^{x_1}\tanh(e^{e^{\tanh(\cosh(\tan(e^{x_1}+0.775)))}}))$ | $2x_0\log(x_1^2)$ | $x_0\cos\left(1.659 - \frac{0.088x_0x_1+x_0}{x_0x_1}\right) + 1.0$ | $\sqrt{x_0}\log(x_1^2)$ |
| TaylorGP | $0.26x_1 + \log\left(\frac{0.984}{|x_0|}\right) - \sqrt{\left|\log\left(\frac{0.593}{|x_0|}\right) + \log\left(\left|\tanh\left(\frac{x_1}{x_0}\right)\right|\right)\right|}$ | $\sin(x_0 e^{x_1})$ | $4x_0\log(|x_1|)$ | $\left(x_0\tanh\left(\frac{0.664}{x_1}\right) + 1\right)\sqrt{|\tanh(x_1)|}$ | $2\log(|x_1|)\sqrt{|x_0|}$ |
| NeSymReS | $\log\left(\frac{0.282|x_1|}{|x_0|}\right)$ | $\sin(x_0 e^{x_1})$ | $x_0\log(x_1^4)$ | $(x_0 + x_1)\sin\left(\frac{1}{x_1}\right)$ | $0.234x_0 + \log(x_1^2)$ |
| E2E | $-0.6\log(10.525(0.003 - x_0)^2 + 0.6) + 1.9 - \frac{0.6}{0.229x_1+0.235}$ | $0.001 - 1.0\sin(0.561x_0(-0.002 + 17.5/((-88.065 - \frac{0.009}{7.629x_0+19.891})(0.001 - 0.025x_1) + 0.364(0.008 x_1 - 1)^2 - 7.514(0.732x_1 - 1)^2 - 5.172)))$ | $-2.266x_0(0.018 - 0.26x_0(13.784(0.002 - (x_1 - 0.024)^2)^2 + 0.008))$ | $0.998 - 6.35\sin((0.006 + \frac{0.673}{0.173x_1+0.007})(-0.04x_0 - 0.002))$ | $(5.75 - 8.24/(1.265(|x_1|+0.008)^{0.5} + 0.135))(2.72\log(0.174x_0 + 0.991) + 0.188)$ |
| uDSR | $0.001x_0^4 - 0.095x_0^2 + 0.001x_0x_1^2 - 0.002x_0x_1 + 0.002x_0 - 0.007x_1^4 + 0.097x_1^3 - 0.497x_1^2 + 1.442x_1 - 2.803 + e^{\cos(x_0)}\sin(x_0)/x_0$ | $\sin(x_0 e^{x_1})$ | $x_0\log(7.389x_1^4) - 2x_0$ | $x_0\sin(1/x_1) + 1.0$ | $0.513e^{0.005x_0^3 - 0.076x_0^2}e^{-0.002x_0x_1}e^{0.588x_0+0.006x_1}\log(x_1^2)$ |
| TPSR | $-0.001x_0 + 1.386x_1 + 23.694\arctan((0.005x_1 + 0.059)(0.155x_0^2 + 37.746(1 - 0.074x_1)^2 - 1.731|x_0| + 25.317) - 4.298) + 12.633$ | $0.9998\sin(0.9994x_0e^{x_1}) - 0.000683$ | $7.994x_0 + 0.052$ | $(0.429 - 0.001x_0)(\arctan(224.422((x_1 + 0.121)^3 - 0.052)^2 + 2.305) - 1.24)(-0.001x_0(-0.049 - \frac{6021.0}{x_1}) + 0.081x_0 + 7.034)$ | $(0.064x_0 + 0.751 - 119.56/((-41.662 - 117.245e^{-0.175x_0})(-0.108|x_1| - 0.014)))(0.313x_1 + 0.362(0.706 - 0.561x_1)(x_1 + 2.798) - 0.513)$ |
| SeTGAP | $-1.0\log(14.156x_0^2 + 3.539) + 1.0|\log(9.499x_1 + 4.75)| - 0.294$ | $1.0\sin(1.0x_0e^{x_1})$ | $2.0x_0\log(x_1^2)$ | $1.0x_0\sin\left(\frac{1}{x_1}\right) + 1.0$ | $\sqrt{1.0x_0}\log(x_1^2)$ |

Table 25: Comparison of predicted equations (E1—E4) with rounded numerical coefficients — Iteration 5

| Method | E1 | E2 | E3 | E4 |
|---|---|---|---|---|
| PySR | $0.608x_0x_1 + 1.1\sin(2.25x_0(x_1 - 0.667) - 1.5x_1 + 1.0)$ | $-0.5x_0 + (0.179x_1 + 2.984)\sin(0.2x_2) + e^{e^{\cos(0.111x_0)}} + 1.041$ | $0.15e^{1.5x_0} + 0.5\cos(3.0x_1)$ | $(\cosh(x_0) + \cosh(x_2))(-0.005x_1 - 0.005x_3 + 0.091)$ |
| TaylorGP | $0.616x_0x_1$ | $-0.494x_0 - 0.005x_1 + 0.204x_2 + 8.593$ | $\frac{-x_0 + e^{x_0} - }{\sqrt{\left|x_0 + 0.838\sqrt{e^{x_0}} - e^{x_0} - \sqrt{|x_0|}\right|}}$ | $\frac{x_2}{\tanh(x_2)} + \log\left(\left|\frac{x_0}{\tanh(x_0)}\right|\right)\sqrt{|x_0|} - 1.797$ |
| NeSymReS | $0.59x_0x_1 + \cos(0.584(0.989 x_0 - x_1)^2)$ | $-x_0 + e^{\sin(x_2)} + 7.456 - 0.05e^{-0.102x_1}$ | $0.585e^{x_0} - 0.437\sin\left(1.612(0.04x_1 + 1)^2\right)$ | — |
| E2E | $0.021x_0(29.332x_1 - 0.194) + 1.1\cos(30.949 x_1 - 0.049) + 0.01$ | $-0.016x_0 + 0.049x_2 + (0.086x_0 - 0.704)(0.75x_0 - 0.015) + (0.003x_1 + 0.052)(-0.97x_1 + 53.2\sin(0.221x_2 + 0.035) + 0.064) + 6.444$ | $0.224e^{0.188x_0}e^{(0.004x_0+0.045)(22.712x_0+0.009x_1-5.15)} + 0.514\cos(3.152x_1 + 0.082) + 0.043$ | $0.002|30.73x_3 + 5.334(x_0 + 0.019)^4 + 6.811(x_1 - 0.94(x_2 - 0.021)^2 + 0.028)^2 + 1.185|$ |
| uDSR | $(0.001x_0^3 - 0.001x_0^2x_1^2 - 0.008x_0^2x_1 + 0.025x_0^2 - 1.824x_0x_1^2 + 0.608x_0x_1 - 0.024x_0 - 0.006x_1^3 + 0.022x_1^2 + 0.145x_1 - 0.225)\sin\left(\frac{1}{1-3x_1}\right)$ | $\log(666.967e^{0.063x_0^2 - 0.503x_0}e^{0.03x_1x_2 - 0.001x_1}e^{-0.002x_2^3+0.541x_2} + 1)$ | $0.075x_0^4 + 0.532x_0^3 + 0.001x_0^2x_1^2 + 0.023x_0^2 + 0.002x_0x_1^2 - 0.002x_0x_1 - 2.746x_0 - 0.129x_1^4 + 0.001x_1^3 + 0.97x_1^2 - 0.003x_1 + e^{\cos(2x_1)} + 3\sin(x_0) - 1.783$ | $0.01x_0^4 - 0.02x_0^2x_1 + 0.01x_1^2 + 0.01x_2^4 - 0.02x_2^2x_3 + 0.01x_3^2$ |
| TPSR | $-0.6\log(12.904(x_0 + 0.008)^2 + 0.6) + 1.9 - \frac{0.6}{0.227x_1+0.236}$ | $(0.372x_0 - 0.368x_2 + 123.99)((0.001x_0 + 0.127)(x_0 - 0.045x_2 - 254.229) + 0.086\arctan(0.002x_2(2.01x_1 + 71.9)) + 32.38)$ | $0.052e^{2.5x_0} + 0.5\cos(4.0x_1 - 0.03) + 0.09$ | $-0.13x_1 + 0.032x_3 + 0.081(0.067x_1 + 0.005x_2 + 0.363x_3 - (1 - 0.032 |7.087x_0 + 28.484|)^2 - 0.334 (x_2 + 0.005)^2 - 0.028)^2 + 0.319$ |
| SeTGAP | $0.607x_0x_1 - 1.1\sin((2.252x_0 - 1.492) (x_1 - 0.666) + 9.432)$ | $-0.5x_0 + 0.063x_0^2 + 3.16\sqrt{0.1x_1 + 1}\sin(0.202x_2) + 6.499$ | $0.15e^{1.5x_0} + 0.5\sin(3.0x_1 + 1.571)$ | $-0.02x_0^2x_1 + 0.001x_0^2 + 0.01x_1^4 + 0.01x_1^2 - 0.02x_2^2x_3 + 0.01x_2^4 + 0.01(-x_3)^2 - 0.001$ |

Table 26: Comparison of predicted equations (E5—E8) with rounded numerical coefficients — Iteration 5

| Method | E5 | E6 | E7 | E8 |
|---|---|---|---|---|
| PySR | $e^{1.2x_3} + \sin(x_0 + x_1x_2)$ | $0.52\cos(0.2x_2^2) + \tanh(x_0)\tan\left(\cos\left(\tanh\left(\frac{1.9}{x_1}\right)\right) + 0.534\right)$ | $4.26x_1^2(\tanh(\sin(6.283 x_0) + 1.8) - 1.1) + 0.967$ | $\sinh(0.406\cosh(\tanh(x_0 \sinh(\tanh(\sinh(x_0)))))) + \tanh(\cosh(0.895x_1)) + 0.69) - 3.031$ |
| TaylorGP | $e^{x_3}|x_3|^{0.5} + 0.487$ | $\frac{x_1\sin(x_2)}{x_2\tanh(x_1)} - 1.185\cos\left(e^{|x_2|^{0.5}}\right)$ | $-x_1^2 + \sqrt{|x_1|}$ | $1.07|x_0x_1|^{\frac{1}{4}}$ |
| NeSymReS | — | $-0.376x_0 + x_1\sin\left(\frac{x_0}{x_1}\right)$ | $\frac{-3967.593x_0 + x_1^2}{\cos(65757.558(-x_1-0.157)^3)+28400.518}$ | $\cos\left(\frac{\sin\left(\frac{x_0}{x_1}\right)}{x_0}\right) + 0.653$ |
| E2E | $0.903e^{1.231x_3} + 0.105 + 1.0\cos(0.764/(0.062 \sin((-0.619x_0 - 0.079)(0.344x_1 - 0.212x_2 - 0.857)) - 0.007))$ | $(-0.001|4.547x_1 - 0.068| - 0.601)\arctan(8.32x_0 - 242.537(0.016x_0 + 1)^3 + 7.105) + 0.005\cos(0.667 (-x_2 - 0.162)^2 - 4.352) - 0.004$ | $(0.09 - 0.083(x_1 - 0.002)^2)(6.724(1 - 0.756 \sin(6.11x_0 - 0.012))^2 + 4.0)$ | $0.536 - 0.903\arctan((0.046 x_1 - 0.991)(-0.367\sin(2.487 \sqrt{(-|32.358x_1 + 1.7| - 0.017)^2 + 0.006)} - 49.4) + 0.017|(0.053(x_0 - 0.007)^2 + 0.016)(0.001x_1 + 425.696 (x_1 + 0.011)^2 - 22.805)| - 0.286))$ |
| uDSR | $0.002x_0x_3 - 0.001x_0 + 0.003x_1x_2 - 0.001x_1x_3^2 + 0.001x_1x_3 + 0.002x_1 - 0.003x_2 + 0.151x_3^4 + 0.565x_3^3 + 0.51x_3^2 + 0.619x_3 + \sin(x_0 + x_1x_2) + 1.086$ | $-0.002x_0^3 + 0.004x_0^2 + 0.001x_0x_2^2 - 0.003x_0x_2 + 0.233x_0 + 0.034x_1^2 - 0.003x_1x_2 - 0.013x_1 + 0.003x_2^4 - 0.264x_2^2 + 0.011x_2 + 3\cos(x_2) - \cos(2x_2) + 2.617$ | $-0.002x_0^4 - 0.001x_0^3x_1 - 0.042x_0^3 + 0.001x_0^2x_1^2 - 0.001x_0^2x_1 + 0.037x_0^2 - 0.001x_0x_1^3 - 0.012x_0x_1^2 + 0.026x_0x_1 + 0.587x_0 - 0.003x_1^3 - 0.915x_1^2 + 0.077x_1 - (x_0 - \sin(x_1))\sin(x_0^2) + 0.867$ | $-0.004x_0^4 + 0.109x_0^2 + 0.008x_1^4 - 0.179x_1^2 - 0.001x_1 - e^{\cos(x_0)} + \cos(x_0) - \cos(x_1) - 0.001x_0 + 2.72$ |
| TPSR | $0.97e^{1.211x_3} + 0.087\cos(x_1 + 3.732x_2 + 10.665) + 0.027$ | $(0.007x_1 + 0.004)(2.087x_1 + 1.792) + 3.762\sin(0.0044x_0 + 2.3489 \cos(0.523x_2 + 0.03) - 0.73097) + 1.077$ | $1.075 - 0.971(-0.022x_0 - x_1 + 0.096\cos(x_0 + 1.524) + 0.078)^2$ | $(-0.004x_1 - 0.001)(x_1 + 0.002) + 2.063 - 1.494e^{-0.001|(5.976x_0-0.065)(59.997x_1+2.475)|}$ |
| SeTGAP | $0.977e^{1.201x_3} + 1.0\sin(x_0 + x_1x_2 + 18.85) + 0.001$ | $1.0\cos(0.2x_2 + 6.282)|x_1| + 1.0\tanh(0.5x_0)$ | $(0.947 - 0.948x_1^2)((\sin(6.283x_0) + 1.478)^{-1.0} + 0.025) + 0.02$ | $2.0 + \frac{15.997}{(0.026x_1^2-15.998x_1^4-15.994)} - \frac{16.587}{(-0.001x_0+0.077x_0^2+0.012x_0^3+16.459x_0^4+16.583)}$ |

Table 27: Comparison of predicted equations (E9—E13) with rounded numerical coefficients — Iteration 5

| Method | E9 | E10 | E11 | E12 | E13 |
|---|---|---|---|---|---|
| PySR | $\sqrt{1.148x_1} + \cosh(1.354\cos(x_0)\cos(\tanh(x_0))) - 2.202\cosh(1.613\tanh(0.481x_0))$ | $\sin\left(x_0 e^{x_1}\right)$ | $x_0(0.943 - 2.768\log((\log(\cosh(1.303\tanh(0.652x_1))))^{0.5})) + 3.385x_0\log(\log(\cosh(x_1)))$ | $1.0x_0\sin\left(\frac{1.0}{x_1}\right) + 1.0$ | $\sqrt{x_0}\log\left(x_1^2\right)$ |
| TaylorGP | $-\log\left(|x_0|\right) + \tanh\left(\log\left(|x_1|\right)\right) - \sqrt{\left|\log\left(\frac{0.603}{|x_0|}\right)\right|}$ | $\frac{\tanh(\sin\left(e^{x_1}\sin\left(x_0\right)\right))}{\sqrt{|\cos\left(\sin\left(x_1\right)\right)|}}$ | $3.947x_0\log\left(|x_1|\right)$ | $\frac{1.217x_0}{1.229x_1 + \frac{0.471}{x_1}} + 0.961$ | $2\log\left(|x_1|\right)\sqrt{|x_0|}$ |
| NeSymReS | $\log\left(\frac{0.28|x_1|}{|x_0|}\right)$ | $\sin\left(x_0 e^{x_1}\right)$ | $x_0\log\left(x_1^4\right)$ | $(x_0 + x_1)\sin\left(\frac{1}{x_1}\right)$ | $0.252x_0 + \log\left(x_1^2\right)$ |
| E2E | $-0.6\log(12.904\left(x_0 + 0.008\right)^2 + 0.6) + 1.9 - \frac{0.6}{0.227x_1 + 0.236}$ | $0.001 - 1.0\sin(0.257x_0\left(0.1 - 3.597e^{1.08x_1}\right))$ | $0.257x_0(30.0\log(0.19(|21.905x_1 + 0.102| + 0.001)^{0.5} - 0.01) + 0.01)$ | $1$ | $(0.031 - 0.44\log(23.696(-x_1 - 0.009)^2(x_1 + 0.201)^2(0.033x_0 + 0.086|0.501x_0 - 7.823| - 1)^2 + 0.02))(-0.001x_0 - 8.705)(0.032x_0 + 0.057)$ |
| uDSR | $\log((x_1 + \cos(1))(0.007x_0^2 x_1^2 + 0.809 - 0.054x_0^2 x_1 + 2.112x_0^2 + 0.003x_0 x_1^2 - 0.006x_0 x_1 + 0.002x_0 + 0.012x_1^4 - 0.145x_1^3 + 0.588x_1^2 - 0.851x_1))$ | $\sin\left(x_0 e^{x_1}\right)$ | $2.0x_0\log\left(x_1^2\right)$ | $\left(1.0x_0^2\sin\left(1/x_1\right) + x_0\right)/x_0$ | $(0.001x_0^3 - 0.025x_0^2 + 0.292x_0 + 0.244)\log\left(x_1^2\right)/\cos(1)$ |
| TPSR | $(0.187 - 0.012x_1)(0.179x_1 - 2.902)((-2.035 + \frac{4.906}{0.282|x_0 + 0.001| + 1.048})(-0.655x_1 - 3.76) - 0.296|x_0| + 8.821)$ | $1.0\sin\left(1.0x_0 e^{x_1}\right)$ | $\frac{(-0.318x_0 - 0.004)(-0.392x_1^2 - 15.221 - \frac{30.882}{-0.017|98.801x_1 + 0.806| - 0.484})}{}$ | $-10.226x_0 + 0.038x_1 + (x_0 + 4639.596 + \frac{371.436}{\arctan(440.318x_1 + 15.611)})(0.006x_0(-0.002x_1 - 141.195) + 0.796x_0 - 0.001) + 6.082$ | $(-0.029 + 19.719/(0.422|(0.45x_0 + 23.971)(4.248x_1 + 0.008)| + 7.133)(3.329 - 3.344(x_1 + 0.002)^2)(-0.207x_0 - 0.669)$ |
| SeTGAP | $-1.001\log\left(18.044x_0^2 + 4.522\right) + 0.994\left|\log\left(18.83x_1 + 9.24\right)\right| - 0.705$ | $1.0\sin\left(1.0x_0 e^{x_1}\right)$ | $2.0x_0\log\left(x_1^2\right)$ | $1.0x_0\sin\left(\frac{1}{x_1}\right) + 1.0$ | $1.0\sqrt{x_0}\log\left(x_1^2\right)$ |

Table 28: Comparison of predicted equations (E1—E4) with rounded numerical coefficients — Iteration 6

| Method | E1 | E2 | E3 | E4 |
|---|---|---|---|---|
| PySR | $0.607x_0 x_1$ | $-0.499x_0 + e^{\tanh\left(e^{x_1}\right)}\tanh\left(x_2\right) + \log\left(\cosh\left(x_0\right) + 29.144\right) + 3.287$ | $0.15e^{1.5x_0} + 0.5\cos\left(3x_1\right)$ | $(0.09 - 0.009x_1)\cosh\left(x_0\right) + (0.091 - 0.009x_3)\cosh\left(x_2\right)$ |
| TaylorGP | $0.62x_0 x_1$ | $-0.497x_0 - 0.002x_1 + 0.208x_2 + 8.571$ | $-x_0 + e^{x_0} - \sqrt{|x_0|} - 0.506 + \tanh(x_0 - e^{x_0} + \sqrt{|e^{x_0} - 0.977|} + 0.977)$ | $\sqrt{\log\left(|x_0|\right)}\sqrt{\left|\frac{x_2\left(-x_2\sqrt{|x_0|} + \log\left(|x_2|\right) + 0.553\right)}{\tanh\left(x_2\right)}\right|}$ |
| NeSymReS | $0.667x_0 x_1 + \cos(0.066(-x_0 - 0.869x_1)^2)$ | $93.845e^{0.002x_2} + e^{\sin\left(\frac{x_0}{x_1}\right)} - 86.601$ | $0.58e^{x_0} - 0.464\sin\left(1.482\left(-0.071x_1 - 1\right)^2\right)$ | — |
| E2E | $0.024x_1(25.373x_0 - 0.001x_1 + 0.057) - 1.1\cos(76.2\tan(2.983x_1 - 4.272) + 0.767) + 0.002$ | $(35.4 - 0.001x_0)(0.022(0.299x_0 - 1)^2 + (0.013x_1 + 0.301)(0.319\sin((0.145x_1 + 3.599)(0.052x_2 - 0.004)) + 0.061) + 0.163)$ | $0.212e^{1.379x_0} + 0.475\cos\left(3.108x_1 - 0.004\right) - 0.013$ | $0.018(0.573x_1 - (0.033x_0 - 0.002)(21.569x_0 - 1.005) + 0.036)^2 + 2.72(0.01\arctan(-(0.079e^{\frac{0.005}{18.338x_2 - 4.735}} - 199.276)e^{-\frac{0.005}{18.338x_2 - 4.735}} + 0.001)((1.851x_2 - 2.372)(0.283x_3 - 0.03) + 0.25((x_2 + 0.005)^2 - 0.003)^2 + 0.075)| + 0.001$ |
| uDSR | $(-0.001x_0^4 + 0.001x_0^3 x_1 - 0.008x_0^3 - 1.822x_0^2 x_1 + 0.023x_0^2 + 0.001x_0 x_1^3 + 0.001x_0 x_1^2 + 0.584x_0 x_1 + 0.089x_0 + 0.003x_1^3 - 0.015x_1^2 - 0.075x_1 + 0.041)\sin\left(\frac{x_0}{-3x_0^2 + x_0}\right)$ | $0.062x_2^2 - 0.5x_0 + 0.029x_1 x_2 - 0.001x_1 - 0.002x_1^2 + 0.001x_2^2 + 0.471x_2 - \cos\left(\left(x_2 + \log\left(x_2 + e^{-x_2}\right)\right)/x_2\right) + 7.094$ | $0.075x_0^4 + 0.335x_0^3 + 0.019x_0^2 + 0.001x_0 x_1^3 - 0.004x_0 x_1 - 0.987x_0 - 0.014x_1^4 + 0.001x_1^3 + 0.231x_1^2 - 0.004x_1 + \sin\left(x_0\right) + \cos\left(x_1\right)\cos\left(2x_1\right) - 0.272$ | $0.01x_1^4 - 0.02x_0^2 x_1 + 0.01x_1^2 + 0.01x_2^4 - 0.02x_2^2 x_3 + 0.01x_3^2 - \sin\left(1\right) + 0.842$ |
| TPSR | $0.074x_1(8.206x_0 + 0.136) + 0.043\sin(4.77x_1 - 0.485) + 0.001$ | $0.037x_1 - 0.134x_2 + 0.766|9.705x_0 + 93.72 + \frac{94.187}{9.425e^{0.022x_0} + 0.035x_1 - 0.524x_2 + 9.862}| - 1026.309 + \frac{117.7}{0.001x_0 + 0.123}$ | $0.15e^{1.5x_0} + 0.5\cos\left(3.0x_1\right)$ | $0.003(-0.005x_2 + (0.363 - 0.003x_3)((-0.941x_2 - 1.086)(-1.089x_1 + x_2 + 0.284) - 0.001(x_0(0.52x_3 + 69.999) + 0.001x_0 + 0.677)^2 - 0.238) - (1 - 0.025x_3)^2(0.148x_1 + x_2 - 0.552)^2 - 0.001)^2 + 0.524$ |
| SeTGAP | $0.608x_0 x_1 - 1.099\sin((2.24x_0 - 1.544)(x_1 - 0.67) - 9.46)$ | $-0.5x_0 + 0.063x_0^2 + 3.162\sqrt{0.1x_1 + 1}\sin\left(0.2x_2\right) + 6.498$ | $0.149e^{1.502x_0} - 0.5\sin\left(3.0x_1 + 4.712\right)$ | $-0.004x_0 - 0.02x_0^2 x_1 + 0.017\left(-x_0\right)^2 + 0.009\left(-x_0\right)^4 + 0.008\left(-x_1\right)^2 + 0.01x_2^4 + 0.011x_3^2 - (0.013x_2^2 + 0.003)(1.575x_3 + 0.614) - 0.023$ |

Table 29: Comparison of predicted equations (E5—E8) with rounded numerical coefficients — Iteration 6

| Method | E5 | E6 | E7 | E8 |
|---|---|---|---|---|
| PySR | $e^{1.2x_3} + \sin\left(x_0 + x_1 x_2\right)$ | $\frac{2.535e^{\tan\left(1.058\cos\left(2.528\tanh\left(\frac{2}{x_1}\right)\right)\right)}}{\cos\left(0.2x_2^2\right) + \tanh\left(x_0\right)}$ | $\frac{1.0 - 1.0x_1^2}{\sin\left(6.283x_0\right) + 1.5}$ | $\sinh(\sinh(\sinh(\tanh(\sinh(\cosh(\tanh(x_0\tanh(x_0))))\cosh(\tanh(x_1\tanh(x_1))) - 0.718)))))$ |
| TaylorGP | $e^{x_3}|x_3|^{0.5} + \tanh((x_3 - 0.891)e^{x_3} + 1.718e^{x_3})$ | $-1.185\cos\left(e^{|x_2|^{0.5}}\right) + \frac{4.854\sin\left(x_2\right)}{x_2}$ | $-x_1^2 + \log(|x_1^2 - \log(|x_1^2 - \sqrt{|x_1\left(x_1 - 0.073\right)|}|)|)$ | $1.102|x_0 x_1|^{\frac{1}{4}}$ |
| NeSymReS | — | $-0.368x_0 + x_1\sin\left(\frac{x_0}{x_1} - 0.001x_2\right)$ | $\frac{0.184x_0 + x_1^2}{\cos\left(1.965x_0 - x_1\right) - 1.94}$ | $\cos\left(\frac{\sin\left(\frac{x_0}{x_1}\right)}{x_0}\right) + 0.643$ |
| E2E | $0.002x_2 + 1.106e^{1.147x_3} + 0.774\cos\left(0.002x_1\right) + 0.942\cos\left(28.703x_2 - 0.956\right) - 0.977$ | $0.942$ | $\left(6.284\left(x_1 - 0.007\right)^2 - 7.49\right)(-0.082(0.01x_1 + \sin(6.457x_0 + 0.14) - 0.705)^2 - 0.098)$ | $2.06 - 30.5/(6.892(0.047 - x_0)^2(x_1 - 0.003)^2 + 18.24((1 - 0.006x_0)^2)^2 - 9.27\cos(1.329x_1 - 44.203) + 10.058)$ |
| uDSR | $0.001x_0 x_3^2 + 0.001x_0 x_3 + 0.002x_0 - 0.001x_1^2 + 0.001x_1 x_2 x_3 + 0.001x_1 x_3 - 0.001x_1 + 0.002x_2^2 + 0.004x_2 x_3 + 0.003x_3 + 0.151x_3^4 + 0.567x_3^3 + 0.519x_3^2 + 0.619x_3 + \sin\left(x_0 + x_1 x_2\right) + 1.083$ | $-0.002x_0^3 + 0.006x_0^2 + 0.003x_0 x_2 + 0.238x_0 - 0.001x_1^2 x_2^2 + 0.001x_1^2 x_2 + 0.042x_1^2 + 0.001x_1 x_2^2 + 0.002x_1 x_2 - 0.029x_1 + 0.003x_2^4 - 0.258x_2^2 - 0.007x_2 + 3\cos\left(x_2\right) - \cos\left(2x_2\right) + 2.536$ | $0.002x_0^4 - 0.004x_0^3 x_1 - 0.019x_0^3 - 0.003x_0^2 x_1 - 0.042x_0^2 - 0.001x_0 x_1^3 - 0.01x_0 x_1^2 + 0.061x_0 x_1 + 0.294x_0 - 0.003x_1^4 + 0.001x_1^3 - 0.851x_1^2 + 0.047x_1 + \sin\left(6x_1\right) + 0.909$ | $0.008x_0^4 - 0.18x_0^2 - 0.001x_0 - 0.004x_1^4 + 0.108x_1^2 - e^{\cos\left(x_1\right)} - \cos\left(x_0\right) + \cos\left(x_1\right) + 2.727$ |
| TPSR | $0.98e^{1.207x_3} - 0.039\sin(-x_1 + 3.302x_2 + 18.63) + 0.022$ | $(-0.068x_0 - 1.576\cos(0.538x_2) - 0.674)(0.016803x_0 + 0.001x_1 + 4.731\sin(0.015(0.326 - x_1)^2 + 3.728) + 2.043)$ | $\frac{412.541 - 411.968}{\left(0.001\left(x_1 + 0.002\right)^2 + 1\right)^2}$ | $(0.001 - 0.003x_1)(x_1 - 0.002) + 2.057 - 1.494e^{-0.001|(5.993x_0 + 0.026)(59.999x_1 - 1.554)|}$ |
| SeTGAP | $0.995e^{1.201x_3} + 1.0\sin(x_0 + x_1 x_2 - 6.283) + 0.002$ | $0.999\cos\left(0.2x_2^2 - 0.011\right)|x_1| + 1.0\tanh\left(0.5x_0\right)$ | $\left(-0.001 + \frac{0.999}{\sin\left(6.283x_0 - 9.424\right) - 1.5}\right)\left(x_1^2 - 0.977\right) + 0.018$ | $-\frac{16.085}{\left(16.038x_1^4 + 2.016.09\right)} + 2.0 + \frac{7.918}{\left(0.158x_0^4 - 8.067x_0^4 - 7.943\right)}$ |

Table 30: Comparison of predicted equations (E9—E13) with rounded numerical coefficients — Iteration 6

| Method | E9 | E10 | E11 | E12 | E13 |
|---|---|---|---|---|---|
| PySR | $\sinh(\sinh(2.7$ $e^{-\frac{0.259x_0^2}{x_1+0.727}-\frac{0.178x_0}{(x_1+1.017)\tanh(x_0)}}$ $-1.154))-1.787$ | $\sin\left(1.0x_0e^{x_1}\right)$ | $2x_0\log\left(x_1^2\right)$ | $x_0\sin\left(\frac{1}{x_1}\right)+1.0$ | $\sqrt{x_0}\log\left(x_1^2\right)$ |
| TaylorGP | $-0.683\log\left(|x_0|\right)-$ $1.618\sqrt{|x_0|}+\sqrt{|x_1|}$ | $\sin\left(x_0e^{x_1}\right)$ | $2.458x_0\log(|x_1|)+$ $(1.632x_0-0.908)\log\left(|x_1|\right)+$ $0.875\cos(3.872x_0+(2.732$ $x_0-2.481)\log(|0.754x_1$ $\tanh(x_1)|x_1|^{1.5}+0.307|)+$ $(4.458x_0-2.481)\log(|x_1|))$ | $0.806x_0\sin\left(\frac{1.229}{x_1}\right)$ $+0.806$ | $(\log\left(|x_0|\right)\log\left(|x_1|\right)+\tanh\left(\log\left(|x_1|\right)\right))$ $(\log(|x_0\tanh(\log(|x_1|))+\log\left(|x_1|\right)-$ $0.13|^{0.5})+|\tanh(x_0\tanh($ $\log(6.162|\log(|x_1|)|)))|^{0.5})$ |
| NeSymReS | $\log\left(\frac{0.281|x_1|}{|x_0|}\right)$ | $\sin\left(x_0e^{x_1}\right)$ | $x_0\log\left(x_1^4\right)$ | $(x_0+x_1)\sin\left(\frac{1}{x_1}\right)$ | $0.245x_0+\log\left(x_1^2\right)$ |
| E2E | $2.964\left(0.2x_1-1\right)^3+4.352-$ $0.792\log(166.831$ $(x_0-0.052)^2+17.6)$ | $1.0\sin(0.005x_0$ $(167.868e^{1.143x_1}-$ $0.01))+0.001$ | $(2.024x_0-0.006)(0.75$ $\log(1.946(-x_1-0.005)^2$ $+0.05)+0.068)$ | $7.12\log(0.001x_0$ $(0.003+\frac{31.2}{0.173x_1+0.001}))$ $+0.996$ | $(0.004-5.12\log(0.175x_0+1.109))$ $(0.945((|0.004-\frac{49.1}{0.326x_1-0.001}|+$ $0.045)^{0.5}-0.898)^{0.5}-3.44)$ |
| uDSR | $\log\left(\frac{2x_1+1}{4.0x_0^2+1.0}\right)$ | $\sin\left(x_0e^{x_1}\right)$ | $x_0\log\left(1.851x_1^4\cos\left(1\right)\right)$ | $\left(1.0x_0^2\sin\left(1/x_1\right)+x_0\right)/x_0$ | $\log\left(x_1^2\right)\log(0.004x_0^3+$ $0.06x_0^2+1.19x_0+0.001x_1+1.419)$ |
| TPSR | $\left(-0.01+\frac{0.001}{x_0+11.073}\right)(3.527-$ $0.23x_1)(x_1-15.288)((3.097$ $-6.725/(0.000341|(x_0-0.003)$ $(0.02x_1+0.399)(0.954x_1+$ $19.312)|+0.016)(-0.005x_1-$ $0.063)+0.665|x_0|-25.78)$ | $1.0\sin\left(0.999x_0e^{x_1}\right)$ | $(26.232-0.008x_1)$ $(-0.523x_0-0.01)$ | $2.867x_0+1.066$ | $(0.023-\frac{0.25}{0.829|x_1-0.001|+0.087})$ $(0.948-0.97x_1)(43.582-0.84x_0)$ $(0.055x_0+0.137)(0.494x_1+0.482)$ |
| SeTGAP | $-1.0\log\left(11.241x_0^2+2.809\right)+$ $1.0\left|\log\left(9.824x_1+4.914\right)\right|$ $-0.559$ | $1.0\sin\left(1.0x_0e^{x_1}\right)$ | $2.0x_0\log\left(x_1^2\right)$ | $1.0x_0\sin\left(\frac{1}{x_1}\right)+1.0$ | $2.0\sqrt{x_0}\log\left(|x_1|\right)$ |

Table 31: Comparison of predicted equations (E1—E4) with rounded numerical coefficients — Iteration 7

| Method | E1 | E2 | E3 | E4 |
|---|---|---|---|---|
| PySR | $0.608x_0x_1+$ $1.1\sin((x_0-0.667)$ $(x_1-0.667))$ | $3.017\sin\left(0.2x_2\right)+$ $3.017\cosh\left(0.162x_0-0.723\right)+2.818$ | $0.15e^{1.5x_0}+0.5\cos\left(3.0x_1\right)$ | $-0.02x_0^2x_1-0.02x_2^2x_3+$ $0.091\cosh\left(x_0\right)+$ $0.091\cosh\left(x_2\right)$ |
| TaylorGP | $0.607x_0x_1$ | $-0.499x_0-0.008x_1+0.207x_2+8.586$ | $-2x_0+e^{x_0}-0.363$ | $0.756x_2\tanh\left(x_2\right)+$ $0.756\log\left(|x_0|\right)\sqrt{|x_0|}$ |
| NeSymReS | $0.61x_0x_1-\sin(0.001$ $(0.965x_0-1)^2)$ | $96.368e^{0.002x_2}+e^{\sin\left(x_0x_1\right)}-89.13$ | $0.579e^{x_0}-$ $0.437\sin\left(1.633\left(0.043x_1+1\right)^2\right)$ | $-$ |
| E2E | $0.598x_0x_1-0.004x_0$ $+0.001x_1+0.143$ | $(0.011x_2+0.445)(0.174x_1-0.051x_2$ $+5.52)(0.387(0.235x_0-1)^2+\sin($ $0.225x_2+0.023)+0.144)+4.88$ | $((7.16(0.001x_0-1)^2+0.01)$ $(0.198e^{1.42x_0}+0.519\cos(3.037x_1$ $-0.034-0.072)+0.057)/$ $(7.16(0.001x_0-1)^2+0.01)$ | $0.002|5.04((x_0+0.033)^2-$ $0.001)^2+5.21(0.818x_1+$ $0.67x_3-(x_2+0.025)^2+$ $0.01)^2-0.11|+0.001$ |
| uDSR | $(-0.001x_0^2+0.606x_0x_1+0.009x_0$ $+0.001x_1^3-0.01x_1^2-0.018x_1$ $+0.022)\cos\left(e^{-3x_0^2+x_0}\right)$ | $0.063x_0^2-0.5x_0+0.029x_1x_2-0.002x_2^3$ $+0.001x_2^2+0.498x_2+\cos\left(\frac{x_2}{x_2e^{x_2}+x_2}\right)+5.677$ | $0.013x_0^4-0.033x_0^3-0.411x_0^2-$ $0.001x_0x_1-0.898x_0-0.014x_1^4$ $+0.23x_1^2+e^{x_0}+$ $\cos\left(x_1\right)\cos\left(2x_1\right)-1.306$ | $0.01x_0^4-0.02x_0^2x_1-$ $1.0x_0^2-1.0x_0x_3+x_0\left(x_0+x_3\right)$ $+0.01x_1^2+0.01x_2^4-$ $0.02x_2^2x_3+0.01x_3^2$ |
| TPSR | $0.011x_1(56.515x_0$ $+1.528)-0.031\sin($ $4.588x_1+0.776)$ $-0.005$ | $(2.621(0.060401x_0-0.019x_2-1.0-$ $\frac{0.1813}{-0.544e^{-2.096x_1+6.406x_2-1.507}})^2+0.508)$ $(0.073x_0+0.011x_1-0.028x_2+2.346)$ | $0.15e^{1.5x_0}+$ $0.5\cos\left(3.0x_1\right)$ | $0.198(-x_2-0.027)^2+$ $0.127(0.236x_1-0.234(0.006$ $-x_0)^2-1)^2-0.662$ |
| SeTGAP | $0.607x_0x_1+$ $1.098\sin((2.24$ $x_0-1.544)(x_1-$ $0.67))-0.002$ | $-0.501x_0+0.064x_0^2+(3.166$ $(0.1x_1+1)^{0.5}+0.001)$ $\sin(0.201x_2)+6.452$ | $0.151e^{1.498x_0}-$ $0.5\sin\left(3.001x_1+4.712\right)$ | $-0.02x_0^2x_1+0.01x_2^4-0.021$ $x_3\left(x_2\right)^2+0.005x_3+0.01\left(x_0\right)^4$ $+0.01\left(x_1\right)^2-(0.004x_0-0.708)$ $\left(0.014x_3^2-0.019\right)+0.028$ |

Table 32: Comparison of predicted equations (E5—E8) with rounded numerical coefficients — Iteration 7

| Method | E5 | E6 | E7 | E8 |
|---|---|---|---|---|
| PySR | $e^{1.2x_3}+\sin\left(x_0+x_1x_2\right)$ | $x_1\cos\left(0.2x_2^2\right)\tanh\left(236.582x_1\right)+$ $\tanh\left(0.496x_0\right)$ | $\left(1.863x_1^2-1.843\right)(\tanh(\sinh($ $\sin(6.281x_0))+1.345)-1.215)$ | $1.644\log(0.976/(\cos(\tanh(1.094x_0$ $\tanh(\sinh(\sinh(x_0)))))$ $\cos(\tanh(0.812\sinh(x_1)))))$ |
| TaylorGP | $2e^{x_3}-\log(|-3e^{x_3}+\log(|$ $2.287e^{x_3}-1.447\log(|\sin(x_3$ $)||)|)+1.447\log(|\sin(x_3)|)|)$ | $\frac{x_1\sin\left(x_2\right)}{x_2\tanh\left(x_1\right)}+\cos\left(x_2\right)+\tanh\left(x_0\right)$ | $-x_1^2+\log\left(\left|x_1^2-\log\left(\left|x_1^2-0.828\sqrt{|x_1^2|}\right|\right)\right|\right)$ | $\tanh\left(\left(x_0+0.504\right)\tanh\left(x_0\right)\right)+$ $\tanh\left(x_1^2-0.017\right)$ |
| NeSymReS | $-$ | $-0.356x_0+x_1\sin\left(\frac{x_0}{x_1}-0.009x_2\right)$ | $\frac{0.08x_0+x_1^2}{\cos\left(2.445(0.924x_1-1)^2\right)-1.916}$ | $\cos\left(\frac{\sin\left(\frac{x_0}{x_1}\right)}{x_0}\right)+0.695$ |
| E2E | $1.138e^{1.163x_3}-0.123+0.953$ $\cos(22.045x_2-14.032+1.11/$ $(-0.742x_0+0.002x_1+19.256)+$ $2.11/((31.854x_2+0.793)(0.001x_3$ $+2.26))-0.033/(3.382x_2+0.147))$ | $0.806$ | $(0.368-\frac{4.78}{8.105-0.291x_1})(0.338x_1+23.594)$ $(0.348x_1-4.637)(0.136(0.019-x_1)^2$ $-0.117)(-0.219(1-0.329$ $\cos(6.186x_0-7.784))^3-0.046)$ | $2.1-0.9/(0.03(0.309x_0+0.074)$ $(0.687x_0-0.059)(2.086x_1-0.139)$ $(17.421x_1-0.429)|+0.6)$ |
| uDSR | $-1.0x_0^2-0.999x_0x_1-0.001x_0x_2+0.001x_0x_3$ $+x_0\left(x_0+x_1\right)-0.001x_1^2x_3-0.002x_1^2$ $-0.002x_1x_2-0.001x_1x_3-0.005x_1-0.001x_2^2$ $-0.003x_2x_3-0.001x_2+0.152x_3^4+0.567x_3^3$ $+0.512x_3^2+0.62x_3+\sin\left(x_0+x_1x_2\right)+1.1$ | $-0.002x_0^3-0.001x_0^2x_1+0.008x_0^2-$ $0.003x_0x_1+0.258x_0+0.035x_1^2+$ $0.004x_1x_2+0.029x_1+0.003x_2^4-$ $0.249x_2^2-0.009x_2+e^{\cos\left(x_2\right)-\cos\left(2x_2\right)}+$ $2\cos\left(x_2\right)+1.093$ | $-0.002x_0^3x_1-0.036x_0^3+0.002x_0^2x_1^2-0.004x_0^2+$ $0.001x_0x_1^3-0.006x_0x_1^2+0.028x_0x_1$ $+0.483x_0-0.001x_1^4-0.011x_1^3-0.892x_1^2$ $+0.114x_1-x_0\sin\left(x_0^2\right)+0.946$ | $0.008x_0^4-0.179x_0^2-0.004x_1^4$ $+0.108x_1^2+0.002x_1-e^{\cos\left(x_1\right)}$ $-\cos\left(x_0\right)+\cos\left(x_1\right)-0.001x_0+2.722$ |
| TPSR | $63.518$ $e^{(0.5-0.15x_3)(-0.068\cos(x_1+1.23x_2+1)-8.195)}$ $-0.024$ | $(-0.396\cos(0.876x_2-0.03)$ $-0.167)((-0.016x_2-0.304)(x_0-$ $13.652x_2+269.341)+\cos(2.997$ $x_1+0.599)+70.235)$ | $-0.09x_0-0.01x_1$ $(90.02x_1+6.533)+0.94$ | $2.011-$ $1.427e^{-0.0039|x_1(98.999x_0+0.009)|}$ |
| SeTGAP | $0.999e^{1.201x_3}+1.0\sin(x_0+$ $x_1x_2-12.566)+0.003$ | $1.0\cos\left(0.2x_2^2-6.275\right)|x_1|+$ $1.0\tanh\left(0.501x_0\right)$ | $\frac{2.907x_1^2-2.919}{2.904\sin\left(6.283x_1+3.148\right)-4.358}$ | $2.0-\frac{14.033}{13.947x_1^4-0.122x_1^3+14.032}-$ $\frac{14.438}{14.532x_1^4+14.434}$ |

Table 33: Comparison of predicted equations (E9—E13) with rounded numerical coefficients — Iteration 7

| Method | E9 | E10 | E11 | E12 | E13 |
|---|---|---|---|---|---|
| PySR | $0.226\cos(\cos(x_0))+\sinh(\sinh(0.974\cos(0.643x_0+0.167)))-\frac{8.714}{x_1+2.537}$ | $\sin\left(1.0x_0e^{x_1}\right)$ | $2.0x_0\log\left(x_1^2\right)$ | $x_0\sin\left(\frac{1}{x_1}\right)+1$ | $\sqrt{x_0}\log\left(x_1^2\right)$ |
| TaylorGP | $0.26x_1+\log\left(\frac{0.952}{\lvert x_0\rvert}\right)-\sqrt{\left\lvert\log\left(\frac{0.593}{\lvert x_0\rvert}\right)+\log\left(\left\lvert\tanh\left(\frac{x_1}{x_0}\right)\right\rvert\right)\right\rvert}$ | $\sin\left(x_0e^{x_1}\right)$ | $4x_0\log\left(\lvert x_1\rvert\right)$ | $\frac{x_0\tanh\left(x_1^2\right)}{x_1}+1$ | $2\log\left(\lvert x_1\rvert\right)\sqrt{\lvert x_0\rvert}$ |
| NeSymReS | $\log\left(\frac{0.274\lvert x_1\rvert}{\lvert x_0\rvert}\right)$ | $\sin\left(x_0e^{x_1}\right)$ | $x_0\log\left(x_1^4\right)$ | $(x_0+x_1)\sin\left(\frac{1}{x_1}\right)$ | $0.23x_0+\log\left(x_1^2\right)$ |
| E2E | $2.0-0.6\log(6.862(1-\frac{0.129}{-0.056x_1-0.02})^2(x_0-0.009)^2+0.8)$ | $-0.998\sin((0.214-33.248x_0)(0.026\,e^{1.125x_1}+0.001))$ | $0.241x_0(31.0\log(0.19(\lvert25.419x_1+0.12\rvert+0.011)^{0.5}-0.01)+0.5)$ | $(0.629-4.58\sin((-0.007+\frac{9.78}{0.014-3.043x_1})(0.042x_0+0.001)))(1.581-0.006x_0)$ | $(-0.052+\frac{0.004}{-0.019-\frac{4.486}{5.98x_0-68.5}})(48.802\log(0.012\lvert0.001-6.156/((0.243x_1-0.002)(0.806\sin(0.085x_0-0.168)+0.054))\rvert+0.025)+0.04$ |
| uDSR | $\log\left(\frac{x_1+0.5}{2.0x_0^2+0.5}\right)$ | $\sin\left(x_0e^{x_1}\right)$ | $x_0\log\left(1.0x_1^4\right)$ | $\left(1.0x_0^2\sin\left(1/x_1\right)+x_0\right)/x_0$ | $(x_0^2\log\left(x_1^2\right))(x_0+(0.089x_0^4+0.223x_1+2.369x_0^3+0.007x_0^2x_1-11.117x_0^2-0.077x_0x_1+25.02x_0-0.001x_1^4+0.002x_1^2-15.166)(x_0^2+x_0))$ |
| TPSR | $0.612x_1+(0.995-1.013\arctan(x_1+0.926))(0.808x_1-1.554)-1.0\log(51.139x_0^2+12.793)+2.955$ | $0.9998\sin\left(1.0x_0e^{x_1}\right)$ | $-0.03x_0(0.196x_1-261.804+40.943/(0.07\lvert1.488x_1+0.003\rvert+0.058))$ | $-0.772x_0+(10.323+\frac{1}{x_0-0.027})(0.029\arctan(15.95x_1-5.238)+0.059)(x_0+(0.128-0.034x_1)(1.088-0.098x_1)(4.661x_0-0.086)-0.027)+0.946$ | $(-1.365+\frac{1.705}{0.0007x_0-0.55113(x_1+0.006)^2+0.376})(-0.457x_0-2.791)$ |
| SeTGAP | $-1.0\log\left(2.785x_0^2+0.696\right)+1.0\left\lvert\log\left(13.845x_1+6.92\right)\right\rvert-2.296$ | $1.0\sin\left(1.0x_0e^{x_1}\right)$ | $4.0x_0\log\left(\lvert x_1\rvert\right)$ | $1.0x_0\sin\left(\frac{1}{x_1}\right)+1.0$ | $2.0\sqrt{x_0}\log\left(\lvert x_1\rvert\right)$ |

Table 34: Comparison of predicted equations (E1—E4) with rounded numerical coefficients — Iteration 8

| Method | E1 | E2 | E3 | E4 |
|---|---|---|---|---|
| PySR | $0.609x_0x_1+0.931\cos(1.488x_0-1.126x_1(2x_0-3.019)-1.9x_1+0.556)$ | $\frac{2.278\tanh\left(0.356x_2\right)+6.388}{\sqrt{\left(0.016x_0^2+e^{-0.402x_0}\right)^{0.5}+0.127\tanh\left(x_1x_2\right)}}$ | $0.15e^{1.499x_0}-0.502\cos\left(2.999x_1+3.143\right)$ | $(\cosh\left(x_0\right)+\cosh\left(x_2\right))(-0.005x_1-0.005x_3+0.091)$ |
| TaylorGP | $0.613x_0x_1$ | $-0.506x_0-0.001x_1+0.206x_2+8.575$ | $-0.612\left(e^{x_0}e^{2\tanh\left(x_0\right)}\right)^{0.5}+e^{x_0}$ | $\left(-\sin\left(x_0\right)+\sqrt{\lvert x_2\rvert}\right)e^{\tanh\left(\log\left(\sqrt{\lvert x_0\rvert\lvert x_2\rvert}\right)\right)}$ |
| NeSymReS | $0.666x_0x_1+\cos(0.068(-x_0-0.83x_1)^2)$ | $e^{\sin\left(x_0x_1\right)}+124.329-116.894e^{-0.002x_2}$ | $0.579e^{x_0}-0.444\cos\left(0.228x_1\right)$ | — |
| E2E | $(0.001x_0-1.463)\sin(0.482\cos(\frac{0.121x_0+78.526}{0.007x_0+0.29})-0.004)+(3.222x_0+0.007)(0.186x_1+0.003)$ | $(0.01x_0-0.099)(5.434x_0+0.438)+6.3+3.0\sin\left((0.001x_1+0.02)(10.39x_2+0.291)\right)$ | $0.143e^{1.497x_0}+0.043+0.506\cos\left(3.559x_1+0.061\right)$ | $0.079\lvert0.162\left(-x_1+0.885(x_0+0.05)^2+0.031\right)^2+0.123(0.135x_0x_1+0.004x_0+0.011x_1-x_2^2-0.022x_2+0.884x_3+0.024)^2+0.523\rvert-0.048$ |
| uDSR | $(0.001x_0^2x_1+0.003x_0^2+0.604x_0x_1-0.003x_0+0.001x_1^3-0.007x_1^2-0.021x_1+0.013)\cos\left(e^{-3x_0^2}\right)$ | $(0.062x_0^2-0.5x_0+0.001x_1^2+0.029x_1x_2-0.001x_1-0.002x_2^2+0.539x_2+6.5)\cos\left(\frac{x_1}{e^4}\right)$ | $0.075x_0^4+0.236x_0^3+0.022x_0^2-0.001x_0x_1^2+0.003x_0x_1-0.104x_0+0.014x_1^4-0.001x_1^3-0.23x_1^2+0.003x_1+\cos\left(x_1\right)\cos\left(2x_1\right)-\cos\left(x_1\right)+0.712$ | $0.01x_0^4-0.02x_0^2x_1+0.01x_1^2+0.01x_2^2-0.02x_2^2x_3+0.01x_3^2$ |
| TPSR | $-2.937x_1(0.001-0.208x_0)+0.04\sin\left(4.355x_1-0.296\right)-0.002$ | $(-20.407-\frac{0.002}{x_1-0.182})(-0.01x_2-1.038+1/(0.014(0.201x_0+0.007x_2-1)^2(0.442x_0+0.001x_2-1)^2+1.342))$ | $0.15e^{1.5x_0}+0.5\cos\left(3.0x_1\right)$ | $-0.00034(0.347x_1-7.955)(x_3-(90.12-0.35x_3)(0.111x_1-2.6)-1.987)+0.01(-0.001x_1+0.995x_2^2-x_3+0.062)^2-0.621$ |
| SeTGAP | $0.608x_0x_1+1.099\sin((2.251x_0-1.52)(x_1-0.655))$ | $-0.5x_0+0.062x_0^2+6.541+\sqrt{0.1x_1+1}(3.166\sin\left(0.201x_2\right)-0.033)$ | $0.15e^{1.5x_0}+0.5\cos\left(3.001x_1\right)$ | $-0.005x_0^2+0.01x_0^4-0.02x_1\left(-x_0\right)^2+0.01x_1^2+0.01x_2^4-0.02x_3\left(-x_2\right)^2+0.01x_3^2+0.015$ |

Table 35: Comparison of predicted equations (E5—E8) with rounded numerical coefficients — Iteration 8

| Method | E5 | E6 | E7 | E8 |
|---|---|---|---|---|
| PySR | $e^{1.2x_3}+\sin\left(x_0+x_1x_2\right)$ | $0.279e^{3.915\cos\left(\frac{\tanh\left(1.761\tanh\left(\sin\left(x_1\right)\right)\right)}{x_1}\right)}\ldots\ldots\cos\left(\tanh\left(\frac{4.667}{x_1}\right)\right)\cos\left(0.2x_2^2\right)+\tanh\left(x_0\right)$ | $\cos\left(x_1\right)+\sinh(\sin\left(6.285x_0\right)+\cos\left(0.35x_1\right)+\cos\left(0.668x_1\right)-2.288)$ | $\log(0.705\sinh(1.975\cosh\left(\tanh\left(x_0\right)\right)-\frac{1.184}{\sinh\left(\cosh\left(x_1\right)\right)}))$ |
| TaylorGP | $0.855e^{x_3}\log(\lvert2x_3+\sqrt{\lvert\tanh\left(e^{2x_3}\right)\rvert}+0.748\rvert)$ | $\cos\left(x_2\right)+0.422+\frac{5.65\log\left(\lvert x_1\rvert^{0.5}\right)\sin\left(x_2\right)}{x_2}$ | $-x_1^2+\log(\lvert x_1^2-\log\left(\lvert x_1^2-\log\left(\lvert x_1^2+0.943\rvert\right)\rvert\right)\rvert)$ | $\frac{0.829\lvert x_0\rvert^{\frac{1}{4}}}{\cos\left(\tanh\left(x_1\right)\right)}$ |
| NeSymReS | — | $0.126x_0+x_1\sin\left(0.294x_1+\frac{x_2}{x_1}\right)$ | $\frac{0.042x_0+x_1^2}{\cos\left(5.419(x_1-0.517)^2\right)-1.896}$ | $\cos\left(\frac{\sin\left(\frac{x_0}{x_1}\right)}{x_0}\right)+0.644$ |
| E2E | $1.392e^{1.085x_3}+0.949\sin(-25.875x_2+100.707+\frac{0.007}{0.055-0.1x_1})-0.484$ | $(6.048\lvert0.168x_1+0.001\rvert+0.14)\cos((0.21-17.355x_2)(-0.012x_2-0.01))+0.8\arctan(0.699x_0+0.269)-0.01$ | $(-10.369(1-0.002x_0)^2(0.346x_1+(0.01-0.003x_1)(0.029x_0-0.003)+0.018)^2+1.7)(0.373(\sin(6.398x_0+0.342)-0.814)^2+0.42)$ | $2.065(-1-0.556/(-0.361\,(x_0+0.028)^2(0.001x_1-1)^2\,(x_1+0.046)^2+0.409\cos(2.271x_1+0.076)-1.369))^2+0.001$ |
| uDSR | $0.001x_0x_1-0.001x_0x_3-0.002x_0+0.001x_1^2+0.001x_1x_2+0.001x_1x_3^2+0.003x_1x_3-0.002x_2^2-0.001x_2x_3-0.004x_2+0.153x_3^4+0.569x_3^3+0.506x_3^2+0.602x_3+\sin\left(x_0+x_1x_2\right)+1.088$ | $-0.002x_0^3-0.002x_0^2+0.007x_0x_1+0.004x_0x_2+0.262x_0+0.039x_1^2+0.002x_1x_2+0.006x_1+0.003x_2^4-0.262x_2^2-0.001x_2+3\cos\left(x_2\right)-\cos\left(2x_2\right)+2.697$ | $0.001x_0^4-0.037x_0^3-0.001x_0^2x_1^2-0.002x_0^2x_1-0.018x_0^2-0.001x_0x_1^3-0.013x_0x_1^2+0.011x_0x_1-x_0\sin\left(x_0^2\right)+0.501x_0+0.001x_1^3-0.896x_1^2+0.032x_1+0.976$ | $-0.004x_0^4+0.107x_0^2-0.002x_0+0.008x_1^4-0.179x_1^2-0.001x_1-e^{\cos\left(x_0\right)}+\cos\left(x_0\right)-\cos\left(x_1\right)+2.723$ |
| TPSR | $0.983e^{-0.001x_2+1.206x_3}e^{0.001\arctan((-59.047x_0}\ldots^{\cdots-117.779)(-70.903x_1-}\ldots^{\cdots32.111x_3+8.033))}+0.01$ | $(0.003\cos(0.044x_0+0.023x_1^2-2.266)+0.003)(5.010^{-6}x_0-0.005x_1-465.626(0.025-\arctan(-0.473-\frac{9.002}{-0.668(x_2-0.082)^2-0.109}))^3-0.002)$ | $1.045-0.926\left(0.012x_0-x_1+0.031\right)^2$ | $2.007-\frac{1.09}{0.179(0.01-x_1)^2(-x_0-0.03)^2+0.823}$ |
| SeTGAP | $0.999e^{1.2x_3}+1.0\sin(x_0+x_1x_2+12.567)+0.003$ | $1.0\cos\left(0.2x_2^2+12.564\right)\lvert x_1\rvert+1.0\tanh\left(0.501x_0\right)$ | $\frac{3.767-3.787x_1^2}{(3.786\sin\left(6.283x_0\right)+5.68)}+0.003$ | $\frac{14.946}{0.121x_0^3-15.228x_0^4+0.176\left(-x_0\right)^2-14.95}-\frac{18.513}{0.135x_1-0.322x_1^3+18.515\left(-x_1\right)^4+18.515}+2.0$ |

Table 36: Comparison of predicted equations (E9—E13) with rounded numerical coefficients — Iteration 8

| Method | E9 | E10 | E11 | E12 | E13 |
|---|---|---|---|---|---|
| PySR | $0.843e^{1.412\cos(x_0)} + \log(x_1 + 0.504)$ $-3.313 + 0.843\cos(\cos(0.866x_0))$ $-0.036\cosh(x_0)$ | $\sin(x_0 e^{x_1})$ | $2x_0 \log(x_1^2)$ | $x_0 \sin(\cos(0.001/$ $((1581.494x_1)/(\cosh(x_1))$ $+\cosh(x_1)))/x_1) + 1$ | $\sqrt{x_0}\log(x_1^2)$ |
| TaylorGP | $\log\left(\frac{0.764}{|x_0|}\right) + \tanh(x_1) - (|\log\left(\frac{0.792}{|x_0|}\right)$ $+\log(|\tanh\left(\frac{x_1}{x_0}\right)|)|)^{0.5}$ | $\sin(x_0 e^{x_1})$ | $4x_0 \log(|x_1|)$ | $0.938 + \frac{-x_0 x_1 - 0.344}{-x_1^2 - 0.307}$ | $0.255x_0 \log(|x_1|) + 2.807\log(0.943$ $|x_1|)|\log(0.826\sqrt{|x_0|}|\log(4.902$ $|x_0|\log(4.902|\log(|x_1|)|)|)|) + 0.13|^{0.5}$ |
| NeSymReS | $\log\left(\frac{0.282|x_1|}{|x_0|}\right)$ | $\sin(x_0 e^{x_1})$ | $x_0 \log(x_1^4)$ | $(x_0 + x_1)\sin\left(\frac{1}{x_1}\right)$ | $0.239x_0 + \log(x_1^2)$ |
| E2E | $-6.44|0.81\arctan(0.295x_0 + 0.004)$ $+0.001|+3.33 + 96.1/((0.021x_0-$ $47.199)(0.062x_1 + 0.262)(0.023x_0$ $+0.692x_1 + 2.901))$ | $0.994\sin((-5.26x_0-$ $0.008)(0.273e^{0.573x_1}+$ $0.006)(-0.31x_1+$ $0.118\log(0.055x_1+$ $0.756) - 0.782))$ | $(0.331x_0 + 0.001)$ $(20.8\log(0.327|20.18$ $x_1 + 0.198|+0.009)^{0.5}$ $-0.003) - 4.11)$ | $7.72\sin(0.003x_0(0.008+$ $\frac{6.35}{0.173x_1+0.002})) + 0.078$ $|0.006x_1 - 0.247|+0.987$ | $(0.034 - 0.052\sqrt{|0.603x_0 + 1|})$ $(0.08 - 57.9\log(|7.09|0.162x_1$ $-0.004|-0.001|))$ |
| uDSR | $\log\left(\frac{2x_1+1}{4.0x_0^2+1.0}\right)$ | $\sin(x_0 e^{x_1})$ | $x_0 \log(1.0x_1^4)$ | $(1.01x_0 + 0.008x_1^2 + 1.017x_1 - 0.161)\sin(1/x_1)$ | $\log(x_1^2)\log(0.004x_0^3 + 0.061x_0^2$ $+1.186x_0 - 0.001x_1 + 1.427)$ |
| TPSR | $0.13x_1 + 2.261\arctan(x_1 + 0.751)$ $-1.0\log(13.419x_0^2 + 3.359) - 0.1994$ | $1.0\sin(1.0x_0 e^{x_1})$ | $-17.697x_0 + (0.185x_0+$ $0.002)(0.673(0.001 - x_1)^2$ $+122.071+$ $\frac{112.993}{-0.116|29.974x_1+0.112|-1.036})$ $-0.157$ | $(7.035 \cdot 10^{-9}x_0 + 8.736 \cdot 10^{-5})$ $(5.096x_1 + 11119.99+$ $\frac{(0.002-0.068x_0)(288275.2x_1+22166.6)}{-1.46(x_1+0.084)^2-0.985})$ | $(0.089 - \frac{0.153}{0.125|x_1|+0.01})(-15.537x_0$ $-72.042)(0.013(0.004 - x_1)^2$ $(1 - 0.005x_0)^2 - 0.01)$ |
| SeTGAP | $-0.999\log(3.882x_0^2 + 0.963)+$ $0.999|\log(10.173x_1 + 5.052)| - 1.656$ | $1.0\sin(1.0x_0 e^{x_1})$ | $2.0x_0 \log(x_1^2)$ | $1.0x_0 \sin\left(\frac{1}{x_1}\right) + 1.0$ | $1.0\sqrt{x_0}\log(x_1^2)$ |

Table 37: Comparison of predicted equations (E1—E4) with rounded numerical coefficients — Iteration 9

| Method | E1 | E2 | E3 | E4 |
|---|---|---|---|---|
| PySR | $0.607x_0 x_1-$ $1.1\cos(x_0(2.25x_1-$ $1.5) - 1.5x_1 + 2.571)$ | $0.178x_1\sin(0.199x_2) + 2.983\sin(0.2x_2)+$ $2.983\log(\cosh(0.464x_0 - 1.809) + 5.42)$ | $0.15e^{1.5x_0} + 0.5\cos(3x_1)$ | $(0.092 - 0.008x_1)(\cosh(x_0) + 6.478)+$ $(0.092 - 0.009x_3)\cosh(x_2) - 0.643$ |
| TaylorGP | $0.613x_0 x_1$ | $\tanh(x_2) + |(x_0(-x_0 + (e^{x_0})^{\frac{1}{4}} + \log(55.556|x_0|)$ $+1.704 + \cos(e^{1.424\log(55.556|x_0|)^{\frac{1}{4}}}))(\tanh(x_0))|^{0.5}$ $+1.305|-0.587x_0 + 0.587(e^{x_0})^{\frac{1}{4}} + 0.587$ $\log(55.556|x_0|) + 0.587\cos(x_0) + 1|^{0.5}$ | $-x_0 + e^{x_0} + \cos(x_0) - 0.815$ | $x_0 + 0.118x_2^2 - 0.412$ |
| NeSymReS | $0.61x_0 x_1 + \sin($ $0.002(-x_0 - 0.698)^2)$ | $93.728e^{0.002x_2} + e^{\sin(x_0 x_1)} - 86.434$ | $0.583e^{x_0} - 0.465\cos(0.204x_1)$ | — |
| E2E | $0.021x_1(29.356x_0+$ $0.027) + 1.1\cos(30.874$ $x_1 - 1.378) - 0.001$ | $0.019x_1 + (0.007x_0 - 0.056)(8.915x_0 + 0.563)+$ $(0.437x_1 + 8.091)$ $(0.371\sin(0.194x_2 + 0.034) - 0.007) + 6.279$ | $0.042x_0 - 0.004x_1 + (0.001x_0$ $+0.216)(2.33\cos(3.43x_1+$ $0.075) - 0.07) + 0.166$ $\sqrt{0.779e^{2.954x_0} + 1} - 0.008$ | $0.004|2.358(x_3 - 0.048)^2 + 2.231((x_0+$ $0.028)^2 + 0.039)^2 + 2.177(-0.921x_1+$ $(x_2 + 0.01)^2 + 0.024)^2 - 0.255|+0.003$ |
| uDSR | $(-0.002x_0^4 + 0.001x_0^3 x_1 + 0.001x_0^3-$ $1.265x_0^2 x_1 + 0.017x_0^2 + 0.001x_0 x_1^3+$ $0.001x_0 x_1^2 + 0.545x_0 x_1 - 0.011x_0+$ $0.001x_1^4 + 0.003x_1^3 - 0.027x_1^2-$ $0.05x_1 + 0.04)\sin\left(\frac{x_0^2}{-2x_0^2+x_0}\right))/x_0$ | $\log(660.383e^{0.063x_0^2 - 0.001x_0 x_1}$ $e^{0.001x_0 x_2 - 0.504x_0 + 0.03x_1 x_2}$ $e^{-0.005x_1 - 0.002x_2^3 + 0.543x_2} + 1)$ | $0.013x_0^4 - 0.033x_0^3 - 0.001x_0^2 x_1$ $-0.414x_0^2 - 0.899x_0 - 0.13x_1^4+$ $0.001x_1^3 + 0.979x_1^2+$ $0.001x_1 + e^{x_0} + e^{\cos(2x_1)} - 2.821$ | $0.01x_0^4 - 0.02x_0^2 x_1 + 0.01x_1^2+$ $0.01x_1^2 - 0.02x_2^2 x_3 + 0.01x_3^2$ |
| TPSR | $0.072x_1(8.488x_0+$ $0.065) + 0.056\sin(6.353$ $x_1 + 1.19) - 0.032$ | $(-0.009 + \frac{1}{-89.564(0.13x_2-1)^2-79.916})$ $(-1126.481((-0.001+$ $\frac{0.002}{0.283(0.259x_2+0.033)(9.015x_2-0.006)|+0.093)}$ $(0.441x_2 + 0.076)(-0.947x_0 - 0.688x_1$ $-0.205x_2 + 0.213) + 1)^3 - 7.999)$ $(-0.002x_1 + 0.122(0.134x_0 - 1)^2 + 0.303)$ | $0.007x_1(82.733x_0 + 27.9)+$ $0.351(0.166x_1 - 1)^3(-0.361$ $\cos((0.014 - 0.032x_1)(33.244$ $-77.329x_0)) - 1)^3 - 0.553$ | $-0.1167x_1 - 0.149x_3 + 0.0016(x_2-$ $0.123)^2 + 0.00512(-x_2 - (-5.082x_0$ $-0.035)(-0.255x_0 - 0.072)$ $-0.002)^2 + 1.551$ |
| SeTGAP | $0.608x_0 x_1 + 1.1$ $\sin((2.256x_0 - 1.491)$ $(x_1 - 0.655) + 0.047)$ $-0.001$ | $-0.499x_0 + 0.064x_0^2+$ $\sqrt{0.1x_1 + 1}(3.161\sin(0.2x_2) + 0.011) + 6.46$ | $0.15e^{1.5x_0} + 0.5\cos(3.0x_1)$ | $-0.02x_0^2 x_1 + 0.004x_0^2 + 0.01x_0^4 + 0.01x_1^2-$ $0.02x_2^2 x_3 + 0.01x_2^4 + 0.01x_3^2 - 0.009$ |

Table 38: Comparison of predicted equations (E5—E8) with rounded numerical coefficients — Iteration 9

| Method | E5 | E6 | E7 | E8 |
|---|---|---|---|---|
| PySR | $(e^{1.2x_3})+$ $\sin(x_0 + x_1 x_2)$ | $\tanh(x_0) + 9.715\cos(0.2x_2^2)$ $\cos\left(\frac{5.902}{\tanh(\cosh(0.095x_1))}\right)$ | $(2.482x_1^2 - 2.467)$ $(\cos(0.435e^{\cos(6.278x_0+1.57)}) - 1.16)$ | $0.8\tan(\sinh(\tanh(\cosh(0.827$ $\cosh(\tan(\tanh(x_1))))\tanh(\cosh(x_0)))))$ $\cosh(\tanh(x_0)) - 0.957$ |
| TaylorGP | $(\cos(\sin(\tanh(1.515e^{x_3})))$ $+\tanh(1.119|(0.799\cos(\sin($ $\tanh(1.515e^{x_3}))) + 0.799\tanh($ $0.534e^{0.5(x_3)}))e^{x_3} - e^{x_3}|^{0.5}))e^{x_3}$ | $\cos(x_2) + \tanh(x_0)+$ $\frac{7.407\log(|x_1|^{0.5})\sin(x_2)}{x_2}$ | $\frac{-x_1^2 + \log(|x_1^2+}{}$ $\frac{-x_1^2 + \sqrt{|x_1|} + \sqrt{\left|x_1 - 0.479\tanh\left(e^{-0.538x_1^2}\right)\right|}}{x_1^2 + 0.392}|)$ | $e^{\tanh\left(\tanh\left(\log\left(0.864\sqrt{|x_0||x_1|}\right)\right)\right)}$ |
| NeSymReS | — | $0.145x_0 + x_1\sin\left(0.302x_1 + \frac{x_2}{x_1}\right)$ | $-0.05632x_0 - 0.8403x_1^2$ | $\cos\left(\frac{\sin\left(\frac{x_0}{x_1}\right)}{x_0}\right) + 0.634$ |
| E2E | $-0.001x_1 + 0.002x_3 + 0.023$ $(0.024 - x_3)^2 + 0.947e^{1.206x_3}+$ $0.933\cos(12297.635(0.404x_1$ $+1)^2 + 0.225(0.001x_0 - 0.001$ $x_2 + 1)^{0.5} - 15.41) + 0.032$ | $(0.035x_1 - 0.002)(4.984\arctan($ $(-0.03 - \frac{4.462}{-0.539x_0 - 0.118})(0.004$ $x_0 + 0.001)) - 0.16) + 0.929$ | $\left(0.09 - 0.082(x_1 + 0.007)^2\right)$ $(6.131(0.766 - \sin(6.516$ $x_0 + 0.216))^2 + 5.0)$ | $2.1-$ $\frac{0.9}{0.002|(3.087x_1 - 0.168)(37.54(x_0 + 0.029)^3 + 6.0)|+0.6}$ |
| uDSR | $0.001x_0 x_1 + 0.001x_0 - 0.002x_1^2-$ $0.002x_1 x_2 - 0.001x_1 x_3^2 - 0.004x_1 x_3+$ $0.007x_1 + 0.003x_2^2 - 0.001x_2 x_3+$ $0.152x_3^4 + 0.569x_3^3 + 0.504x_3^2+$ $0.601x_3 + \sin(x_0 + x_1 x_2) + 1.088$ | $-0.002x_0^3 + 0.003x_0^2 - 0.003x_0 x_1+$ $0.002x_0 x_2 + 0.238x_0 + 0.038x_1^2+$ $0.002x_1 x_2 + 0.029x_1 + 0.003x_2^4-$ $0.264x_2^2 + 0.022x_2 + 3\cos(x_2)$ $-\cos(2x_2) + 2.717$ | $0.001x_0^4 - 0.037x_0^3 + 0.001x_0^2 x_1^2+$ $0.002x_0^2 x_1 - 0.019x_0^2 + 0.001x_0 x_1^3-$ $0.009x_0 x_1^2 - 0.009x_0 x_1 - x_0\sin(x_0^2)+$ $0.485x_0 + 0.004x_1^3 - 0.897x_1^2$ $-0.04x_1 + 0.951$ | $-0.004x_0^4 + 0.107x_0^2 + 0.008x_1^4$ $-0.18x_1^2 - 0.002x_1 - e^{\cos(x_0)}$ $+\cos(x_0) - \cos(x_1) + 2.725$ |
| TPSR | $2.627214 - 0.004x_2$ | $(0.397\cos(0.017x_1 + 0.524x_2+$ $0.076) + 0.24)(0.234x_0 + \cos($ $1.716x_1 - 2.174) + 6.18)$ | $-0.934(0.086 - x_1)^2(0.001x_0$ $+1 + \frac{0.116}{x_1})^2 + 1.098$ | $0.001x_1 + 0.61(0.002 - \arctan($ $-0.01x_0(89.997x_1 - 2.242))^2 + 0.654$ |
| SeTGAP | $1.001e^{1.2x_3} + 1.0\sin($ $x_0 + x_1 x_2 + 12.566) - 0.008$ | $-1.0\cos(0.2x_2^2 - 9.425)|x_1|+$ $1.0\tanh(0.5x_0)$ | $\frac{8.18x_1^2 - 8.247}{(-8.185\sin(6.283x_0 - 6.283) - 12.273)}$ $-0.006$ | $\frac{16.205}{(-0.015x_0 - 16.207x_0^4 - 0.033(-x_0)^3 - 16.206)} + 2.0+$ $\frac{11.914}{(0.013x_1 - 0.118x_1^2 - 0.016x_1^3 - 11.817x_1^4 - 11.901)}$ |

Table 39: Comparison of predicted equations (E9—E13) with rounded numerical coefficients — Iteration 9

| Method | E9 | E10 | E11 | E12 | E13 |
|---|---|---|---|---|---|
| PySR | $\log\left(\frac{x_1+0.5}{2x_2^2+0.5}\right)$ | $\sin\left(x_0 e^{x_1}\right)$ | $2.0 x_0 \log\left(x_1^2\right)$ | $x_0 \sin\left(\frac{1}{x_1}\right) + 1.0$ | $\sqrt{x_0}\log\left(x_1^2\right)$ |
| TaylorGP | $-\log\left(|x_0|\right) + \tanh\left(x_1\right) - 0.298 -$ $\sqrt{\left|\log\left(\frac{0.873}{|x_0|}\right) + \tanh\left(x_1\right) - \sqrt{|x_0|}\right|}$ | $\frac{\sin\left(\tanh\left(x_0\right)\right)}{\cos\left(\sqrt{|x_1|}\right)}$ | $4 x_0 \log\left(|x_1|\right)$ | $\frac{0.998 x_0 \sin\left(\frac{x_0}{x_1(x_0+0.048)}\right)}{\sqrt{|\log\left(\sqrt{|x_0|}\right)|} + 0.998}$ | $\left(\sqrt{x_0}\log\left(\sqrt{|x_1|}\right)^2 + 0.689\right)$ $\left(\log\left(\sqrt{|x_1|}\right) + 0.446\right) +$ $\log\left(|x_1|\right)\left(\log\left(|x_0|\right) + 1\right)$ $+ \log\left(|x_1 \tanh\left(x_1\right)|\right)$ |
| NeSymReS | $\log\left(\frac{0.281|x_1|}{|x_0|}\right)$ | $\sin\left(x_0 e^{x_1}\right)$ | $x_0 \log\left(x_1^4\right)$ | $(x_0 + x_1)\sin\left(\frac{1}{x_1}\right)$ | $0.243 x_0 + \log\left(x_1^2\right)$ |
| E2E | $-0.6\log\left(10.053(0.009 - x_0)^2 + 0.6\right)$ $+ 1.9 - \frac{0.7}{0.274 x_1 + 0.301}$ | $-0.994\sin((3.608 x_0 +$ $0.013)(-0.009 -$ $0.234 e^{(0.595 - 0.053 x_1)}$ $...(1.976 x_1 + 0.061))) - 0.001$ | $(2.922 x_0 + 0.006)$ $(4.14\arctan(0.336 -$ $0.008|0.26 x_1 + 0.695 -$ $\frac{84.4}{3.242 x_1 - 0.071}|) - 0.047)$ | $0.588 + 7.1\sin(0.058 -$ $9.51/((0.01 - \frac{103.409}{13.96 x_0 - 0.071})$ $(-0.001 x_0 + 10.994 x_1$ $-0.059)))$ | $(0.741 - 4.65\arctan(0.175$ $x_0 + 0.35))$ $(0.174 e^{0.175 x_0} - 20.8)(0.008$ $\log(0.001 x_1 + 12.9) + 0.072)$ |
| uDSR | $\log((0.002 x_0^4 + 0.002 x_0^3 x_1 + 0.001 x_0^3$ $+ 0.046 x_0^2 x_1 - 0.016 x_0^2 + 0.002 x_0 x_1^3 -$ $0.009 x_0 x_1^2 + 0.213 x_0 x_1 + 0.101 x_0 - 0.001 x_1^4 +$ $0.01 x_1^3 - 0.035 x_1^2 + 2.146 x_1 + 1.143)(4 x_0^2 + e^{x_0}))$ | $\sin\left(x_0 e^{x_1}\right)$ | $x_0 \log\left(1.0 x_1^4\right)$ | $x_0\sin\left(1/x_1\right) + 1.0$ | $(\log\left(x_1^2\right))(x_0 + x_1 +$ $(0.003 x_0^3 - 1.047 x_0^2 - 0.999 x_0 x_1$ $+ 0.54 x_0 + 0.455)/x_0)$ |
| TPSR | $59.31(-0.003 x_1 - 1)^2 - 66.389 -$ $82.366/(-4.5|(31.371 + \frac{17.165}{0.218 x_1 + 0.124})$ $(0.301 x_0 - 0.001)(0.007$ $x_1 + 0.021)| - 10.235$ | $1.0\sin\left(0.9992 x_0 e^{x_1}\right)$ | $7.953 x_0 - 0.188$ | $0.827 + 5.558\sin((0.056 x_0$ $-4.862)(0.006 x_0(0.1 +$ $\frac{8.807}{0.002 - 1.613 x_1}) - 0.009))$ | $-0.684 x_0 + 97.57 +$ $\frac{50.936}{-0.009 x_0 - 0.535} + 15.513/(($ $66.734 + 207.296 e^{-0.219 x_0})$ $(-0.012(x_1 - 0.002)^2 - 0.007))$ |
| SeTGAP | $-1.001\log\left(13.041 x_0^2 + 3.264\right) +$ $0.998|\log\left(7.365 x_1 + 3.647\right)| - 0.108$ | $1.0\sin\left(1.0 x_0 e^{x_1}\right)$ | $4.0 x_0 \log\left(|x_1|\right)$ | $1.0 x_0\sin\left(\frac{1}{x_1}\right) + 1.0$ | $2.0\sqrt{x_0}\log\left(|x_1|\right)$ |

Table 40: Comparison of predicted equations (E1—E4) with rounded numerical coefficients — Iteration 10

| Method | E1 | E2 | E3 | E4 |
|---|---|---|---|---|
| PySR | $0.608 x_0 x_1$ | $x_0 + 2.969\sin\left(0.2 x_2\right) +$ $\cosh\left(0.083 x_0 - 3.519\right) - 10.302$ | $0.15 e^{1.5 x_0} + 0.5\sin\left(3.0 x_1 + 1.571\right)$ | $-0.165 x_1 - 0.165 x_3 + 1.564\log($ $(\cosh(\sinh(0.083 x_2^2)) - 0.553)$ $\cosh(x_2)) + 1.564\cos(x_2)$ |
| TaylorGP | $\frac{0.604 x_0 x_1 (x_0^2 x_1^2)^{\frac{1}{4}}}{\sqrt{|x_0 x_1|}}$ | $-0.513 x_0 - 0.001 x_1 + 0.204 x_2 + 8.572$ | $-x_0\tanh\left(0.632 e^{x_0} - 1\right) + e^{x_0} -$ $\log\left(e^{x_0}\right) - 0.682$ | $x_0 + x_2 - 2$ |
| NeSymReS | $0.589 x_0 x_1 + \cos($ $0.598(0.966 x_0 - x_1)^2)$ | $-x_0 + e^{\sin\left(x_1\right)} + 63.876 - 56.599 e^{-0.004 x_2}$ | $0.575 e^{x_0} - 0.418\sin\left(1.451(0.052 x_1 + 1)^2\right)$ | — |
| E2E | $(23.064 x_0 - 0.034)$ $(0.026 x_1 + 0.001) - 0.001$ $+ 1.09\sin(0.158 x_0 + (1.872$ $x_1 + 0.125)(0.003|0.021$ $x_1 - 2.239| + 81.7) + 0.021)$ | $-0.008 x_0(0.172 x_1 - 1.607)(0.007 x_1 -$ $40.299) + 6.26 + (0.008 x_1 + 0.149)$ $(19.2\sin((0.001 x_2 + 0.073)$ $(3.077 x_2 + 0.001)) + 0.003)$ | $0.003 x_0 + 0.06 e^{1.804 x_0} -$ $0.499\cos(0.003 x_0 - 2.985 x_1 +$ $59.557) + 0.159$ | $0.843|0.07 x_3 - 0.013(0.836 x_1 -$ $(x_0 + 0.059)^2(0.004 x_1 - 1)^2 +$ $0.039)^2 - 0.01(0.96 x_3 - (x_2 -$ $0.046)^2 + 0.046)^2 + 0.017|$ |
| uDSR | $(-0.003 x_0^4 - 0.002 x_0^3 x_1 + 0.002 x_0^3 -$ $0.004 x_0^2 x_1^2 + 0.002 x_0^2 x_1 + 0.084 x_0^2$ $-0.002 x_0 x_1^3 + 2.43 x_0 x_1^2 - 1.168 x_0 x_1$ $-0.003 x_1^4 - 0.003 x_1^3 + 0.069 x_1^2 +$ $0.022 x_1 - 0.329)\sin\left(\frac{x_1}{4 x_1^2 - 2 x_1}\right)$ | $\log(658.504 e^{0.062 x_0^2 - 0.001 x_0 x_1}$ $e^{0.001 x_0 x_2 - 0.502 x_0 + 0.03 x_1 x_2}$ $e^{-0.002 x_1 - 0.002 x_2^2 + 0.542 x_2} + \cos\left(x_1/x_2\right))$ | $0.013 x_0^4 - 0.033 x_0^3 - 0.412 x_0^2 + 0.001 x_0 x_1$ $-0.898 x_0 - 0.129 x_1^4 - 0.001 x_1^3 + 0.973 x_1^2$ $+0.004 x_1 + e^{x_0} + e^{\cos(2 x_1)} - 2.814$ | $0.01 x_0^4 - 0.02 x_0^2 x_1 + 0.01 x_1^4$ $+0.01 x_2^4 - 0.02 x_2^2 x_3 + 0.01 x_3^4$ |
| TPSR | $-x_0 + (1.009 - 73.098 x_0)$ $(-0.008 x_1 - 0.013) + 0.007$ | $(0.098 + \frac{1}{x_0 + 44.83})(0.217 - 0.991 x_2)(-$ $0.006 x_0 - (-0.303 + \frac{23.261}{x_2 - 0.219})(0.033 x_1 +$ $38.009)(-0.003 x_2 - 0.213)(-0.613 x_2 +$ $(0.499 - 0.007 x_0)(27.627(-0.01 x_0 - 1)^2$ $+7.021) - 17.446) + 111.696 + \frac{0.001}{x_1})$ | $-0.008 x_0(5.0 - 75.398 x_1) + 0.04 x_0 +$ $1.1\cos(8.906(0.168 - 0.253 x_1)$ $(x_0 - 0.667) + 1.567)$ | $-0.182 x_1 + 0.005(1 - 0.015$ $(0.007 - |9.838 x_0 + 0.02|)^2)^2$ $+0.04(0.011 x_0^2 + 0.45 x_2^2$ $-0.425 x_3 + 1)^2 + 0.045$ |
| SeTGAP | $0.607 x_0 x_1 + 1.099\sin($ $(2.242 x_0 - 1.492)$ $(x_1 - 0.685))$ | $-0.5 x_0 + 0.062 x_0^2 +$ $3.162\sqrt{0.1 x_1 + 1}\sin\left(0.201 x_2\right) + 6.501$ | $0.147 e^{1.507 x_0} - 0.5\sin\left(3.001 x_1 + 4.712\right)$ | $-0.02 x_0^2 x_1 + 0.01 x_0^4 + 0.01 x_1^2 +$ $0.01 x_2^4 + 0.01 x_3^2 - (0.02 x_2^2$ $+0.002)(x_3 + 0.212) + 0.009$ |

Table 41: Comparison of predicted equations (E5—E8) with rounded numerical coefficients — Iteration 10

| Method | E5 | E6 | E7 | E8 |
|---|---|---|---|---|
| PySR | $1.0 e^{1.2 x_3} +$ $\sin\left(x_0 + x_1 x_2\right)$ | $|x_1|\cos\left(0.2 x_2^2\right) + \tanh\left(0.5 x_0\right)$ | $\frac{2.119 - 2.118 x_1^2}{2.118\sin\left(6.283 x_0\right) + 3.176}$ | $\log(\tan(1.442(\tanh(\cosh(x_1)\tanh($ $\sqrt{\cosh(x_0)}))\tanh(\cosh(x_0)))^{0.5}) - 0.635)$ |
| TaylorGP | $2 e^{x_3} - 0.24 - \tanh(e^{x_3} +$ $\sin((1.598 - 0.126\sqrt{|x_3|})$ $(e^{x_3} - 0.24)))\sqrt{|x_3 + e^{x_3}|}$ | $\cos\left(x_2\right) + \tanh\left(x_0\right) + \frac{4.95\sin\left(x_2\right)}{x_2}$ | $-x_1^2 + 0.802\sqrt{|x_1^2 + \sin\left(x_0\right)|}$ | $2.037\tanh\left(0.521\sqrt{|x_0||x_1|}\right)$ |
| NeSymReS | — | $-0.368 x_0 + x_1\sin\left(\frac{x_0}{x_1} + 0.005 x_2\right)$ | $\frac{0.166 x_0 + x_1^2}{\cos\left(2.498(0.137 x_1 - 1)^2\right) - 2.073}$ | $\cos\left(\frac{\sin\left(\frac{x_0}{x_1}\right)}{x_0}\right) + 0.64$ |
| E2E | $0.002 x_2 + 0.273 + 0.845 e^{1.25 x_3}$ $e^{0.008\arctan\left(-0.065 x_0 + 0.002 x_1 - 14.76\right)} +$ $0.993\cos(-16.985 x_2 + (-0.008 x_0 -$ $37.102)(67.0|0.925 + \frac{0.008}{-0.573 x_3 - 0.015}|$ $-59.1) + 0.088 + \frac{46.3}{0.012 - 0.001 x_0})$ | $(-0.007 x_1 - 0.015)(9.191\sin((0.021$ $-0.906 x_2)(21.206|3.697 x_0 + 11.042$ $(-x_2 - 0.077)^2 + 0.689| + 0.008))$ $-0.121) - 0.219\arctan(-0.489 x_0$ $-0.094) + 0.566$ | $0.003 x_0 + 0.144 +$ $(0.002 - 0.005(-x_1 - 0.014)^2)$ $/(0.006 - 0.001(-\sin($ $7.343 x_0 + 3.499) - 0.628)^2)$ | $2.0 - 0.9/(0.006|(2.068 x_0 - 0.4)$ $(3.413 x_0 + 0.229)(2.073 x_1 + 0.038)$ $(3.109 x_1 + 0.206)| + 0.5)$ |
| uDSR | $0.001 x_0 x_1 + 0.001 x_0 + 0.001 x_1^2 +$ $0.003 x_1 x_3 + 0.002 x_1 + 0.003 x_2^2 +$ $0.001 x_2 x_3^2 + 0.004 x_2 x_3 + 0.001 x_2 +$ $0.153 x_3^4 + 0.568 x_3^3 + 0.503 x_3^2 +$ $0.613 x_3 + \sin\left(x_0 + x_1 x_2\right) + 1.086$ | $-0.001 x_0^3 - 0.002 x_0^2 - 0.002 x_0 x_1 -$ $0.002 x_0 x_2 + 0.242 x_0 + 0.037 x_1^2 +$ $0.01 x_1 + 0.003 x_2^4 - 0.259 x_2^2 +$ $3\cos\left(x_2\right) - \cos\left(2 x_2\right) + 2.688$ | $0.001 x_0^4 - 0.022 x_0^3 + 0.002 x_0^2 x_1 -$ $0.001 x_0^2 x_1 - 0.028 x_0^2 - 0.001 x_0 x_1^3 -$ $0.007 x_0 x_1^2 + 0.02 x_0 x_1 + 0.321 x_0 +$ $0.002 x_1^4 + 0.005 x_1^3 - 0.933 x_1^2 -$ $0.051 x_1 + e^{\sin\left(6 x_0\right)} - 0.234$ | $0.005 x_0^4 - 0.001 x_0^3 - 0.306 x_0^2 - 0.001 x_0 x_1$ $-0.816 x_0 + 0.008 x_1^4 - 0.18 x_1^2 +$ $\log\left(-x_0 + e^{2 x_0}\right) - \cos\left(x_1\right) + 0.914$ |
| TPSR | $14.264(0.045 e^{x_3} + 0.002\cos($ $7.628 x_2 - 12.078) + 1)^2 - 14.456$ | $1.56(0.05 x_0 + 1)^2 - 0.88$ | $1.0 - 0.89(-x_1 - 0.05)^2$ | $2.015 -$ $\frac{0.793}{0.124(0.001 - x_0)^2(-x_1 - 0.005)^2 + 0.595}$ |
| SeTGAP | $1.002 e^{1.2 x_3} +$ $0.999\sin\left(x_0 + x_1 x_2 - 6.281\right)$ | $0.998\cos\left(0.201 x_2^2 - 18.884\right)|x_1| +$ $1.0\tanh\left(0.501 x_0\right)$ | $\frac{1.005 x_1^2 - 0.873}{(-\sin\left(6.283 x_0 - 6.283\right) - 1.507)}$ $+0.084$ | $-\frac{11.534}{(11.542 x_1^4 + 11.535)} + 2.0 -$ $\frac{11.534}{(0.045 x_0 + 0.062 x_0^2 + 11.474 x_0^4 + 11.528)}$ |

Table 42: Comparison of predicted equations (E9—E13) with rounded numerical coefficients — Iteration 10

| Method | E9 | E10 | E11 | E12 | E13 |
|---|---|---|---|---|---|
| PySR | $\sqrt{1.148x_1} - 4.723 + \frac{4.49}{\cosh\left(x_0 \tanh\left(\frac{2.296}{1.849 - 0.771\cos(x_0)}\right)\right)}$ | $\sin\left(x_0 e^{x_1}\right)$ | $2x_0 \log\left(x_1^2\right)$ | $1.0 x_0 \sin\left(\frac{1.0}{x_1}\right) + 1.0$ | $\log(3.711 \sinh(\sqrt{|\log(|\cosh(0.381x_1)|)|}))$ $\tanh(\sqrt{e^{-2.032\cosh(0.209x_1)}} + 2.45)2.015\sqrt{|x_0|}$ |
| TaylorGP | $\log\left(\sqrt{\left|\frac{x_1}{x_0}\right|}\right) - 1.486\sqrt{|x_0|} + 0.889$ | $\frac{\cos\left(\sqrt{|x_1|}\right)}{\tanh\left(\tanh(x_0)\right)}$ | $4x_0 \log\left(|x_1|\right)$ | $\log\left(e^{\tanh\left(\frac{x_0}{x_1}\right)}\right) + \tanh\left(\frac{x_0}{x_1}\right) + \sqrt{|\tanh(x_1)|}$ | $2\log\left(|x_1|\right)\sqrt{|x_0|}$ |
| NeSymReS | $\log\left(\frac{0.284|x_1|}{|x_0|}\right)$ | $\sin\left(x_0 e^{x_1}\right)$ | $x_0 \log\left(x_1^4\right)$ | $(x_0 + x_1)\sin\left(\frac{1}{x_1}\right)$ | $0.23x_0 + \log\left(x_1^2\right)$ |
| E2E | $-0.986\log(26.495(-0.179 - \frac{1}{0.691x_1+7.596})^2(-0.063 - \frac{1}{0.691x_1+0.626})^2(0.031 - x_0)^2 + 0.224) - 0.087$ | $-1.0\sin(0.026x_0$ $(1.0 - 36.638e^{1.091x_1}$ $)) - 0.001$ | $(-0.214x_0 - 0.001)$ $(6.0\log(1742.4$ $(0.02 - \frac{1}{1.244x_1+0.023})^2$ $+0.01) - 50.0)$ | $0.996 - 7.12\sin((0.01 + \frac{19.5}{0.006x_0 - 9.741x_1 - 0.305})$ $(0.067x_0 + 0.001))$ | $(2.27 - 0.465(0.086(|(92.8$ $\arctan(0.172x_0 + 20.71) + 29.3)$ $(-0.015x_1 + (0.913 + \frac{61.081}{x_1})(0.056$ $x_0 - 2.916) + 0.736)|+0.001)^{0.5}$ $-1)^{0.5})(3.87\log(|21.928|0.027x_0$ $-1|^{0.5} - 26.9|) + 0.05)$ |
| uDSR | $\log\left(\frac{2x_1+1}{4.0x_0^2+1.0}\right)$ | $\sin\left(x_0 e^{x_1}\right)$ | $x_0 \log\left(0.368x_1^4\right) + x_0$ | $x_0 \sin\left(1/x_1\right) + 1.0$ | $(0.003x_0^3 - 0.048x_0^2 + 0.547x_0$ $+0.001x_1 + 0.443)\log\left(x_1^2\right)$ |
| TPSR | $-97.127x_1 - 0.029(-x_0 - 0.024)^2 +$ $(-0.013x_0 + x_1 - 1169.375)(-0.084x_1$ $-0.002\arctan(0.29 - 5.115$ $|0.129x_0 + 0.001|) + 0.118) + 137.562$ | $1.003$ $\sin\left(0.99667x_0 e^{x_1}\right)$ | $(-0.524x_0 - 0.02)$ $(-0.277x_1^2 - 7.971 -$ $\frac{32.033}{-5.252|0.701x_1+0.001|-0.726})$ | $\frac{0.026x_0(-0.068 +}{-0.634(-x_1 - 0.964)^2 - 0.887)}$ $(-36.911\arctan(44.002x_1$ $-16.916) - 34.003)$ $-0.075x_0 + 0.978$ | $(-0.015 + \frac{1}{0.539|9.48x_1+0.014|+0.511})$ $(0.417 - 0.44(0.001 - x_1)^2)$ $(1.325 - \frac{2.356}{x_0+1.916})$ $(-1.316x_0 - 22.185)$ |
| SeTGAP | $-1.0\log\left(10.831x_0^2 + 2.707\right) +$ $1.0|\log\left(21.911x_1 + 10.961\right)| - 1.399$ | $1.0\sin\left(1.0x_0 e^{x_1}\right)$ | $4.0x_0 \log\left(|x_1|\right)$ | $1.0x_0 \sin\left(\frac{1}{x_1}\right) + 1.0$ | $1.0\sqrt{x_0}\log\left(x_1^2\right)$ |

Table 43: Comparison of expressions learned by SeTGAP Under Noisy Conditions

| Problem | $\sigma_a = 0.01$ | $\sigma_a = 0.03$ | $\sigma_a = 0.05$ |
|---|---|---|---|
| E1 | $0.608\,x_0 x_1 + 1.089\sin((2.263$ $x_0 - 1.525)(x_1 - 0.665) - 0.011)$ | $(0.611\,x_0 + 0.001)(x_1 - 0.043) - 0.885$ $\sin((2.253\,x_0 - 1.544)(x_1 - 0.721) - 3.214) + 0.006$ | $0.61\,x_1(x_0 - 0.006) -$ $0.47\sin((2.381\,x_0 - 1.171)(x_1 - 0.798)$ $+8.501) + 0.006$ |
| E2 | $0.063\,x_0^2 - 0.499\,x_0 + (3.078$ $\sqrt{0.098\,x_1 + 1} + 0.062)$ $(\sin(0.199\,x_2) - 0.002) + 6.481$ | $0.063\,x_0^2 - 0.499\,x_0 + (3.222\sqrt{0.099\,x_1 + 1} -$ $0.058)(\sin(0.201\,x_2) - 0.007) + 6.511$ | $(-(0.507\,x_0 - 6.557)(0.181\,x_1 - 18.566) -$ $(0.374\,x_0^2 + 18.893\sin(0.198\,x_2))$ $\log(x_1 + 19.824))/(0.181\,x_1 - 18.566)$ |
| E3 | $0.143e^{1.516\,x_0} + 0.506$ $\sin(2.999\,x_1 + 1.572) + 0.009$ | $0.148e^{1.505\,x_0} - 0.498\sin(3.0\,x_1 - 1.572) + 0.003$ | $0.149e^{1.502\,x_0} -$ $0.496\sin(3.0\,x_1 - 1.581) + 0.003$ |
| E4 | $0.01x_0^4 - 0.02x_0^2x_1 - 0.007x_0^2$ $+0.009x_1^2 + 0.011x_2^4 - 0.02x_2^2x_3$ $-0.02x_2^2 + 0.008x_3^2 + 0.113$ | $0.01x_0^4 + 0.01x_2^4 - 0.02x_2^2x_3 + 0.01x_3^2 +$ $0.242\cosh(10.69\sqrt{1 - 0.016x_1} - 8.186) - 1.473$ | $0.01x_1^4 - 0.021x_3^2x_4 + 0.01x_4^4 +$ $0.011x_4 + 0.01x_4^2 + (0.093x_1^2 + 0.007$ $|x_4 + 1|)\sin(0.248x_2 - 2.981) + 0.124$ |
| E5 | $1.004e^{1.199\,x_3} + 0.997$ $\sin(1.0\,x_0 + 1.0\,x_1 x_2) - 0.005$ | $(-(0.015\sin(0.508\,x_2 + 1.574) - 0.001)(|4.779$ $\sin(1.0\,x_0 + 3.141) - 6.283| - 6.283) +$ $(6.226\sin(0.508\,x_2 + 1.574) - 6.283)(0.002$ $e^{1.202|x_3+5.282|} + 0.145\sin(1.0|x_1 - 6.283| -$ $1.847) + 0.067\sin(1.0|x_1 + 6.202| - 4.425)|2.649$ $\sin(1.0\,x_0 + 3.24) - 2.097| + 0.005))/$ $(6.226\sin(0.508\,x_2 + 1.574) - 6.283)$ | $0.996e^{1.202\,x_3} +$ $1.0\sin(1.0\,x_0 + 1.0\,x_1 x_2) + 0.004$ |
| E6 | $1.0\cos(0.2\,x_2^2 + 0.008)|x_1|$ $+1.0\tanh(0.5\,x_0) + 0.005$ | $\cos(0.199\,x_2^2 + 0.028)(1.001|x_1| - 0.001) +$ $1.002\tanh(0.486\,x_0) + 0.02$ | $1.0\cos(0.2\,x_2^2)|x_1| + 0.997\tanh(0.502\,x_0)$ |
| E7 | $\frac{1.001\,x_1^2 - 0.9914}{\sin(6.283\,x_0 + 3.15) - 1.501} + 0.011$ | $\frac{-1.001\,x_1^2 + 1.006}{\sin(6.283\,x_0) + 1.501} + 0.007$ | $-0.982\frac{x_1^2 - 0.993}{\sin(6.283\,x_0 + 3.139) - 1.492} + 0.019$ |
| E8 | $2.0 - 11.356/(11.313x_1^4 - 0.116x_1^3$ $+0.047x_1^2 + 0.078x_1 + 11.352) -$ $\frac{10.636}{10.804x_0^4 - 0.113x_0^2 + 10.646}$ | $2.0 - \frac{15.531}{15.414x_1^4 - 0.051x_1^3 + 0.165x_1^2 + 0.018x_1 + 15.517} -$ $\frac{12.561}{12.523x_0^4 - 0.085x_0^3 + 0.062x_0^2 + 0.061x_0 + 12.553}$ | $2.0 - \frac{12.837}{12.956x_0^4 + 0.007x_0^3 - 0.125x_0^2 + 12.861} -$ $\frac{16.19}{16.217x_1^4 - 0.007x_1^3 + 16.202}$ |
| E9 | $-1.0\log(4.884\,x_0^2 + 1.22) +$ $1.0|\log(9.415\,x_1 + 4.709)| - 1.35$ | $-1.0\log(17.004\,x_0^2 + 4.251) +$ $0.999|\log(10.158\,x_1 + 5.072)| - 0.176$ | $5.965\sqrt{0.625\log(9.36\,x_1 + 6.35) + 1} -$ $0.999\log(11.259\,x_0^2 + 2.817) - 7.713$ |
| E10 | $1.0\sin(1.0\,x_0 e^{x_1})$ | $1.0\sin(1.0\,x_0 e^{x_1})$ | $1.0\sin(1.0\,x_0 e^{1.0\,x_1})$ |
| E11 | $2.0\,x_0 \log(x_1^2)$ | $(4.001\,x_0 - 0.004)\log(|x_1|) + 0.003$ | $(4.001\,x_0 - 0.008)\log(|x_1|) + 0.006$ |
| E12 | $1.0\,x_0 \sin(1/x_1) + 1.0$ | $1.0\,x_0 \sin(1/x_1) + 1.0$ | $1.0\,x_0 \sin(1/x_1) + 1.0$ |
| E13 | $(1.001\sqrt{x_0} - 0.001)\log(x_1^2)$ | $(0.993\sqrt{x_0} - 0.002)\log(x_1^2) + 0.029$ | $(1.001\sqrt{x_0} - 0.003)\log(x_1^2)$ |

## G  SeTGAP Under Noisy Conditions

This appendix presents the expressions learned by SeTGAP under noisy conditions, shown in Table 43, where shaded cells indicate an incorrectly identified functional form. While the expressions obtained in the noiseless setting are provided in Tables 3–5, here we focus on analyzing the impact of noise on the recovered

functional forms. These expressions correspond to the results reported in Table 8, where different noise levels were introduced to assess the robustness of SeTGAP.

## H  Runtime Analysis

In this appendix, we provide a detailed breakdown of the runtime performance for the main SeTGAP components across the 13 tested synthetic problems. The execution pipeline consists of four primary stages: (1) the initial generation of univariate symbolic skeletons using the Multi-Set Transformer, (2) the subsequent evaluation of these skeletons through a GA, (3) the iterative merging process where symbolic skeletons are combined in a cascade fashion by adding one variable at a time, and (4) a final fitting stage where a GA optimizes the numerical coefficients. The time allocations for each of these phases are reported in Table 44. All experiments were conducted on a High-Performance Computing (HPC) node equipped with an AMD EPYC 7H12 processor with 32 physical cores, 30 GB of RAM, and no GPU acceleration.

The computational cost of generating a single univariate skeleton candidate for a given data collection $\tilde{\mathbf{D}}_v$ (as described in Sec. 3.2.2) requires approximately 1.47 seconds and 392 GFLOPs, and incurs a peak memory footprint of 149.47 MB. In the worst-case scenario, this generation process is repeated $n_{\text{cand}} n_B = 9$ times per variable. Because these transformer inference passes are executed sequentially for each variable, the peak memory footprint remains strictly bounded at 149.47 MB, while the cumulative computational cost scales linearly to 3,533 GFLOPs per variable.

The subsequent skeleton evaluation phase, the computational overhead is highly dynamic and depends on the structural complexity of the generated candidates. As demonstrated by the variance in the second column of Table 44, the runtime of the GA varies based on the number of coefficient placeholders that must be optimized for a given candidate skeleton. For instance, a basic linear skeleton such as $c_1 x + c_2$ requires the GA to fit only two parameters. In contrast, a more complex non-linear skeleton, such as $c_1 \sin(c_2 x + c_3)(c_4 x + c_5) + c_6$, requires optimizing six coefficients, which significantly expands the parameter search space. Therefore, the computational effort and convergence rates of the GA populations vary naturally across iterations, depending on the specific functional relationships and mathematical operators identified for each variable.

Regarding the running time observed during the merging stages, there is a general trend where search time increases alongside the number of variables, as seen when transitioning from two-variable to three-variable merges. This is a reasonable expectation given the expanded search space. However, the actual runtime is heavily influenced by the specific problem structure and the functional topologies discovered by the algorithm. For instance, in problem E5, the time required for merging three variables was actually greater than the time taken to merge four variables. This occurs because there were fewer viable candidate combinations of skeletons when merging four variables than when merging three variables, which reduced the search effort despite the higher dimensionality of the problem. As a consequence, the actual duration and complexity remain highly dependent on the specific skeletons generated for each variable, which, in turn, can vary across iterations depending on the uncertainty of the model $\hat{f}$, and the particular combinations of skeletons selected during each stage of the cascade merging process.

Finally, it is important to emphasize that these reported runtimes are anecdotal and should be regarded as illustrative rather than absolute measures of computational complexity. Beyond the internal algorithmic dependencies noted above, these readings may vary significantly depending on external factors such as system architecture, programming optimizations, and ambient compute loads. Furthermore, the algorithm possesses significant potential for further performance gains. For instance, in Algorithm 3, each subpopulation is evolved independently, making the process highly suitable for advanced parallelization techniques that have not yet been implemented in this version.

## I  Experiments on SRBench++ Functions

In this section, we evaluate SeTGAP on synthetic functions from the SRBench++ benchmark (de Franca et al., 2025). Specifically, we focus on the four functions F1–F4 from the "Rediscovery of Exact Expressions" task, presented in Table 45. Here, SeTGAP follows the same configurations described in Sec. 4.

Table 44: Detailed breakdown of SeTGAP runtime (seconds)

| Eq. | Skel. Gen. | Skel. Eval. | Merge (2 vars.) | Merge (3 vars.) | Merge (4 vars.) | Final Fit | Total (s) |
|---|---|---|---|---|---|---|---|
| E1 | 340.93 | 23,953.48 | 130,419.00 | — | — | 200.00 | 154,913.41 |
| E2 | 127.97 | 5,724.99 | 14,006.93 | 27,223.29 | — | 281.82 | 47,365.00 |
| E3 | 176.59 | 10,743.94 | 37,641.43 | — | — | 681.63 | 49,243.59 |
| E4 | 296.60 | 32,803.91 | 37,165.82 | 83,737.50 | 209,343.75 | 800.00 | 364,147.58 |
| E5 | 453.79 | 25,177.78 | 40,501.43 | 67,372.18 | 34,619.85 | 441.53 | 168,566.56 |
| E6 | 243.33 | 9,922.87 | 45,161.15 | 76,081.50 | — | 485.18 | 131,894.03 |
| E7 | 67.15 | 2,762.79 | 28,586.82 | — | — | 544.68 | 31,961.44 |
| E8 | 220.00 | 8,816.38 | 61,630.77 | — | — | 359.92 | 71,027.07 |
| E9 | 213.27 | 10,005.72 | 73,890.96 | — | — | 406.63 | 84,516.58 |
| E10 | 137.23 | 13,527.42 | 35,391.66 | — | — | 748.78 | 49,805.09 |
| E11 | 224.76 | 16,344.42 | 23,427.18 | — | — | 480.56 | 40,476.92 |
| E12 | 302.95 | 18,109.08 | 48,321.79 | — | — | 477.04 | 67,210.86 |
| E13 | 183.43 | 15,028.31 | 22,734.28 | — | — | 466.16 | 38,412.18 |

Table 45: SRBench++ synthetic equations

| Eq. | Underlying equation | Domain range |
|---|---|---|
| F1 | $0.4x_0x_1 - 1.5x_0 + 2.5x_1 + 1$ | $[-5,5]^2$ |
| F2 | $0.4x_0x_1 - 1.5x_0 + 2.5x_1 + 1 + +\log(30x_2^2)$ | $[-5,5]^3$ |
| F3 | $\frac{0.4x_0x_1 - 1.5x_0 + 2.5x_1 + 1}{0.2(x_0^2 + x_1^2) + 1}$ | $[-20,20]^2$ |
| F4 | $\frac{0.4x_0x_1 - 1.5x_0 + 2.5x_1 + 1 + 5.5\sin(x_0 + x_1)}{0.2(x_0^2 + x_1^2) + 1}$ | $[-20,20]^2$ |

Table 46: Expressions learned by SeTGAP on SRBench++ functions

| It. | F1 | F2 | F3 | F4 |
|---|---|---|---|---|
| 1 | $0.403x_0x_1$ $-1.504x_0$ $+2.5x_1 + 1.0$ | $0.401x_0x_1 - 1.5x_0+$ $2.5x_1 + 0.994\log\left(x_2^2\right)$ $+4.411$ | $\frac{(16.43x_0 + 101.498)(x_1 - 3.751) + 420.837}{8.23x_0^2 + 8.145x_1^2 + 40.317}$ | $\frac{37.563(-0.232x_0 - 1.467)(x_1 - 3.255) - 192.406}{-0.519x_0 - 4.32x_1^2 + \left(0.354x_0^2 + 0.224\right)(-0.127x_1 - 13.04) - 10.342}$ $+0.023$ |
| 2 | $0.403x_0x_1$ $-1.501x_0$ $+2.5x_1 + 1.0$ | $0.4x_0x_1 - 1.5x_0+$ $2.501x_1 + 0.994\log\left(x_2^2\right)$ $+4.415$ | $\frac{(5.225x_0 + 32.182)(x_1 - 3.679) + 131.061}{2.599x_0^2 + 2.591x_1^2 + 12.653}$ | $\frac{45.613\cdot(0.73x_0 + 4.5)(x_1 - 3.22) + 713.964}{16.491x_1^2 + \left(0.339x_0^2 + 0.744\right)(0.556x_1 + 51.947) + 11.18}$ $+0.027$ |
| 3 | $0.402x_0x_1$ $-1.5x_0$ $+2.5x_1 + 1.0$ | $0.4x_0x_1 - 1.5x_0+$ $2.5x_1 + 1.999\log\left(|x_2|\right)$ $+4.402$ | $\frac{(3.518x_0 + 22.14)(x_1 - 3.804) + 92.972}{1.755x_0^2 + 1.771x_1^2 + 8.998}$ | $0.019-$ $\frac{14.122(-2.3x_0 - 14.317)(x_1 - 3.208) - 716.019}{16.077x_1^2 + \left(0.578x_0^2 + 1.851\right)(0.251x_1 + 29.437)}$ |
| 4 | $0.4x_0x_1$ $-1.5x_0+$ $2.5x_1 + 1.0$ | $0.4x_0x_1 - 1.5x_0+$ $2.5x_1 + 1.006\log\left(x_2^2\right)$ $+4.39$ | $\frac{-20.847(-0.437x_0 - 2.699)(x_1 - 3.68) + 229.698}{4.523x_0^2 + 4.524x_1^2 + 22.284}$ | $0.25-$ $\frac{50.0}{-3.823x_0^2 - 4.555x_1^2 + (8.072x_0 + 66.932)(x_1 - 6.951) - 327.87}$ |
| 5 | $0.4x_0x_1$ $-1.5x_0+$ $2.5x_1 + 1.0$ | $0.4x_0x_1 - 1.5x_0+$ $2.5x_1 + 0.995\log\left(x_2^2\right)$ $+4.411$ | $\frac{8.168(0.507x_0 + 3.191)(x_1 - 3.779) + 109.02}{2.076x_0^2 + 2.078x_1^2 + 10.51}$ | $0.256-$ $\frac{50.0}{-3.411x_0^2 - 4.472x_1^2 + (7.662x_0 + 64.337)(x_1 - 6.769) - 332.132}$ |
| 6 | $0.4x_0x_1$ $-1.5x_0+$ $2.5x_1 + 1.0$ | $0.4x_0x_1 - 1.5x_0+$ $2.5x_1 + 1.987\log\left(|x_2|\right)$ $+4.411$ | $\frac{4.819(2.48x_0 + 15.215)(x_1 - 3.665) + 298.03}{5.929x_0^2 + 5.917x_1^2 + 28.938}$ | $0.254-$ $\frac{50.0}{-3.174x_0^2 - 3.987x_1^2 + (7.231x_0 + 60.398)(x_1 - 6.93) - 355.795}$ |
| 7 | $0.4x_0x_1$ $-1.5x_0$ $+2.5x_1 + 1.0$ | $0.4x_0x_1 - 1.5x_0+$ $2.5x_1 + 1.004\log\left(x_2^2\right)$ $+4.395$ | $\frac{-6.653(0.808x_0 + 5.044)(x_1 - 3.708) - 137.9}{-2.675x_0^2 - 2.687x_1^2 - 13.337}$ | $0.26+$ $\frac{50.0}{4.443x_0^2 + 5.439x_1^2 - (9.653x_0 + 79.204)(x_1 - 6.977) + 362.483}$ |
| 8 | $0.4x_0x_1$ $-1.5x_0$ $+2.5x_1 + 1.0$ | $0.4x_0x_1 - 1.5x_0+$ $2.501x_1 + 0.993\log\left(x_2^2\right)$ $+4.413$ | $\frac{29.656(0.745x_0 + 4.661)(x_1 - 3.756) + 574.829}{11.069x_0^2 + 11.043x_1^2 + 55.471}$ | $0.236+$ $\frac{50.0}{2.314x_0^2 + 2.99x_1^2 - (5.12x_0 + 43.79)(x_1 - 6.84) + 277.643}$ |
| 9 | $0.4x_0x_1$ $-1.5x_0$ $+2.5x_1 + 1.0$ | $0.4x_0x_1 - 1.501x_0+$ $2.5x_1 + 1.975\log\left(|x_2|\right)$ $+4.424$ | $\frac{6.325(2.199x_0 + 13.381)(x_1 - 3.567) + 335.564}{6.827x_0^2 + 6.865x_1^2 + 32.789}$ | $0.274-$ $\frac{50.0}{-6.504x_0^2 - 8.41x_1^2 + (14.504x_0 + 119.585)(x_1 - 6.792) - 474.742}$ |
| 10 | $0.4x_0x_1$ $-1.5x_0$ $+2.5x_1 + 1.0$ | $0.4x_0x_1 - 1.5x_0+$ $2.499x_1 + 0.995\log\left(x_2^2\right)$ $+4.41$ | $\frac{34.748(-0.581x_0 - 3.638)(x_1 - 3.715) - 519.981}{-10.04x_0^2 - 10.11x_1^2 - 50.332}$ | $\frac{42.93(-0.59x_0 - 3.686)(x_1 - 3.267) - 562.393}{-12.503x_1^2 + \left(0.977x_0^2 + 2.913\right)(-0.118x_1 - 13.727) + 1.471}$ $+0.021$ |

Table 46 shows the expressions learned, where shaded cells indicate an incorrectly identified functional form. The evaluation was repeated 10 times, each on a newly generated dataset and with random seeds. Results indicate that SeTGAP identifies the correct functional form of problems F1–F3 across all iterations, while it

Table 47: Extrapolation MSE on SRBench++ functions

| F1 | F2 | F3 | F4 |
|---|---|---|---|
| 6.480e-03 ± 1.055e-02 | 5.949e-04 ± 5.012e-04 | 4.759e-05 ± 7.400e-05 | 6.172e-01 ± 4.915e-01 |

Table 48: Selected problems from the Feynman dataset

| Problem | Underlying Expression | Problem | Underlying Expression | Problem | Underlying Expression |
|---|---|---|---|---|---|
| I.6.2a | $\frac{\sqrt{2}e^{-\frac{x_0^2}{2}}}{2\sqrt{\pi}}$ | I.8.14 | $\sqrt{(-x_0+x_1)^2+(-x_2+x_3)^2}$ | I.9.18 | $\frac{x_0 x_3 x_4}{(-x_3+x_4)^2+(-x_5+x_6)^2+(-x_7+x_8)^2}$ |
| I.11.19 | $x_0 x_3 + x_1 x_4 + x_2 x_5$ | I.12.1 | $x_0 x_1$ | I.12.2 | $\frac{x_0 x_1}{4\pi x_2 x_3^2}$ |
| I.12.4 | $\frac{x_0}{4\pi x_1 x_2^2}$ | I.12.5 | $x_0 x_1$ | I.12.11 | $x_0\left(x_1 + x_2 x_3 \sin(x_4)\right)$ |
| I.13.4 | $\frac{x_0\left(x_1^2+x_2^2+x_3^2\right)}{2}$ | I.13.12 | $x_0 x_1 x_4 \cdot \left(\frac{1}{x_3}-\frac{1}{x_2}\right)$ | I.14.3 | $x_0 x_1 x_2$ |
| I.14.4 | $\frac{x_0 x_1^2}{2}$ | I.18.4 | $\frac{x_0 x_2 + x_1 x_3}{x_0 + x_1}$ | I.18.12 | $x_0 x_1 \sin(x_2)$ |
| I.18.14 | $x_0 x_1 x_2 \sin(x_3)$ | I.24.6 | $\frac{x_0 x_3^2\left(x_1^2+x_2^2\right)}{4}$ | I.25.13 | $\frac{x_0}{x_1}$ |
| I.29.4 | $\frac{x_0}{x_1}$ | I.29.16 | $\sqrt{x_0^2 - 2x_0 x_1 \cos(x_2-x_3)+x_1^2}$ | I.30.3 | $\frac{x_0 \sin^2\left(\frac{x_1 x_2}{2}\right)}{\sin^2\left(\frac{x_1}{2}\right)}$ |
| I.32.5 | $\frac{x_0^2 x_1^2}{6\pi x_2 x_3^3}$ | I.34.8 | $\frac{x_0 x_1 x_2}{x_3}$ | I.34.27 | $\frac{x_0 x_1}{2\pi}$ |
| I.37.4 | $x_0 + x_1 + 2\sqrt{x_0 x_1}\cos(x_2)$ | I.38.12 | $\frac{x_2^2 x_3}{\pi x_0 x_1^2}$ | I.39.1 | $\frac{3 x_0 x_1}{2}$ |
| I.39.11 | $\frac{x_1 x_2}{x_0 - 1}$ | I.39.22 | $\frac{x_0 x_1 x_3}{x_2}$ | I.43.16 | $\frac{x_0 x_1 x_2}{x_3}$ |
| I.43.31 | $x_0 x_1 x_2$ | I.43.43 | $\frac{x_1 x_3}{x_2(x_0-1)}$ | I.44.4 | $x_0 x_1 x_2 \log\left(\frac{x_4}{x_3}\right)$ |
| I.47.23 | $\sqrt{\frac{x_0 x_1}{x_2}}$ | I.50.26 | $x_0\left(x_3\cos^2(x_1 x_2)+\cos(x_1 x_2)\right)$ | II.2.42 | $\frac{x_0 x_3(-x_1+x_2)}{x_4}$ |
| II.3.24 | $\frac{x_0}{4\pi x_1^2}$ | II.4.23 | $\frac{x_0}{4\pi x_1 x_2}$ | II.6.11 | $\frac{x_1\cos(x_2)}{4\pi x_0 x_2^2}$ |
| II.6.15b | $\frac{3x_1\sin(x_2)\cos(x_2)}{4\pi x_0 x_3^3}$ | II.8.7 | $\frac{3x_0^2}{20\pi x_1 x_2}$ | II.8.31 | $\frac{x_0 x_1^2}{2}$ |
| II.10.9 | $\frac{x_0}{x_1(x_2+1)}$ | II.11.3 | $\frac{x_0 x_1}{x_2\left(x_3^2-x_4^2\right)}$ | II.11.7 | $x_0 \cdot \left(1+\frac{x_4 x_5 \cos(x_3)}{x_1 x_2}\right)$ |
| II.11.20 | $\frac{x_0 x_1^2 x_2}{3 x_3 x_4}$ | II.11.27 | $\frac{x_0 x_1 x_2 x_3}{-\frac{x_0 x_1}{3}+1}$ | II.11.28 | $\frac{x_0 x_1}{-\frac{x_0 x_1}{3}+1}+1$ |
| II.13.17 | $\frac{x_2}{2\pi x_0 x_1^2 x_3}$ | II.15.4 | $x_0 x_1 \cos(x_2)$ | II.15.5 | $x_0 x_1 \cos(x_2)$ |
| II.24.17 | $\sqrt{\frac{x_0^2}{x_1^2}-\frac{\pi^2}{x_2^2}}$ | II.27.16 | $x_0 x_1 x_2^2$ | II.27.18 | $x_0 x_1^2$ |
| II.34.2a | $\frac{x_0 x_1}{2\pi x_2}$ | II.34.2 | $\frac{x_0 x_1 x_2}{2}$ | II.34.11 | $\frac{x_0 x_1 x_2}{2 x_3}$ |
| II.34.29a | $\frac{x_0 x_1}{4\pi x_2}$ | II.34.29b | $\frac{2\pi x_0 x_2 x_3 x_4}{x_1}$ | II.35.21 | $x_0 x_1 \tanh\left(\frac{x_1 x_2}{x_3 x_4}\right)$ |
| II.36.38 | $\frac{x_0 x_1}{x_2 x_3}+\frac{x_0 x_4 x_7}{x_2 x_3 x_5 x_6^2}$ | II.37.1 | $x_0 x_1(x_2+1)$ | II.38.3 | $\frac{x_0 x_1 x_3}{x_2}$ |
| II.38.14 | $\frac{x_0}{2 x_1+2}$ | III.4.32 | $\frac{1}{e^{\frac{x_0 x_1}{2\pi x_2 x_3}}-1}$ | III.4.33 | $\frac{x_0 x_1}{2\pi\left(e^{\frac{x_0 x_1}{2\pi x_2 x_3}}-1\right)}$ |
| III.7.38 | $\frac{4\pi x_0 x_1}{x_2}$ | III.8.54 | $\sin^2\left(\frac{2\pi x_0 x_1}{x_2}\right)$ | III.9.52 | $\frac{8\pi x_0 x_1 \sin^2\left(\frac{x_2(x_4-x_5)}{2}\right)}{x_2 x_3(x_4-x_5)^2}$ |
| III.10.19 | $x_0\sqrt{x_1^2+x_2^2+x_3^2}$ | III.12.43 | $\frac{x_0 x_1}{2\pi}$ | III.13.18 | $\frac{4\pi x_0 x_1^2 x_2}{x_3}$ |
| III.15.12 | $2x_0 \cdot (1-\cos(x_1 x_2))$ | III.15.14 | $\frac{x_0^2}{8\pi^2 x_1 x_2^2}$ | III.15.27 | $\frac{2\pi x_0}{x_1 x_2}$ |
| III.17.37 | $x_0(x_1\cos(x_2)+1)$ | III.19.51 | $-\frac{x_0 x_1^4}{8 x_2^2 x_3^2 x_4^2}$ | III.21.20 | $\frac{x_0 x_1 x_2}{x_3}$ |

fails to recover the functional form of problem F4. This outcome is expected, as the corresponding univariate skeleton with respect to variable $x_0$ is $\mathbf{e}(x_0) = (cx_0 + \sin(x_0 + c))/(cx_0^2 + c)$, which involves eight operators, including three unary operators (`sin`, `sqr`, and `inv`). Such complexity exceeds the representational capacity of our approach, since the Multi-Set Transformer used in the experiments was trained only on expressions with up to seven operators, at most two unary operators, and up to one nested unary operator [Ref. 1][2]. Finally, we assessed the extrapolation capability of the learned expressions by evaluating them on an extended domain. As in Sec.4, the extrapolation range was defined as twice the size of the original domain, excluding the original interval. Each extrapolation set consisted of 10,000 points sampled within this range. Table 47 reports the mean and standard deviation of the extrapolation MSE.

## J  Experiments on the Feynman Dataset

This section presents the experiments conducted on the Feynman dataset from the SRBench benchmark. The set of Feynman problems considered in this study is listed in Table 48, selected according to the criteria described in Sec. 4.2. SeTGAP is evaluated against a suite of benchmark methods.

Table 49: Mean and standard deviation of $R^2$ for each Feynman problem (Part 1 of 2)

| Problem | AFP | AFP_FE | AIFeynman | BSR | DSR | EPLEX | FEAT | FFX | GP-GOMEA | gplearn | ITEA | MRGP | Operon | SBP-GP | SeTGAP |
|---|---|---|---|---|---|---|---|---|---|---|---|---|---|---|---|
| I.6.2a | 1.000 ± 0.000 | 1.000 ± 0.000 | 1.000 ± 0.000 | 0.991 ± 0.015 | 0.998 ± 0.003 | 1.000 ± 0.000 | 1.000 ± 0.000 | 0.999 ± 0.000 | 1.000 ± 0.000 | 0.987 ± 0.015 | 1.000 ± 0.000 | 1.000 ± 0.000 | 1.000 ± 0.000 | 1.000 ± 0.000 | 1.000 ± 0.000 |
| I.8.14 | 0.834 ± 0.183 | 0.953 ± 0.044 | 0.998 ± 0.006 | 0.593 ± 0.215 | 0.265 ± 0.071 | 0.975 ± 0.027 | 0.987 ± 0.012 | 0.861 ± 0.091 | 0.964 ± 0.012 | 0.779 ± 0.207 | 0.491 ± 0.002 | 1.000 ± 0.000 | 0.989 ± 0.026 | 0.995 ± 0.004 | 1.000 ± 0.001 |
| I.9.18 | 0.859 ± 0.052 | 0.964 ± 0.017 | 0.955 ± 0.010 | 0.605 ± 0.191 | 0.598 ± 0.109 | 0.980 ± 0.008 | 0.922 ± 0.013 | 0.992 ± 0.001 | 0.986 ± 0.010 | 0.752 ± 0.096 | 0.988 ± 0.000 | 0.999 ± 0.001 | 0.996 ± 0.002 | 0.996 ± 0.001 | 0.993 ± 0.005 |
| I.11.19 | 0.966 ± 0.021 | 0.987 ± 0.009 | 1.000 ± 0.000 | 0.348 ± 0.298 | 0.787 ± 0.191 | 0.993 ± 0.007 | 1.000 ± 0.000 | 0.996 ± 0.000 | 0.997 ± 0.005 | 0.985 ± 0.032 | 0.957 ± 0.001 | 1.000 ± 0.000 | 1.000 ± 0.000 | 1.000 ± 0.001 | 1.000 ± 0.000 |
| I.12.1 | 1.000 ± 0.000 | 1.000 ± 0.000 | 1.000 ± 0.000 | 1.000 ± 0.000 | 0.988 ± 0.033 | 1.000 ± 0.000 | 1.000 ± 0.000 | 0.998 ± 0.000 | 1.000 ± 0.000 | 1.000 ± 0.000 | 1.000 ± 0.000 | 1.000 ± 0.000 | 1.000 ± 0.000 | 1.000 ± 0.000 | 1.000 ± 0.000 |
| I.12.2 | 0.984 ± 0.015 | 0.999 ± 0.002 | 1.000 ± 0.000 | 0.465 ± 0.214 | 0.883 ± 0.176 | 0.997 ± 0.005 | 0.957 ± 0.001 | 0.951 ± 0.001 | 1.000 ± 0.000 | 0.751 ± 0.163 | 0.972 ± 0.001 | 1.000 ± 0.001 | 1.000 ± 0.000 | 1.000 ± 0.000 | 0.994 ± 0.000 |
| I.12.4 | 0.999 ± 0.001 | 0.999 ± 0.001 | 1.000 ± 0.000 | 0.423 ± 0.279 | 0.933 ± 0.181 | 0.999 ± 0.001 | 0.994 ± 0.006 | 0.979 ± 0.001 | 1.000 ± 0.000 | 0.254 ± 0.151 | 0.970 ± 0.001 | 1.000 ± 0.001 | 1.000 ± 0.000 | 1.000 ± 0.000 | 0.961 ± 0.062 |
| I.12.5 | 1.000 ± 0.000 | 1.000 ± 0.000 | 1.000 ± 0.000 | 0.866 ± 0.351 | 0.988 ± 0.035 | 1.000 ± 0.000 | 1.000 ± 0.000 | 0.998 ± 0.000 | 1.000 ± 0.000 | 1.000 ± 0.000 | 1.000 ± 0.000 | 1.000 ± 0.000 | 1.000 ± 0.000 | 1.000 ± 0.000 | 1.000 ± 0.000 |
| I.12.11 | 0.988 ± 0.012 | 1.000 ± 0.000 | 1.000 ± 0.000 | 0.415 ± 0.344 | 0.960 ± 0.112 | 0.995 ± 0.007 | 0.995 ± 0.004 | 0.856 ± 0.045 | 1.000 ± 0.000 | 0.987 ± 0.019 | 0.919 ± 0.001 | 1.000 ± 0.000 | 1.000 ± 0.000 | 1.000 ± 0.000 | 1.000 ± 0.000 |
| I.13.4 | 0.976 ± 0.007 | 0.993 ± 0.003 | 1.000 ± 0.000 | 0.390 ± 0.337 | 0.918 ± 0.079 | 0.995 ± 0.005 | 1.000 ± 0.000 | 0.997 ± 0.000 | 0.996 ± 0.004 | 0.992 ± 0.004 | 0.957 ± 0.000 | 1.000 ± 0.000 | 1.000 ± 0.000 | 1.000 ± 0.000 | 1.000 ± 0.000 |
| I.13.12 | 0.949 ± 0.030 | 0.980 ± 0.017 | 1.000 ± 0.000 | 0.403 ± 0.196 | 0.740 ± 0.095 | 0.998 ± 0.004 | 0.937 ± 0.027 | 0.654 ± 0.003 | 1.000 ± 0.000 | 0.963 ± 0.062 | 1.000 ± 0.000 | 1.000 ± 0.000 | 1.000 ± 0.000 | 1.000 ± 0.000 | 0.996 ± 0.007 |
| I.14.3 | 1.000 ± 0.000 | 1.000 ± 0.000 | 1.000 ± 0.000 | 0.847 ± 0.186 | 0.969 ± 0.089 | 1.000 ± 0.000 | 1.000 ± 0.000 | 0.994 ± 0.000 | 1.000 ± 0.000 | 1.000 ± 0.000 | 1.000 ± 0.000 | 1.000 ± 0.000 | 1.000 ± 0.000 | 1.000 ± 0.000 | 1.000 ± 0.000 |
| I.14.4 | 1.000 ± 0.000 | 1.000 ± 0.000 | 1.000 ± 0.000 | 0.999 ± 0.002 | 0.974 ± 0.052 | 0.999 ± 0.003 | 1.000 ± 0.000 | 0.998 ± 0.000 | 1.000 ± 0.000 | 0.999 ± 0.001 | 0.979 ± 0.000 | 1.000 ± 0.000 | 1.000 ± 0.000 | 1.000 ± 0.000 | 1.000 ± 0.000 |
| I.18.4 | 0.945 ± 0.021 | 0.959 ± 0.023 | 1.000 ± 0.000 | 0.968 ± 0.008 | 0.869 ± 0.029 | 0.988 ± 0.010 | 1.000 ± 0.001 | 0.987 ± 0.000 | 0.996 ± 0.004 | 0.893 ± 0.028 | 0.929 ± 0.004 | 1.000 ± 0.000 | 1.000 ± 0.000 | 1.000 ± 0.000 | 0.975 ± 0.004 |
| I.18.12 | 1.000 ± 0.000 | 1.000 ± 0.000 | 1.000 ± 0.000 | 0.510 ± 0.322 | 0.965 ± 0.099 | 1.000 ± 0.000 | 0.997 ± 0.004 | 0.929 ± 0.031 | 1.000 ± 0.000 | 1.000 ± 0.000 | 0.785 ± 0.001 | 1.000 ± 0.000 | 1.000 ± 0.000 | 1.000 ± 0.000 | 1.000 ± 0.000 |
| I.18.14 | 1.000 ± 0.000 | 1.000 ± 0.000 | 1.000 ± 0.000 | 0.229 ± 0.281 | 0.964 ± 0.102 | 0.999 ± 0.001 | 0.987 ± 0.017 | 0.768 ± 0.090 | 1.000 ± 0.000 | 1.000 ± 0.000 | 0.950 ± 0.001 | 1.000 ± 0.000 | 1.000 ± 0.000 | 1.000 ± 0.000 | 1.000 ± 0.000 |
| I.24.6 | 0.986 ± 0.006 | 0.994 ± 0.005 | 1.000 ± 0.000 | 0.588 ± 0.305 | 0.914 ± 0.138 | 0.998 ± 0.002 | 0.999 ± 0.000 | 0.993 ± 0.000 | 0.999 ± 0.001 | 0.987 ± 0.010 | 0.970 ± 0.001 | 1.000 ± 0.000 | 1.000 ± 0.000 | 1.000 ± 0.000 | 1.000 ± 0.000 |
| I.25.13 | 1.000 ± 0.000 | 1.000 ± 0.000 | 1.000 ± 0.000 | 0.995 ± 0.005 | 0.977 ± 0.064 | 1.000 ± 0.000 | 1.000 ± 0.000 | 0.997 ± 0.000 | 1.000 ± 0.000 | 1.000 ± 0.000 | 1.000 ± 0.000 | 1.000 ± 0.000 | 1.000 ± 0.000 | 1.000 ± 0.000 | 1.000 ± 0.000 |
| I.29.4 | 1.000 ± 0.000 | 1.000 ± 0.000 | 1.000 ± 0.000 | 0.992 ± 0.013 | 0.967 ± 0.092 | 1.000 ± 0.000 | 1.000 ± 0.001 | 0.995 ± 0.000 | 1.000 ± 0.000 | 1.000 ± 0.000 | 1.000 ± 0.000 | 1.000 ± 0.000 | 1.000 ± 0.000 | 1.000 ± 0.000 | 1.000 ± 0.000 |
| I.29.16 | 0.917 ± 0.041 | 0.951 ± 0.024 | 0.930 ± 0.001 | 0.246 ± 0.081 | 0.553 ± 0.073 | 0.964 ± 0.016 | 0.987 ± 0.017 | 0.797 ± 0.008 | 0.961 ± 0.009 | 0.891 ± 0.037 | 0.514 ± 0.007 | 1.000 ± 0.000 | 0.983 ± 0.025 | 0.997 ± 0.002 | 0.599 ± 0.088 |
| I.30.3 | 0.801 ± 0.125 | 0.919 ± 0.132 | 0.900 ± 0.316 | 0.443 ± 0.262 | 0.492 ± 0.120 | 0.853 ± 0.346 | 0.449 ± 0.192 | 0.585 ± 0.032 | 0.987 ± 0.016 | 0.783 ± 0.153 | 0.724 ± 0.006 | 1.000 ± 0.000 | 0.844 ± 0.133 | 0.998 ± 0.001 | 0.839 ± 0.064 |
| I.32.5 | 0.992 ± 0.005 | 0.998 ± 0.002 | 1.000 ± 0.000 | 0.719 ± 0.176 | 0.831 ± 0.233 | 0.996 ± 0.007 | 0.827 ± 0.337 | 0.830 ± 0.009 | 1.000 ± 0.000 | 0.970 ± 0.047 | 0.797 ± 0.005 | 0.998 ± 0.003 | 1.000 ± 0.000 | 0.998 ± 0.002 | 0.997 ± 0.002 |
| I.34.8 | 0.999 ± 0.002 | 1.000 ± 0.000 | 1.000 ± 0.000 | 0.406 ± 0.315 | 0.935 ± 0.184 | 0.996 ± 0.007 | 0.995 ± 0.004 | 0.977 ± 0.000 | 0.998 ± 0.005 | 0.953 ± 0.133 | 1.000 ± 0.000 | 1.000 ± 0.000 | 1.000 ± 0.000 | 1.000 ± 0.000 | 1.000 ± 0.000 |
| I.34.27 | 1.000 ± 0.000 | 1.000 ± 0.000 | 1.000 ± 0.000 | 1.000 ± 0.000 | 0.959 ± 0.023 | 1.000 ± 0.001 | 1.000 ± 0.000 | 0.998 ± 0.000 | 1.000 ± 0.000 | 0.988 ± 0.014 | 1.000 ± 0.000 | 1.000 ± 0.000 | 1.000 ± 0.000 | 1.000 ± 0.000 | 1.000 ± 0.000 |
| I.37.4 | 0.975 ± 0.027 | 0.983 ± 0.034 | 0.999 ± 0.000 | 0.950 ± 0.025 | 0.941 ± 0.060 | 0.992 ± 0.005 | 0.999 ± 0.002 | 0.994 ± 0.006 | 0.998 ± 0.001 | 0.840 ± 0.340 | 0.919 ± 0.001 | 1.000 ± 0.000 | 0.999 ± 0.001 | 1.000 ± 0.000 | 1.000 ± 0.000 |
| I.38.12 | 0.880 ± 0.302 | 0.951 ± 0.151 | 1.000 ± 0.000 | 0.719 ± 0.178 | 0.911 ± 0.190 | 0.995 ± 0.013 | 0.825 ± 0.337 | 0.906 ± 0.016 | 0.998 ± 0.006 | 0.905 ± 0.263 | 0.920 ± 0.002 | 0.999 ± 0.001 | 1.000 ± 0.000 | 0.999 ± 0.001 | 0.998 ± 0.001 |
| I.39.1 | 1.000 ± 0.000 | 1.000 ± 0.000 | 1.000 ± 0.000 | 0.980 ± 0.057 | 0.987 ± 0.034 | 1.000 ± 0.000 | 1.000 ± 0.000 | 0.998 ± 0.000 | 1.000 ± 0.000 | 0.996 ± 0.010 | 1.000 ± 0.000 | 1.000 ± 0.000 | 1.000 ± 0.000 | 1.000 ± 0.000 | 1.000 ± 0.000 |
| I.39.11 | 0.996 ± 0.011 | 1.000 ± 0.000 | 1.000 ± 0.000 | 0.855 ± 0.347 | 0.959 ± 0.115 | 0.996 ± 0.006 | 1.000 ± 0.000 | 0.993 ± 0.000 | 1.000 ± 0.001 | 0.982 ± 0.049 | 0.994 ± 0.000 | 1.000 ± 0.000 | 1.000 ± 0.000 | 1.000 ± 0.000 | 1.000 ± 0.000 |
| I.39.22 | 0.988 ± 0.023 | 1.000 ± 0.000 | 1.000 ± 0.000 | 0.437 ± 0.303 | 0.932 ± 0.192 | 0.996 ± 0.009 | 0.994 ± 0.003 | 0.977 ± 0.000 | 0.998 ± 0.005 | 0.972 ± 0.069 | 1.000 ± 0.000 | 1.000 ± 0.000 | 1.000 ± 0.000 | 1.000 ± 0.000 | 1.000 ± 0.000 |
| I.43.16 | 0.993 ± 0.021 | 1.000 ± 0.000 | 1.000 ± 0.000 | 0.510 ± 0.225 | 0.932 ± 0.192 | 0.987 ± 0.016 | 0.991 ± 0.010 | 0.977 ± 0.000 | 0.997 ± 0.009 | 0.954 ± 0.128 | 1.000 ± 0.000 | 1.000 ± 0.000 | 1.000 ± 0.000 | 1.000 ± 0.000 | 1.000 ± 0.000 |
| I.43.31 | 1.000 ± 0.001 | 1.000 ± 0.000 | 1.000 ± 0.000 | 0.877 ± 0.163 | 0.968 ± 0.090 | 1.000 ± 0.001 | 1.000 ± 0.000 | 0.994 ± 0.001 | 1.000 ± 0.001 | 0.981 ± 0.054 | 1.000 ± 0.000 | 1.000 ± 0.000 | 1.000 ± 0.000 | 1.000 ± 0.000 | 1.000 ± 0.000 |
| I.43.43 | 0.992 ± 0.015 | 1.000 ± 0.001 | 1.000 ± 0.000 | 0.839 ± 0.138 | 0.911 ± 0.218 | 0.989 ± 0.016 | 0.985 ± 0.011 | 0.975 ± 0.001 | 0.998 ± 0.005 | 0.948 ± 0.139 | 0.995 ± 0.000 | 1.000 ± 0.000 | 1.000 ± 0.000 | 1.000 ± 0.000 | 1.000 ± 0.000 |
| I.44.4 | 0.956 ± 0.041 | 0.992 ± 0.017 | 1.000 ± 0.000 | 0.422 ± 0.224 | 0.906 ± 0.127 | 0.998 ± 0.002 | 0.979 ± 0.019 | 0.630 ± 0.067 | 0.999 ± 0.004 | 0.933 ± 0.136 | 0.991 ± 0.000 | 1.000 ± 0.000 | 1.000 ± 0.000 | 1.000 ± 0.000 | 1.000 ± 0.000 |
| I.47.23 | 0.998 ± 0.004 | 0.991 ± 0.029 | 1.000 ± 0.000 | 0.907 ± 0.213 | 0.895 ± 0.055 | 0.997 ± 0.003 | 0.990 ± 0.026 | 0.996 ± 0.000 | 0.999 ± 0.002 | 0.980 ± 0.038 | 1.000 ± 0.000 | 1.000 ± 0.001 | 1.000 ± 0.000 | 1.000 ± 0.000 | 0.999 ± 0.001 |
| I.50.26 | 0.894 ± 0.069 | 0.966 ± 0.021 | 1.000 ± 0.000 | 0.331 ± 0.019 | 0.537 ± 0.109 | 0.938 ± 0.076 | 0.587 ± 0.199 | 0.751 ± 0.128 | 0.959 ± 0.070 | 0.822 ± 0.215 | 0.413 ± 0.005 | 1.000 ± 0.000 | 0.951 ± 0.070 | 0.994 ± 0.010 | 0.773 ± 0.047 |
| II.2.42 | 0.930 ± 0.101 | 0.998 ± 0.005 | 1.000 ± 0.000 | 0.273 ± 0.196 | 0.911 ± 0.147 | 0.995 ± 0.010 | 0.979 ± 0.029 | 0.613 ± 0.004 | 1.000 ± 0.000 | 0.910 ± 0.140 | 1.000 ± 0.000 | 1.000 ± 0.000 | 1.000 ± 0.000 | 1.000 ± 0.000 | 1.000 ± 0.000 |
| II.3.24 | 1.000 ± 0.001 | 1.000 ± 0.000 | 1.000 ± 0.000 | 0.712 ± 0.163 | 0.966 ± 0.096 | 1.000 ± 0.000 | 0.999 ± 0.002 | 0.996 ± 0.000 | 1.000 ± 0.000 | 0.913 ± 0.114 | 0.967 ± 0.000 | 1.000 ± 0.000 | 1.000 ± 0.000 | 1.000 ± 0.000 | 1.000 ± 0.000 |
| II.4.23 | 1.000 ± 0.000 | 0.999 ± 0.001 | 1.000 ± 0.000 | 0.562 ± 0.149 | 0.939 ± 0.157 | 0.998 ± 0.003 | 0.972 ± 0.040 | 0.988 ± 0.000 | 1.000 ± 0.000 | 0.598 ± 0.344 | 1.000 ± 0.000 | 1.000 ± 0.000 | 1.000 ± 0.000 | 1.000 ± 0.000 | 1.000 ± 0.000 |
| II.6.11 | 0.959 ± 0.040 | 0.993 ± 0.015 | 1.000 ± 0.000 | 0.407 ± 0.269 | 0.785 ± 0.094 | 0.998 ± 0.001 | 0.976 ± 0.025 | 0.938 ± 0.019 | 1.000 ± 0.000 | 0.135 ± 0.257 | 0.909 ± 0.001 | 1.000 ± 0.000 | 1.000 ± 0.000 | 0.999 ± 0.001 | 0.993 ± 0.001 |
| II.6.15b | 0.954 ± 0.060 | 0.990 ± 0.012 | 1.000 ± 0.000 | 0.200 ± 0.167 | 0.634 ± 0.116 | 0.996 ± 0.005 | 0.860 ± 0.120 | 0.886 ± 0.012 | 1.000 ± 0.000 | 0.000 ± 0.000 | 0.708 ± 0.016 | 0.999 ± 0.002 | 1.000 ± 0.000 | 0.992 ± 0.004 | 0.809 ± 0.071 |
| II.8.7 | 0.998 ± 0.004 | 1.000 ± 0.000 | 1.000 ± 0.000 | 0.664 ± 0.094 | 0.920 ± 0.119 | 0.998 ± 0.005 | 0.955 ± 0.112 | 0.977 ± 0.001 | 1.000 ± 0.000 | 0.977 ± 0.038 | 0.983 ± 0.000 | 1.000 ± 0.000 | 1.000 ± 0.000 | 1.000 ± 0.000 | 1.000 ± 0.000 |
| II.8.31 | 0.998 ± 0.004 | 1.000 ± 0.000 | 1.000 ± 0.000 | 0.983 ± 0.048 | 0.977 ± 0.040 | 1.000 ± 0.000 | 1.000 ± 0.000 | 0.998 ± 0.000 | 1.000 ± 0.000 | 0.999 ± 0.001 | 0.980 ± 0.000 | 0.999 ± 0.001 | 1.000 ± 0.000 | 1.000 ± 0.000 | 1.000 ± 0.000 |
| II.10.9 | 0.999 ± 0.002 | 0.999 ± 0.002 | 1.000 ± 0.000 | 0.882 ± 0.042 | 0.962 ± 0.106 | 0.999 ± 0.001 | 0.995 ± 0.006 | 0.992 ± 0.000 | 1.000 ± 0.000 | 0.981 ± 0.041 | 0.995 ± 0.000 | 1.000 ± 0.001 | 1.000 ± 0.000 | 1.000 ± 0.000 | 1.000 ± 0.000 |
| II.11.3 | 0.982 ± 0.011 | 0.996 ± 0.003 | 0.995 ± 0.002 | 0.759 ± 0.051 | 0.863 ± 0.138 | 0.991 ± 0.010 | 0.996 ± 0.002 | 0.987 ± 0.002 | 1.000 ± 0.000 | 0.892 ± 0.059 | 0.966 ± 0.001 | 1.000 ± 0.000 | 1.000 ± 0.000 | 0.999 ± 0.001 | 0.995 ± 0.003 |
| II.11.20 | 0.993 ± 0.008 | 0.999 ± 0.001 | 1.000 ± 0.000 | 0.308 ± 0.163 | 0.909 ± 0.245 | 0.998 ± 0.004 | 0.948 ± 0.035 | 0.920 ± 0.007 | 1.000 ± 0.000 | 0.976 ± 0.062 | 0.986 ± 0.000 | 0.998 ± 0.002 | 1.000 ± 0.000 | 0.999 ± 0.001 | 1.000 ± 0.000 |
| II.11.27 | 0.999 ± 0.001 | 1.000 ± 0.000 | 1.000 ± 0.000 | 0.976 ± 0.020 | 0.970 ± 0.080 | 1.000 ± 0.000 | 0.999 ± 0.000 | 0.987 ± 0.000 | 1.000 ± 0.000 | 0.997 ± 0.003 | 0.998 ± 0.000 | 1.000 ± 0.000 | 1.000 ± 0.000 | 1.000 ± 0.000 | 0.991 ± 0.021 |
| II.11.28 | 1.000 ± 0.000 | 1.000 ± 0.000 | 1.000 ± 0.000 | 0.991 ± 0.007 | 0.997 ± 0.000 | 1.000 ± 0.000 | 1.000 ± 0.000 | 0.997 ± 0.000 | 1.000 ± 0.000 | 0.994 ± 0.012 | 1.000 ± 0.000 | 1.000 ± 0.000 | 1.000 ± 0.000 | 1.000 ± 0.000 | 0.999 ± 0.001 |

Table 50: Mean and standard deviation of $R^2$ for each Feynman problem (Part 2 of 2)

| Problem | AFP | AFP_FE | AIFeynman | BSR | DSR | EPLEX | FEAT | FFX | GP-GOMEA | gplearn | ITEA | MRGP | Operon | SBP-GP | SeTGAP |
|---|---|---|---|---|---|---|---|---|---|---|---|---|---|---|---|
| II.13.17 | 0.996 ± 0.006 | 0.997 ± 0.007 | 1.000 ± 0.000 | 0.205 ± 0.238 | 0.910 ± 0.215 | 0.998 ± 0.002 | 0.830 ± 0.337 | 0.936 ± 0.006 | 1.000 ± 0.000 | 0.044 ± 0.063 | 0.973 ± 0.001 | 0.999 ± 0.001 | 1.000 ± 0.000 | 0.999 ± 0.000 | 0.999 ± 0.001 |
| II.15.4 | 0.999 ± 0.001 | 1.000 ± 0.000 | 1.000 ± 0.000 | 0.421 ± 0.280 | 0.956 ± 0.125 | 0.999 ± 0.002 | 0.996 ± 0.004 | 0.987 ± 0.000 | 1.000 ± 0.000 | 0.953 ± 0.094 | 0.806 ± 0.002 | 1.000 ± 0.000 | 1.000 ± 0.000 | 1.000 ± 0.000 | 1.000 ± 0.000 |
| II.15.5 | 0.999 ± 0.003 | 1.000 ± 0.000 | 1.000 ± 0.000 | 0.794 ± 0.109 | 0.955 ± 0.127 | 1.000 ± 0.000 | 0.995 ± 0.008 | 0.986 ± 0.000 | 1.000 ± 0.000 | 0.945 ± 0.084 | 0.804 ± 0.001 | 1.000 ± 0.000 | 0.999 ± 0.003 | 1.000 ± 0.000 | 1.000 ± 0.000 |
| II.24.17 | 0.999 ± 0.002 | 0.998 ± 0.004 | 1.000 ± 0.000 | 0.994 ± 0.003 | 0.984 ± 0.019 | 0.998 ± 0.005 | 1.000 ± 0.000 | 0.998 ± 0.000 | 1.000 ± 0.000 | 0.993 ± 0.006 | 1.000 ± 0.000 | 1.000 ± 0.000 | 1.000 ± 0.000 | 1.000 ± 0.000 | 1.000 ± 0.000 |
| II.27.16 | 1.000 ± 0.000 | 1.000 ± 0.000 | 1.000 ± 0.000 | 0.638 ± 0.400 | 0.966 ± 0.095 | 1.000 ± 0.001 | 1.000 ± 0.000 | 0.989 ± 0.000 | 1.000 ± 0.000 | 1.000 ± 0.000 | 0.981 ± 0.000 | 1.000 ± 0.000 | 1.000 ± 0.000 | 1.000 ± 0.000 | 1.000 ± 0.000 |
| II.27.18 | 1.000 ± 0.000 | 1.000 ± 0.000 | 1.000 ± 0.000 | 0.969 ± 0.059 | 0.986 ± 0.039 | 1.000 ± 0.000 | 1.000 ± 0.000 | 0.998 ± 0.000 | 1.000 ± 0.000 | 1.000 ± 0.000 | 0.980 ± 0.000 | 1.000 ± 0.000 | 1.000 ± 0.000 | 1.000 ± 0.000 | 1.000 ± 0.000 |
| II.34.2 | 1.000 ± 0.000 | 1.000 ± 0.000 | 1.000 ± 0.000 | 0.821 ± 0.296 | 0.969 ± 0.087 | 0.999 ± 0.002 | 1.000 ± 0.000 | 0.994 ± 0.000 | 1.000 ± 0.000 | 0.998 ± 0.002 | 1.000 ± 0.000 | 1.000 ± 0.000 | 1.000 ± 0.000 | 1.000 ± 0.000 | 1.000 ± 0.000 |
| II.34.2a | 1.000 ± 0.001 | 1.000 ± 0.000 | 1.000 ± 0.000 | 0.964 ± 0.026 | 0.931 ± 0.118 | 0.999 ± 0.003 | 0.998 ± 0.001 | 0.992 ± 0.000 | 1.000 ± 0.000 | 0.953 ± 0.089 | 1.000 ± 0.000 | 1.000 ± 0.000 | 1.000 ± 0.000 | 1.000 ± 0.000 | 1.000 ± 0.000 |
| II.34.11 | 0.998 ± 0.003 | 1.000 ± 0.000 | 1.000 ± 0.000 | 0.519 ± 0.308 | 0.921 ± 0.200 | 0.998 ± 0.003 | 0.994 ± 0.005 | 0.977 ± 0.000 | 1.000 ± 0.000 | 0.992 ± 0.007 | 1.000 ± 0.000 | 1.000 ± 0.000 | 1.000 ± 0.000 | 1.000 ± 0.000 | 1.000 ± 0.000 |
| II.34.29a | 1.000 ± 0.000 | 1.000 ± 0.000 | 1.000 ± 0.000 | 0.867 ± 0.093 | 0.935 ± 0.109 | 0.998 ± 0.004 | 0.998 ± 0.002 | 0.991 ± 0.000 | 1.000 ± 0.000 | 0.983 ± 0.009 | 1.000 ± 0.000 | 1.000 ± 0.000 | 1.000 ± 0.000 | 1.000 ± 0.000 | 1.000 ± 0.000 |
| II.34.29b | 0.981 ± 0.013 | 0.996 ± 0.014 | 0.900 ± 0.316 | 0.338 ± 0.224 | 0.869 ± 0.235 | 0.990 ± 0.008 | 0.926 ± 0.094 | 0.957 ± 0.001 | 1.000 ± 0.000 | 0.997 ± 0.005 | 1.000 ± 0.000 | 1.000 ± 0.000 | 1.000 ± 0.000 | 1.000 ± 0.000 | 1.000 ± 0.000 |
| II.35.21 | 0.965 ± 0.027 | 0.985 ± 0.008 | 0.998 ± 0.000 | 0.750 ± 0.315 | 0.857 ± 0.055 | 0.991 ± 0.006 | 0.978 ± 0.009 | 0.986 ± 0.001 | 0.989 ± 0.004 | 0.977 ± 0.015 | 0.971 ± 0.000 | 1.000 ± 0.000 | 0.997 ± 0.001 | 0.996 ± 0.001 | 0.981 ± 0.010 |
| II.36.38 | 0.917 ± 0.041 | 0.948 ± 0.038 | 0.994 ± 0.001 | 0.839 ± 0.045 | 0.662 ± 0.110 | 0.976 ± 0.019 | 0.912 ± 0.062 | 0.962 ± 0.007 | 0.988 ± 0.005 | 0.946 ± 0.059 | 0.976 ± 0.000 | 0.999 ± 0.001 | 0.997 ± 0.002 | 0.990 ± 0.004 | 0.989 ± 0.002 |
| II.37.1 | 1.000 ± 0.000 | 1.000 ± 0.000 | 1.000 ± 0.000 | 0.801 ± 0.339 | 0.972 ± 0.079 | 0.999 ± 0.002 | 1.000 ± 0.000 | 0.996 ± 0.000 | 1.000 ± 0.000 | 1.000 ± 0.001 | 1.000 ± 0.000 | 1.000 ± 0.000 | 1.000 ± 0.000 | 1.000 ± 0.000 | 1.000 ± 0.000 |
| II.38.3 | 0.999 ± 0.004 | 1.000 ± 0.000 | 1.000 ± 0.000 | 0.621 ± 0.110 | 0.924 ± 0.214 | 0.997 ± 0.004 | 0.988 ± 0.015 | 0.977 ± 0.000 | 1.000 ± 0.000 | 0.980 ± 0.041 | 1.000 ± 0.000 | 1.000 ± 0.000 | 1.000 ± 0.000 | 1.000 ± 0.000 | 1.000 ± 0.000 |
| II.38.14 | 1.000 ± 0.001 | 1.000 ± 0.000 | 1.000 ± 0.000 | 0.951 ± 0.021 | 0.986 ± 0.041 | 0.998 ± 0.003 | 0.999 ± 0.004 | 0.997 ± 0.000 | 1.000 ± 0.000 | 0.981 ± 0.018 | 0.992 ± 0.000 | 1.000 ± 0.000 | 1.000 ± 0.000 | 1.000 ± 0.000 | 1.000 ± 0.000 |
| III.4.32 | 0.994 ± 0.014 | 0.997 ± 0.010 | 1.000 ± 0.000 | 0.536 ± 0.174 | 0.896 ± 0.200 | 0.999 ± 0.003 | 0.992 ± 0.004 | 0.972 ± 0.001 | 1.000 ± 0.000 | 0.995 ± 0.007 | 1.000 ± 0.000 | 1.000 ± 0.000 | 1.000 ± 0.000 | 1.000 ± 0.000 | 1.000 ± 0.000 |
| III.4.33 | 0.999 ± 0.002 | 1.000 ± 0.001 | 1.000 ± 0.000 | 1.000 ± 0.000 | 0.983 ± 0.033 | 1.000 ± 0.000 | 1.000 ± 0.000 | 0.998 ± 0.000 | 1.000 ± 0.000 | 0.998 ± 0.002 | 1.000 ± 0.000 | 1.000 ± 0.000 | 1.000 ± 0.000 | 1.000 ± 0.000 | 1.000 ± 0.000 |
| III.7.38 | 0.999 ± 0.002 | 1.000 ± 0.000 | 1.000 ± 0.000 | 0.612 ± 0.328 | 0.946 ± 0.114 | 0.992 ± 0.016 | 0.987 ± 0.034 | 0.992 ± 0.000 | 1.000 ± 0.000 | 1.000 ± 0.000 | 1.000 ± 0.000 | 0.999 ± 0.001 | 1.000 ± 0.000 | 1.000 ± 0.000 | 1.000 ± 0.000 |
| III.8.54 | 0.259 ± 0.127 | 0.476 ± 0.164 | — | 0.055 ± 0.022 | 0.030 ± 0.010 | 0.773 ± 0.186 | 0.009 ± 0.005 | 0.039 ± 0.011 | 0.988 ± 0.017 | 0.000 ± 0.000 | 0.082 ± 0.003 | 0.962 ± 0.053 | 0.392 ± 0.442 | 0.579 ± 0.098 | 0.641 ± 0.186 |
| III.9.52 | 0.683 ± 0.208 | 0.939 ± 0.032 | 0.000 ± 0.000 | 0.181 ± 0.117 | 0.315 ± 0.014 | 0.842 ± 0.341 | 0.750 ± 0.166 | 0.595 ± 0.020 | 0.964 ± 0.017 | 0.937 ± 0.048 | 0.505 ± 0.004 | 0.999 ± 0.001 | 0.983 ± 0.020 | 0.987 ± 0.005 | 0.833 ± 0.065 |
| III.10.19 | 0.982 ± 0.017 | 0.995 ± 0.005 | 1.000 ± 0.000 | 0.938 ± 0.068 | 0.935 ± 0.056 | 0.994 ± 0.003 | 1.000 ± 0.000 | 0.997 ± 0.000 | 0.998 ± 0.001 | 0.984 ± 0.011 | 0.968 ± 0.000 | 1.000 ± 0.000 | 1.000 ± 0.000 | 1.000 ± 0.000 | 0.997 ± 0.002 |
| III.12.43 | 1.000 ± 0.000 | 1.000 ± 0.000 | 1.000 ± 0.000 | 0.996 ± 0.008 | 0.959 ± 0.023 | 0.999 ± 0.002 | 1.000 ± 0.000 | 0.998 ± 0.000 | 1.000 ± 0.000 | 0.988 ± 0.012 | 1.000 ± 0.000 | 1.000 ± 0.000 | 1.000 ± 0.000 | 1.000 ± 0.000 | 1.000 ± 0.000 |
| III.13.18 | 0.984 ± 0.013 | 0.997 ± 0.006 | 0.875 ± 0.354 | 0.516 ± 0.225 | 0.905 ± 0.183 | 0.994 ± 0.007 | 0.983 ± 0.017 | 0.963 ± 0.001 | 1.000 ± 0.000 | 0.999 ± 0.002 | 0.984 ± 0.000 | 1.000 ± 0.000 | 1.000 ± 0.000 | 1.000 ± 0.000 | 1.000 ± 0.000 |
| III.15.12 | 0.866 ± 0.188 | 1.000 ± 0.001 | 1.000 ± 0.000 | 0.233 ± 0.010 | 0.592 ± 0.229 | 0.994 ± 0.014 | 0.392 ± 0.200 | 0.349 ± 0.016 | 1.000 ± 0.000 | 0.927 ± 0.091 | 0.899 ± 0.001 | 1.000 ± 0.000 | 0.919 ± 0.187 | 1.000 ± 0.000 | 1.000 ± 0.000 |
| III.15.14 | 0.983 ± 0.036 | 0.999 ± 0.001 | 1.000 ± 0.000 | 0.329 ± 0.225 | 0.882 ± 0.149 | 0.997 ± 0.001 | 0.963 ± 0.044 | 0.954 ± 0.001 | 1.000 ± 0.000 | 0.034 ± 0.080 | 0.956 ± 0.003 | 0.994 ± 0.012 | 1.000 ± 0.000 | 1.000 ± 0.000 | 0.983 ± 0.008 |
| III.15.27 | 0.998 ± 0.005 | 1.000 ± 0.000 | 1.000 ± 0.000 | 0.814 ± 0.334 | 0.798 ± 0.080 | 0.997 ± 0.005 | 0.996 ± 0.002 | 0.988 ± 0.001 | 1.000 ± 0.000 | 0.998 ± 0.002 | 1.000 ± 0.000 | 1.000 ± 0.000 | 1.000 ± 0.000 | 1.000 ± 0.000 | 1.000 ± 0.000 |
| III.17.37 | 0.986 ± 0.026 | 1.000 ± 0.000 | 1.000 ± 0.000 | 0.568 ± 0.227 | 0.965 ± 0.099 | 0.995 ± 0.012 | 0.828 ± 0.406 | 0.961 ± 0.038 | 1.000 ± 0.000 | 0.990 ± 0.013 | 0.766 ± 0.001 | 1.000 ± 0.000 | 1.000 ± 0.000 | 1.000 ± 0.000 | 1.000 ± 0.001 |
| III.19.51 | 0.967 ± 0.025 | 0.989 ± 0.017 | 0.000 ± 0.000 | 0.092 ± 0.072 | 0.337 ± 0.185 | 0.978 ± 0.018 | 0.566 ± 0.323 | 0.511 ± 0.020 | 1.000 ± 0.000 | 0.932 ± 0.167 | 0.784 ± 0.064 | 0.983 ± 0.023 | 1.000 ± 0.001 | 0.993 ± 0.008 | 0.904 ± 0.074 |
| III.21.20 | 0.998 ± 0.004 | 1.000 ± 0.000 | 1.000 ± 0.000 | 0.559 ± 0.259 | 0.935 ± 0.167 | 0.997 ± 0.006 | 0.990 ± 0.015 | 0.977 ± 0.001 | 1.000 ± 0.000 | 0.995 ± 0.007 | 1.000 ± 0.000 | 1.000 ± 0.000 | 1.000 ± 0.000 | 1.000 ± 0.000 | 1.000 ± 0.000 |

Tables 49 and 50 report the average $R^2$ obtained by each method across the 10 data partitions recommended by SRBench, together with the corresponding standard error. Bold entries denote methods whose performance is either the lowest or statistically comparable to the lowest according to Tukey's Honestly Significant Difference (HSD) test at the 0.05 significance level. Tables 51 and 52 present the average MSE and associated standard deviation under the same evaluation protocol.

Considering that the obtained MSE distributions often exhibit strong skewness, heavy tails, and differences in variance across methods, the assumptions underlying ANOVA and Tukey's HSD may be violated. For this reason, statistical comparisons for the MSE tables are performed using a nonparametric two-step procedure. First, a Kruskal–Wallis test is applied to determine whether the distributions of MSE across algorithms are statistically indistinguishable for a given dataset. If the null hypothesis of equal distributions is rejected at the 0.05 significance level, Dunn's post hoc test is performed to conduct pairwise comparisons between the best-performing algorithm and the remaining methods. The resulting $p$-values are adjusted using the Holm procedure to control the family-wise error rate under multiple comparisons. Bold entries in the MSE tables therefore indicate algorithms whose MSE is either the lowest or not significantly worse than the lowest according to this Kruskal–Wallis and Dunn testing procedure with Holm correction.

Table 51: Mean and standard deviation of MSE for each Feynman problem (Part 1 of 2)

| Prob. | AFP | AFP_FE | AIFeynman | BSR | DSR | EPLEX | FEAT | FFX | GP-GOMEA | gplearn | ITEA | MRGP | Operon | SBP-GP | SeTGAP |
|---|---|---|---|---|---|---|---|---|---|---|---|---|---|---|---|
| I.6.2a | 6.57e-07 ± 4.61e-07 | 6.17e-07 ± 3.98e-07 | **1.30e-26 ± 1.17e-28** | 4.63e-05 ± 7.47e-05 | 9.83e-06 ± 1.29e-05 | 7.60e-07 ± 1.42e-07 | **8.08e-08 ± 9.73e-08** | 5.36e-06 ± 1.23e-07 | **5.21e-09 ± 5.27e-09** | 6.53e-05 ± 7.13e-05 | 2.79e-07 ± 2.65e-09 | 5.49e-07 ± 9.81e-07 | **3.13e-11 ± 4.93e-11** | **4.55e-12 ± 8.16e-12** | **4.47e-09 ± 1.41e-08** |
| I.8.14 | 1.65e-01 ± 1.83e-01 | **4.63e-02 ± 4.41e-02** | **1.81e-03 ± 5.43e-03** | 4.03e-01 ± 2.13e-01 | 7.27e-01 ± 7.29e-02 | **2.52e-02 ± 2.69e-02** | **1.26e-02 ± 1.18e-02** | 1.37e-01 ± 8.90e-02 | **3.56e-02 ± 1.20e-02** | 2.18e-01 ± 2.05e-01 | 5.03e-01 ± 2.32e-03 | **1.23e-04 ± 1.13e-04** | **1.04e-02 ± 2.54e-02** | **4.71e-03 ± 4.04e-03** | **5.38e-04 ± 1.16e-03** |
| I.9.18 | 2.12e-03 ± 7.77e-04 | 5.46e-04 ± 2.51e-04 | 6.82e-04 ± 1.51e-04 | 5.98e-03 ± 2.96e-03 | 6.06e-03 ± 1.67e-03 | **2.99e-04 ± 1.27e-04** | 1.18e-03 ± 2.07e-04 | **1.18e-04 ± 1.23e-05** | **2.07e-04 ± 1.52e-04** | 3.73e-03 ± 1.40e-03 | **1.85e-04 ± 4.27e-06** | **1.02e-05 ± 1.02e-05** | **5.26e-05 ± 2.28e-05** | **6.69e-05 ± 2.27e-05** | **1.62e-02 ± 4.40e-02** |
| I.11.19 | 2.61 ± 1.64 | **9.78e-01 ± 7.03e-01** | **5.45e-05 ± 1.09e-04** | 78.47 ± 83.67 | 16.40 ± 14.79 | **5.07e-01 ± 5.02e-01** | **1.00e-12 ± 3.70e-13** | **3.10e-01 ± 2.95e-02** | **2.31e-01 ± 4.21e-01** | 1.18 ± 2.50 | 3.31 ± 6.25e-02 | **3.79e-03 ± 1.09e-03** | **9.06e-10 ± 1.99e-09** | **2.89e-02 ± 4.74e-02** | **3.86e-03 ± 8.05e-03** |
| I.12.1 | **4.51e-30 ± 8.83e-32** | **4.64e-30 ± 4.74e-31** | **4.50e-30 ± 8.83e-32** | 1.77e-04 ± 4.65e-04 | **3.05e-01 ± 8.63e-01** | **1.80e-10 ± 5.10e-10** | **2.25e-13 ± 6.03e-14** | 5.47e-02 ± 1.30e-03 | **4.90e-30 ± 3.29e-31** | **4.51e-30 ± 8.83e-32** | **3.73e-29 ± 3.56e-29** | 9.84e-04 ± 2.44e-04 | **2.17e-10 ± 5.25e-10** | **4.97e-30 ± 5.28e-31** | **1.22e-08 ± 1.57e-08** |
| I.12.2 | **1.28e-04 ± 1.22e-04** | **6.27e-06 ± 1.61e-05** | **4.91e-27 ± 1.64e-28** | 5.59e-02 ± 1.48e-01 | 9.27e-04 ± 1.40e-03 | **2.63e-05 ± 4.43e-05** | 3.41e-04 ± 4.09e-04 | 3.90e-04 ± 1.80e-05 | **8.84e-20 ± 2.42e-19** | 1.94e-03 ± 5.03e-06 | 2.23e-04 ± 5.03e-06 | **3.30e-06 ± 5.37e-06** | **4.92e-11 ± 7.95e-11** | **3.34e-06 ± 2.70e-06** | 1.16e-04 ± 8.16e-05 |
| I.12.4 | **3.95e-07 ± 4.16e-07** | **5.50e-07 ± 3.81e-07** | **4.79e-28 ± 9.46e-30** | 4.86e-04 ± 3.29e-04 | 4.69e-05 ± 1.26e-04 | **8.94e-07 ± 9.17e-07** | 4.23e-06 ± 4.36e-06 | 1.54e-05 ± 4.85e-07 | **1.01e-21 ± 2.86e-21** | 5.69e-04 ± 1.68e-04 | 2.20e-05 ± 4.22e-07 | **2.55e-07 ± 4.86e-07** | **9.46e-12 ± 1.78e-11** | **8.21e-08 ± 1.35e-07** | 7.47e-05 ± 1.15e-04 |
| I.12.5 | **4.53e-30 ± 6.49e-32** | **4.45e-10 ± 1.41e-09** | **4.52e-30 ± 6.03e-32** | 10.73 ± 29.65 | **3.14e-01 ± 8.88e-01** | **4.80e-19 ± 1.36e-18** | **2.64e-13 ± 1.35e-13** | 5.46e-02 ± 6.47e-04 | **4.71e-30 ± 2.14e-31** | **4.53e-30 ± 6.49e-32** | **5.33e-29 ± 4.01e-29** | 1.02e-03 ± 2.90e-08 | 1.13e-08 ± 3.47e-04 | **4.89e-30 ± 1.65e-31** | **2.70e-06 ± 3.89e-06** |
| I.12.11 | 7.99 ± 7.80 | **5.91e-02 ± 1.87e-01** | **9.72e-29 ± 1.90e-30** | 406.33 ± 249.34 | **26.59 ± 75.22** | 3.27 ± 4.49 | 3.49 ± 2.76 | 97.47 ± 30.51 | **9.89e-08 ± 1.94e-07** | 8.68 ± 12.64 | 54.52 ± 7.18e-01 | **3.46e-02 ± 1.09e-02** | **6.58e-04 ± 1.96e-03** | **2.18e-01 ± 2.22e-01** | **2.52e-01 ± 5.75e-01** |
| I.13.4 | 16.42 ± 4.97 | 4.83 ± 1.78 | **1.58e-28 ± 1.82e-30** | 969.19 ± 1.19e+03 | 56.94 ± 3.26 | 3.82 ± **5.96e-03** | **9.06e-03 ± 5.96e-03** | 2.44 ± 2.48e-01 | 2.60 ± 2.47 | 5.34 ± 3.00 | 30.15 ± 4.45e-01 | **4.00e-02 ± 1.75e-02** | **3.77e-02 ± 1.12e-01** | **9.90e-02 ± 9.90e-02** | **5.59e-01 ± 9.30e-01** |
| I.13.12 | 4.28 ± 2.49 | 1.68 ± 1.47 | **1.06e-29 ± 3.68e-31** | 50.01 ± 16.02 | 21.78 ± 7.98 | **1.93e-01 ± 3.61e-01** | 5.26 ± 2.27 | 29.02 ± 8.57e-01 | **5.45e-12 ± 1.27e-11** | 3.07 ± 5.13 | **1.27e-28 ± 1.54e-28** | **1.05e-02 ± 1.39e-02** | **1.97e-04 ± 5.98e-04** | **1.22e-02 ± 1.36e-02** | 6.93e-01 ± 1.10 |
| I.14.3 | **7.11e-08 ± 2.01e-07** | **6.47e-29 ± 5.78e-30** | **6.03e-29 ± 9.71e-31** | 57.33 ± 69.62 | 11.73 ± 33.19 | **5.11e-02 ± 1.45e-01** | **3.15e-12 ± 9.25e-13** | 2.40 ± 2.48e-02 | **6.81e-29 ± 7.59e-30** | **6.05e-29 ± 1.05e-30** | **2.77e-28 ± 1.88e-28** | 1.64e-02 ± 2.48e-03 | **6.91e-07 ± 1.43e-06** | **6.85e-29 ± 3.10e-30** | **3.23e-04 ± 3.22e-03** |
| I.14.4 | **1.64e-03 ± 4.64e-03** | **6.11e-03 ± 1.84e-02** | **3.29e-29 ± 5.91e-31** | **1.89e-01 ± 4.05e-01** | 4.24 ± 8.35 | 1.69e-01 ± 4.33e-01 | **1.94e-12 ± 6.56e-13** | 3.71e-01 ± 1.27e-02 | **3.67e-29 ± 2.13e-30** | 1.32e-01 ± 1.06e-01 | 3.34 ± 2.24e-02 | **6.44e-03 ± 1.33e-03** | **4.25e-08 ± 1.27e-07** | **3.72e-29 ± 1.63e-30** | **1.84e-03 ± 1.93e-03** |
| I.18.4 | 3.99e-02 ± 1.50e-02 | 2.94e-02 ± 1.66e-02 | **3.04e-31 ± 3.86e-33** | 2.32e-02 ± 6.11e-03 | 9.48e-02 ± 2.07e-02 | **8.88e-03 ± 7.32e-03** | **2.61e-04 ± 5.50e-04** | **9.50e-03 ± 1.62e-04** | **3.08e-03 ± 2.97e-03** | 5.16e-02 ± 2.01e-02 | 5.16e-02 ± 2.64e-03 | **3.64e-05 ± 9.12e-06** | **5.43e-07 ± 8.88e-07** | **2.36e-04 ± 2.88e-04** | 3.54e-02 ± 5.53e-03 |
| I.18.12 | **1.00e-07 ± 2.83e-07** | **1.27e-08 ± 2.61e-08** | **6.59e-30 ± 7.16e-32** | 27.31 ± 18.00 | **1.92 ± 5.44** | **6.70e-04 ± 1.39e-03** | 1.91e-01 ± 2.39e-01 | 3.98 ± 1.74 | **7.02e-30 ± 2.41e-31** | **6.71e-30 ± 1.54e-31** | 11.95 ± 8.03e-02 | 2.75e-03 ± 1.37e-03 | **1.01e-08 ± 3.18e-08** | **7.12e-30 ± 2.21e-31** | **1.87e-04 ± 3.17e-04** |
| I.18.14 | **1.13e-07 ± 3.13e-07** | **3.14e-07 ± 4.44e-07** | **9.11e-29 ± 2.45e-30** | 503.15 ± 180.51 | 24.15 ± 68.29 | **3.35e-01 ± 6.35e-01** | 8.36 ± 10.81 | 152.17 ± 60.69 | **9.71e-29 ± 2.65e-30** | **2.58e-02 ± 7.30e-02** | 32.39 ± 4.21e-01 | 1.08e-01 ± 2.15e-01 | **8.65e-06 ± 1.47e-05** | **8.42e-07 ± 2.38e-06** | **3.77e-03 ± 4.29e-03** |
| I.24.6 | 2.89 ± 1.32 | 1.22 ± 1.13 | **1.47e-04 ± 4.41e-04** | 125.87 ± 165.85 | 17.95 ± 28.58 | **4.97e-01 ± 3.76e-01** | **1.20e-01 ± 7.13e-02** | 1.39 ± 8.94e-02 | **1.11e-01 ± 1.08e-01** | 2.65 ± 2.16 | 6.42 ± 1.00e-01 | **9.72e-03 ± 1.99e-03** | **1.73e-04 ± 5.42e-04** | **1.16e-02 ± 1.22e-02** | **8.97e-02 ± 1.65e-01** |
| I.25.13 | **7.66e-32 ± 1.21e-32** | **7.25e-32 ± 1.42e-33** | **7.25e-32 ± 1.42e-33** | 3.41e-03 ± 3.26e-03 | **1.42e-02 ± 4.02e-02** | **5.21e-06 ± 1.47e-05** | 3.17e-05 ± 4.11e-05 | 1.83e-03 ± 1.87e-04 | **7.43e-27 ± 2.10e-26** | **7.24e-32 ± 1.54e-33** | **9.31e-31 ± 1.80e-30** | 2.73e-05 ± 1.04e-05 | **2.75e-10 ± 8.68e-10** | **1.57e-10 ± 4.43e-10** | **6.13e-04 ± 6.53e-04** |
| I.29.4 | **9.68e-09 ± 2.74e-08** | **3.97e-11 ± 1.26e-10** | **1.34e-31 ± 2.71e-33** | 1.35e-02 ± 2.38e-02 | **5.83e-02 ± 1.65e-01** | **1.34e-31 ± 3.00e-33** | 5.82e-04 ± 9.47e-04 | 8.18e-03 ± 5.97e-04 | **1.88e-31 ± 5.28e-32** | **1.34e-31 ± 3.00e-33** | **6.42e-31 ± 5.68e-31** | 6.96e-05 ± 2.24e-05 | **2.38e-09 ± 6.89e-09** | **9.30e-20 ± 2.63e-19** | **6.81e-05 ± 2.11e-04** |
| I.29.16 | 3.06e-01 ± 1.53e-01 | **1.82e-01 ± 8.80e-02** | 2.57e-01 ± 3.74e-03 | 2.78 ± 2.96e-01 | 1.65 ± 2.79e-01 | **1.32e-01 ± 5.83e-02** | **4.77e-02 ± 2.83e-02** | 7.48e-01 ± 6.17e-02 | **1.43e-01 ± 3.45e-02** | 4.01e-01 ± 1.34e-01 | 1.79 ± 2.78e-02 | **1.01e-03 ± 1.22e-03** | **6.35e-02 ± 9.18e-02** | **1.23e-02 ± 6.16e-03** | 2.36 ± 4.22e-01 |
| I.30.3 | **1.32 ± 8.18e-01** | **5.37e-01 ± 8.77e-01** | **1.09 ± 3.45** | 4.43 ± 3.55 | 3.37 ± 7.96e-01 | **3.78 ± 10.23** | 35.95 ± 92.55 | 2.76 ± 2.26e-01 | **8.67e-02 ± 1.07e-01** | **1.44 ± 1.01** | **1.84 ± 2.84e-02** | **2.21e-03 ± 5.33e-03** | **1.03 ± 8.69e-01** | **1.24e-02 ± 8.03e-03** | 2.09 ± 7.59e-01 |
| I.32.5 | 4.81e-03 ± 2.69e-03 | **9.29e-04 ± 1.30e-03** | **2.11e-23 ± 1.74e-24** | 1.69e-01 ± 1.12e-01 | 9.22e-02 ± 1.15e-01 | **2.47e-03 ± 3.91e-03** | 1.26e+03 ± 3.57e+03 | 1.00e-01 ± 1.24e-02 | **1.91e-15 ± 2.70e-15** | 1.82e-02 ± 2.89e-02 | 1.20e-01 ± 1.12e-02 | **1.22e-03 ± 1.34e-03** | **2.96e-06 ± 7.48e-06** | **9.43e-04 ± 1.02e-03** | **3.96e-03 ± 2.35e-03** |
| I.34.8 | **6.53e-02 ± 1.85e-01** | **3.87e-08 ± 1.20e-07** | **1.51e-29 ± 2.26e-31** | 75.86 ± 61.58 | **6.89 ± 19.48** | 4.05e-01 ± 6.90e-01 | 4.79e-01 ± 3.67e-01 | 2.37 ± 7.56e-02 | **1.84e-01 ± 5.21e-01** | 4.96 ± 14.03 | **7.79e-29 ± 8.42e-29** | 1.38e-02 ± 1.86e-02 | **9.43e-08 ± 2.00e-07** | **1.24e-03 ± 2.07e-03** | **1.22e-02 ± 1.41e-02** |
| I.34.27 | **1.56e-07 ± 9.49e-08** | **4.07e-07 ± 3.19e-07** | **1.24e-31 ± 3.14e-33** | **1.57e-06 ± 4.43e-06** | 2.68e-02 ± 1.49e-02 | 2.58e-04 ± 3.88e-04 | **5.66e-15 ± 1.54e-15** | 1.30e-03 ± 2.78e-05 | **1.27e-31 ± 4.58e-33** | 8.16e-03 ± 8.89e-03 | **5.06e-31 ± 3.73e-31** | 1.29e-04 ± 1.94e-04 | **4.60e-14 ± 8.20e-14** | **1.28e-31 ± 5.23e-33** | **6.28e-06 ± 1.10e-05** |
| I.37.4 | 2.03e-01 ± 2.22e-01 | 1.38e-01 ± 2.84e-01 | **6.51e-03 ± 3.87e-04** | 4.12e-01 ± 2.06e-01 | 4.85e-01 ± 4.96e-01 | 6.78e-02 ± 4.36e-02 | **1.21e-02 ± 1.81e-02** | **5.19e-02 ± 4.73e-02** | **1.64e-02 ± 1.02e-02** | 1.34 ± 2.86 | 6.65e-01 ± 1.07e-02 | **3.49e-04 ± 1.95e-05** | **5.53e-03 ± 1.18e-02** | **1.27e-03 ± 1.15e-03** | **7.26e-03 ± 7.36e-03** |
| I.38.12 | 2.73e-01 ± 6.98e-01 | **1.01e-01 ± 3.06e-01** | **1.14e-07 ± 3.41e-07** | 5.86e-01 ± 3.64e-01 | 1.86e-01 ± 3.95e-01 | **1.04e-02 ± 2.62e-02** | 5.26e+66 ± 1.49e+67 | 1.97e-01 ± 3.16e-02 | **4.45e-03 ± 1.26e-02** | **1.97e-01 ± 5.47e-01** | 1.67e-01 ± 8.96e-03 | **1.96e-03 ± 2.64e-03** | **2.65e-05 ± 8.36e-05** | **1.92e-03 ± 2.18e-03** | 1.60e-02 ± 2.45e-02 |
| I.39.1 | 2.69e-04 ± 7.61e-04 | 2.93e-07 ± 2.80e-07 | **1.07e-29 ± 1.25e-31** | 1.17 ± 3.31 | 7.82e-01 ± 2.00 | 7.12e-04 ± 1.99e-03 | **4.93e-13 ± 1.50e-13** | 1.16e-01 ± 5.64e-03 | **3.10e-10 ± 8.77e-10** | 2.17e-01 ± 6.02e-01 | **7.20e-29 ± 6.39e-29** | 2.27e-03 ± 1.84e-04 | **1.10e-07 ± 3.42e-07** | **1.25e-29 ± 8.89e-31** | **2.29e-07 ± 2.33e-05** |
| I.39.11 | 3.98e-02 ± 1.04e-01 | **8.50e-04 ± 1.71e-03** | **2.18e-30 ± 3.83e-32** | 3.52 ± 9.36 | **3.81e-01 ± 1.08** | 3.63e-02 ± 5.84e-02 | 2.21e-03 ± 2.23e-03 | 6.61e-02 ± 1.46e-03 | **3.51e-03 ± 9.94e-03** | 1.66e-01 ± 4.61e-01 | 5.75e-02 ± 6.58e-04 | 5.26e-04 ± 2.19e-04 | **1.04e-07 ± 1.84e-07** | **1.95e-05 ± 5.31e-05** | **1.44e-11 ± 4.13e-11** |
| I.39.22 | **1.24 ± 2.40** | **6.50e-08 ± 2.06e-07** | **1.60e-29 ± 2.90e-31** | 77.54 ± 66.66 | **6.93 ± 19.59** | 4.10e-01 ± 9.19e-01 | 6.09e-01 ± 3.37e-01 | 2.35 ± 6.10e-02 | **1.90e-01 ± 5.37e-01** | **2.90 ± 7.01** | **7.72e-29 ± 5.07e-29** | 5.54e-03 ± 3.01e-03 | **1.03e-06 ± 2.34e-06** | **2.08e-04 ± 5.74e-04** | **4.45e-04 ± 3.83e-04** |
| I.43.16 | **7.33e-01 ± 2.07** | **3.53e-08 ± 1.12e-07** | **1.49e-29 ± 4.09e-31** | 60.87 ± 53.12 | 6.83 ± 19.31 | 1.32 ± 1.61 | 8.99e-01 ± 1.02 | 2.37 ± 5.49e-02 | **3.29e-01 ± 9.30e-01** | 4.66 ± 12.85 | **8.14e-29 ± 6.70e-29** | 5.05e-03 ± 2.04e-03 | **4.71e-10 ± 5.98e-10** | **1.55e-04 ± 3.60e-04** | **9.66e-05 ± 1.88e-04** |
| I.43.31 | **1.74e-01 ± 4.92e-01** | **6.56e-29 ± 4.92e-30** | **6.49e-29 ± 8.96e-31** | 45.87 ± 60.78 | **11.99 ± 33.91** | **1.80e-01 ± 2.34e-01** | **3.22e-12 ± 9.52e-13** | 2.28 ± 2.18e-01 | **9.92e-02 ± 2.81e-01** | 7.18 ± 20.31 | **2.31e-28 ± 2.04e-28** | 1.61e-02 ± 2.16e-03 | **6.15e-07 ± 1.33e-06** | **7.42e-29 ± 7.46e-30** | **2.47e-04 ± 2.72e-04** |
| I.43.43 | 1.89e-02 ± 3.66e-02 | **6.42e-04 ± 1.80e-03** | **5.01e-31 ± 1.27e-32** | 4.08e-01 ± 3.45e-01 | 2.22e-01 ± 5.39e-01 | 2.75e-02 ± 4.24e-02 | 3.85e-02 ± 2.80e-02 | 6.27e-02 ± 1.68e-03 | **4.78e-03 ± 1.35e-02** | 1.30e-01 ± 3.43e-01 | 1.14e-02 ± 3.17e-04 | **1.39e-04 ± 1.01e-04** | **3.32e-08 ± 2.75e-08** | **6.21e-04 ± 1.03e-03** | **3.75e-04 ± 8.06e-04** |
| I.44.4 | 18.76 ± 17.56 | **3.32 ± 7.09** | **3.57e-05 ± 1.13e-04** | 244.73 ± 97.24 | 39.70 ± 54.25 | **6.51e-01 ± 7.05e-01** | 8.97 ± 8.13 | 156.65 ± 28.80 | **5.61e-01 ± 1.57** | 28.55 ± 57.79 | 3.93 ± 8.98e-02 | **5.48e-02 ± 4.62e-02** | **5.39e-03 ± 1.28e-02** | **5.46e-02 ± 7.71e-02** | **8.31e-02 ± 1.33e-01** |
| I.47.23 | **9.70e-04 ± 1.81e-03** | **4.14e-03 ± 1.31e-02** | **8.76e-32 ± 1.08e-33** | 4.17e-02 ± 9.57e-02 | 4.75e-02 ± 2.46e-02 | 1.21e-03 ± 1.51e-03 | 4.62e-03 ± 1.18e-02 | 1.65e-03 ± 1.30e-04 | **2.97e-04 ± 8.39e-04** | **9.15e-03 ± 1.69e-02** | **1.83e-30 ± 2.12e-30** | 1.98e-04 ± 5.06e-04 | **1.57e-06 ± 4.29e-06** | **1.71e-05 ± 1.43e-05** | **2.14e-02 ± 5.89e-02** |
| I.50.26 | 4.09e-01 ± 2.62e-01 | **1.31e-01 ± 8.22e-02** | **6.78e-30 ± 2.03e-31** | 2.58 ± 7.01e-02 | 1.78 ± 4.13e-01 | **2.39e-01 ± 2.93e-01** | 1.59 ± 7.66e-01 | 9.58e-01 ± 4.89e-01 | **1.59e-01 ± 2.67e-01** | 6.85e-01 ± 8.22e-01 | 2.26 ± 1.39e-02 | **1.68e-04 ± 3.58e-05** | **1.89e-01 ± 2.73e-01** | **2.38e-02 ± 3.95e-02** | 1.55 ± 2.70e-01 |
| II.2.42 | 3.96 ± 5.64 | **9.76e-02 ± 2.44e-31** | **8.95e-30 ± 2.44e-31** | 41.08 ± 11.01 | **5.04 ± 8.24** | **2.80e-01 ± 5.90e-01** | 1.18 ± 1.65 | 21.87 ± 5.01e-01 | **8.91e-16 ± 2.52e-15** | 5.10 ± 7.91 | **7.26e-29 ± 2.98e-29** | 2.72e-02 ± 2.79e-02 | **2.74e-07 ± 5.01e-07** | **2.35e-04 ± 4.29e-04** | **7.18e-04 ± 1.26e-03** |
| II.3.24 | 1.01e-06 ± 2.07e-06 | 4.87e-07 ± 3.11e-07 | **2.37e-27 ± 4.05e-29** | 9.17e-04 ± 3.08e-04 | 1.10e-04 ± 1.02e-06 | 8.45e-07 ± 5.32e-06 | **2.42e-06 ± 5.34e-07** | 1.36e-05 ± 3.03e-35 | **1.27e-33 ± 3.66e-04** | 2.80e-04 ± 1.71e-06 | 1.06e-04 ± 8.57e-09 | **1.28e-07 ± 2.07e-11** | **7.15e-12 ± 1.29e-08** | **8.15e-09 ± 5.22e-08** | **2.73e-08 ±** |
| II.4.23 | **4.74e-07 ± 3.50e-07** | **6.02e-07 ± 1.85e-06** | **1.12e-27 ± 9.92e-30** | 4.84e-04 ± 1.65e-04 | 6.74e-05 ± 1.74e-04 | 2.50e-06 ± 3.22e-06 | 3.15e-05 ± 4.45e-05 | 1.29e-05 ± 4.16e-07 | **1.04e-33 ± 1.36e-34** | 4.55e-04 ± 4.00e-04 | **1.89e-33 ± 7.85e-34** | **1.28e-07 ± 1.28e-07** | **4.98e-12 ± 8.59e-12** | **3.03e-08 ± 5.67e-08** | 8.68e-03 ± 2.21e-02 |
| II.6.11 | 1.69e-05 ± 1.66e-05 | **3.02e-06 ± 6.29e-06** | **2.23e-28 ± 2.94e-30** | 2.53e-04 ± 1.29e-04 | 8.81e-05 ± 3.87e-05 | **8.66e-07 ± 5.30e-07** | **9.90e-06 ± 1.02e-05** | 2.54e-05 ± 8.12e-06 | **1.36e-18 ± 3.43e-18** | 3.71e-04 ± 1.16e-04 | 3.72e-05 ± 4.94e-07 | **1.32e-07 ± 1.85e-07** | **5.65e-10 ± 1.53e-09** | **6.03e-07 ± 4.21e-07** | **8.06e-07 ± 2.69e-06** |
| II.6.15b | **4.06e-05 ± 5.32e-05** | **8.67e-06 ± 1.05e-05** | **4.27e-28 ± 6.90e-30** | 8.53e-04 ± 4.62e-04 | 3.25e-04 ± 1.00e-04 | **3.65e-06 ± 4.27e-06** | 1.25e-04 ± 1.07e-04 | 1.02e-04 ± 1.04e-05 | **1.61e-17 ± 4.08e-17** | 9.09e-04 ± 1.68e-05 | 2.60e-04 ± 1.35e-05 | **7.31e-07 ± 1.67e-06** | **1.38e-09 ± 3.69e-09** | **6.97e-06 ± 3.26e-06** | 3.42e-04 ± 1.10e-04 |
| II.8.7 | **1.52e-05 ± 3.10e-05** | **9.77e-07 ± 1.57e-06** | **1.17e-24 ± 1.70e-26** | 2.67e-03 ± 7.70e-04 | 6.32e-04 ± 9.47e-04 | **1.83e-05 ± 3.99e-05** | 3.48e-04 ± 8.60e-04 | 1.80e-04 ± 6.60e-06 | **5.57e-19 ± 1.58e-18** | 1.80e-04 ± 3.01e-04 | 1.32e-04 ± 2.25e-06 | **4.60e-07 ± 2.84e-07** | **1.34e-11 ± 4.14e-11** | **2.26e-07 ± 2.71e-07** | **2.38e-03 ± 3.86e-03** |
| II.8.31 | 2.50e-01 ± 7.07e-01 | **2.89e-07 ± 2.42e-07** | **3.26e-29 ± 6.38e-31** | 2.85 ± 7.92 | 3.79 ± 6.44 | 1.95e-02 ± 4.66e-02 | **1.92e-12 ± 6.19e-13** | 3.56e-01 ± 1.48e-02 | **3.93e-29 ± 9.95e-02** | 3.34 ± 1.36e-01 | 8.56e-02 ± 3.40e-02 | **3.77e-08 ± 2.23e-01** | **3.94e-29 ± 1.19e-07** | **1.81e-04 ± 6.31e-30** | 2.04e-04 ± |
| II.10.9 | 7.04e-05 ± 1.14e-04 | **3.75e-05 ± 1.18e-04** | **8.45e-33 ± 1.24e-34** | 7.22e-03 ± 2.62e-03 | **2.30e-03 ± 6.52e-03** | **4.79e-05 ± 6.04e-05** | 2.93e-04 ± 3.82e-04 | 4.64e-04 ± 3.48e-05 | **2.88e-16 ± 8.15e-16** | 1.14e-03 ± 2.49e-03 | 2.80e-04 ± 2.98e-06 | **2.73e-05 ± 6.89e-05** | **5.11e-10 ± 6.30e-10** | **4.09e-07 ± 5.21e-07** | 1.99e-02 ± 5.27e-02 |
| II.11.3 | 2.75e-04 ± 1.72e-04 | **6.46e-05 ± 4.49e-05** | **7.89e-05 ± 3.11e-05** | 3.67e-03 ± 7.42e-04 | 2.09e-03 ± 2.14e-03 | **1.36e-04 ± 1.58e-04** | **6.85e-05 ± 2.62e-05** | **1.93e-04 ± 3.65e-05** | **1.87e-06 ± 1.82e-06** | 1.64e-03 ± 8.94e-04 | 5.16e-04 ± 1.94e-05 | **2.18e-06 ± 3.27e-06** | **1.90e-06 ± 5.63e-06** | **2.01e-05 ± 8.46e-06** | 1.52e-02 ± 4.20e-03 |
| II.11.20 | 3.28e-01 ± 4.03e-01 | **2.59e-02 ± 3.66e-02** | **7.38e-23 ± 1.57e-24** | 36.62 ± 15.47 | 4.57 ± 12.39 | **1.11e-01 ± 1.82e-01** | 2.50 ± 1.69 | 3.89 ± 3.76e-01 | **1.73e-13 ± 2.83e-13** | 1.19 ± 3.02 | 6.74e-01 ± 1.55e-02 | **9.00e-02 ± 8.15e-02** | **9.86e-05 ± 3.11e-04** | **3.39e-02 ± 3.16e-02** | **7.29e-03 ± 8.77e-03** |
| II.11.27 | 3.05e-04 ± 3.80e-04 | **1.13e-04 ± 1.58e-04** | **5.16e-07 ± 1.31e-06** | 1.19e-02 ± 9.94e-03 | 1.49e-02 ± 3.94e-02 | **6.55e-05 ± 1.00e-04** | **5.37e-04 ± 2.47e-04** | 6.41e-03 ± 1.83e-04 | **4.66e-05 ± 3.67e-05** | 1.64e-03 ± 1.45e-03 | 1.01e-03 ± 4.32e-05 | **2.58e-05 ± 1.06e-05** | **1.08e-07 ± 3.06e-07** | **9.75e-06 ± 1.10e-05** | 3.58e-01 ± 2.84e-01 |
| II.11.28 | 1.29e-05 ± 2.13e-05 | **7.67e-06 ± 2.02e-05** | **4.56e-32 ± 6.10e-34** | 7.58e-04 ± 5.91e-04 | 2.27e-04 ± 3.77e-06 | **1.84e-06 ± 1.84e-06** | 1.70e-05 ± 1.36e-05 | 2.21e-04 ± 2.13e-05 | **4.63e-08 ± 5.48e-08** | 5.46e-04 ± 1.03e-03 | **2.06e-07 ± 5.63e-09** | 3.55e-06 ± 1.44e-07 | **8.87e-10 ± 7.54e-10** | **1.50e-09 ± 2.94e-09** | **1.08e-01 ± 2.93e-01** |
| II.13.17 | **2.50e-06 ± 4.00e-06** | **1.98e-06 ± 4.68e-06** | **3.74e-28 ± 7.56e-30** | 5.92e-04 ± 2.42e-04 | 5.61e-05 ± 1.34e-04 | **1.25e-06 ± 1.16e-06** | 1.90e-04 ± 4.45e-04 | 4.01e-05 ± 3.42e-06 | **1.12e-21 ± 3.13e-21** | 6.51e-04 ± 8.62e-05 | 1.71e-05 ± 3.17e-07 | **5.05e-07 ± 8.05e-07** | **9.11e-08 ± 2.88e-07** | **3.34e-07 ± 2.15e-07** | **2.25e-01 ± 7.01e-01** |
| II.15.4 | **1.38e-02 ± 3.92e-02** | **4.41e-07 ± 3.68e-07** | **1.10e-29 ± 1.75e-31** | 17.88 ± 12.07 | **1.22 ± 3.44** | **2.64e-02 ± 6.24e-02** | 1.16e-01 ± 1.22e-01 | 3.68e-01 ± 5.31e-03 | **1.13e-29 ± 3.03e-31** | 1.28 ± 2.56 | 5.31 ± 5.56e-02 | **1.07e-03 ± 6.18e-05** | **1.51e-07 ± 2.74e-07** | **2.82e-26 ± 7.99e-26** | **1.27e-03 ± 2.33e-03** |
| II.15.5 | **2.90e-02 ± 7.83e-02** | **3.02e-05 ± 9.43e-05** | **1.10e-29 ± 3.25e-31** | 5.64 ± 3.00 | **1.23 ± 3.48** | **3.06e-04 ± 7.97e-04** | 1.34e-01 ± 2.35e-01 | 3.79e-01 ± 8.11e-03 | **1.15e-29 ± 3.96e-31** | **1.52 ± 2.30** | 5.36 ± 5.46e-02 | **2.86e-03 ± 4.01e-03** | **2.63e-02 ± 8.30e-02** | **1.14e-29 ± 3.25e-31** | **2.72e-03 ± 4.44e-03** |
| II.24.17 | 9.59e-04 ± 1.38e-03 | **1.22e-03 ± 3.01e-03** | 2.05e-04 ± 8.89e-06 | 4.74e-03 ± 2.63e-03 | 1.21e-02 ± 1.44e-02 | **1.44e-03 ± 3.50e-03** | **1.29e-04 ± 1.11e-04** | 1.83e-03 ± 7.93e-05 | **3.49e-05 ± 3.58e-05** | 4.85e-03 ± 4.38e-03 | **1.88e-04 ± 2.15e-06** | **2.92e-05 ± 1.37e-06** | **2.90e-07 ± 2.97e-07** | **6.52e-06 ± 3.06e-06** | 7.42e-01 ± 3.22e-01 |

Table 52: Mean and standard deviation of MSE for each Feynman problem (Part 2 of 2)

| Prob. | AFP | AFP_FE | AIFeynman | BSR | DSR | EPLEX | FEAT | FFX | GP-GOMEA | gplearn | ITEA | MRGP | Operon | SBP-GP | SeTGAP |
|---|---|---|---|---|---|---|---|---|---|---|---|---|---|---|---|
| II.27.16 | 3.51e-10 ± 9.94e-10 | 6.50e-09 ± 2.05e-08 | 1.62e-27 ± 2.64e-29 | 7.61e+04 ± 2.06e+05 | 275.99 ± 780.61 | 3.82 ± 7.04 | 2.98e-01 ± 7.25e-01 | 89.31 ± 1.21 | 1.90e-27 ± 1.26e-28 | 2.73e-02 ± 7.71e-02 | 150.47 ± 1.17 | 3.16e-01 ± 4.53e-02 | 1.01e-07 ± 2.99e-07 | 2.41e-27 ± 1.35e-27 | 1.43e-02 ± 1.66e-02 |
| II.27.18 | 1.34e-28 ± 1.26e-29 | 1.37e-28 ± 1.91e-29 | 1.27e-28 ± 2.53e-30 | 19.76 ± 37.98 | 9.01 ± 25.48 | 2.34e-09 ± 5.10e-09 | 7.87e-12 ± 2.51e-12 | 1.44 ± 4.08e-02 | 1.38e-28 ± 6.99e-30 | 1.28e-28 ± 1.71e-30 | 13.40 ± 6.13e-02 | 2.84e-02 ± 4.02e-04 | 1.24e-06 ± 3.77e-06 | 1.37e-28 ± 3.68e-30 | 1.71e-03 ± 1.62e-03 |
| II.34.2 | 2.54e-05 ± 7.13e-05 | 5.59e-05 ± 1.76e-04 | 1.60e-29 ± 2.94e-31 | 16.87 ± 27.85 | 2.88 ± 8.07 | 1.07e-01 ± 1.95e-01 | 7.76e-13 ± 2.22e-13 | 5.90e-01 ± 5.30e-03 | 1.85e-29 ± 1.42e-30 | 1.65e-01 ± 1.80e-01 | 6.93e-29 ± 5.78e-29 | 4.03e-03 ± 5.41e-04 | 1.24e-08 ± 3.36e-08 | 1.83e-29 ± 1.21e-30 | 1.09e-03 ± 7.83e-04 |
| II.34.2a | 8.32e-05 ± 2.35e-04 | 2.96e-06 ± 8.10e-06 | 2.62e-26 ± 7.86e-26 | 7.71e-03 ± 5.60e-03 | 1.47e-02 ± 2.52e-02 | 2.94e-04 ± 7.27e-04 | 3.20e-04 ± 2.26e-04 | 1.80e-03 ± 3.95e-05 | 2.44e-27 ± 6.89e-27 | 9.88e-03 ± 1.89e-02 | 1.59e-31 ± 1.67e-31 | 1.41e-05 ± 1.50e-05 | 5.34e-10 ± 1.42e-09 | 3.76e-08 ± 1.05e-07 | 9.51e-06 ± 6.29e-06 |
| II.34.11 | 4.61e-02 ± 7.11e-02 | 3.12e-07 ± 2.78e-07 | 3.72e-30 ± 8.12e-32 | 15.91 ± 16.72 | 1.99 ± 5.05 | 5.60e-02 ± 6.47e-02 | 1.45e-01 ± 1.23e-01 | 5.94e-01 ± 1.04e-02 | 1.53e-26 ± 4.34e-26 | 2.10e-01 ± 1.75e-01 | 2.36e-29 ± 2.12e-29 | 2.96e-03 ± 2.45e-03 | 4.31e-09 ± 1.25e-08 | 1.58e-03 ± 4.27e-03 | 6.71e-05 ± 7.02e-05 |
| II.34.29a | 3.66e-07 ± 2.46e-07 | 2.74e-07 ± 1.98e-07 | 3.52e-26 ± 3.03e-26 | 6.96e-03 ± 4.89e-03 | 3.41e-03 ± 5.70e-03 | 9.87e-05 ± 2.17e-04 | 1.13e-04 ± 1.29e-04 | 4.49e-04 ± 6.70e-06 | 1.01e-32 ± 1.16e-33 | 8.79e-04 ± 4.78e-04 | 8.00e-32 ± 4.70e-32 | 2.37e-06 ± 6.09e-07 | 1.91e-10 ± 4.70e-10 | 3.76e-10 ± 7.12e-10 | 1.04e-05 ± 6.32e-06 |
| II.34.29b | 901.30 ± 639.70 | 217.88 ± 670.30 | 9.10e+03 ± 2.88e+04 | 3.60e+04 ± 1.84e+04 | 6.29e+03 ± 1.13e+04 | 468.19 ± 381.51 | 3.58e+03 ± 4.55e+03 | 2.05e+03 ± 34.87 | 2.34e-13 ± 6.62e-13 | 139.78 ± 224.46 | 2.05e-26 ± 9.55e-27 | 9.30 ± 11.19 | 1.84e-04 ± 1.46e-04 | 6.20e-01 ± 1.00e+00 | 1.02e-03 ± 8.11e-04 |
| II.35.21 | 8.87e-01 ± 6.89e-01 | 3.81e-01 ± 1.90e-01 | 4.13e-02 ± 2.08e-03 | 12.05 ± 23.99 | 3.63 ± 1.41 | 2.22e-01 ± 1.55e-01 | 5.51e-01 ± 2.23e-01 | 3.62e-01 ± 3.10e-02 | 2.84e-01 ± 9.24e-02 | 5.82e-01 ± 3.86e-01 | 7.35e-01 ± 1.16e-02 | 3.61e-03 ± 2.47e-03 | 6.57e-02 ± 3.59e-02 | 8.97e-02 ± 1.67e-02 | 4.55 ± 8.53 |
| II.36.38 | 1.03e-01 ± 5.13e-02 | 6.51e-02 ± 4.75e-02 | 7.95e-03 ± 1.36e-03 | 1.99e-01 ± 5.64e-02 | 4.21e-01 ± 1.42e-01 | 3.02e-02 ± 2.32e-02 | 1.09e-01 ± 7.63e-02 | 4.68e-02 ± 8.75e-03 | 1.49e-02 ± 6.77e-03 | 6.59e-02 ± 7.26e-02 | 3.01e-02 ± 5.94e-04 | 7.32e-04 ± 1.13e-03 | 3.78e-03 ± 2.69e-03 | 1.29e-02 ± 5.17e-03 | 8.53e-01 ± 6.43e-01 |
| II.37.1 | 1.22e-07 ± 3.45e-07 | 6.52e-08 ± 1.10e-07 | 1.02e-28 ± 1.32e-30 | 272.06 ± 641.25 | 15.60 ± 44.13 | 6.21e-01 ± 1.15 | 4.77e-03 ± 1.35e-02 | 2.38 ± 4.08e-02 | 1.51e-28 ± 8.07e-29 | 2.27e-01 ± 4.57e-01 | 4.43e-28 ± 2.76e-28 | 2.34e-02 ± 3.11e-03 | 1.91e-08 ± 5.93e-08 | 3.24e-05 ± 9.08e-05 | 3.58e-04 ± 2.35e-04 |
| II.38.3 | 1.48e-01 ± 4.17e-01 | 2.35e-07 ± 3.56e-07 | 1.08e-02 ± 3.42e-02 | 38.01 ± 11.09 | 7.48 ± 21.15 | 2.62e-01 ± 3.79e-01 | 1.25 ± 1.52 | 2.32 ± 4.68e-02 | 2.29e-24 ± 4.15e-24 | 1.95 ± 4.10 | 5.26e-29 ± 1.98e-29 | 1.24e-02 ± 1.29e-02 | 2.54e-10 ± 4.18e-10 | 9.99e-04 ± 2.32e-03 | 1.69e-05 ± 1.57e-05 |
| II.38.14 | 1.59e-05 ± 2.88e-05 | 2.98e-06 ± 7.50e-06 | 6.15e-33 ± 1.06e-34 | 2.24e-03 ± 9.85e-04 | 6.60e-04 ± 1.87e-03 | 8.68e-05 ± 1.53e-04 | 6.63e-05 ± 1.72e-04 | 1.14e-04 ± 1.02e-05 | 8.82e-19 ± 2.49e-18 | 8.74e-04 ± 8.18e-04 | 3.59e-04 ± 6.09e-06 | 1.93e-06 ± 1.49e-07 | 3.12e-10 ± 4.64e-10 | 1.31e-10 ± 2.52e-10 | 5.35e-05 ± 3.72e-05 |
| III.4.32 | 4.93e-01 ± 1.20 | 2.84e-01 ± 8.53e-01 | 6.14e-03 ± 1.09e-02 | 39.58 ± 14.65 | 8.81 ± 16.86 | 1.26e-01 ± 2.29e-01 | 6.71e-01 ± 3.01e-01 | 2.39 ± 6.09e-02 | 1.02e-05 ± 2.54e-05 | 4.44e-01 ± 5.66e-01 | 2.32e-07 ± 3.76e-08 | 1.02e-02 ± 7.75e-03 | 9.01e-04 ± 2.83e-03 | 3.63e-03 ± 6.97e-03 | 4.57e-02 ± 4.46e-02 |
| III.4.33 | 2.69e-02 ± 4.51e-02 | 9.24e-03 ± 2.22e-02 | 1.90e-03 ± 3.72e-04 | 5.88e-03 ± 4.64e-03 | 4.31e-01 ± 8.62e-01 | 1.65e-03 ± 2.39e-03 | 8.64e-04 ± 1.49e-03 | 5.74e-02 ± 9.63e-04 | 5.60e-04 ± 5.62e-04 | 5.87e-02 ± 4.85e-02 | 1.09e-03 ± 3.25e-05 | 1.05e-03 ± 7.87e-05 | 1.02e-05 ± 2.26e-05 | 6.05e-06 ± 6.71e-06 | 7.93e-01 ± 1.14 |
| III.7.38 | 1.70 ± 2.85 | 1.35e-03 ± 4.11e-03 | 6.33e-25 ± 6.66e-27 | 508.66 ± 436.91 | 70.50 ± 148.91 | 9.84 ± 21.27 | 16.42 ± 42.54 | 10.98 ± 1.97e-01 | 2.88e-28 ± 6.56e-29 | 3.56e-01 ± 5.55e-01 | 1.21e-27 ± 9.24e-28 | 7.26e-01 ± 1.20 | 1.97e-09 ± 6.04e-09 | 1.71e-06 ± 3.85e-06 | 6.79e-06 ± 6.85e-06 |
| III.8.54 | 9.27e-02 ± 1.56e-02 | 6.56e-02 ± 2.06e-02 | — | 1.18e-01 ± 3.02e-03 | 1.21e-01 ± 1.35e-03 | 2.84e-02 ± 2.32e-02 | 1.24e-01 ± 1.03e-03 | 1.20e-01 ± 1.33e-03 | 1.54e-03 ± 2.11e-03 | 1.25e-01 ± 8.81e-04 | 1.15e-01 ± 7.44e-04 | 4.72e-03 ± 6.60e-03 | 4.04e+76 ± 4.51e+76 | 5.27e-02 ± 1.22e-02 | 1.11e-01 ± 2.72e-02 |
| III.9.52 | 65.32 ± 43.03 | 12.52 ± 6.50 | 430.88 ± 16.31 | 194.20 ± 89.97 | 140.38 ± 2.46 | 35.16 ± 77.45 | 51.06 ± 33.61 | 82.98 ± 4.75 | 7.30 ± 3.51 | 12.81 ± 9.81 | 101.47 ± 1.48 | 1.74e-01 ± 2.17e-01 | 3.57 ± 4.14 | 2.72 ± 1.08 | 71.58 ± 32.89 |
| III.10.19 | 9.75e-01 ± 8.95e-01 | 2.37e-01 ± 2.42e-01 | 1.29e-29 ± 1.73e-31 | 3.26 ± 3.58 | 3.46 ± 3.02 | 3.17e-01 ± 1.84e-01 | 1.56e-02 ± 1.64e-02 | 1.38e-01 ± 1.08e-02 | 9.33e-02 ± 2.88e-02 | 1.69 ± 2.39e-02 | 4.93e-03 ± 2.14e-03 | 1.37e-03 ± 9.66e-04 | 1.18e-02 ± 4.39e-03 | 1.35 ± 1.18 | |
| III.12.43 | 2.71e-07 ± 2.69e-07 | 3.83e-07 ± 3.18e-07 | 1.22e-31 ± 3.25e-33 | 2.29e-03 ± 5.07e-03 | 2.68e-02 ± 1.54e-02 | 4.36e-04 ± 9.97e-04 | 5.49e-15 ± 1.65e-15 | 1.31e-03 ± 1.25e-05 | 1.25e-31 ± 2.17e-33 | 8.01e-03 ± 7.70e-03 | 1.65e-30 ± 1.53e-30 | 3.15e-05 ± 1.13e-05 | 2.37e-12 ± 6.82e-12 | 1.27e-31 ± 2.92e-33 | 7.36e-09 ± 8.63e-09 |
| III.13.18 | 4.98e+03 ± 4.14e+03 | 845.90 ± 2.02e+03 | 6.61e+04 ± 1.87e+05 | 1.77e+05 ± 1.41e+05 | 2.90e+04 ± 5.51e+04 | 1.83e+03 ± 2.18e+03 | 5.23e+03 ± 5.08e+03 | 1.14e+04 ± 429.86 | 2.22e-11 ± 6.29e-11 | 425.22 ± 476.43 | 4.84e+03 ± 104.60 | 27.18 ± 31.96 | 1.81e-03 ± 3.24e-03 | 34.91 ± 50.63 | 1.10e-03 ± 8.56e-04 |
| III.15.12 | 3.51 ± 4.91 | 1.27e-02 ± 3.05e-02 | 6.29e-29 ± 1.14e-30 | 20.12 ± 2.12e-01 | 10.72 ± 6.03 | 1.64e-01 ± 3.59e-01 | 15.93 ± 5.17 | 17.08 ± 1.90 | 9.22e-29 ± 6.98e-29 | 1.90 ± 2.38 | 2.65 ± 2.29e-02 | 3.98e-03 ± 8.38e-03 | 2.12 ± 4.89 | 4.18e-11 ± 1.18e-10 | 1.15e-04 ± 3.21e-04 |
| III.15.14 | 5.07e-06 ± 1.07e-05 | 4.36e-07 ± 3.99e-07 | 3.48e-11 ± 1.10e-10 | 2.00e-04 ± 6.91e-05 | 3.51e-05 ± 4.43e-05 | 7.83e-07 ± 2.39e-07 | 1.09e-05 ± 1.27e-05 | 1.37e-05 ± 7.27e-07 | 5.38e-21 ± 1.01e-20 | 3.21e-04 ± 4.90e-05 | 1.32e-05 ± 1.19e-06 | 1.87e-06 ± 3.36e-06 | 2.49e-10 ± 7.84e-10 | 2.27e-08 ± 2.04e-08 | 7.69e-05 ± 1.39e-04 |
| III.15.27 | 1.64e-02 ± 3.91e-02 | 3.22e-05 ± 1.01e-04 | 1.66e-29 ± 3.85e-30 | 1.35 ± 2.42 | 1.45 ± 5.86e-01 | 1.83e-02 ± 3.40e-02 | 2.75e-02 ± 1.11e-02 | 8.74e-02 ± 4.13e-03 | 1.61e-30 ± 6.11e-31 | 1.20e-02 ± 1.31e-02 | 6.82e-30 ± 4.52e-30 | 9.84e-04 ± 1.00e-03 | 5.71e-08 ± 8.41e-08 | 5.22e-05 ± 1.09e-04 | 7.53e-08 ± 5.20e-08 |
| III.17.37 | 3.47e-01 ± 6.44e-01 | 1.10e-07 ± 2.32e-07 | 1.03e-29 ± 2.48e-31 | 10.87 ± 5.79 | 8.82e-01 ± 2.49 | 1.15e-01 ± 2.94e-01 | 6.64 ± 15.86 | 9.72e-01 ± 9.53e-01 | 2.04e-29 ± 2.38e-29 | 2.63e-01 ± 3.19e-01 | 5.89 ± 4.25e-02 | 9.82e-04 ± 7.15e-05 | 1.58e-07 ± 2.49e-07 | 1.46e-04 ± 4.12e-04 | 1.06e-02 ± 2.87e-02 |
| III.19.51 | 1.35e-01 ± 9.26e-02 | 5.62e-02 ± 8.79e-02 | 4.54 ± 6.43e-01 | 3.98 ± 2.14 | 3.31 ± 2.14 | 9.43e-02 ± 7.58e-02 | 1.87 ± 1.52 | 2.12 ± 3.18e-01 | 3.20e-12 ± 4.27e-12 | 2.92e-01 ± 7.23e-01 | 9.63e-01 ± 4.14e-01 | 7.96e-02 ± 1.14e-01 | 7.53e-04 ± 1.75e-03 | 3.06e-02 ± 3.22e-02 | 3.41 ± 2.14 |
| III.21.20 | 2.16e-01 ± 4.31e-01 | 3.56e-07 ± 3.32e-07 | 1.53e-29 ± 4.84e-31 | 56.33 ± 56.34 | 6.76 ± 17.39 | 3.48e-01 ± 6.67e-01 | 1.02 ± 1.57 | 2.38 ± 8.86e-02 | 7.59e-29 ± 1.30e-28 | 5.14e-01 ± 7.62e-01 | 1.03e-28 ± 8.95e-29 | 7.27e-03 ± 2.57e-03 | 2.86e-07 ± 9.03e-07 | 3.75e-03 ± 5.65e-03 | 1.34e-05 ± 1.88e-05 |

## K  Cascade Merging Ordering

This appendix provides a sensitivity analysis of the variable ordering strategy within the cascade merging process to evaluate its impact on the final symbolic expressions. As described in Sec. 3.3.3, the suggested heuristic ranks variables based on the MSE values obtained by their skeletons. To assess this choice, we conducted experiments on problems I.50.26, II.6.11, and II.13.17, comparing the MSE-based ordering against reversed sequences and slight variations of these permutations. These problems were selected because SeTGAP failed to identify the correct functional form for variable $x_2$ in problem I.50.26, variable $x_2$ in problem II.6.11, and variable $x_1$ in problem II.13.17. By varying the order, we aim to observe whether the inclusion of an incorrectly identified skeleton at the beginning versus the end of the cascade influences the final expression's fitness. For each configuration, we utilized the test data $\tilde{\mathbf{D}}_{S'}^{(\text{test})} = (\tilde{\mathbf{X}}_{S'}^{(\text{test})}, \tilde{\mathbf{y}}^{(\text{test})})$ (see Sec. 3.3.2), and reported the coefficient of determination, $R^2$, between the target $\tilde{\mathbf{y}}^{(\text{test})}$ and the output predicted by evaluating the intermediate expressions on $\tilde{\mathbf{X}}_{S'}^{(\text{test})}$ after each variable merge, as shown in Table 53. The use of $R^2$ as a goodness-of-fit metric provides a more interpretable scale (0 to 1) in comparison to raw MSE values. In addition, both $R^2$ and MSE values are reported for the final expressions to facilitate a direct comparison of the global performance across the different variable orderings.

The results in Table 53 indicate that the impact of variable ordering is problem-dependent, revealing a nuanced relationship between skeleton quality and structural preservation. In the case of I.50.26, there is an advantage to leaving the incorrectly identified functional form, corresponding to variable $x_2$, at the end of the cascade, as evidenced by the higher intermediate $R^2$ values. Specifically, until the final step, the algorithm successfully identifies the functional form $\hat{\mathbf{e}}(x_0, x_3, x_1) = c_1 x_0 x_3 \sin^2(c_2 x_1 + c_3) + c_4 x_0 \sin(c_5 x_1 + c_6) + c_7 x_3 \sin(c_8 x_1 + c_9)$, which is equivalent to the target skeleton involving these three variables when $c_7$ is set to 0. A similar preservation of the skeleton is observed for the $[x_3, x_0, x_1, x_2]$ sequence. However, including

Table 53: Sensitivity analysis of variable ordering in cascade merging

| Eq. | Ordering | Merge 2 vars. ($R^2$) | Merge 3 vars. ($R^2$) | Merge 4 vars. ($R^2$) | MSE | Final Function |
|---|---|---|---|---|---|---|
| I.50.26 | $[x_0, x_3, x_1, x_2]^\dagger$ | 0.9999 | 0.9973 | 0.9566 | 0.5768 | $0.43x_0x_3\sin^2(0.04x_1+1) + 0.122(8.225 - 10.08\sin(2x_1x_2+23.5))(x_3+0.13) + +0.122(1.79x_3 +5.56)(2.40\sin(x_1x_2-11.07)+0.27) - 1.648$ |
| | $[x_2, x_1, x_3, x_0]^{\dagger\dagger}$ | 0.8170 | 0.8249 | 0.8221 | 1.3749 | $0.231x_0(2.92x_3+6.28\sin^2(1.1x_1-0.82)- 8.20\sin(-3.62x_1x_2+5.07x_1+5.88x_2+6.28)) +0.629x_3\sin^2(2.66x_1+3.31) - 2.182$ |
| | $[x_3, x_0, x_1, x_2]$ | 0.9999 | 0.9995 | 0.8269 | 0.8574 | $0.848x_0x_3\sin^2(0.24x_1+19.03)- 0.679x_0\sin(4.37x_1+4.12x_2+2.22) + 0.068$ |
| | $[x_1, x_2, x_0, x_3]$ | 0.8179 | 0.8664 | 0.8970 | 0.6785 | $0.719x_0\sin^2(1.11x_1-1.36) - 0.327(0.0088x_3 +0.198)(-5.08x_0\sin(-x_1x_2+1.94x_1+x_2+19.98) -0.21) - 0.327(0.48x_3-0.066)(2.60x_0 \sin(x_1x_2+2.22x_1+2.08x_2+0.28) - 6.01) - 1.282$ |
| II.6.11 | $[x_1, x_0, x_3, x_2]^\dagger$ | 0.9902 | 0.9933 | 0.9931 | 1.33e-02 | $\frac{7.46\cdot10^{-7}x_1(6.58-4.18x_2)}{x_3^2(-31.54x_0-0.08)} - 0.0066$ |
| | $[x_2, x_3, x_0, x_1]^{\dagger\dagger}$ | 0.9926 | 0.9934 | 0.9931 | 1.43e-02 | $\frac{-1.98x_1(6.26-3.98x_2)}{(-111.57x_0-2.89)(x_3^2+0.03)+4.25} + 1.0\cdot10^{-4}$ |
| | $[x_0, x_1, x_2, x_3]$ | 0.9902 | 0.9908 | 0.9931 | 1.68e-02 | $0.0212 - \frac{0.0063(8.48x_2-13.34)[\frac{91.07x_1+0.07}{-0.77x_3^2-0.0016}+0.009]}{-20.64x_0-0.05}$ |
| | $[x_2, x_1, x_3, x_0]$ | 0.9920 | 0.9931 | 0.9931 | 9.25e-03 | $0.0029 - \frac{0.0066(-175.80x_1x_2+274.94x_1+0.87x_2)}{-18.64x_3^2-0.22}$ |
| II.13.17 | $[x_0, x_3, x_2, x_1]^\dagger$ | 0.9985 | 0.9982 | 0.9998 | 1.95e-02 | $-0.10 + \frac{0.005(1.19-386.5/x_3)(19.90x_2-0.42)e^{-2.2\sqrt{x_1-0.78}}}{0.25-20.89x_0}$ |
| | $[x_1, x_2, x_3, x_0]^{\dagger\dagger}$ | 0.9989 | 0.9995 | 0.9998 | 1.20e-02 | $\frac{-0.0002x_2 e^{-2.12\sqrt{x_1-0.81}}}{16.16x_0x_3-0.12} + 0.0071$ |
| | $[x_0, x_2, x_3, x_1]$ | 0.9880 | 0.9981 | 0.9991 | 1.20e-02 | $0.007 - \frac{0.004(0.23-2.98x_2)e^{-2.69\sqrt{x_1-0.47}}}{(0.20-6.75x_0)(23.11x_3-0.69)}$ |
| | $[x_1, x_3, x_2, x_0]$ | 0.9996 | 0.9995 | 0.9998 | 1.20e-02 | $\frac{-0.0079x_2 e^{-2.19\sqrt{x_1-0.78}}}{(0.58-6.86x_0)(0.75-7.95x_3)+6.14} + 0.0071$ |

$\dagger$: MSE-based ordering. $\dagger\dagger$: Inverse ordering.

the final variable, $x_2$, in the merging process causes a drop in fitness, as it forces the algorithm to fit the data using a mismatched functional form. Interestingly, while other permutations achieve lower intermediate and final $R^2$ values, they arrive at similar final MSE values. This suggests that the misidentification of $x_2$ leads to similar global performance limits regardless of the order, though the recommended ordering better preserves the functional forms of $x_0$, $x_1$, and $x_3$ during the process. Fig. 9 illustrates the discovery process when considering the $[x_3, x_0, x_1, x_2]$ ordering. For brevity, we show only the results obtained using the best-performing skeleton for each variable. During the univariate symbolic skeleton prediction step, green boxes indicate that the selected candidate matches the corresponding univariate target skeleton, while red boxes indicate a mismatch. Similarly, in the cascade merging step, green boxes denote that the combined skeleton matches the target skeleton for that specific subset of variables, whereas red boxes highlight mismatches.

Conversely, problems II.6.11 and II.13.17 arrive at expressions that are functionally similar independently of the chosen ordering. This behavior challenges the intuition that including the worst-performing skeletons at the start of the merging process inevitably propagates structural uncertainty. In these cases, the algorithm reached comparable $R^2$ and MSE results across all tested permutations.

The primary takeaway from these experiments is that while the cascade approach is often robust to the merging sequence, skeleton structural fidelity can be improved by prioritizing high-confidence skeletons when a system contains variables with varying degrees of identification uncertainty. Therefore, future work

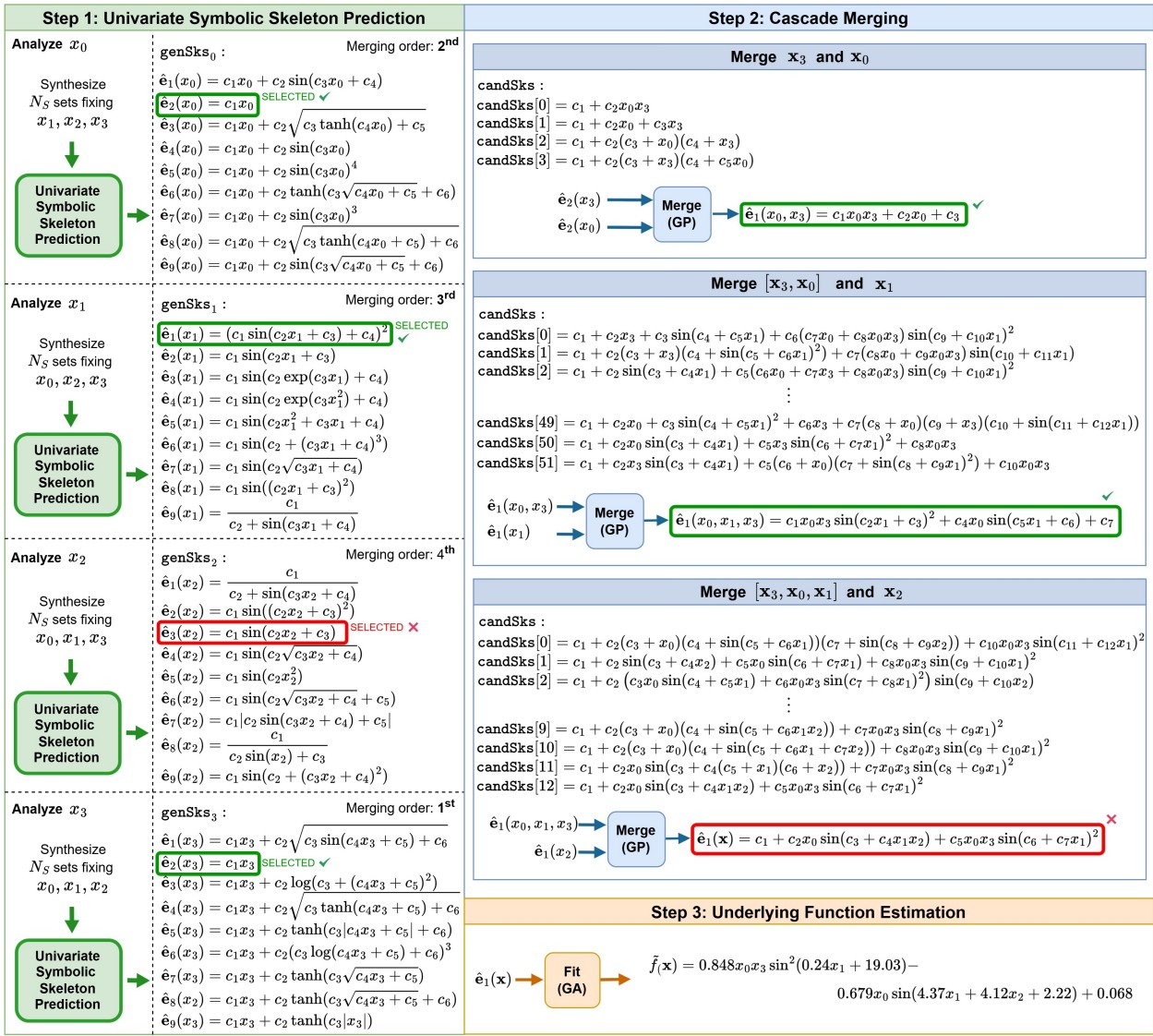

Figure 9: SeTGAP result for Feynman problem `I.50.26` with $f(\mathbf{x}) = x_0 \left(x_3 \cos^2\left(x_1 x_2\right) + \cos\left(x_1 x_2\right)\right)$

will focus on analyzing and formalizing the specific conditions and impacts that variable ordering has on the final merged expressions to better understand when a heuristic ordering is most critical.

