# OpenReview forum: "Decomposable Neural Symbolic Regression"
_TMLR — Under review for TMLR_

### Review · Reviewer_a95w · 2026-04-22

**Summary Of Contributions:**

This paper introduces SeTGAP, a decomposable symbolic regression (SR) method to discover interpretable mathematical expressions from data. The central contribution is a three-stage pipeline: (1) a Multi-Set Transformer generates multiple univariate symbolic skeletons (2) these skeletons are incrementally merged via a recursive GP-based cascade procedure that preserves the original univariate structures, and (3) a GA-based coefficient optimization refines the final multivariate expressions.

Strength:

1. The method shows consistent ground-truth recovery and novelty. It correctly recovered the functional form of all 13 synthetic problems across all 10 iterations.

2. The idea of decomposing multivariate SR into univariate skeleton prediction followed by structure-preserving merging is novel and well-motivated. The cascade approach provides interpretability at each stage of the construction process. This is importance for the users in practice.

3. The evaluation spans 13 custom synthetic problems with comparisons against 14 benchmark methods using appropriate statistical tests. The benchmarking is comprehensive.

Weakness:

1. There is no computational cost analysis. Since the paper acknowledges SeTGAP is less efficient than all compared methods, It's better to provide the runtime data for feasibility assessment.

2. The GA-based coefficient optimization consistently introduces small but nonzero errors even when the correct functional form is recovered, leading to statistically worse MSE than methods that achieve exact coefficient matches on simpler problems.

**Audience:**

Yes

**Audience Explanation:**

I think audience will be interested in it.

- researchers working on interpretable ML will find the post-hoc interpretability framing valuable. The idea of distilling opaque neural networks into faithful mathematical expressions addresses a need in high-stakes scientific applications.

- the community working on equation discovery in physics, chemistry, and engineering will appreciate both the method and the extensive Feynman benchmark evaluation.

**Broader Impact Concerns:**

There is no broader impact concern in the paper and I think there is no major ethical concerns are identified in this topic.

**Claims And Evidence:**

Yes

**Claims Explanation:**

I think the paper's central claim  is convincingly supported. Tables 2–4 and the extensive appendix tables (Tables 13–39) provide iteration-by-iteration expression comparisons across all methods, with shaded cells clearly indicating correct functional form recovery.

- The experimental protocol is sound: all methods are trained and evaluated on the same data partitions with consistent random seeds, and the evaluation is repeated 10 times per problem.

- The extrapolation superiority claim (Table 5) is well-supported, with the extended domain evaluation providing a meaningful test of whether learned expressions capture true functional relationships versus overfitting to in-domain patterns.

- The noise robustness claims (Table 6) are supported with a suitable experimental design, and the authors honestly report failure cases.

**Requested Changes:**

- The complete absence of runtime information is an omission. it could further provide: (a) wall-clock time per problem for SeTGAP and each competing method under equivalent hardware, (b) a breakdown of SeTGAP's runtime across pipeline stages , and (c) a discussion of how runtime scales with the number of variables.

- The variable ordering in the cascade is a core design choice and the evaluation is necessary. Conduct an experiment on at least 2–3 problems comparing: (a) the default MSE-based ordering, (b) reversed ordering, and (c) random orderings across multiple runs. It will be helpful to report whether the final recovered expressions and their quality metrics change meaningfully.

- The iteration-by-iteration expression tables (Tables 2–4, 13–39) are thorough but difficult to digest. Add a single summary table showing, for each method and each problem, the fraction of iterations (out of 10) where the correct functional form was recovered. This would make the paper's strongest result more accessible.

---

> ### Author Response · Authors · 2026-05-29
> **Response to Reviewer a95w**
>
> **Ch. 1: Computational Cost Analysis**
>
> **R:** We thank the reviewer for the recommendation. We have added a new Appendix H, which provides a detailed breakdown of the execution time for SeTGAP across the different SeTGAP pipeline stages for all tested synthetic problems.
>
> As we discuss in that appendix, while runtime generally increases with the number of variables due to the expanded search space, the relationship is highly dependent on the problem's functional topology. For instance, in problem E5, merging four variables was faster than merging three, as the number of viable skeleton combinations decreased, reducing the search effort. We clarify that the reported runtimes should be regarded as illustrative rather than absolute measures of computational complexity, as they are influenced by the specific skeletons generated and the inherent uncertainty of the initial opaque model $\hat{f}$ and the particular combinations of skeletons selected during each stage of the cascade merging process.
>
> **Ch. 2: Cascade Merging Ordering**
>
> **R:** Following the reviewer’s suggestion, we conducted an ablation study on the impact of variable ordering, now detailed in Appendix K. We compared our default MSE-based ordering against reversed and modified sequences across three problems from the Feynman dataset.
>
> Our results (Table 53) indicate that while the impact is problem-dependent, the cascade approach shows robustness to the merging ordering. In particular, for problem I.50.26, placing the best-identified skeletons first helps preserve structural fidelity throughout the process. However, for the problems, II.6.11 and II.13.17, the algorithm reached functionally similar results regardless of the sequence. This challenges the intuition that placing the worst-performing skeletons at the start of the merging process inevitably propagates structural uncertainty, and suggests that our merging logic is resilient to the merging sequence in some cases. Future work will focus on analyzing and formalizing the specific conditions and impacts that variable ordering has on the final merged expressions.
>
> **Ch. 3: Summary Table of Results**
>
> **R:** We thank the reviewer for this constructive suggestion. We have added Table 6, which provides a single, concise summary of the success rates (i.e., fraction of iterations where the correct functional form was recovered) for SeTGAP and all compared methods across the 10 iterations. This table allows for a more immediate comparison of the symbolic recovery performance, which we believe is the core strength of our decomposable approach.
>
> **All changes related to this review are highlighted in yellow in the paper.**
> **EDIT**: Table numbers were updated

---

### Review · Reviewer_AG2M · 2026-05-11

**Summary Of Contributions:**

The paper proposes a methods based for symbolic regression, which in its heart uses genetic algorithms + custom mutation and combination tools. The method works by fitting skeleton of functions and then fitting constants by genetic algorithm. The key novelty, in my view, is the way how skeletons are discovered, which is done by series of 1D fits.

**Audience:**

Yes

**Audience Explanation:**

Possibly some people in symbolic regression might be interested.

**Broader Impact Concerns:**

No concerns

**Claims And Evidence:**

No

**Claims Explanation:**

I do not know, what authors mean by term "explainable". The way how i understand it is that we can understand how the algorithms works. I think that what authors mean here is that it returns "short expression". I think this should be clarified.

Table 2-4 are quite difficult to follow. As I understand, the goal was to identify equations in Table 1. Why is not the ground truth shown in Table 2-4 as well?

What I do not understand is following. In Table 4, PySR has exactly recovered E11 and E13. How comes the extrapolation error in Table 5 is not zero for these cases?

I think that the tested examples are trivial and I do not know, if they are of practical interest. Can the authors demonstrate some practical example of their method? The method is designed for post-hoc explanation. What are the typical functions a user of the method should be interested in? I think they have to be restricted to simple equations.

Are there any other methods, which are explainable outside of symbolic regression methods? For example Kormogorov-arnoldi networks?

**Requested Changes:**

I would like authors to answer me the above questions.
1. First,  I would like to see some practically useful applications.
2. Is there some theory explaining why the proposed method is better than the other prior art?
3. What if the transformer used to fit one-dimensional function is trained on different distribution, i.e. it needs to extrapolate?
4. I would like a discussion, if there are functions which cannot be represented by a current method.

---

> ### Author Response · Authors · 2026-05-29
> **Response to Reviewer AG2M's Comments**
>
> ## Reviewer’s Comments to the Authors:
>
> **C1: I do not know what authors mean by term "explainable"**
>
> **R1:** We agree with the reviewer that “explainability” is a broad term. In this work, we follow the established definition of post-hoc global explanation. In that sense, SR can be used to approximate the internal function computed by an already trained opaque model using human-readable expressions as explanations.
>
> Specifically, SeTGAP distills a complex, opaque model $\hat{f}$ into a human-readable mathematical surrogate $\tilde{f}$ that describes the global functional mapping learned by the original model. As the reviewer correctly noted, this involves prioritizing short or parsimonious expressions, which are inherently more interpretable to human experts. We have updated the Introduction section to clarify this distinction and specify that our explanations take the form of these symbolic surrogates.
>
> **C2: Table 2-4 are quite difficult to follow?**
>
> **R2:** Thank you for the observation. These tables explicitly compare the expressions learned by different SR methods. However, we added a new Table 6 that summarizes how often these methods identified the correct functional form across the 10 iterations.
>
> **C3: PySR has exactly recovered E11 and E13. How comes the extrapolation error in Table 5 is not zero for these cases?**
>
> **R2:** Table 5 shows that PySR was successful during the first iteration. However, PySR has only recovered the correct functional form for E11 nine times and for E13 eight times. This can be verified by checking the expressions learned by PySR across the other iterations. For example, PySR failed in Iteration 5 for E11 (new Table 27) and Iteration 10 for E13 (new Table 42). Table 7 (previously, Table 5) shows the average extrapolation MSE and the std across iterations; thus, the average is different than zero for these cases.
>
> **C4: explainable outside of SR methods?**
>
> **R2:** This work aims to demonstrate that the decomposability approach leads to robust results in SR when the goal is to recover a human-readable mathematical expression that describes the function computed by an opaque model or a given dataset. Thus, it is compared against relevant SR approaches. A broad discussion and comparison against other explainability methods falls outside the scope of this paper. KANs, for example, do not share the same objective. Instead, they represent an architectural shift with respect to MLPs, replacing fixed linear weights with learnable univariate functions on the edges. While KANs can be “symbolified” after training (applying pruning repeatedly), they are fundamentally a modeling framework rather than a post-hoc symbolic discovery tool.
>
>
> **EDIT**: Table numbers were updated

---

> > ### Author Response · Authors · 2026-05-29
> > **Response to Reviewer AG2M's Requested Changes 1/2**
> >
> > ## Response to the requested changes:
> >
> > **Ch. 1: Practically useful application**
> >
> > **R:** We have updated the Introduction to include a broader discussion of the real-world applications of SR. It is also worth mentioning that our previous work (hidden [Ref. 1]) was successfully employed to address a precision-agriculture challenge. That study demonstrated the Multi-Set Transformer's ability to accurately identify the functional form of fertilizer-response curves in winter wheat dryland farming, providing interpretable and actionable models for agricultural management. This application was extended and reported in another recently published paper (hidden [Ref. 2]) that uses SeTGAP as a backbone.
> >
> > However, adding new practical experiments to the current paper goes beyond its intended scope. Our primary objective is to demonstrate that a decomposable SR approach is feasible and leads to more robust results than traditional SR methods that treat all variables simultaneously. We rely on established benchmarks for this validation because they offer the known ground-truth functional forms required to measure this robustness. This level of verification is generally not possible with real-world datasets where the underlying function is unknown.
> >
> > **Ch. 2: Theory explaining why the proposed method is better**
> >
> > **R:** We appreciate the reviewer’s request for a deeper theoretical grounding. While we do not provide a formal convergence proof, the superiority of SeTGAP is rooted in the theoretical principle of functional decomposability and constrained search.
> >
> > As established in [Ref 1], the Multi-Set Transformer is highly effective at identifying the correct univariate skeleton for each variable within a multivariate system. Traditional SR methods often fail because they attempt to discover the entire multivariate structure simultaneously, which leads to a combinatorial explosion of the search space and a high risk of getting trapped in local minima.
> >
> > The fundamental hypothesis of SeTGAP is that if the univariate functional forms of all variables have been successfully identified, a multivariate expression equivalent to the system’s underlying function must exist within a search space that respects these pre-identified structures. While prior work focuses on minimizing empirical error, often at the cost of structural accuracy, SeTGAP uses the initially identified skeletons as structural constraints. This ensures that the final expression remains aligned with the individual functional relationships identified by the Multi-Set Transformer, even when noise or low variable sensitivity might lead other methods to select incorrect but numerically close operators. As a consequence, we transform a high-dimensional symbolic search into a series of guided, incremental operations.

---

> > > ### Author Response · Authors · 2026-05-29
> > > **Response to Reviewer AG2M's Requested Changes 2/2**
> > >
> > > **Ch. 3: What if the transformer used to fit one-dimensional function is trained on different distribution, i.e. it needs to extrapolate?**
> > >
> > > **R:** We would like to clarify that the Multi-Set Transformer is a pre-trained model and does not require re-training specific problem instances. The model is specifically designed to be domain-agnostic and scale-invariant. During its pre-training phase on millions of synthetic symbolic skeletons, the transformer is exposed to a wide range of input domains and coefficient scales.
> > >
> > > For a given skeleton, such as $\mathbf{e}(x)=c_1 \sin(c_2 x) + c_3$, the training process involves sampling coefficients $(c_1, c_2, c_3)$ from random ranges. This effectively forces the transformer to learn the underlying symbolic identity rather than overfitting to specific numerical intervals. For example, during a single training epoch, $\mathbf{e}(x)$ might yield functions such as $1.2 \sin(x) + 0.8$, $-5.1 \sin(2x) + 0.45$, and $2.9 \sin(3x) - 1.48$. By sampling varied coefficients, the model effectively observes the sinusoidal form across multiple scales and shifts. If $x \in [-10, 10]$, the variation in the frequency coefficient (e.g., $2x$ or $3x$) makes the task equivalent to analyzing the function over expanded domains such as $[-20, 20]$ or $[-30, 30]$.
> > >
> > > In practice, the Multi-Set Transformer does not perform extrapolation in the traditional sense (i.e., predicting values outside a known range). Instead, it performs pattern recognition on the functional form of the data. Because it analyzes sets of input-response pairs that have been normalized or projected across various scales during training, the model is robust to cases where the test data distribution differs from the typical training examples. Its task is to retrieve the common functional form (i.e., the skeleton) which remains invariant regardless of whether the input $x$ is sampled from $[-1, 1]$ or $[-100, 100]$.
> > >
> > > **Ch. 4: functions which cannot be represented by a current method**
> > >
> > > **R:.** We have explained in the Discussion section (and Appendix I) that the representational capacity of SeTGAP is primarily constrained by two factors:
> > >
> > > 1. The transformer $g$ is pre-trained on a specific vocabulary and complexity threshold (up to seven operators and a maximum of two unary operators). Functions that exceed these bounds, such as problem F4 in the SRBench++ benchmark, and that require operators outside the considered vocabulary (e.g., differential equations) are currently outside the model's recovery range.
> > >
> > > 2. Since our approach distills a trained regression model $\hat{f}$, any failure of the NN to accurately approximate the underlying data, due to high epistemic uncertainty or insufficient training, directly affects the skeleton prediction. If $\hat{f}$ does not correctly capture the functional relationships, the Multi-Set Transformer will naturally produce incorrect univariate skeletons.
> > >
> > > **All changes related to this review are highlighted in green in the paper.**

---

### Review · Reviewer_9Vry · 2026-06-11

**Summary Of Contributions:**

This paper proposes SeTGAP, a symbolic regression framework that combines a Multi-Set Transformer, genetic algorithms (GA), and genetic programming (GP) to recover multivariate symbolic expressions from a trained opaque regression model. The key idea is to first identify univariate symbolic skeletons for each variable independently and then progressively merge them into a multivariate expression through a cascade procedure.

Strengths: The paper is generally well written, and the motivation of improving symbolic structure recovery is important. The proposed decomposition strategy is intuitive and the empirical results demonstrate promising performance on a collection of synthetic benchmark problems.
Weaknesses: I have several concerns regarding the problem formulation, applicability, novelty, methodological justification, and experimental validation.

**Audience:**

Yes

**Audience Explanation:**

The paper addresses a topic that is likely to be of interest to portions of the TMLR audience, particularly researchers working on symbolic regression, interpretable machine learning, scientific machine learning, equation discovery, and post-hoc model explanation. The proposed decomposition-based framework provides an alternative perspective on symbolic structure recovery and symbolic distillation, which may stimulate further discussion and future research in these areas.

**Broader Impact Concerns:**

No.

**Claims And Evidence:**

No

**Claims Explanation:**

(1). The proposed framework solves a substantially different problem from standard symbolic regression. A fundamental assumption underlying SeTGAP is the ability to generate an arbitrary number of new input-output pairs by repeatedly querying a trained opaque model (\hat{f}). The proposed univariate skeleton discovery procedure constructs artificial datasets in which one variable is varied while all remaining variables are fixed to randomly selected values. This process is repeated many times to generate the collections required by the Multi-Set Transformer.
While this assumption is valid in the symbolic distillation setting considered in the paper, it is considerably stronger than the assumptions typically made in symbolic regression.
In standard symbolic regression, one is given a finite dataset {x_i, y_i}, and no additional observations can be generated. In contrast, SeTGAP effectively gains unlimited access to new intervention-style data through model queries. This distinction is particularly important in scientific applications, such as biomedical imaging, where observations are expensive and fixed in advance. In such settings, practitioners generally cannot generate arbitrary new samples or independently vary one variable while holding all others constant.
Therefore, the proposed method relies on information that would not normally be available in standard symbolic regression settings. This raises an important question regarding the fairness of comparisons with existing SR methods, which operate only on the original dataset.
More importantly, it suggests that SeTGAP may be better characterized as a symbolic distillation or symbolic explanation framework rather than a conventional symbolic regression method.
I believe the paper should explicitly discuss this distinction and evaluate the method under restricted-query or fixed-dataset scenarios.

(2). Limited conceptual novelty relative to existing decomposition-based symbolic regression methods. A more thorough comparison with decomposition-based baselines would help clarify the novelty of the contribution.

(3).  The recursive merging strategy introduced in Section 3.3 is arguably the central technical component of the paper. However, many aspects of the procedure appear highly heuristic. Consequently, it remains unclear whether the observed performance gains arise from a principled decomposition framework or from carefully engineered search heuristics. Please explain the contributions of the each part in this framework.

(4). Scalability remains insufficiently demonstrated. Most experiments involve only two- to four-dimensional symbolic regression problems. The method generates multiple skeleton candidates per variable and performs repeated pairwise merging operations, which means that computational complexity may increase rapidly with dimensionality. Experiments involving larger-scale systems (e.g., 8–20 variables, PDE discovery tasks, or more challenging SRBench problems) would be necessary to establish practical scalability.

(5). Limited theoretical guarantees. Theoretical analysis remains relatively weak, including correctness of the merge procedure, convergence of the cascade process, recovery of the true symbolic expression, identifiability of recovered solutions. A discussion of identifiability conditions would significantly strengthen the work.

(6) Robustness to noise is only partially evaluated. Many real-world scientific datasets exhibit substantially higher levels of noise, measurement bias, missing observations, and distribution shifts. Please test the robustness to noise in these scenarios.

**Requested Changes:**

(1) The notation in Section 3.3 is difficult to follow due to the large number of nested indices and symbolic operators.

(2) Additional visualizations illustrating successful and unsuccessful merge operations would improve readability.

(3) Runtime and memory consumption should be reported more comprehensively.

(4) The computational overhead associated with generating and evaluating multiple skeleton candidates needs further discussion.

---

> ### Author Response · Authors · 2026-06-14
> **Response to Reviewer 9Vry’s comments 1/2**
>
> **C1: Comparison Fairness**
>
> **R1:** We thank the reviewer for this comment. The distinction raised regarding the data requirements and the operational framework of SeTGAP compared to traditional SR is an important point for clarifying the scope of this work. To address this, a discussion has been added to the Experimental Results section to contextualize the fairness of the comparisons and define the boundaries of our work.
>
> The comparison with standard SR methods remains fundamentally fair because all approaches, including SeTGAP, operate under identical initial information constraints for each tested problem. The standard methods map the fixed dataset directly to an expression via a data-to-expression pipeline. In contrast, SeTGAP implements a distillation pipeline where **the same fixed dataset is first used to train the opaque model** $\hat{f}$, which is subsequently queried by the explainable SR method to extract the final expression $\tilde{f}$. Because the opaque model is entirely bound by the constraints, sample size, and distribution of the original training data, **SeTGAP does not possess an unfair advantage regarding the underlying system information**. If the initial dataset is limited or poorly distributed, the opaque model will naturally yield an inadequate approximation of function $f$, and the resulting distilled expression will inherently reflect that mismatch.
>
> The challenge of limited data availability and scenarios with restricted experimentation addressed by the reviewer is indeed a critical constraint in many real-world applications. This exact problem was investigated in [Ref. 2], where the behavior of the framework was evaluated under conditions of severe epistemic uncertainty (e.g., regions with incomplete data) and high heteroskedastic noise. When restricted to an incomplete and small initial dataset, the framework produces expressions that achieve low prediction errors on the available data but fail to recover the underlying functions. To mitigate this, we introduced an adaptive sampling mechanism that recommends small batches of additional samples that are estimated to reduce epistemic uncertainty across the input domain. As such, we show how the discovered expressions evolve and quickly converge towards the expected underlying functions (more on this in our response R6).
>
> **C2: Conceptual novelty relative to existing decomposition-based SR methods**
>
> **R2:** We appreciate the opportunity to clarify the conceptual novelty of our framework relative to other decomposition-based approaches. We have added a new Section 3.4 (Comparison to Decomposition-based Methods) in the revised manuscript, which explicitly outlines the methodological advantages of SeTGAP compared to Control Variable Genetic Programming (CVGP) (Jiang & Xue, 2023) and ScaleSR (Chu et al., 2024).
>
> **C3: Please explain the contributions of each part in this framework**
>
> **R3:** Thank you for the suggestion. We added a brief summary at the beginning of Section 3.3 that outlines the specific contribution and purpose of each component.
>
> **C4: Scalability**
>
> **R4:** While we acknowledge the reviewer's concerns regarding the computational overhead of combinatorial merging, the characterization of our evaluation as being limited to four variables is inaccurate. As detailed in Section 4.2 and Appendix J, **SeTGAP was evaluated on 78 problems from the Feynman dataset within the SRBench framework, containing problems with up to 9 variables.**
>
> Nevertheless, we agree that extreme scaling remains an inherent challenge for this methodology due to the computational cost of the merging process, a limitation that is addressed in the Discussion section. When evaluating these computational demands, it is necessary to consider the primary objective of this work, which is to demonstrate that a decomposable neural SR approach is feasible and leads to more robust structural results than traditional methods that attempt to treat all variables simultaneously.
>
> Furthermore, the scalability of SeTGAP should be contextualized within the current state of neural SR. Existing baseline neural SR approaches are severely constrained by strict architectural caps dictated by the maximum number of variables used during their transformer pre-training phases. For example, NeSymReS and E2E are limited to a maximum of three and five variables, respectively.
>
> Finally, PDE discovery tasks fall outside both the scope of this work and standard conventional SR benchmarks. However, we are currently developing an extension of our framework for ODEs and, potentially, for PDEs in future work. Also, evaluating additional datasets from SRBench is not currently feasible, as the only two problem suites that feature ground-truth symbolic expressions are the Feynman and the ODE-Strogatz databases, the latter of which is not applicable to SeTGAP in its current form

---

> ### Author Response · Authors · 2026-06-14
> **Response to Reviewer 9Vry’s comments 2/2**
>
> **C5: Limited theoretical analysis**
>
> **R5:** We appreciate the reviewer’s request for a deeper theoretical grounding. While we do not provide a formal correctness or convergence proof, we argue that the superiority of SeTGAP is rooted in the principle of functional decomposability and constrained search. In particular, traditional SR methods often attempt to discover the entire multivariate expressions using all variables simultaneously. This approach triggers a combinatorial explosion of the search space, making the optimization landscape highly prone to getting stuck in local minima. SeTGAP mitigates this vulnerability by operating on the hypothesis that if the univariate functional forms of all variables are successfully isolated and identified by the Multi-Set Transformer, an expression equivalent to the system’s underlying function must exist within a search space bounded by these pre-identified structures. By using these univariate structures to constrain the subsequent search, we transform an unconstrained multivariate symbolic search into a series of guided and incremental operations. This has been added to the second paragraph of the Discussion section.
>
> We acknowledge that a formal discussion of identifiability conditions and theoretical guarantees has not been established in this work, and we agree that it constitutes a potential path for future research. Having demonstrated empirically the feasibility and robustness of our decomposable neural SR approach, our next step is to provide a theoretical analysis which will focus on characterizing the boundaries of identifiability relative to acceptable epistemic and aleatoric uncertainty levels, the optimal number of input sets $N_S$, the impact of the variable merging sequence, and the required number of skeleton candidates per variable.
>
> **C6: Robustness to noise is only partially evaluated**
>
> **R6:** We appreciate the reviewer’s suggestion to evaluate our method under more challenging conditions. As noted in our previous response R1, we have studied the behavior of SeTGAP under varying levels of epistemic uncertainty and heteroskedastic noise in [Ref. 2]. To explicitly address this comment within this manuscript, we have added an illustrative example at the end of Section 4.1. This addition details how the framework performs when restricted to an initially incomplete and noisy dataset, and demonstrates how it successfully recovers the underlying function throughout an adaptive sampling process.

---

> > ### Author Response · Authors · 2026-06-14
> > **Response to Reviewer 9Vry’s Requested Changes**
> >
> > **CH1: Notation in Section 3.3**
> >
> > **R:** As pointed out in our response R3, we added a brief summary at the beginning of the section that helps to follow our notation convention.
> >
> >
> > **CH2: Additional visualizations**
> >
> > **R:** Thanks for the suggestion. While Fig. 1 already illustrates a successful instance of the merging process, we have included a new diagram, Fig. 9, which provides a more detailed trace of an unsuccessful execution path.
> >
> >
> > **CH3: Runtime and memory consumption should be reported more comprehensively**
> >
> > **R:** We would like to point out that Table 44 in the Appendix already provides a stage-by-stage breakdown of the runtime performance across all 13 synthetic problems. The table delineates the runtime of each core phase of the SeTGAP framework, including univariate skeleton generation, skeleton evaluation, cascade merging at progressive variable dimensions, and final coefficient optimization. This report helps us to illustrate how the computational overhead is distributed across different problem topologies and dimensionalities.
> > Regarding memory consumption, precise hardware profiling logs were not recorded during the execution of these problems, as tracking memory footprints is generally not standard practice within the SR literature. However, we can provide further discussion about memory consumption during the skeleton generation step in the next question.
> >
> >
> > **CH4: Computational overhead associated with generating and evaluating multiple skeleton candidates**
> >
> > **R:** The computational cost of generating a single univariate skeleton candidate for a given data collection $\tilde{\mathbf{D}}\_v$ (as described in Sec. 3.2.1) requires approximately 1.47 seconds and 392 GFLOPs, and incurs a peak memory footprint of 149.47 MB. In the worst-case scenario, this generation process is repeated $n_{{cand}} n_B = 9$ times per variable. Because these transformer inference passes are executed sequentially for each variable, the peak memory footprint remains strictly bounded at 149.47 MB, while the cumulative computational cost scales linearly to 3,533 GFLOPs per variable.
> >
> > Regarding the subsequent skeleton evaluation phase, the computational overhead is highly dynamic and depends on the structural complexity of the generated candidates. As demonstrated by the variance in the "Skel. Eval." column of Table 44, the runtime of the GA varies based on the number of coefficient placeholders that must be optimized for a given candidate skeleton. For instance, a basic linear skeleton such as $c_1 x + c_2$ requires the GA to fit only two parameters. In contrast, a more complex non-linear skeleton, such as $c_1 \sin(c_2 x + c_3)(c_4 x + c_5) + c_6$, requires optimizing six coefficients, which significantly expands the parameter search space. Therefore, the computational effort and convergence rates of the GA populations vary naturally across iterations, depending on the specific functional relationships and mathematical operators identified for each variable.
> >
> >
> > **All changes related to this review are highlighted in blue in the paper.**